# Bi-Level Offline Policy Optimization with Limited Exploration

**Wenzhuo Zhou**
Department of Statistics
University of California Irvine
wenzhuz3@uci.edu

## Abstract

We study offline reinforcement learning (RL) which seeks to learn a good policy based on a fixed, pre-collected dataset. A fundamental challenge behind this task is the distributional shift due to the dataset lacking sufficient exploration, especially under function approximation. To tackle this issue, we propose a bi-level structured policy optimization algorithm that models a hierarchical interaction between the policy (upper-level) and the value function (lower-level). The lower level focuses on constructing a confidence set of value estimates that maintain sufficiently small weighted average Bellman errors, while controlling uncertainty arising from distribution mismatch. Subsequently, at the upper level, the policy aims to maximize a conservative value estimate from the confidence set formed at the lower level. This novel formulation preserves the maximum flexibility of the implicitly induced exploratory data distribution, enabling the power of model extrapolation. In practice, it can be solved through a computationally efficient, penalized adversarial estimation procedure. Our theoretical regret guarantees do not rely on any data-coverage and completeness-type assumptions, only requiring realizability. These guarantees also demonstrate that the learned policy represents the "best effort" among all policies, as no other policies can outperform it. We evaluate our model using a blend of synthetic, benchmark, and real-world datasets for offline RL, showing that it performs competitively with state-of-the-art methods.

## 1   Introduction

Offline reinforcement learning (RL) is a task to learn a good policy using only a pre-collected, fixed dataset, without further exploration with the environment. This distinctive characteristic positions offline RL as a promising approach for solving real-world sequential decision-making problems in healthcare [35, 61], financial marketing [46], robotics [47] and education [32], as acquiring diverse or expert-quality data in these fields can be costly or practically unattainable.

Arguably, two of the biggest challenges in offline RL are the distributional shift between the data-generating distribution and those induced by candidate policies, and the stringent requirements on the properties of function approximation [28]. It has been observed that, in practice, the distributional mismatch often results in unsatisfactory performance of many existing algorithms, and even amplifying with function approximation [18, 27]. Many prior works [39, 13, 2, 14] crucially rely on a global data-coverage assumption and completeness-type function approximation condition in a technical sense. The former necessitates that the dataset to contain any state-action pair with a lower bounded probability so that the distributional shift can be well calibrated. The latter requires the function class to be closed under Bellman updates. Both assumptions are particularly strong and are likely to be violated in practice [58]. Consequently, algorithms that depend on these assumptions may experience performance degradation and instability [52]. Therefore, it is crucial to develop novel algorithms that relax these assumptions, offering robust and widely applicable solutions for real-world scenarios.

37th Conference on Neural Information Processing Systems (NeurIPS 2023).

To address the aforementioned challenges in offline settings, one fundamental principle is the concept of pessimism, which aims to maximize rewards in the worst possible MDP consistent with the offline dataset [18, 53]. In practice, these methods have generally been shown to be more robust when coverage assumptions are violated [27].

Although many such pessimistic algorithms have been developed, very few works can tackle data-coverage and function approximation issues simultaneously, while establishing strong regret guarantees. For instance, deep offline RL algorithms [18, 25, 23] exhibit impressive empirical performance, but their theoretical consistency guarantees are limited to tabular Markov decision processes (MDPs). The works of [30, 41, 22, 53, 56, 58, 10] relax the global coverage to a partial coverage condition, wherein the offline data only covers a single comparator policy. However, all of these methods require Bellman completeness for the function class. The most recent works [9, 59] take a significant step towards relaxing Bellman completeness to realizability, that the function class can capture the target ground-truth function. Nonetheless, these algorithms are unable to provide a meaningful regret guarantee without any data-coverage assumption (when both global and partial coverage fails), and also empirical evaluations are absent. Even without additional conditions, the learned policies of these algorithms can only compete with the (Bellman flow) optimal policy, resulting in a lack of robustness when the optimal policy is not covered by data, a situation that frequently occurs. Due to page limit, we have only discussed the closest related work here, and the rest is deferred to Appendix.

**Our contribution.** In this paper, we develop a provably sample-efficient offline RL framework. Our information-theoretic algorithm is designed based on the concept of bi-level (upper and lower level) structured optimization, which leads to a hierarchical interpretation and naturally enjoys learning stability and algorithmic convergence from a game-theoretic perspective. In particular, at the lower level, one component is to construct a confidence set with consistent value estimates regarding the appropriately small weighted average Bellman error, effectively preventing overly pessimistic evaluation. Meanwhile, the second component, which deals with uncertainty control, implicitly enhances the power of model extrapolation. In addition to the information-theoretic algorithm, we also develop a computationally efficient counterpart that is solved by a penalized adversarial estimation algorithm with proximal-mapping updating, allowing both non-linear and linear function approximation. From a theoretical standpoint, we establish a strong regret guarantee for both information-theoretical and practical algorithms under only realizability *without* requiring any data-coverage (neither global nor partial coverage) and completeness-type assumptions. As a special case study, we further refine our developed mixture density ratio-based concentrability coefficient to a relative condition number in linear MDP settings. The sample complexity of our regret bound improves or at least matches the prior results in the fully exploratory or partial coverage settings where the Bellman-completeness holds. Notably, compared with existing works, either focusing on theoretical or empirical development, we provide a comprehensive theoretical analysis of the proposed framework and also conduct synthetic, benchmark, and real data experiments for empirical evaluation.

## 2 Preliminaries and Notations

**Markov decision process.** We consider an infinite-horizon discounted MDP $\mathcal{M} = \{\mathcal{S}, \mathcal{A}, \mathbb{P}, \gamma, r, s^0\}$ [40], where $\mathcal{S}$ is the state space, $\mathcal{A}$ is the action space, $\mathbb{P} : \mathcal{S} \times \mathcal{A} \to \Delta(\mathcal{S})$ is the Markov transition kernel for some probabilistic simplex $\Delta$, $r : \mathcal{S} \times \mathcal{A} \to [0, \bar{R}]$ is the reward function for $\bar{R} \geq 0$, $\gamma \in [0, 1)$ is the discounted factor and $s^0$ is the initial state. A policy $\pi : \mathcal{S} \to \Delta(\mathcal{A})$ induces a distribution of the trajectory $s^0, a^0, r^0, s^1, \ldots$, where $a^t \sim \pi(\cdot|s^t), r^t = r(s^t, a^t), s^{t+1} \sim \mathbb{P}(\cdot|s^t, a^t)$ for any $t \geq 0$. The expected discounted return of a policy is defined as $J(\pi) = \mathbb{E}[\sum_{t=0}^{\infty} \gamma^t r^t | \pi]$. The discounted return when the trajectory starts with $(s, a)$ and all remaining actions are taken according to $\pi$ is called q-function $q^\pi : \mathcal{S} \times \mathcal{A} \to [0, \bar{V}]$. The $q^\pi$ is the unique fixed point of the Bellman operator $\mathcal{B}^\pi$, satisfying the Bellman equation [45]: $\mathcal{B}^\pi q(s, a) := r(s, a) + \gamma \mathbb{E}_{s' \sim \mathbb{P}(\cdot|s,a)}[q(s', \pi)]$. Here $q(s', \pi)$ is denoted as shorthand for $\mathbb{E}_{a' \sim \pi(\cdot|s')}[q(s', a')]$, and we define $\mathbb{P}^\pi q(s, a) := \mathbb{E}_{s' \sim \mathbb{P}(\cdot|s,a)}[q(s', \pi)]$. Additionally, it is helpful to remember that $J(\pi) = q^\pi(s^0, \pi)$. Another important notion is the normalized discounted visitation of $\pi$, defined as $d_\pi(s, a) := (1 - \gamma) \sum_{t=0}^{\infty} \gamma^t d_{\pi,t}(s, a)$, where $d_{\pi,t}$ is the marginal state-action distribution at the time-step $t$.

**Offline RL under function approximation.** In the offline RL setting, there exists an unknown offline data-generating distribution $\mu$ induced by behavior policies. Despite the unknowns of $\mu$, we can observe a set of transition pairs, as offline dataset $\mathcal{D}_{1:n} := \{s_i, a_i, r_i, s_i'\}_{i=1}^n$ sampling from $\mu$. For a given policy $\pi$, the density-ratio (importance-weight), $\tau_{d_\pi/\mu}(s, a) = d_\pi(s, a)/\mu(s, a)$, measures how

effectively $\mu$ covers the visitation induced by $\pi$. The primary objective of offline policy optimization is to learn an optimal policy that maximizes the return, $J(\pi)$, using the offline dataset. Under the function approximation setting, we assume access to two function classes $\mathcal{Q} : \mathcal{S} \times \mathcal{A} \to \mathbb{R}$ and $\Omega : \mathcal{S} \times \mathcal{A} \to \mathbb{R}$, which are utilized to capture $q^\pi$ and $\tau_{d_\pi/\mu}$, respectively.

**Exploration and coverage.** In general, when saying an offline dataset is well-explored, it means that a well-designed behavior policy has been executed, allowing for comprehensive exploration of the MDP environment. As a result, the dataset is likely to contain possibly all state-action pairs. This implicitly requires $\mu$ has the global coverage [18, 49]. In this context, the global coverage means that the density ratio-based concentrability coefficient, $\sup_{s,a}\{d_\pi(s,a)/\mu(s,a)\}$, is upper-bounded by a constant $c \in \mathbb{R}^+$ for all policies $\pi \in \Pi$, where $\Pi$ is some policy class. This condition is frequently employed in offline RL [2, 8, 12]. However, in practice, this assumption may not hold true, as devising an exploratory policy is a challenging task for large-scale RL problems. Instead, our goal is to learn a good policy with strong theoretical guarantees that can compete against any arbitrarily covered comparator policy under much weaker conditions than the global coverage.

## 3 Bi-Level Offline Policy Optimization Algorithm

In this section, we introduce our bi-level offline policy optimization framework. The development of the framework consists of three major steps.

**Step 1: robust interval learning.** In this step, we aim to provide a robust off-policy interval evaluation. The major advantage of this interval formulation is its robustness to the model-misspecification of the importance-weight class $\Omega$, and the encoding of distributional-shift information in the policy evaluation process. First, we define a detection function $\mathbb{D}(\cdot)$, which is used to measure the degree of the distributional-shift in terms of density ratio.

**Definition 3.1.** *For $x, c_1, c_2, C \in \mathbb{R}^+$ and $C \geq 1$, the detection function $\mathbb{D}(\cdot)$ satisfies the following conditions: (1) 1-minimum: $\mathbb{D}(1) = 0$. (2) Non-negativity: $\mathbb{D}(x) \geq 0$. (3) Boundedness on first-order derivative: $|\mathbb{D}'(x)| \leq c_2$ if $x \in [0, C]$. (4) Boundedness on value: $|\mathbb{D}(x)| \leq c_1$ for $x \in [0, C]$. (5) Strong convexity: $\mathbb{D}(x)$ is $M$-strongly convex with respect to $x$.*

The family of Rényi entropy [42], Bhattacharyya distance [11], and simple quadratic form functions [60]all satisfy the conditions outlined in Definition 3.1. Under this definition, it can easily observe that $\mathbb{D}$ has a convex conjugate function [6], $\mathbb{D}_*$ with $\mathbb{D}_*(x_*) = \sup_x \{x \cdot x_* - \mathbb{D}(x)\}$, that satisfies $\mathbb{D}_*(0) = 0$. It follows from Bellman equation $\mathcal{B}^\pi q^\pi(s, a) = q^\pi(s, a)$ for any $s, a$, then $J(\pi) = q^\pi(s^0, \pi) + \mathbb{E}_\mu[\lambda \mathbb{D}_*((\mathcal{B}^\pi q^\pi(s, a) - q^\pi(s, a)/\lambda))/(1 - \gamma)]$ for $\lambda \geq 0$. Applying Fenchel-Legendre transformation [36, 21], and model $x$ in a restricted importance weight class $\Omega$ for any $s, a$, we obtain

$$J(\pi) = q^\pi(s^0, \pi) + \mathbb{E}_\mu[\sup_x x \cdot (\mathcal{B}^\pi q^\pi(s, a) - q^\pi(s, a)) - \lambda \mathbb{D}(x)]/(1 - \gamma) \tag{1}$$

$$\geq q^\pi(s^0, \pi) + \mathbb{E}_\mu[\tau(s, a)(r(s, a) + \gamma q^\pi(s', \pi) - q^\pi(s, a)) - \lambda \mathbb{D}(\tau(s, a))]/(1 - \gamma). \tag{2}$$

Suppose $q^\pi$ is well-sepcified, i.e., $q^\pi \in \mathcal{Q}$, we can find a lower bound of (2), which is valid for any $\tau \in \Omega$, via replacing $q^\pi$ with $\inf_{q \in \mathcal{Q}}$ as follows:

$$J(\pi) \geq \inf_{q \in \mathcal{Q}} \left\{ \underbrace{(\mathbb{E}_\mu [\tau(s, a)(r(s, a) + \gamma q(s', \pi) - q(s, a))] + q(s^0, \pi))/(1 - \gamma)}_{:= H(\tau, q, \pi)} \right.$$
$$\left. - \underbrace{\lambda/(1 - \gamma)\mathbb{E}_\mu[\mathbb{D}(\tau(s, a))]}_{:= \lambda \xi(\mathbb{D}, \tau)} \right\}.$$

After following a similar derivation, we can establish an upper bound for $J(\pi)$ as well, and thus construct a value interval for $J(\pi)$. This interval holds for any $\tau$ and is therefore robust against model-misspecification of $\Omega$. In order to obtain a tighter interval, we can shrink the interval width by maximizing the lower bound and minimizing the upper bound, both with respect to $\tau$. This procedure can be interpreted as searching for some good $\tau \in \Omega$ to minimize the function approximation error.

$$J(\pi) \in \left[ \sup_{\tau \in \Omega} \inf_{q \in \mathcal{Q}} H(\tau, q, \pi) - \lambda \xi(\mathbb{D}, \tau), \inf_{\tau \in \Omega} \sup_{q \in \mathcal{Q}} H(\tau, q, \pi) + \lambda \xi(\mathbb{D}, \tau) \right], \tag{3}$$

While the interval offers a robust method for dealing with the bias introduced by function approximation when estimating $J(\pi)$, it lacks a crucial and non-trivial step for handling statistical uncertainty.

**Step 2: uncertainty quantification.** In this step, we quantify the uncertainty of the interval (3), and establish a non-asymptotic confidence interval (CI) for $J(\pi)$ which integrates bias and uncertainty quantifications in a single interval inspired by [60]. Given offline data $\mathcal{D}_{1:n}$, our formal result for quantifying sampling uncertainty in order to establish the CI for $J(\pi)$.

**Theorem 3.1** (Non-asymptotic confidence interval). *For a target policy $\pi$, the return $J(\pi)$ is within a CI for any $\tau \in \Omega$ with probability at least $1 - \delta$, i.e., $J(\pi) \in [\widehat{J}_n^-(\pi; \tau), \widehat{J}_n^+(\pi; \tau)]$ for*

$$\widehat{J}_n^-(\pi; \tau) := \frac{1}{n} \sum_{i=1}^{n} \frac{r_i \tau(s_i, a_i)}{1 - \gamma} - \sup_{q \in \mathcal{Q}} \widehat{M}_n(-q, \tau) - \lambda \xi_n(\mathbb{D}, \tau) - \sigma_n,$$

$$\widehat{J}_n^+(\pi; \tau) := \frac{1}{n} \sum_{i=1}^{n} \frac{r_i \tau(s_i, a_i)}{1 - \gamma} + \sup_{q \in \mathcal{Q}} \widehat{M}_n(q, \tau) + \lambda \xi_n(\mathbb{D}, \tau) + \sigma_n, \qquad (4)$$

*if the uncertainty deviation $\sigma_n$ satisfies*

$$P\left( \sup_{\tau \in \Omega} \left| \frac{1}{n(1 - \gamma)} \sum_{i=1}^{n} \tau(s_i, a_i)(r_i + \gamma q^\pi(s_i', \pi) - q^\pi(s_i, a_i)) - \lambda \xi_n(\mathbb{D}, \tau) \right| \leq \sigma_n \right) \geq 1 - \delta,$$

*where $\widehat{M}_n(q, \tau) := \sum_{i=1}^{n} \tau(s_i, a_i)(\gamma q(s_i', \pi) - q(s_i, a_i))/(1 - \gamma)n + q(s^0, \pi)$.*

Similar to the value interval, the CI $[\widehat{J}_n^-(\pi; \tau), \widehat{J}_n^+(\pi; \tau)]$ also holds for any $\tau \in \Omega$. Therefore, we can optimize the confidence lower and upper bounds in (4) over $\tau \in \Omega$ to tighten the CI, and obtain:

$$P\left( J(\pi) \in [\sup_{\tau \in \Omega} \widehat{J}_n^-(\pi; \tau), \inf_{\tau \in \Omega} \widehat{J}_n^+(\pi; \tau)] \subseteq [\widehat{J}_n^-(\pi; \tau), \widehat{J}_n^+(\pi; \tau)) \right) \geq 1 - \delta.$$

**Step 3: bridge policy evaluation to policy optimization.** In this step, we aim to formulate a policy optimization based on the derived high-confidence policy evaluation from the previous steps. Given the consistent CI estimation of $J(\pi)$, we can naturally incorporate the pessimism principle, i.e., using the CI lower bounds of $J(\pi)$ as the value estimate of the policy evaluation of $\pi$ [22]. With such a procedure, our objective is to maximize these lower bounds over some family $\Pi$ of policies:

$$\max_{\pi \in \Pi} \left\{ \sup_{\tau \in \Omega} \widehat{J}_n^-(\pi; \tau) \right\}. \qquad (5)$$

Although (5) is algorithmically feasible for obtaining a policy solver $\widehat{\pi}$, it lacks direct interpretation without taking advantage of the bi-level optimization structure in hindsight. Therefore, we propose to reformulate (5) via a *dual-to-prime conversion* (shown in Theorem 3.2), which naturally lends itself to lower-upper optimization with guaranteed convergence. Specifically, we formulate (5) as a bi-level framework problem:

(Upper Level) $\quad \min_{\pi \in \Pi} -\underline{q}^\pi(s^0, \pi),$ $\qquad\qquad\qquad\qquad\qquad\qquad\qquad (6)$

(Lower Level) $\quad s.t. \ \underline{q}^\pi \in \arg\min_{q \in \mathcal{Q}_{\varepsilon_n}} q(s^0, \pi),$ $\qquad\qquad\qquad\qquad\qquad (7)$

**$\underline{Consistency}$** : $\quad \mathcal{Q}_{\varepsilon_n} = \left\{ q \in \mathcal{Q} : \sup_{\tau \in \widetilde{\Omega}_{\widetilde{\sigma}_n}} \left| n^{-1} \sum_{i=1}^{n} \tau(s_i, a_i)(r_i + \gamma q(s_i', \pi) - q(s_i, a_i)) \right| \leq \varepsilon_n \right\},$

**$\underline{Uncertainty\ Control}$** : $\quad \widetilde{\Omega}_{\widetilde{\sigma}_n} = \left\{ \tau_\circ / \sup_{\tau_\circ \in \Omega} \|\tau_\circ\|_\Omega \text{ for } \tau_\circ \in \Omega : \xi_n(\mathbb{D}, \tau_\circ)) \leq \widetilde{\sigma}_n \right\}.$

At the upper level, the learned policy $\widehat{\pi}$ attempts to maximize the value estimate of $\underline{q}^\pi$ over some policy class $\Pi$, while at the lower level, $\underline{q}^\pi$ is to seek the $q$-function with the pessimistic policy evaluation value from the confidence set $\overline{\mathcal{Q}}_{\varepsilon_n}$ with consistency guarantee and uncertainty control. For *consistency*, whenever $q^\pi$ or its good approximator is included in $\mathcal{Q}$ (realizability for $\mathcal{Q}$ class is satisfied), the set $\mathcal{Q}_{\varepsilon_n}$ ensures the estimation consistency of $q^\pi$ in terms of "sufficiently small" weighted average Bellman error. For *uncertainty control*, the constrained set $\widetilde{\Omega}_{\widetilde{\sigma}_n}$ attempts to control the uncertainty arising from distributional shift via a user-specific thresholding hyperparameter $\widetilde{\sigma}_n$. The feasible (uncertainty controllable) candidates $\tau \in \widetilde{\sigma}_n$ are used as weights for the average Bellman error, helping to construct the consistent set $\mathcal{Q}_{\varepsilon_n}$. Risk-averse users can specify a lower value for the thresholding hyperparameter or consider a higher $\widetilde{\sigma}_n$ to tolerate a larger distribution shift. In other words, the chosen value of $\widetilde{\sigma}_n$ depends on the degree of pessimism users want to incorporate in the policy optimization.

**Theorem 3.2.** *There must exist some threshold values $\varepsilon_n$ and $\widetilde{\sigma}_n$, the return policy of (5) $\widehat{\pi}$ satisfies the minimization problem in (6), indicating the solution of the optimization (5) and (6) is equivalent.*

Interestingly, the new form in (6) characterizes our policy optimization framework as a two-player general-sum game [16], which is a sequential game involving two players. Each player aims to maximize their own payoffs while considering the decisions of other players. Our bi-level optimization framework has been demonstrated to improve learning stability and ensure algorithmic convergence, benefiting from the existence of a local equilibrium [51].

To close this section, we remark that the establishment of *consistency* with respect to the weighted average Bellman error is the key point for us to relax the completeness-type assumptions. In the famous API/AVI-type algorithms [13, 14, 8], they target to minimize a squared or minimax Bellman error for finding $q \in \mathcal{Q}$ so that $\|q - \mathcal{B}^\pi q\|_{L_2(\mu)}^2 \approx 0$ to obtain $q \approx q^\pi$. Unfortunately, even with the infinite amount of data, the empirical estimate of $\|q - \mathcal{B}^\pi q\|_{L_2(\mu)}^2$, i.e., squared empirical Bellman error) is biased due to the appearance of unwanted conditional variance, i.e., the *double sampling* issue [3]. The API/AVI-type algorithms need a separate helper function class $\widetilde{\mathcal{Q}}$ for modeling $\mathcal{B}^\pi q$, and [8] has shown that when the class $\widetilde{\mathcal{Q}}$ realizes the Bayes optimal regressor $\mathcal{B}q$ (Bellman-completeness condition), the estimation is consistent and unbiased. In contrast, thanks to not using the squared loss, our weighted average Bellman error can be estimated from an unbiased estimate without concern about the *double sampling* issue, and thus no need for any completeness-type conditions.

# 4 Information-Theoretic Results

In this section, we provide theoretical analyses of our algorithm, which reveals the advantages of the proposed policy optimization method from a technical standpoint.

Notably, to the best of our knowledge, Theorem 4.1 is the first result of regret guarantee under only realizability *without* requiring any data coverage or completeness-type assumptions. Additionally, in contrast to most existing works that assume finite function classes, we carefully quantify the space complexities for infinite function classes (e.g., a class of real-valued functions generated by neural networks) using Pollard's pseudo-dimension [38]. The formal definition is provided in Appendix. It notices that the pseudo-dimension is a generalization of the well-known VC dimension [50]. In the following, we first introduce the necessary assumptions before presenting the guarantees for our algorithm.

**Assumption 1** (Realizability for $q$-function class). *For any policy $\pi \in \Pi$, we have $q^\pi \in \mathcal{Q}$. When this assumption holds approximately, we measure violation by $\inf_{q \in \mathcal{Q}} \sup_\rho \mathbb{E}_\rho[(q(s,a) - \mathcal{B}^\pi q(s,a))^2] \leq \varepsilon_{\mathcal{Q}}$, where $\varepsilon_{\mathcal{Q}} \geq 0$ and $\rho$ is some data distribution such that $\rho \in \{d_{\widetilde{\pi}} : \widetilde{\pi} \in \Pi\}$.*

We would like to emphasize that we do not require Bellman-completeness condition [53, 58], which is much stronger than the realizability condition. In addition, we do not impose realizability on the importance-weight class $\Omega$, thereby allowing model misspecification on $\Omega$. Having stated the major assumptions, we now turn to the routine ones on boundedness before presenting the main results.

**Assumption 2** (Boundedness on $\mathcal{Q}$). *There exists a non-negative constant $\bar{V} < \infty$, the function $q(s,a) \in [0, \bar{V}], \forall q \in \mathcal{Q}, s \in \mathcal{S}, a \in \mathcal{A}$.*

**Assumption 3** (Boundedness on $\Omega$). *There exists a non-negative constant $1 \leq \mathcal{U}_\infty^\tau < \infty$, the function $\tau(s,a) \in [0, \mathcal{U}_\infty^\tau], \forall \tau \in \Omega, s \in \mathcal{S}, a \in \mathcal{A}$.*

**Theorem 4.1.** *Under Assumptions 1-3 and denote supremum of $\mu$-weighted $L_2$ norm of $\Omega$, i.e., $\sup_{\tau \in \Omega} \|\tau(s,a)\|_{L_2(\mu)}$, as $\mathcal{U}_2^\tau$. Let $\widehat{\pi}$ be the output of solving (6) when we set $\varepsilon_n = \widetilde{\mathcal{O}}(n^{-1/2}\mathcal{U}_2^\tau(\sqrt{\ln\{\text{Vol}(\Theta)/\delta\}} + \mathcal{U}_\infty^\tau\sqrt{\varepsilon_{\mathcal{Q}}})$ and $\widetilde{\sigma}_n = \widetilde{\mathcal{O}}(n^{-1/2}\mathcal{U}_2^\tau L\sqrt{\ln\{\text{Vol}(\Theta)/\delta\}} + M(\mathcal{U}_2^\tau - 1)^2)$, then for any policy $\pi \in \Pi$ and some constant $\mathcal{U}_2^\star \in [1, \mathcal{U}_2^\tau)$, w.p. $\geq 1 - \delta$,*

$$J(\pi) - J(\widehat{\pi}) \leq \frac{1}{1-\gamma}\widetilde{\mathcal{O}}\left( \underbrace{\mathcal{U}_2^\star \mathfrak{C}_{\bar{V},L}\sqrt{\frac{\ln\{\text{Vol}(\Theta)/\delta\}}{nM}}}_{\epsilon_\sigma} + \underbrace{\sqrt{\frac{\mathfrak{C}_{\mathcal{U}_\infty^\tau}}{M}}\max\{(\varepsilon_{\mathcal{Q}})^{1/2}, (\varepsilon_{\mathcal{Q}})^{3/4}\}}_{\epsilon_{mis}} \right.$$

$$\left. + \min_{\left\{\rho: \left\|\frac{\rho}{\mu}\right\|_{L_2(\mu)} \leq \mathcal{U}_2^\star\right\}}\left\{\mathbb{E}_{(d_\pi - \rho)^+}\left[\underbrace{\mathbb{1}_{\mu=0}(\mathbb{I} - \gamma\mathbb{P}^\pi)\Delta_{\overline{q^\pi} - \underline{q^\pi}}(s,a)}_{\epsilon_{off}} + \underbrace{\mathbb{1}_{\mu>0}\mathfrak{C}_{\bar{V},\gamma}\sqrt{\frac{\ln\{\text{Vol}(\Theta)/\delta\}}{n}}}_{\epsilon_b}\right]\right\}\right).$$

*Here $\Delta_{\overline{q^\pi} - \underline{q^\pi}}(s,a) = \overline{q^\pi}(s,a) - \underline{q^\pi}(s,a)$ for $\overline{q^\pi} := \arg\max_{q \in \mathcal{Q}_{\varepsilon_n}} q(s^0, \pi)$ and $\underline{q^\pi} := \arg\min_{q \in \mathcal{Q}_{\varepsilon_n}} q(s^0, \pi)$. For Pollard's pseudo-dimensions $D_\Omega, D_\mathcal{Q}, D_\Pi$, $\mathrm{Vol}(\Theta) = (e^D \max\{D_\Omega, D_\mathcal{Q}, D_\Pi\} + 1)^3 (\{1 \vee L\}\mathcal{U}_2^\tau)^{2D}$ with the effective pseudo dimension $D = D_\Omega + D_\mathcal{Q} + D_\Pi$, where $L$ is Lipschitz constant of $M$-strongly convex function $\mathbb{D}(\cdot)$. Moreover, $\mathfrak{C}_x$ and $\widetilde{\mathcal{O}}$ denote constant terms depending on $x$, and big-Oh notation ignoring high-order terms, respectively.*

In the upper bound of Theorem 4.1, we split the regret into four different parts: the on-support intrinsic uncertainty $\epsilon_\sigma$, the on-support bias $\epsilon_b$, the violation of realizability $\epsilon_{\mathrm{mis}}$, and the off-support extrapolation error $\epsilon_{\mathrm{off}}$. Recall that we require $q^\pi \in \mathcal{Q}$ as in Assumption 1, in fact, we can further relax the condition to requiring $q^\pi$ to be in the linear hull of $\mathcal{Q}$ [48], which is more robust to the realizability error $\epsilon_{\mathrm{mis}}$. In the following, we focus on investigating the roles of the on-support and off-support error terms in the regret bound.

**On-support errors: bias and uncertainty tradeoff.** The on-support error consists of two terms: $\epsilon_b$ and $\epsilon_\sigma$. The on-support uncertainty deviation, $\epsilon_\sigma$, is scaled by a weighted $L_2$-based concentrability coefficient $\mathcal{U}_2^\star := \|\rho/\mu\|_{L_2(\mu)}$, which measures the distribution mismatch between the implicit exploratory data distribution and the baseline data distribution $\mu$. Meanwhile, $\epsilon_b$ depends on the probability mass of $(d_\pi - \rho)^+ \mathbb{1}_{\mu > 0}$, and represents the bias weighted by the probability mass difference between $d_\pi$ and $\rho$ in the support region of $\mu$. In general, a small value of $\mathcal{U}_2^\star$ necessitates choice of the distribution $\rho$ to be closer to $\mu$ which reduces $\epsilon_\sigma$, reducing $\epsilon_\sigma$ but potentially increasing the on-support bias $\epsilon_b$ due to the possible mismatch between $d_\pi$ and $\rho$. Consequently, within the on-support region, there is a trade-off between $\epsilon_\sigma$ and $\epsilon_b$, which is adjusted through $\mathcal{U}_2^\star$.

**Off-support error: enhanced model extrapolation.** One of our main algorithmic contributions is that the off-support extrapolation error $\epsilon_{\mathrm{off}}$ can be minimized by selecting the best possible $\rho$ *without* worrying about balancing the error trade-off, unlike the on-support scenario. This desirable property is essential for allowing the model to harness its extrapolation capabilities to minimize $\epsilon_{\mathrm{off}}$, while simultaneously achieving a good on-support estimation error. As a result, the model attains a small regret. Recall the bi-level formulation; at the lower level, (7) addresses uncertainty arising from the distributional shift using $L_2(\mu)$ control rather than $L_\infty$ control. This plays an important role in enhancing the power of the model extrapolation. In particular, Specifically, there exists an implicit exploratory data distribution $\rho$ with on-support behavior $(\rho \mathbb{1}_{\mu > 0})$ close to $\mu$, such that $\|\rho/\mu\|_{L_2(\mu)}$ is small. On the other hand, its off-support behavior $(\rho \mathbb{1}_{\mu = 0})$ can be arbitrarily flexible, ensuring that $d_\pi \mathbb{1}_{\mu = 0}$ is close to $\rho \mathbb{1}_{\mu = 0}$. Consequently, $(d_\pi - \rho)^+ \mathbb{1}_{\mu = 0}$ is small, as is $\epsilon_{\mathrm{off}}$.

When a dataset with partial coverage, as indicated in [49], it is necessary to provide a guarantee: learn the policy with "best efforts" which is competitive to any policy as long as it is covered. Before we state the near-optimal regret guarantee of our algorithm, we formally define a notion of covered policies according to a newly-defined concentrability coefficient.

**Definition 4.1** ($\mathcal{U}_2^\tau$-covered policy class). *Let $\Pi(\mathcal{U}_2^\tau)$ denote the $\mathcal{U}_2^\tau$-covered policy class of $\mu$ for $\mathcal{U}_2^\tau \geq 1$, defined as*

$$\Pi(\mathcal{U}_2^\tau) := \left\{ \pi \in \Pi : \left\| \frac{d_\pi(s,a)\mathbb{1}_{\mu(s,a)>0}}{\mu(s,a)} \right\|_{L_2(\mu)} \leq \mathcal{U}_2^\tau \text{ and } \sup_{s,a} \frac{d_\pi(s,a)\mathbb{1}_{\mu(s,a)=0}}{\mu(s,a)} < +\infty \right\}.$$

Note that this mixture density ratio concentrability coefficient is always bounded by the $L_\infty$-based concentrability coefficient. Thus such single-policy concentrability assumption in terms of the mixture density ratio is weaker than the standard $L_\infty$ density ratio-based assumption.

**Corollary 4.1** (Near-optimal regret). *Under Assumptions 1-3 with $\varepsilon_\mathcal{Q} \in [0, 1)$, and we set $\varepsilon_n, \widetilde{\sigma}_n$ as in Theorem 4.1, then for any good comparator policy $\pi^\diamond \in \Pi(\mathcal{U}_2^\tau)$ (not necessary the optimal policy $\pi^*$), w.p. $\geq 1 - \delta$, the output policy $\widehat{\pi}$ of (6) satisfies*

$$J(\pi^\diamond) - J(\widehat{\pi}) \leq \frac{1}{1-\gamma} \widetilde{\mathcal{O}}\left( \mathcal{U}_2^\star(\bar{V} + L)\sqrt{\frac{\ln\{\mathrm{Vol}(\Theta)/\delta\}}{nM}} + \sqrt{(1 + \mathcal{U}_\infty^\tau + \mathcal{U}_\infty^\tau/M)\varepsilon_\mathcal{Q}} \right).$$

A close prior result to Corollary 4.1 is that of [9], which develops a pessimistic algorithm based on a nontrivial performance gap condition. Their regret guarantees only hold if the data covers the optimal policy $\pi^*$, in particular, requiring a bounded $L_\infty$ single-policy concentrability with respect to $\pi^*$. In comparison, our guarantee can still provide a meaningful guarantee even when $\pi^*$ is not covered by data. In the following, we include the sample complexity of our algorithm when $\varepsilon_\mathcal{Q} = 0$.

**Corollary 4.2** (Polynomial sample complexity). *Under the conditions in Corollary 4.1, the output policy $\widehat{\pi}$ of solving (6) satisfies $J(\pi^\diamond) - J(\widehat{\pi}) \leq \varepsilon$ w.p. $\geq 1 - \delta$, if*

$$n = \mathcal{O}\left( \left( \frac{(\mathcal{U}_2^\star(\bar{V} + L)/\sqrt{M})^2}{\varepsilon^2(1 - \gamma)^2} + \frac{(\mathcal{U}_2^\tau \bar{V}^2(\bar{V} + L)/M)^{0.67}}{\varepsilon^{1.33}(1 - \gamma)^{1.33}} + \frac{\mathcal{U}_\infty^\tau(\bar{V} + L)}{\varepsilon(1 - \gamma)} \right) \ln \frac{\mathrm{Vol}(\Theta)}{\delta} \right).$$

The sample complexity consists of three terms corresponding to the slow rate $\mathcal{O}(n^{-1/2})$ and the two faster rate $\mathcal{O}(n^{-1})$ and $\mathcal{O}(n^{-3/4})$ terms in Corollary 4.1. When $\mathcal{U}_2^\tau$ and $\mathcal{U}_\infty^\tau$ are not too much larger than $\mathcal{U}_2^\star$, the fast rate terms are dominated, and the sample complexity is of order $\mathcal{O}(1/\varepsilon^2)$, which is much faster than $\mathcal{O}(1/\varepsilon^6)$ in the close work of [59]. It is worth noting that even in exploratory settings where the global coverage assumption holds, our sample complexity rate matches the fast rate in popular offline RL frameworks with general function approximation [8, 54, 12].

In addition to the near-optimal regret guarantee, in safety-critical applications, an offline RL algorithm should consistently improve upon the baseline (behavior) policies that collected the data [19, 26]. Our algorithm also achieves this improvement guarantee with respect to the baseline policy.

**Theorem 4.2** (Baseline policy improvement). *Under Assumptions 1-3 with $\varepsilon_{\mathcal{Q}} = 0$ and set $\varepsilon_n, \widetilde{\sigma}_n$ as in Theorem 4.1. Suppose $1 \in \Omega$ and the baseline policy $\pi_b \in \Pi$ such that $d_{\pi_b} = \mu$, then the regret $(1 - \gamma)(J(\pi_b) - J(\widehat{\pi}))$ for the output policy $\widehat{\pi}$ of solving (6), w.p. $\geq 1 - \delta$, is upper bounded by*

$$\mathcal{O}\left( \sqrt{\frac{(\bar{V} + L)^2 \ln\{\mathrm{Vol}(\Theta)/\delta\}}{nM}} + \sqrt{\frac{(\bar{V}^3 + \bar{V}^2 L)}{M}} \left( \frac{\ln\{\mathrm{Vol}(\Theta)/\delta\}}{n} \right)^{\frac{3}{4}} + \frac{(\bar{V} + L) \ln\{\mathrm{Vol}(\Theta)/\delta\}}{n} \right).$$

The aforementioned information-theoretic results enhance the understanding of the developed algorithm, in terms of the function approximation and coverage conditions, sample complexity, horizon dependency, and bound tightness. In practice, although the information-theoretic algorithm offers a feasible solution to the problem, it is not yet tractable and computationally efficient due to the need to solve constrained optimization. In the following section, we develop a practical algorithm as a computationally efficient counterpart for the information-theoretic algorithm.

# 5 Penalized Adversarial Estimation Algorithm

Although the information-theoretic algorithm offers a feasible solution to the problem, it is not yet tractable and computationally efficient due to the need to solve constrained optimization. In this section, we develop an adversarial estimation proximal-mapping algorithm that still adheres to the pessimism principle, but through penalization. Specifically, the adversarial estimation loss is constructed as follows: $\max_\tau \min_q \mathcal{L}(q, \tau, \pi, c^*, \lambda)$ for solving

$$q(s^0, \pi) + \frac{1}{(1 - \gamma)n} \left\{ c^* \left| \sum_{i=1}^n \tau(s_i, a_i) \left( q(s_i, a_i) - r_i - \gamma q(s_i', \pi) \right) \right| - \lambda \sum_{i=1}^n \mathbb{D}(\tau(s_i, a_i)) \right\}.$$

We observe that the inner minimization for solving $q$ is relatively straightforward, as we can obtain a closed-form global solver using the maximum mean discrepancy principle [20, 44]. In contrast, optimizing $\tau_\psi$ is more involved, often requiring a sufficiently expressive non-linear function approximation class, e.g., neural networks. However, concavity typically does not hold for such a class of functions [21]. From this perspective, our problem can be viewed as solving a non-concave maximization problem, conditional on the solved global optimizer $\bar{q} := \arg\min_q \mathcal{L}(q, \tau, \pi, c^*, \lambda)$. At each iteration, we propose to update $\tau$ by solving the proximal mapping [37] using the Euclidean distance to reduce the computational burden. As a result, the pre-iteration computation is quite low.

---

**Algorithm 1** Adversarial proximal-mapping algorithm

1: **Input** observed data $\mathcal{D}_{1:n} = \{(s_i, a_i, r_i, s_i')\}_{i=1}^n$ and parameters $q^0, \tau^0, \pi^0, c^*, \lambda$ and $\zeta$.
2: **For** $k = 1$ to $\bar{K}$:
3:      Update $\tau^k$ and $q^k$ by solving $\max_\tau \min_q \mathcal{L}(q, \tau, \pi^{k-1}, c^*, \lambda)$
4:      Update $\pi^k$ by solving $\pi^k(\cdot|s) = \arg\max_{\pi \in \Pi} \zeta \left\langle q^k(\cdot, s), \pi(\cdot|s) \right\rangle - D_{\mathrm{NegEntropy}} \left( \pi(\cdot|s), \pi^k(\cdot|s) \right)$.
5: **Return** the policy $\widehat{\pi}$, which randomly selects a policy from the set $\{\pi^k\}_{k=1}^{\bar{K}}$.

---

Once $q$ and $\tau$ are solved, we apply mirror descent in terms of the negative entropy $D_{\text{NegEntropy}}$ [5]. That is, given a stochastic gradient direction of $\pi$ we solve the prox-mapping in each iteration as outlined in step 4 of Algorithm 1. A detailed version of Algorithm 1 with extended discussions on convergence and complexity is provided in Appendix. In the following, we establish the regret guarantee for the policy output by Algorithm 1.

**Theorem 5.1.** *Under Assumptions 1-3 with $\varepsilon_{\mathcal{Q}} = 0$, we properly choose $\lambda = \lambda(\mathcal{U}_2^\tau)$, i.e., $\lambda$ well depends on $\mathcal{U}_2^\tau$, and $c^* = \widetilde{\mathcal{O}}\big(\sqrt{n\bar{V}/(\lambda L \mathcal{U}_2^\tau \ln\{\text{Vol}(\Theta^\dagger)/\delta\})}\big)$. After running $\bar{K} \geq \log|\mathcal{A}|$ rounds of Algorithm 1 with the stepsize $\zeta = \sqrt{\log|\mathcal{A}|/(2\bar{V}\bar{K})}$, for any policy $\pi \in \Pi$, the output policy $\widehat{\pi}$ of the algorithm, w.p $\geq 1 - \delta$, satisfies,*

$$
J(\pi) - J(\widehat{\pi}) \leq \frac{1}{1-\gamma} \widetilde{\mathcal{O}}\bigg( \sqrt[4]{\frac{(\mathcal{U}_2^\star)^2 \mathfrak{C}_{\bar{V},\lambda,L}^1 \ln\{\text{Vol}(\Theta^\dagger)/\delta\}}{n}} + \sqrt{\frac{\bar{V}\log|\mathcal{A}|}{\bar{K}}}
$$

$$
+ \frac{1}{\bar{K}}\sum_{k=1}^{\bar{K}} \min_{\rho_k \in \Delta_{\mathcal{U}_2^\star}} \mathbb{E}_{(d_\pi - \rho_k)^+}\bigg[ \mathbb{1}_{\mu=0}\Big(\mathcal{B}^{\pi^k}q^k(s,a) - q^k(s,a)\Big) + \mathbb{1}_{\mu>0}\sqrt{\frac{\mathfrak{C}_{\bar{V},\lambda,L}^2 \ln\{\text{Vol}(\Theta^\dagger)/\delta\}}{n}} \bigg]\bigg),
$$

*where $\Delta_{\mathcal{U}_2^\star} := \{\rho_k : \|\frac{\rho_k}{\mu}\|_{L_2(\mu)} < \mathcal{U}_2^\star\}$, $\mathfrak{C}_{\bar{V},\lambda,L}^1, \mathfrak{C}_{\bar{V},\lambda,L}^2$ are some constant terms, and the function class complexity $\text{Vol}(\Theta^\dagger) = (e^D \max\{D_\Omega, D_\mathcal{Q}, D_\Pi\}+1)^3(\{1 \vee L\}\mathcal{U}_2^\tau)^{2D}$ for $D = D_\Omega + D_\mathcal{Q} + D_\Pi$.*

**Trajectory-adaptive exploratory data distribution.** Similar to Theorem 4.1, the penalized algorithm also exhibits a desirable extrapolation property for minimizing extrapolation error while simultaneously preserving small on-support estimation errors. This is achieved through adaptations of the implicit exploratory data distributions, $\rho_k$ for $k \in [\bar{K}]$. In contrast to the information-theoretic algorithm, the automatic splitting by $\rho_k$ now depends on the optimization trajectory. At each iteration $k$, the penalized algorithm allows each implicit exploratory data distribution $\rho_k$ to adapt to the comparator policy $\pi$. This results in a more flexible adaptation than the one in the information-theoretic algorithm, either for balancing the trade-off between on-support bias and uncertainty incurred by the distributional mismatch between $d_\pi$ and $\rho_k$, or for selecting the best implicit exploratory to minimize model extrapolation error.

**Opimization error.** Blessed by the reparametrization in the proximal-mapping policy update, which projects the mixture policies into the parametric space $\Pi_\omega$, the complexity of the restricted policy class is independent of the class of $\mathcal{Q}$ and the horizon optimization trajectory $\bar{K}$. As a result, the optimization error $\mathcal{O}(\sqrt{\bar{V}\log|\mathcal{A}|/\bar{K}})$ can be reduced arbitrarily by increasing the maximum number of iterations, $\bar{K}$, without sacrificing overall regret to balance statistical error and optimization error. This allows for the construction of tight regret bounds. This distinguishes our algorithm from API-style algorithms, which do not possess a policy class that is independent of $\mathcal{Q}$ [2, 43, 53].

## 5.1 An Application to Linear MDPs with Refined Concentrability Coefficient

In this section, we conduct a case study in linear MDPs with insufficient data coverage. The concept of the linear MDP is initially developed in the fully exploratory setting [55]. Let $\phi : \mathcal{S} \times \mathcal{A} \to \mathbb{R}^d$ be a $d$-dimensional feature mapping. We assume throughout that these feature mappings are normalized, such that $\|\phi(s,a)\|_{L_2} \leq 1$ uniformly for all $(s,a) \in \mathcal{S} \times \mathcal{A}$. We focus on action-value functions that are linear in $\phi$ and consider families of the following form: $\mathcal{Q}_\theta := \{(s,a) \mapsto \langle\phi(s,a),\theta\rangle \mid \|\theta\|_{L_2} \leq c_\theta\}$, where $c_\theta \in [0,\bar{V}]$. For stochastic policies, we consider the soft-max policy class $\Pi_\omega := \{\pi_\omega(a|s) \propto e^{\langle\phi(s,a),\omega\rangle} \mid \|\omega\|_{L_2} \leq c_\omega\}$, where $c_\omega \in (0,\infty)$. Note that the softmax policy class is consistent with the implicit policy class produced by the mirror descent updates with negative entropy in Algorithm 1, where the exponentiated gradient update rule is applied in each iteration. For the importance-weight class, we also consider the following form: $\Omega_\psi := \{(s,a) \mapsto \langle\phi(s,a),\psi\rangle \mid \|\psi\|_{L_2} \leq c_\psi\}$ where $c_\psi \in (0,\infty)$. To simplify the analysis, we assume the realizability condition for $\mathcal{Q}_\theta$ is exactly met. In this linear MDP setting, we further refine the density ratio to a relative condition number to characterize partial coverage. This concept is recently introduced in the policy gradient literature [1] and is consistently upper-bounded by the $L_\infty$-based density ratio concentrability coefficient.

**Definition 5.1** (Relative condition number). *For any policy $\pi \in \Pi_\omega$ and behavior policy $\pi_b$ such that $d_{\pi_b} = \mu$, the relative condition number is defined as $\iota(d_\pi, \mu) = \sup_{x \in \mathbb{R}^d} \frac{x^T \mathbb{E}_{d_\pi}[\phi(s,a)\phi(s,a)^\top]x}{x^\top \mathbb{E}_\mu[\phi(s,a)\phi(s,a)^\top]x}$.*

**Assumption 4** (Bounded relative condition number). *For any $\pi \in \Pi_\omega$, $\iota(d_\pi, \mu) < \infty$.*

Intuitively, this implies that as long as a high-quality comparator policy exists, which only visits the subspace defined by the feature mapping $\phi$ and is covered by the offline data, our algorithm can effectively compete against it [49]. This partial coverage assumption, in terms of the relative condition number, is considerably weaker than density ratio-based assumptions. In the following, we present our main near-optimal guarantee in linear MDPs. In addition, we design and conduct numerical experiments to empirically validate Theorem 5.2 in terms of the regret rate of convergence.

**Theorem 5.2.** *Under Assumption 4, if we set properly choose $\lambda = \lambda(c_\psi)$ and $c^* = \widetilde{\mathcal{O}}\big(\sqrt[4]{n/d\ln\{(1+e\sqrt{n}(1\vee L)\bar{V}c_\psi c_\omega)/\delta\}}\big)$, and suppose $\widehat{\pi}^{lr}$ is returned by Algorithm 1 with linear function approxiamiton after running $\bar{K} \gg \log|\mathcal{A}|$ rounds, then for any policy in $\pi \in \Pi_\omega(\mathcal{U}_2^{lr})$ for $\mathcal{U}_2^{lr} \geq 1$, w.p. $\geq 1 - \delta$, $J(\pi) - J(\widehat{\pi}^{lr})$ is bounded by*

$$\widetilde{\mathcal{O}}\bigg(\frac{\sqrt{\min\{\kappa^2 c_\psi^2\{\mathcal{U}_2^{lr}\}d^2, \iota(d_\pi, \mu)d\}}}{1-\gamma}\sqrt[4]{\frac{\mathfrak{C}_{\bar{V},\lambda,L}d\ln\{(1+e\sqrt{n}(1\vee L)\bar{V}c_\psi c_\omega)/\delta\}}{n}}\bigg),$$

*where $\kappa = trace(\mathbb{E}_\mu[\phi(s,a)\phi(s,a)^\top])$ and $c_\psi\{\mathcal{U}_2^{lr}\} = \sup_{\{\psi: \|\phi(s,a)^\top\psi\|_{L_2(\mu)}=\mathcal{U}_2^{lr}\}} \|\psi\|_{L_\infty}$.*

To the best of our knowledge, this is the first result PAC guarantees for an offline model-free RL algorithm in linear MDPs, requiring only realizability and single-policy concentrability. The regret bound we obtain is at least linear and, at best, sub-linear with respect to the feature dimension $d$. Our approach demonstrates a sample complexity improvement in terms of feature dimension compared to prior work by [22], with a complexity of $\mathcal{O}(d^{1/2})$ versus $\mathcal{O}(d)$. It is worth noting that [22] only establishes results that compete with the optimal policy, and when specialized to linear MDPs, assumes the offline data has global coverage. Another previous study by [53] achieves a similar sub-linear rate in $d$ as our approach; however, their algorithm is computationally intractable, relying on a much stronger Bellman-completeness assumption and requiring a small action space.

## 6 Experiments

In this section, we evaluate the performance of our practical algorithm by comparing to the model-free offline RL baselines including CQL [25], BEAR [24], BCQ [18], OptiDICE [27], ATAC [10], IQL [23], and TD3+BC [17]. We also compete with a popular model-based approach COMBO [57].

**Synthetic data.** We consider two synthetic environments: a synthetic CartPole environment from the OpenAI Gym [7] and a simulated environment. Detailed discussions on the experimental designs are deferred to the Appendix. In both settings, following [48], we first learn a sub-optimal policy using DQN [34] and then apply softmax to its $q$-function, divided by a temperature parameter $\alpha$ to set the action probabilities to define a behavior policy $\pi_b$.

A smaller $\alpha$ implies $\pi_b$ is less explored, and thus the support of $\mu = d_{\pi_b}$ is relatively small. We vary different values of $\alpha$ for evaluating the algorithm performance in "low", "medium" and "relatively high" offline data exploration scenarios. We use $\gamma = 0.95$ with the sample-size $n = 1500$ in all experiments. Tuning parameter selection is an open problem in offline policy optimization. Fortunately, Theorem 5.2 suggests an offline selection rule for hyperparameters $\lambda$ and $c^*$. In the following

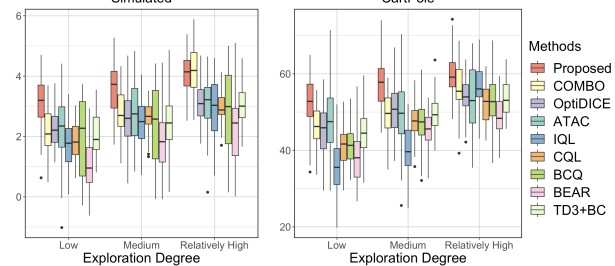

Figure 1: The boxplot of the discounted return over 50 repeated experiments.

experiments, we set the hyper-parameters satisfying the condition $\mathcal{O}(\frac{n^{1/4}}{d\log(\bar{V}\sqrt{n})})$. Figure 1 shows that our algorithm almost consistently outperforms competing methods in different settings. This performance mainly benefits from the advantages exposed in our theoretical analysis, such as model extrapolation enhancement, relaxation of completeness-type assumptions on function approximation, etc. The only exception is the slightly poorer performance compared to COMBO in a high exploration setting, where COMBO may learn a good dynamic model with relatively sufficient exploration. We provide the experiment details in Appendix due to page limit.

**Benchmark data.** We evaluate our proposed approach on the D4RL benchmark of OpenAI Gym locomotion (walker2d, hopper, halfcheetah) and Maze2D tasks [15], which encompasses a variety

of dataset settings and domains and positions our algorithm within the existing baselines. We take the results of COMBO, OptiDICE and ATAC from their original papers for Gym locomotion, and run COMBO and ATAC using author-provided implementations for Maze2D. The results of BCQ, BEAR methods from the D4RL original paper. In addition, CQL, IQL and TD3+BC are re-run to ensure a fair evaluation process for all tasks. As shown in Table 1, the proposed algorithm achieves the best performance in 7 tasks and is comparable to the baselines in the remaining tasks. In addition to the evaluation of the policy performance, we also conduct sensitivity analyses on the hyperparameter-tuning and study the regret rate of convergence.

Table 1: The normalized score of the policy at the last iteration of training, averaged over 5 random seeds. The highest performing scores are highlighted. The *med*, *med-rep*, and *med-exp* is shorthand for *medium*, *medium-replay*, and *medium-expert*, respectively.

| Tasks | Proposed | COMBO | BCQ | BEAR | OptiDICE | ATAC | CQL | IQL | TD3+BC |
|---|---|---|---|---|---|---|---|---|---|
| walker2d-med | $80.8 \pm 5.1$ | $81.9 \pm 2.8$ | 53.1 | 59.1 | $21.8 \pm 7.1$ | **89.6** | $77.2 \pm 4.2$ | $78.3 \pm 4.3$ | $81.7 \pm 2.3$ |
| hopper-med | $94.9 \pm 4.3$ | $97.2 \pm 2.2$ | 54.5 | 52.1 | $94.1 \pm 3.7$ | 85.6 | $74.3 \pm 5.8$ | $66.3 \pm 6.4$ | **98.4** $\pm 1.6$ |
| halfcheetah-med | **58.1** $\pm 1.4$ | $54.2 \pm 1.5$ | 40.7 | 41.7 | $38.2 \pm 0.1$ | 53.3 | $37.2 \pm 0.3$ | $47.4 \pm 1.1$ | $27.8 \pm 0.7$ |
| walker2d-med-rep | **99.6** $\pm 2.9$ | $56.0 \pm 8.6$ | 15.0 | 19.2 | $21.6 \pm 2.1$ | 92.5 | $20.8 \pm 1.6$ | $73.9 \pm 2.8$ | $34.4 \pm 4.2$ |
| hopper-med-rep | **113.0** $\pm 2.1$ | $89.5 \pm 1.8$ | 33.1 | 33.7 | $36.4 \pm 1.1$ | 102.5 | $32.6 \pm 1.9$ | $94.7 \pm 1.5$ | $44.4 \pm 3.7$ |
| halfcheetah-med-rep | $49.3 \pm 2.1$ | **55.1** $\pm 1.0$ | 38.2 | 38.6 | $39.8 \pm 0.3$ | 48.0 | $41.9 \pm 1.1$ | $44.2 \pm 2.5$ | $48.3 \pm 0.7$ |
| walker2d-med-exp | $108.2 \pm 7.4$ | $103.3 \pm 5.6$ | 57.5 | 40.1 | $74.8 \pm 9.2$ | **114.2** | $103.8 \pm 6.9$ | $109.6 \pm 7.0$ | $100.5 \pm 8.9$ |
| hopper-med-exp | $117.8 \pm 1.9$ | $111.1 \pm 2.9$ | 110.9 | 96.3 | $111.5 \pm 0.6$ | **119.2** | $111.4 \pm 1.2$ | $91.5 \pm 2.2$ | $112.4 \pm 0.3$ |
| halfcheetah-med-exp | **98.5** $\pm 3.8$ | $90.0 \pm 5.6$ | 64.7 | 53.4 | $91.1 \pm 3.7$ | 94.8 | $66.7 \pm 8.9$ | $86.7 \pm 3.6$ | $95.9 \pm 3.9$ |
| walker2d-random | **11.2** $\pm 3.8$ | $7.0 \pm 3.6$ | 4.9 | 7.3 | $9.9 \pm 4.3$ | 6.8 | $4.7 \pm 1.5$ | $5.8 \pm 2.8$ | $3.4 \pm 1.7$ |
| hopper-random | **18.7** $\pm 1.5$ | $17.9 \pm 1.4$ | 10.6 | 11.4 | $11.2 \pm 1.1$ | 17.5 | $10.7 \pm 0.1$ | $10.8 \pm 0.6$ | $11.1 \pm 0.2$ |
| halfcheetah-random | $37.6 \pm 2.4$ | **38.8** $\pm 3.7$ | 2.2 | 25.1 | $11.6 \pm 1.2$ | 3.9 | $26.7 \pm 1.4$ | $22.4 \pm 1.8$ | $26.1 \pm 1.8$ |
| maze2d-umaze | $96.5 \pm 27.8$ | $34.2 \pm 8.6$ | 12.8 | 3.4 | **111.0** $\pm 8.3$ | $84.4 \pm 24.8$ | $50.5 \pm 7.9$ | $41.5 \pm 4.7$ | $13.8 \pm 22.8$ |
| maze2d-med | $137.5 \pm 18.9$ | $49.9 \pm 13.9$ | 8.3 | 29.0 | $145.2 \pm 17.5$ | **152.3** $\pm 34.6$ | $28.6 \pm 9.2$ | $38.5 \pm 4.2$ | $59.1 \pm 44.7$ |
| maze2d-large | **187.8** $\pm 15.2$ | $128.2 \pm 17.3$ | 6.2 | 4.6 | $155.7 \pm 33.4$ | $142.1 \pm 33.8$ | $46.2 \pm 16.2$ | $54.2 \pm 18.1$ | $87.6 \pm 15.4$ |

**Real-world application.** The Ohio Type 1 Diabetes (OhioT1DM) dataset [33] comprises a cohort of patients with Type-1 diabetes, where each patient exhibits different dynamics and 8 weeks of life-event data, including health status measurements and insulin injection dosages. Clinicians aim to adjust insulin injection dose levels [33, 4] based on a patient's health status in order to maintain glucose levels within a specific range for safe dose recommendations. The state variables consist of health status measurements, and the action space is a bounded insulin dose range. The glycemic index serves as a reward function to assess the quality of dose suggestions. Since the data-generating process is unknown, we follow [31, 29] to utilize the Monte Carlo approximation of the estimated value function on the initial state of each trajectory to evaluate the performance of each method. The mean and standard deviation of the improvements on the Monto Carlo discounted returns are presented in Table 2. As a result, our algorithm achieves the best performance for almost all patients, except for Patient 552. The main reason for the desired performance in real data is from the enhanced model extrapolation and relaxed function approximation requirements and outperforms the competing methods. This finding is consistent with the results in the synthetic and benchmark datasets, demonstrating the potential applicability of the proposed algorithm in real-world environments.

Table 2: The baseline policy improvements over 50 repeated experiments in the OhioT1DM dataset.

| Patient ID | Proposed | COMBO | BCQ | BEAR | OptiDICE | ATAC | CQL | IQL | TD3+BC |
|---|---|---|---|---|---|---|---|---|---|
| 596 | **6.5** $\pm 1.1$ | $4.1 \pm 0.8$ | $3.8 \pm 0.9$ | $2.7 \pm 1.1$ | $4.7 \pm 1.1$ | $5.1 \pm 2.0$ | $4.6 \pm$ **0.6** | $3.4 \pm 0.7$ | $4.8 \pm 1.3$ |
| 584 | **33.1** $\pm 1.8$ | $27.0 \pm 1.3$ | $20.3 \pm 1.2$ | $22.9 \pm 1.6$ | $27.7 \pm 1.9$ | $26.9 \pm 2.6$ | $21.6 \pm$ **1.2** | $22.7 \pm 1.3$ | $22.4 \pm 1.7$ |
| 567 | **36.9** $\pm 1.3$ | $30.6 \pm 2.0$ | $24.3 \pm 1.4$ | $25.6 \pm 1.4$ | $28.8 \pm 2.2$ | $29.7 \pm 2.8$ | $26.5 \pm 1.4$ | $25.8 \pm 1.4$ | $27.8 \pm 1.5$ |
| 552 | $7.9 \pm 0.9$ | $6.8 \pm 0.7$ | $5.7 \pm 0.5$ | $5.0 \pm 0.8$ | **8.1** $\pm 0.9$ | $7.2 \pm 1.5$ | $6.7 \pm$ **0.4** | $6.1 \pm 0.5$ | $7.4 \pm 0.8$ |
| 544 | **13.2** $\pm 1.9$ | $9.8 \pm 1.5$ | $7.5 \pm 2.5$ | $5.9 \pm$ **0.8** | $10.3 \pm 1.8$ | $10.1 \pm 2.1$ | $8.7 \pm 1.0$ | $7.8 \pm 0.9$ | $9.7 \pm 0.8$ |
| 540 | **20.4** $\pm$ **0.5** | $17.5 \pm 0.9$ | $14.3 \pm 0.6$ | $12.7 \pm 0.5$ | $17.9 \pm 0.9$ | $18.2 \pm 1.4$ | $16.5 \pm 0.5$ | $14.0 \pm 0.6$ | $17.1 \pm 0.8$ |

# 7  Conclusion

We study offline RL with limited exploration in function approximation settings. We propose a bi-level policy optimization framework, which can be further solved by a computationally practical penalized adversarial estimation algorithm, offering strong theoretical and empirical guarantees. Regarding limitations and future work, while the penalized adversarial estimation is more computationally efficient than the previously constrained problem, it may still be more challenging to solve than single-stage optimization problems. Another future direction is to explore environments with unobservable confounders. It will be interesting to address these limitations in future works.

## 8 Acknowledgments

The author is grateful to the five anonymous reviewers and the area chair for their valuable comments and suggestions.

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
