# Supplementary Material to Bi-Level Offline Policy Optimization with Limited Exploration

**Wenzhuo Zhou**
Department of Statistics
University of California Irvine
wenzhuz3@uci.edu

## Contents

## A  Discussion on Algorithm 1

In this section, we provide a pronounced discussion on Algorithm 1 in maintext by offering more details on solving adversarial estimation over $q$ and $\tau$, as well as establishing the theoretical convergence guarantee in Theorem A.1. The convergence of the step provides a basis for us to use the mirror descent for policy updating. The detailed version of Algorithm 1 in maintext is summarized in Algorithm A.1.

In function approximation settings, the $\Omega, \mathcal{Q}, \Pi$ are often represented by compact parametric functions in practice, either in linear or non-linear function classes [53]. In the following, we denote these parameters as $\psi$ and $\theta$ and $\omega$ corresponding to $\Omega_\psi$ and $\mathcal{Q}_\theta$, and $\Pi_\omega$ respectively.

Under this parametric setting, we focus on solving the adversarial loss $\mathcal{L}^\circ(q, \tau, \pi, c^*, \lambda)$, which can be expressed as:

$$
\mathcal{L}(q_\theta, \tau_\psi, \pi_\omega, c^*, \lambda) = q_\theta(s^0, \pi) + \frac{1}{(1-\gamma)n}\left\{ c^* \Big| \sum_{i=1}^n \tau_\psi(s_i, a_i)\left(q_\theta(s_i, a_i) - r_i - \gamma q_\theta(s_i', \pi_\omega)\right) \Big|
$$
$$
- \lambda \sum_{i=1}^n \mathbb{D}(\tau_\psi(s_i, a_i)) \right\}. \tag{A.1}
$$

As stated in Algorithm 1, at each iteration, we aim to solve $\max_\psi \min_\theta \mathcal{L}^\circ(q_\theta, \tau_\psi, \pi_\omega, c^*, \lambda)$, which forms a saddle-point formulation, and we denote the saddle point as $(\psi^*, \theta^*)$ (should depend on $\pi_\omega$, but we omit here for simplifying the notation). At the same time, we denote the population loss as

$$
\mathcal{L}^\circ(q_\theta, \tau_\psi, \pi_\omega, c^*, \lambda) = q_\theta(s^0, \pi) + \frac{1}{(1-\gamma)}\left\{ c^* \Big| \mathbb{E}_\mu\left[\tau_\psi(s, a)\left(q_\theta(s, a) - r(s, a) - \gamma q_\theta(s', \pi_\omega)\right)\right] \Big|
$$
$$
- \lambda \mathbb{E}_\mu[\mathbb{D}(\tau_\psi(s, a))] \right\}. \tag{A.2}
$$

In the following, we omit the arguments $\pi_\omega, c^*, \lambda$ in the expression for simplicity, and thus using

$$\mathcal{L}^\circ(q_\theta, \tau_\psi) \tag{A.3}$$

to denote the population loss for $\mathcal{L}^\circ(q_\theta, \tau_\psi, \pi_\omega, c^*, \lambda)$.

We can observe that the inner minimization problem is relatively easy to solve. In addition to the closed-form solution as discussed in maintext, the feature mapping class is sufficient for modeling $\mathcal{Q}$, as demonstrated in [8]. The feature mapping class simplifies the optimization, making it efficiently solvable by various algorithms as discussed in [52]. In contrast, the more challenging aspect is optimizing $\tau_\psi$. Due to its complex structure, it demands a sufficiently flexible non-linear function approximation class, e.g., deep neural networks, for optimization [18]. Unfortunately, concavity typically does not hold for non-linear function approximation classes, and thus the outer maximization of $\max_\psi \min_\theta \mathcal{L}^\circ(q_\theta, \tau_\psi)$ is also affected. As a result, we need to develop a more efficient and convergent algorithm. Therefore, we regard solving a non-concave maximization problem, conditional on the solved global optimizer $\bar{q}_\theta := \arg\min_\theta \mathcal{L}^\circ(q_\theta, \tau_\psi)$. Under this framework, we first study the gradients of the objective function with respect to $\psi$. Define $\bar{\mathcal{L}}^\circ(\tau_\psi) = \mathcal{L}^\circ(\bar{q}_\theta, \tau_\psi)$, then the gradient of $\bar{\mathcal{L}}^\circ(\tau_\psi)$ with respect to $\psi$ satisfies

$$\nabla_\psi \bar{\mathcal{L}}^\circ(\tau_\psi) = \frac{\mathbb{E}_\mu[(r(s,a) + q_\theta(s,a) - \gamma q_\theta(s',\pi))(\tau_\psi(s,a)/|\tau_\psi(s,a)|)\nabla_\psi \tau_\psi(s,a)]}{1-\gamma} \tag{A.4}$$

$$- \frac{\lambda \mathbb{E}_\mu[\mathbb{D}'(\tau(s,a))\nabla_\psi \tau_\psi(s,a)]}{1-\gamma}. \tag{A.5}$$

With the gradients provided in (A.5), we propose a stochastic approximation algorithm to update $\tau_\psi$. At each iteration, we update $\tau_\psi$ by solving the proximal mapping [41]:

$$\text{Proj}_\psi(\psi^*, \nabla; D_{Berg}) := \arg\max_\psi \{\langle \psi, \nabla \rangle - D_{Berg}(\psi^*, \psi)\}, \tag{A.6}$$

where $\psi^*$ can be viewed as the current update of the parameter, $D_{Berg}(\cdot, \cdot)$ denotes the Bregman divergence as discussed in [45], and $\nabla$ represents the scaled stochastic gradient of the parameter of interest. In practice, we may consider using the Euclidean distance to reduce the computational burden. Once $q$ and $\tau$ are solved, we apply mirror descent in terms of the negative entropy $D_{\text{NegEntropy}}$ [4]. That is, given a stochastic gradient direction of $\pi$ we solve the prox-mapping in each iteration. Note that, it follows from [41], step 4 in Algorithm 1 (step 13 in Algorithm A.1) has a closed-form exponential updating rule, particularly with the negative entropy $D_{\text{NegEntropy}}$, as

$$\pi_{w^k}(\cdot|s) \propto \pi_{w^{k-1}} \exp(\zeta q^k(s, \cdot)),$$

for any $s$. The detailed version of the proposed optimization algorithm is presented in Algorithm A.1.

---

**Algorithm A.1** Adversarial proximal-mapping algorithm (detailed version)

---

1: **Input** observed data $\mathcal{D}_{1:n} = \{(s_i, a_i, r_i, s_i')\}_{i=1}^n$ and and the initial state $s^0$.
2: **Initialize** the parameters $\theta^{(0)}, \psi^{(0)}, \omega^{(0)}, c^*, \lambda, \zeta, \eta^0, \bar{K}$ and $\bar{T}$.
3: **For** $k = 1$ to $\bar{K}$:
4:     **Update** $\psi^{(k)}$ and $\theta^{(k)}$:
5:         **Initialize** $\psi^0 = \psi^{(k-1)}$ and $\theta^0 = \theta^{(k-1)}$ and $\eta^0 = \eta^0$.
6:         **For** $t = 1$ to $t = T$:
7:             Update $\theta^t$ by solving $\mathcal{L}(q_\theta, \tau_{\psi^{t-1}}, \pi_{\omega^{k-1}}, c^*, \lambda)$ in (A.1).
8:             Decay the stepsize $\eta^t$ of the rate $\mathcal{O}(t^{-1/4})$.
9:             Compute the stochastic gradient with respect to $\psi$ as $\widetilde{\nabla}_\psi \mathcal{L}^\circ(\tau_\psi, q_{\theta^t})$ in (A.3).
10:           Update $\psi^t$ by solving: $\psi^t = \text{Proj}_\psi(\psi^{t-1}, \eta^t \widetilde{\nabla}_\psi \mathcal{L}^\circ(\tau_\psi, q_{\theta^t}); D_{Berg})$ in (A.6).
11:     **End for**
12:     **Output** $\psi^{(k)} = \psi^T$ and $\theta^{(k)} = \theta^T$.
13:     **Update** $\omega^k$ by solving $\arg\max_\omega \zeta \langle q^k(s, \cdot), \pi_\omega(\cdot|s) \rangle - D_{\text{NegEntropy}}(\pi_\omega(\cdot|s), \pi_{\omega^{k-1}}(\cdot|s))$.
14: **Return** the policy $\widehat{\pi}$, which randomly selects a policy from the set $\{\pi^k\}_{k=1}^{\bar{K}}$.

---

In the following, we demonstrate that our algorithm is convergent with a sublinear rate even under non-linear (non-concave) settings regarding solving the steps 6 to 12 in Algorithm A.1. Before we state our convergence guarantee, we make the following regular assumptions as stated in [64].

**Assumption A.1** ($L_0$-Lipschitz continuity on gradient). *For any $\tau_\psi \in \Omega_\psi$, $\tau_\psi$ is differentiable (not necessarily convex or concave), bounded from below, $\|\nabla_\psi \tau_{\psi_1}(s,a) - \nabla_\psi \tau_{\psi_2}(s,a)\| \leq L_0 \|\psi_1 - \psi_2\|$, for any $s, a$, where $L_0 < \infty$ is some universal Lipschitz constant and $\|\cdot\|$ denotes the Euclidean norm.*

Assumption A.1 imposes the first-order smoothness condition on the specified function class.

**Assumption A.2** (Smooth function class). *$|q_{\theta_1}(s,a) - q_{\theta_2}(s,a)| \leq L_0 \|\theta_1 - \theta_2\|$, for any $s, a$, and $q_\theta \in \mathcal{Q}_\theta$.*

Assumption A.2 holds for a wide range of function approximation classes, including feature mapping space with smooth basis functions, non-linear approximation classes, DNNs with Leaky ReLU activation function, or spectral normalization on ReLU activation [17].

**Assumption A.3.** *The gradient of function $\tau_\psi(\cdot)$ evaluated at saddle point $\psi^*$ is bounded above; i.e., $\nabla_\psi \tau_{\psi^*}(s,a) < c_3$ uniformly over $(s,a)$ for some finite and positive constant $c_3$.*

Assumption A.3 is a much weaker assumption compared to the bounded variance of stochastic gradients assumption which is commonly made in the existing literature [44, 37]. In the following, we derive the convergence rate, which holds for non-concave function approximation class $\Omega_\psi$.

**Theorem A.1** (Convergence to a stationary point [64]). *Under Assumption 3 in maintext, and Assumptions A.1-A.3 above, suppose the steps 6-12 in Algorithm A.1 runs $T \geq 1$ rounds with stepsize*

$$\eta^t = \min\{\sqrt[4]{tT4\mathbb{G}^2/\sigma_{\max}^4 C_1}, 1/C_1\},$$

*for $t = 1, ..., T$ and Euclidean distance is used for Bergman divergence. If we pick up the solution output $\psi^{T^\star}$ following the probability mass function*

$$P(T^\star = t) = \frac{2\eta^t - (\eta^t)^2 C_1}{\sum_{t=1}^{T}(2\eta^t - (\eta^t)^2 C_1)},$$

*then it follows that*

$$\mathbb{E}[\|\nabla_\psi \bar{\mathcal{L}}^\circ(\tau_{\psi^{T^\star}})\|^2] \leq \sqrt{\frac{2\mathbb{G}C_1\sigma_{\max}^2}{T}} + \frac{\sqrt{2\mathbb{G}C_1\sigma_{\max}^2}}{T^{3/4}} + \frac{2\mathbb{G}C_1}{T}, \tag{A.7}$$

*where $\bar{\mathcal{L}}^\circ(\tau_\psi)$ is defined in (A.5) and $\mathbb{G} := \bar{\mathcal{L}}^\circ(\tau_{\psi^0}) - \min_\psi \bar{\mathcal{L}}^\circ(\tau_\psi)$ measures the distance of the initial and optimal solution, $C_1$ is Lipschitz constant depending on $c^*, c_2, c_3, M, L_0, \bar{V}$ and $\lambda$. Recall that $c_2$ and $M$ are from the definition of $\mathbb{D}$. Here the variance of the stochastic gradient is bounded above by $\sigma_{\max} := \max_{t \in 1:T} \sqrt{c_4 \|\widetilde{\theta}(\psi^t) - \theta^*\|^2 + c_5 \|\psi^t - \psi^*\|^2}$, for some constants $c_4, c_5$ depending on $c^*, c_2, c_3, L_0, \bar{V}, \lambda$ and $\gamma$. Here, $\widetilde{\theta}(\psi^t)$ is the optimizer for $\mathcal{L}^\circ(q_\theta, \tau_{\psi^t})$.*

Theorem A.1 is adapted from Theorem 6.5 in [64] on local convergence. Theorem A.1 implies that the steps 6-12 in Algorithm A.1 can converge sublinearly to a stationary point if the $\sigma_{\max}$ is sufficiently small. The rate of convergence is also affected by the smoothness of the class $\Omega_\psi$ and the distance of the initial and optimal solution.

# B Experiment Details

We include our source code for experiments and algorithm, and the guideline for access to the OhioT1DM dataset in this GitHub repository.

## B.1 Environment Settings

**Simulated environment.** For the simulated environment setting, the system dynamics are given by

$$s^{t+1} = \begin{pmatrix} 0.75\,(2a^t - 1) & 0 \\ 0 & 0.75\,(1 - 2a^t) \end{pmatrix} s^t + \begin{pmatrix} 0 & 1 \\ 1 & 0 \end{pmatrix} \odot s^t {s^t}^\top \mathbb{I}_{2\times 1} + \varepsilon^t,$$

$$r^t = {s^{t+1}}^\top \begin{pmatrix} 2 \\ 1 \end{pmatrix} - \frac{1}{4}\,(2a^t - 1) + ({s^{t+1}}^\top s^{t+1})^{\frac{3}{2}} \odot \begin{pmatrix} 0.25 \\ 0.5 \end{pmatrix},$$

for $t \geq 0$, where $\odot$ denotes the Hadamard product, $\mathbb{I}$ is the identity matrix, the noise $\{\varepsilon^t\}_{t\geq 0} \overset{iid}{\sim} N(0_{2\times 1}, 0.25\mathbb{I}_{2\times 2})$ and the initial state variable $s^0 \sim N(0_{2\times 1}, 0.25\mathbb{I}_{2\times 2})$. The transition dynamic mainly follows the design in [50], but the reward function we consider here is more complex. In this setting, we consider a binary action space $a^t = \{0, 1\}$.

**CartPole environment.** We utilize the CartPole environment from OpenAI Gym [6], a standard benchmark in RL for evaluating policies. The 4-dimensional state space in this environment is represented as $s^t = (s^t_{[1]}, s^t_{[2]}, s^t_{[3]}, s^t_{[4]})$, encompassing both the cart's position and velocity and the pole's angle and angular velocity. The action space is binary, with actions $\{0, 1\}$, representing pushes to the left or right, respectively. To enhance the differentiation between various policy values, we adopt a modified reward function, as in [49, 64]. The reward function is defined as:

$$r^t = -1 + \left| 2 - \frac{s^t_{[1]}}{s^t_{[1]}(\text{clip})} \right| \left| 2 - \frac{s^t_{[3]}}{s^t_{[3]}(\text{clip})} \right|.$$

Here, $s^t_{[1]}$ and $s^t_{[3]}$ represent the cart's position and the pole's angle, respectively. The terms $s^t_{[1]}(\text{clip})$ and $s^t_{[3]}(\text{clip})$ denote the thresholds at which the episode terminates (done = True) if either $|s^t_{[1]}| \geq s^t_{[1]}(\text{clip})$ or $|s^t_{[3]}| \geq s^t_{[3]}(\text{clip})$ is satisfied. Under this definition, a higher reward is obtained when the cart is closer to the center and the pole's angle is closer to the perpendicular position.

**D4RL benchmark environments.** We use Maze2D and Gym-locomotion environments of D4RL benchmark [12, 28] to evaluate the proposed algorithm in continuous control tasks. We summarize the descriptions of different task settings in [12] in the following:

**Maze2D** is a navigation task set within a 2D state space where the agent aims to reach a predetermined goal location. By leveraging previously collected trajectories, the agent's objective is to determine the shortest path to the destination. The complexity of the mazes increases in the sequence: "maze2d-umaze," "maze2d-medium," and "maze2d-large."

**Gym-locomotion.** For each task within the Gym-locomotion continuous controls set, which includes {hopper, walker2d, halfcheetah}. We refer the readers to [12] for detailed background for the above-mentioned tasks. In our experiments, data is generated and collected in the following manners:

- random: This dataset is produced using a policy initialized at random for each task.
- medium: This dataset is derived from a policy trained with the SAC algorithm in [15]. The training is stopped prematurely through early stopping.
- medium-replay: This combines two subsets. The "replay" subset consists of samples collected during the training of the policy for the "medium" dataset. Therefore, the "medium-replay" dataset encompasses both the "medium" and "replay" data.
- medium-expert: This dataset supplements an equal number of expert trajectories with suboptimal trajectories. The suboptimal samples are sourced either from a uniformly random policy or from a medium-performance policy.

**Real world enviroment: OhioT1DM offline dataset.**

We applied the proposed algorithm on the Ohio Type 1 Diabetes Mobile Health (OhioT1DM) study [35]. This dataset comprises six patients with type 1 diabetes, each contributing eight weeks of life-event data—spanning health status measurements to insulin injection dosages. Given the unique glucose dynamics of each patient, we treat each patient's data as an individual dataset, in line with [66]. Thus, daily data is seen as an individual trajectory. Data points are aggregated over 60-minute intervals, ensuring a maximum horizon length of 24. After the exclusion of missing samples and outliers, the total number of transition pairs for each patient's dataset approximates $n = 360$. The state variable $s^t$ is set to be a three-dimensional vector including the average blood glucose levels $s^t_{[1]}$, the average heart rate $s^t_{[2]}$ and the total carbohydrates $s^t_{[3]}$ intake during the period time $[t-1, t]$. Here, the reward is defined as the average of the index of glycemic control [47, 31] between time $t-1$ and $t$, measuring the health status of the patient's glucose level. That is

$$r^t = -\frac{\mathbb{I}(s^t_{[1]} > 140)|s^t_{[1]} - 140|^{1.10} + \mathbb{I}(s^t_{[1]} < 80)(s^t_{[1]} - 80)^2}{30},$$

which implies that reward $r^t$ is non-positive and a larger value is preferred. Then we estimate the optimal policy by treating each day as an independent sample. We study the individualized dose-finding problem by selecting the optimal continuous dose level for intervention options. For model performance evaluation, since the data-generating process is unknown, we follow [34] to utilize the Monte Carlo approximation of the estimated function of the initial state of each trajectory to evaluate the performance of each method. To better evaluate the stability and performance of each method, we randomly select 20 trajectories from each individual based on available trajectories 50 times and apply all methods to the selected data. The mean and standard deviation of the improvements on the Monto Carlo discounted returns are presented in Table 2 in maintext.

## B.2 Implementation Details

In the synthetic environments, we first learn a sub-optimal policy using DQN [36] and then apply softmax to its $q$-function, divided by a temperature parameter $\alpha$ to set the action probabilities to define a behavior policy $\pi_b$. In particular, we set $\alpha = 0.1, 0.5, 1$ for the three degree of exploration "Low", "Medium", and "Relatively High", respectively. For the implementation, we set the detection function as a quadratic form, i.e., $\mathbb{D}(x) = \frac{1}{2}(x - 1)^2$, which satisfies the definition of $\mathbb{D}(x)$ in Definition 3.1 in maintext. To evaluate the policy obtained from the proposed method in synthetic experiments, we generate 100 independent trajectories, each with a length of 100 based on the learned policy. We sample each action by the learned policy $\pi(a|s)$ and calculate the discounted sum of reward for each trajectory. We compare the discounted return of each method and output the results in maintext.

For function approximation in $\mathcal{Q}_\theta$ class in our practical implementation, we set the function spaces $\mathcal{Q}_\theta$ to RKHSs to facilitate the computation. For function modeling in $\Omega_\psi$, we model $\Omega_\psi$ by feedforward neural networks to handle the complex behavior of $\tau$. The radius of the function class is selected to be sufficiently large to ensure the flexibility of the $\Omega_\psi$. For the feedforward neural networks modeling, we are parameterized by a two-layer neural network with a layer width 256 and using ReLU as activation functions. For the RKHS modeling, we use the Gaussian RBF kernel. RBF kernel, for any sample $x$ and $x'$

$$K\left(x; x'\right) := \exp\left(-\frac{\|x - x'\|^2}{2\mathrm{bw}^2}\right),$$

where bw is the bandwidth. In our numerical experiments, we use Silverman's rule of thumb for bandwidth selection [51]. In particular, we apply the finite representer theorem in RKSH to model $\theta \in \mathcal{Q}$ as $q_\theta(s, a) = \sum_{i=1}^n K(\{s, a\}, \{s_i, a_i\})\theta_i$, for the parameters of interest $\{\theta_i\}_{i=1}^n$. In step 7 in Algorithm A.1, we optimize $\theta^t$ with a fixed $\psi^{t-1}$ using stochastic gradient descent with learning rate $5 \times 10^{-3}$, and set the stepsize $\eta^0 = 1 \times 10^{-3}$. We set the decay learning rate $\eta^t$ for the $t$th iteration be $\frac{\eta^0}{1 + 0.3 \cdot t^{1/4}}$, where $\alpha_0$ is the learning rate of the initial iteration for optimizating $\psi^t$. For updating the policy, we model the policy class $\Pi_\omega$ by a softmax policy class or Gaussian distribution a two-layer neural network with a layer width 64. The updating rate $\zeta$ is also set to $3 \times 10^{-3}$. The class $\Omega_\omega$ and $\Pi_\omega$ and $\mathcal{Q}_\theta$ are optimized with Adam [20]. For hyperparameters-tuning, we set hyper-parameters satisfying the condition $\lambda = c^* = \frac{2 \cdot n^{1/4}}{3 \cdot d \log(V \sqrt{n})}$ via a offline selection rule inspired from Theorem 5.2.

For the implementation of competing methods, we implement the methods BEAR, CQL, IQL, BCQ, and COMBO mainly based on the popular offline deep reinforcement learning library [48]. For the general optimization and function approximation settings, we use a multi-layer perceptron (MLP) with 2 hidden layers, each with 256 units for function approximation. We set the batch size to be 64, and use ReLU function as the activation function. In addition to the explicitly mentioned in the following, we choose the learning rate from the set of $\{3 \times 10^{-4}, 1 \times 10^{-4}, 3 \times 10^{-5}\}$. We use Adam as the optimizer for learning the neural network parameters. Specifically, for BEAR, the MMD constraint parameter is tuned over the candidate set $\{0.1, 0.25, 0.5, 0.75, 1\}$ as in [24]. The samples of MMD is tuned over the set $5, 10, 15$. The KL-control baseline uses automatic temperature tuning as in [24]. For CQL, we follow the author-released default settings but we modify the actor learning rate and use a fixed $\alpha$ instead of the Lagrange variant. This modification is to match the hyperparameters defined in their paper as [13] found the original hyperparameters performed better. For IQL, we use cosine schedule for the actor learning rate. For COMBO, we selected the conservative coefficient from the set $\{0.5, 1, 2.5\}$ and found 1 is the best. We choose $\rho(s, a)$ in [62] as the soft-maximum of the $q$-values and estimated with log-sum-exp. In addition, we set up the learning rate for policy

and value function updates as $1 \times 10^{-4}$ and $3 \times 10^{-5}$, respectively. For the implementation of the methods, ATAC and OptiDICE, we use the source code provided by the authors [7] and [28]. In particular, we follow the basic implementation for OptiDICE setup in [28], we model the value function class, the advantage function class and the policy class using fully-connected MLPs with two hidden layers and ReLU activations, where the number of hidden units on each layer is equal to 256. For the optimization of each network, we use Adam optimizer and its learning rate 0.0003. The batch size is set to be 32. We select the regularization coefficient to be 0.1. Before training neural networks, we preprocess the dataset $\mathcal{D}_{1:n}$ by standardizing observations and rewards. In terms of the details for implementing ATAC, we follow [7], employing separate 3-layer fully connected neural networks for realizing the policy and the critics. Each hidden layer comprises 256 neurons and utilizes a ReLU activation function, while the output layer employs a linear function. We use a softmax policy class for the policy. Optimization is performed using Adam with a minibatch size of 64, and we set the two-timescale stepsizes in [7] as $\eta_{\text{fast}} = 0.0005$ and $\eta_{\text{slow}} = 10^{-3}\eta_{\text{fast}}$, with values $\eta_{\text{fast}} = 5 \times 10^{-4}$ and $\eta_{\text{slow}} = 5 \times 10^{-5}$. The mixing weights in a combination of the temporal difference (TD) losses of the critic and its delayed targets are set to $w = 0.5$ to ensure stability. Finally, for TD3+BC, we follow the default implementation in the original paper but we make a flexible choice on the hyperparameter $\lambda$ not fix $\lambda = \alpha$ in the original paper. We set and implement $\lambda = \frac{\alpha}{\frac{1}{n}\sum_{(s,a)}|q(s,a)|}$, which decreases the value of $\lambda$ when the function estimate is divergent due to extrapolation error [13, 14]. We found this setup helps to improve the performance of the algorithm.

## B.3  Addittion Experiments Results

**Sensitivity Analyses** Tuning parameter selection is an open problem in offline policy optimization. Fortunately, our algorithm has desired robustness to choices of hyperparameters, when we set the hyperparameters satisfying the conditions in Theorem 5.2, i.e., $\mathcal{O}(\frac{n^{1/4}}{d\log(\bar{V}\sqrt{n})})$. To validate the robustness of the proposed algorithm with respect to the hyperparameter-tuning, we conduct sensitivity analyses on the walker2d, hopper, and halfcheetah datasets. Figure B.1 shows that the policy performance is robust over a wide value range of $c^*$ and $\lambda$ ($\lambda, c^*$ in $[1, 0.01]$), and the performance of under our choice ($c^* = 0.1, \lambda = 0.1$) shown in Table B.3 is close the best.

In Tables B.3-B.3, we report the results of the experiments for sensitivity analyses on the values of the hyperparameters vs policy performance on the additional D4RL benchmarks (hopper, walker2d, maze2d), in addition to the results (halfcheetah) we previously presented. Each number in the following tables is the normalized score of the policy at the last iteration of training, averaged over 3 random seeds. From the tables, we can see that, our algorithm demonstrates robustness over a wide value range of hyperparameters. Also, the policy performance under our hyperparameter choice is close to the best performance in the table, which indicates the effectiveness of our proposed hyperparameter selection rule.

Table B.1: Hopper-medium-replay: Our selection rule chooses $\lambda = c^* = 0.25$ with the policy performance $114.0 \pm 2.4$.

| $c^*$(col), $\lambda$(row) | 2.5 | 1 | 0.1 | 0.01 | 0.0025 | 0.001 |
|---|---|---|---|---|---|---|
| 2.5 | $108.1 \pm 2.7$ | $109.7 \pm 2.4$ | $111.6 \pm 3.1$ | $111.1 \pm 2.7$ | $109.5 \pm 2.1$ | $108.3 \pm 4.4$ |
| 1 | $109.3 \pm 1.7$ | $111.8 \pm 2.4$ | $113.2 \pm 2.6$ | $112.9 \pm 2.2$ | $112.0 \pm 3.0$ | $110.7 \pm 3.3$ |
| 0.1 | $112.6 \pm 2.1$ | $113.3 \pm 2.0$ | $114.4 \pm 2.9$ | $114.6 \pm 2.1$ | $113.2 \pm 2.9$ | $112.5 \pm 3.4$ |
| 0.01 | $111.8 \pm 2.8$ | $112.0 \pm 3.6$ | $114.6 \pm 2.9$ | $114.2 \pm 3.3$ | $113.1 \pm 2.7$ | $110.2 \pm 3.4$ |
| 0.0025 | $109.7 \pm 2.6$ | $111.5 \pm 3.2$ | $113.8 \pm 2.6$ | $113.3 \pm 4.4$ | $112.6 \pm 3.7$ | $110.1 \pm 3.7$ |
| 0.001 | $108.2 \pm 3.0$ | $109.5 \pm 3.8$ | $111.9 \pm 3.5$ | $111.2 \pm 2.6$ | $109.8 \pm 3.1$ | $108.4 \pm 4.6$ |

Table B.2: Walker2d-medium-replay: Our selection rule chooses $\lambda = c^* = 0.1$ with the policy performance $101.2 \pm 3.2$.

| $c^*$(col), $\lambda$ (row) | 2.5 | 1 | 0.1 | 0.01 | 0.0025 | 0.001 |
|---|---|---|---|---|---|---|
| 2.5 | $95.8 \pm 2.5$ | $97.4 \pm 2.8$ | $98.4 \pm 2.5$ | $99.1 \pm 3.2$ | $97.8 \pm 3.0$ | $97.9 \pm 3.4$ |
| 1 | $97.3 \pm 2.7$ | $98.0 \pm 3.1$ | $98.8 \pm 2.8$ | $99.4 \pm 3.4$ | $98.7 \pm 2.7$ | $98.1 \pm 3.2$ |
| 0.1 | $97.4 \pm 2.8$ | $98.3 \pm 2.9$ | $101.2 \pm 3.2$ | $101.3 \pm 3.4$ | $98.9 \pm 3.6$ | $97.5 \pm 4.2$ |
| 0.01 | $98.2 \pm 2.8$ | $99.5 \pm 2.9$ | $101.7 \pm 3.9$ | $102.6 \pm 3.4$ | $100.2 \pm 3.1$ | $98.4 \pm 3.3$ |
| 0.0025 | $98.0 \pm 3.6$ | $97.5 \pm 4.2$ | $100.1 \pm 3.6$ | $100.8 \pm 3.5$ | $99.2 \pm 4.2$ | $97.4 \pm 4.0$ |
| 0.001 | $97.2 \pm 3.8$ | $98.5 \pm 3.3$ | $98.2 \pm 3.8$ | $99.3 \pm 3.6$ | $97.8 \pm 5.2$ | $98.2 \pm 4.1$ |

Table B.3: Maze2d-medium: Our selection rule chooses $\lambda = c^* = 2.25$ with the policy performance $138.1 \pm 7.6$.

| $c^*$(col), $\lambda$ (row) | 15 | 10 | 5 | 2.5 | 1 | 0.5 |
|---|---|---|---|---|---|---|
| 15 | $134.5 \pm 4.6$ | $133.9 \pm 5.8$ | $134.8 \pm 4.5$ | $136.7 \pm 6.2$ | $134.8 \pm 6.0$ | $134.9 \pm 5.4$ |
| 10 | $133.7 \pm 4.2$ | $136.7 \pm 5.1$ | $135.8 \pm 6.8$ | $138.4 \pm 7.4$ | $135.7 \pm 12.2$ | $137.5 \pm 8.2$ |
| 5 | $133.9 \pm 5.8$ | $137.3 \pm 6.9$ | $136.6 \pm 5.5$ | $138.3 \pm 9.2$ | $134.9 \pm 7.0$ | $135.1 \pm 5.2$ |
| 2.5 | $137.5 \pm 6.3$ | $135.8 \pm 5.9$ | $140.7 \pm 10.9$ | $138.9 \pm 9.2$ | $132.2 \pm 8.1$ | $133.7 \pm 6.5$ |
| 1 | $134.0 \pm 4.2$ | $137.2 \pm 10.7$ | $133.8 \pm 6.9$ | $137.3 \pm 9.5$ | $138.2 \pm 5.2$ | $137.6 \pm 8.0$ |
| 0.5 | $135.2 \pm 11.8$ | $133.7 \pm 8.3$ | $136.1 \pm 7.8$ | $134.5 \pm 6.7$ | $137.2 \pm 9.2$ | $135.5 \pm 7.1$ |

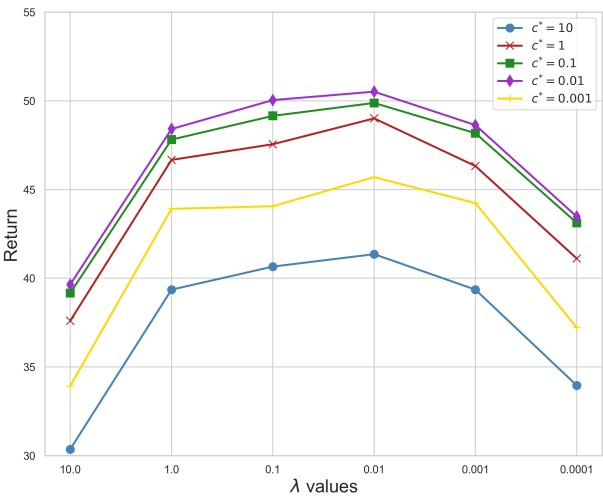

Figure B.1: Sensitivity analysis on the effects of hyperparameters $\lambda$ and $c^*$ for model performance with halfcheetah-medium-replay dataset.

**Empirical evaluation on theoretical results.** We also empirically validate the regret bound in Theorem 5.2. In general, we have no information on the optimal policy and whether it is covered by offline datasets, which makes it challenging to accurately compute the regret in order to verify our theoretic bound. Thus, we carefully design a synthetic environment. We describe the environment in the following: the reward $r(s, a) = (a - \beta s)^\top \Lambda (a - \beta s)$ with coefficient matrix $\beta$ and the negative definite matrix $\Lambda$. Therefore, the optimal policy has an analytical form $\pi^\star(s) = s^\top \beta$, which is important to calculate precise regret. The dataset is generated following $\pi_b$ such that $a = \beta s + \mathcal{N}\left(0, \sigma_0^2 I\right)$, indicating the behavior policy is more different from the optimal one and the data is more explored when $\sigma_0$ is large.

In Figure B.3 we study the convergence rate of regret, which validates the $\mathcal{O}(n^{-1/4})$ rate in Theorem 5.1 and 5.2. The plot shows that the convergence rate is close to $\mathcal{O}(n^{-1/4})$ in all scenarios, which validates the theoretical regret bound of our practical algorithm in Theorem 5.1 and 5.2.

Table B.4: Hyperparameter values for D4RL benchmark.

| Gym locomotion Tasks | Hypereparameters |
|---|---|
| walker2d-medium | 0.25 |
| walker2d-medium-replay | 0.1 |
| walker2d-medium-expert | 0.35 |
| walker2d-random | 0.25 |
| hopper-medium | 0.4 |
| hopper-medium-replay | 0.25 |
| hopper-medium-expert | 0.5 |
| hopper-random | 0.4 |
| halfcheetah-medium | 0.25 |
| halfcheetah-medium-replay | 0.1 |
| halfcheetah-medium-expert | 0.35 |
| halfcheetah-random | 0.25 |
| Maze2d Tasks | Hyperparamters |
| maze2d-umaze | 2 |
| maze2d-medium | 2.25 |
| maze2d-large | 2.5 |

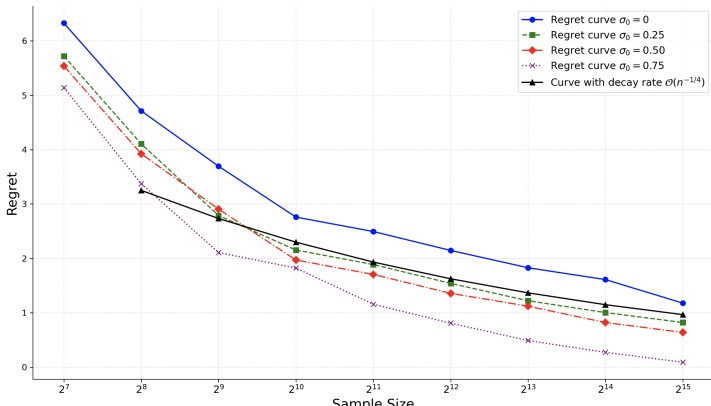

Figure B.2: Convergence rate of the near-optimal regret (compete to the optimal policy) on the synthetic dataset with different degrees of exploration $\sigma_0$. A smaller $\sigma_0$ indicates the training data is less explored.

## C  Proof of Theorem 3.1

### C.1  Proof of Lemma C.1

**Lemma C.1** ([64]). *For any target policy $\pi \in \Pi$ and $\tau \in \Omega$,*

$$\frac{\mathbb{E}_\mu[\tau(s,a)r(s,a)]}{1-\gamma} - J^\mathbb{D}(\pi) = \frac{\mathbb{E}_\mu\left[\tau(s,a)\left(q^\pi(s,a) - \gamma q^\pi(s',\pi)\right) - \lambda\mathbb{D}(\tau(s,a))\right]}{1-\gamma} - q^\pi(s^0,\pi),$$

*where $J^\mathbb{D}(\pi) := J(\pi) - \lambda\xi(\mathbb{D},\tau)$ for $\xi(\mathbb{D},\tau) := \mathbb{E}_\mu[\frac{\mathbb{D}(\tau(s,a))}{1-\gamma}]$, and $q^\pi$ is the unique fixed point of Bellman equation $\mathcal{B}^\pi q = q$.*

*Proof of Lemma C.1.* It follows the definition $J^\mathbb{D}(\pi) = J(\pi) + \lambda\xi(\mathbb{D},\tau)$. Then it is sufficient to show

$$\frac{\mathbb{E}_\mu[\tau(s,a)r(s,a)]}{1-\gamma} - J(\pi) = \frac{\mathbb{E}_\mu\left[\tau(s,a)\left(q^\pi(s,a) - \gamma q^\pi(s',\pi)\right)\right]}{1-\gamma} - q^\pi(s^0,\pi).$$

We rearrange the equation as

$$J(\pi) - q^\pi(s^0,\pi) = \frac{\mathbb{E}_\mu\left[\tau(s,a)\left(-q^\pi(s,a) + r(s,a) + \gamma q^\pi(s',\pi)\right)\right]}{1-\gamma}.$$

Following the definition of $J(\pi) = \mathbb{E}\left[\sum_{t=0}^{\infty} \gamma^t r^t | \pi\right] = q^\pi(s^0, \pi)$, Therefore, it leaves to show the $\frac{\mathbb{E}_\tau[r + \gamma q^\pi(s', \pi) - q^\pi(s,a)]}{1-\gamma} = 0$. As $q^\pi(s,a) = r(s,a) + \mathbb{E}_{s' \sim \mathbb{P}(\cdot|s,a)}[q^\pi(s', \pi)]$ by Bellman evaluation equation, thus we concldue that

$$\frac{\mathbb{E}_\mu\left[\tau(s,a)\left(r(s,a) + \gamma q^\pi(s', \pi) - q^\pi(s,a)\right)\right]}{1-\gamma}$$

$$= \frac{\mathbb{E}_\mu\left[\tau(s,a)\left(r + \gamma q^\pi(s', \pi) - q^\pi(s,a)\right)\right]}{1-\gamma}$$

$$= \frac{\int_{s,a} \mu(s,a)\left[\tau(s,a)\mathbb{E}_{s' \sim \mathbb{P}(\cdot|s,a)}\left[\left(r(s,a) + \gamma q^\pi(s', \pi) - q^\pi(s,a)\right)\right]\right]}{1-\gamma}$$

$$= \frac{\int_{s,a} \mu(s,a)\left[\tau(s,a)\left(r(s,a) + \gamma \mathbb{E}_{s' \sim \mathbb{P}(\cdot|s,a)}\left[q^\pi(s', \pi)\right] - q^\pi(s,a)\right)\right]}{1-\gamma} = 0.$$

This completes the proof. $\square$

## C.2 Proof of Theorem 3.2

*Proof.* To prove the theorem, we follow the proof of Theorem 3.4 in [64]. We need to establish appropriate confidence in upper and lower bounds at the same time. To simplify the notation, we denote $\mathbb{E}_\tau[\cdot] = \mathbb{E}_\mu[\tau(s,a)\cdot]$. At first, we prove for the confidence lower bound. It follows Lemma C.1 and for any $\lambda > 0$, we have

$$\frac{\mathbb{E}_\tau[r(s,a) + \gamma q^\pi(s', \pi) - q^\pi(s,a) - \lambda \mathbb{D}(\tau(s,a))/\tau(s,a)]}{1-\gamma} = J(\pi) - q^\pi(s^0, \pi) - \lambda \xi(\mathbb{D}, \tau).$$

This immediately implies that

$$J(\pi) - q^\pi(s^0, \pi) \geq \frac{\mathbb{E}_\tau[r(s,a) + \gamma q^\pi(s', \pi) - q^\pi(s,a) - \lambda \mathbb{D}(\tau(s,a))/\tau(s,a)]}{1-\gamma}$$

$$\iff J(\pi) \geq \frac{\mathbb{E}_\tau[r(s,a) - \gamma q^\pi(s', \pi) - q^\pi(s,a) - \lambda \mathbb{D}(\tau(s,a))/\tau(s,a)]}{1-\gamma} + q^\pi(s^0, \pi).$$

The above equation helps to obtain the lower bound for the bias evaluation but without concern about the uncertainty quantification due to sampling. To construct the sample estimator for the lower bound and incorporate the uncertainty deviation, we first observe that it suffices to approximation

$$\frac{\mathbb{E}_\tau[r(s,a) - \gamma q^\pi(s', \pi) - q^\pi(s,a) - \lambda \mathbb{D}(\tau(s,a))/\tau(s,a)]}{1-\gamma},$$

by its sample counterparts. That is,

$$\frac{\frac{1}{n}\sum_{i=1}^{n} \tau(s_i, a_i)(r_i + \gamma q^\pi(s_i', \pi) - q^\pi(s_i, a_i))}{1-\gamma} - \lambda \xi_n(\mathbb{D}, \tau)$$

To bound below the uncertainty, this is equivalent to finding a good $\sigma_n$ such that for any $\tau \in \Omega$,

$$\frac{\mathbb{E}_\tau[r(s,a) - \gamma q^\pi(s', \pi) - q^\pi(s,a) - \lambda \mathbb{D}(\tau(s,a))/\tau(s,a)]}{1-\gamma}$$

$$\geq \frac{\frac{1}{n}\sum_{i=1}^{n} \tau(s_i, a_i)(r_i + \gamma q^\pi(s_i', \pi) - q^\pi(s_i, a_i))}{1-\gamma} - \lambda \xi_n(\mathbb{D}, \tau) - \sigma_n^\circ \qquad \text{(C.1)}$$

with probability at least $1 - \delta/2$. Note that the power of $\delta/2$ is due to that we need to further consider the upper confidence bound with also $\delta/2$ power so that the confidence interval holds w.p. $\geq 1 - \delta$.

According to Bellman equation, we know that $r(s,a) - \gamma \mathbb{E}_{s' \sim \mathbb{P}(\cdot|s,a)}[q^\pi(s', \pi)] - q^\pi(s,a) = 0$ for any $s, a$. This implies that

$$\frac{\mathbb{E}_\tau[r(s,a) - \gamma q^\pi(s', \pi) - q^\pi(s,a) - \lambda \mathbb{D}(\tau(s,a))/\tau(s,a)]}{1-\gamma}$$

$$= \frac{\mathbb{E}_\tau[-\lambda \mathbb{D}(\tau(s,a))/\tau(s,a)]}{1-\gamma} \leq 0 \qquad \text{(C.2)}$$

where the last inequality comes from the fact of Definition 3.1 on the detection function $\mathbb{D}(\cdot)$ which is always non-negative.

Combine the inequalities (C.1) and (C.2), it is sufficient to obtain $\sigma_n^\circ$ satisfying the following condition:

$$\sigma_n^\circ \geq \frac{\frac{1}{n}\sum_{i=1}^n \tau(s_i, a_i)\left(r_i + \gamma q^\pi(s_i', \pi) - q^\pi(s_i, a_i)\right)}{1 - \gamma} - \lambda \xi_n(\mathbb{D}, \tau),$$

for any $\tau \in \Omega$. We can rewrite it to use a uniform argument, that is

$$\sup_{\tau \in \Omega}\left\{\frac{1}{n}\sum_{i=1}^n \frac{\tau(s_i, a_i)\left(r_i + \gamma q^\pi(s_i', \pi) - q^\pi(s_i, a_i)\right)}{1 - \gamma} - \lambda \xi_n(\mathbb{D}, \tau)\right\} \leq \sigma_n^\circ$$

Now, recall that we have a condition that

$$\sup_{\tau \in \Omega}\left|\frac{1}{n(1-\gamma)}\sum_{i=1}^n \tau(s_i, a_i)\left(r_i + \gamma q^\pi(s_i', \pi) - q^\pi(s_i, a_i)\right) - \lambda \xi_n(\mathbb{D}, \tau)\right| \leq \sigma_n,$$

which directly implies that

$$\sup_{\tau \in \Omega}\left\{\frac{1}{n}\sum_{i=1}^n \frac{\tau(s_i, a_i)\left(r_i + \gamma q^\pi(s_i', \pi) - q^\pi(s_i, a_i)\right)}{1 - \gamma} - \lambda \xi_n(\mathbb{D}, \tau)\right\} \leq \sigma_n.$$

Therefore, we set $\sigma_n^\circ = \sigma_n$, and combine with (C.1), it obtains that for any $\tau \in \Omega$,

$$J(\pi) \geq \frac{1}{n}\sum_{i=1}^n \frac{\tau(s_i, a_i)\left(r_i + \gamma q^\pi(s_i', \pi) - q^\pi(s_i, a_i)\right)}{1 - \gamma} + q^\pi(s^0, \pi) - \lambda \xi_n(\mathbb{D}, \tau) - \sigma_n$$

$$J(\pi) \geq \frac{1}{n}\sum_{i=1}^n \frac{r_i \tau(s_i, a_i)}{1 - \gamma} + \inf_{q \in \mathcal{Q}} \frac{\frac{1}{n}\sum_{i=1}^n \tau(s_i, a_i)(\gamma q(s_i', \pi) - q(s_i, a_i)) + (1-\gamma)q(s^0, \pi)}{1 - \gamma}$$

$$- \lambda \xi_n(\mathbb{D}, \tau) - \sigma_n^L$$

$$J(\pi) \geq \frac{1}{n}\sum_{i=1}^n \frac{r_i \tau(s_i, a_i)}{1 - \gamma} - \sup_{q \in \mathcal{Q}} \underbrace{\frac{\frac{1}{n}\sum_{i=1}^n \tau(s_i, a_i)(q(s_i, a_i) - \gamma q(s_i', \pi)) - (1-\gamma)q(s^0, \pi)}{1 - \gamma}}_{\widehat{M}_n(-q, \tau)}$$

$$- \lambda \xi_n(\mathbb{D}, \tau) - \sigma_n^L.$$

This completes the proof for the confidence lower bound.

Now, it remains to prove the result for the confidence upper bound. According to the value interval in (3) of the maintext, we observe

$$J(\pi) \leq \frac{\mathbb{E}_\tau\left[r(s, a) - \gamma q^\pi(s', \pi) - q^\pi(s, a) + \lambda \mathbb{D}(\tau(s, a))/\tau(s, a)\right]}{1 - \gamma} + q^\pi(s^0, \pi).$$

To construct the sample estimator for the lower bound and incorporate the uncertainty deviation, we first observe that it suffices to approximation

$$\frac{\mathbb{E}_\tau\left[r(s, a) - \gamma q^\pi(s', \pi) - q^\pi(s, a) + \lambda \mathbb{D}(\tau(s, a))/\tau(s, a)\right]}{1 - \gamma},$$

by its sample counterparts. That is,

$$\frac{\frac{1}{n}\sum_{i=1}^n \tau(s_i, a_i)\left(r_i + \gamma q^\pi(s_i', \pi) - q^\pi(s_i, a_i)\right)}{1 - \gamma} + \lambda \xi_n(\mathbb{D}, \tau)$$

To bound below the uncertainty, this is equivalent to finding a good $\sigma_n$ such that for any $\tau \in \Omega$,

$$\frac{\mathbb{E}_\tau\left[r(s, a) - \gamma q^\pi(s', \pi) - q^\pi(s, a) + \lambda \mathbb{D}(\tau(s, a))/\tau(s, a)\right]}{1 - \gamma}$$

$$\leq \frac{\frac{1}{n}\sum_{i=1}^n \tau(s_i, a_i)\left(r_i + \gamma q^\pi(s_i', \pi) - q^\pi(s_i, a_i)\right)}{1 - \gamma} + \lambda \xi_n(\mathbb{D}, \tau) + \sigma_n^* \quad \text{(C.3)}$$

with probability at least $1 - \delta/2$. According to Bellman equation, we know that $r(s,a) - \gamma\mathbb{E}_{s'\sim\mathbb{P}(\cdot|s,a)}[q^{\pi}(s',\pi)] - q^{\pi}(s,a) = 0$ for any $s, a$. This implies that

$$\frac{\mathbb{E}_{\tau}\left[r(s,a) - \gamma q^{\pi}(s',\pi) - q^{\pi}(s,a) + \lambda\mathbb{D}(\tau(s,a))/\tau(s,a)\right]}{1-\gamma}$$

$$= \frac{\mathbb{E}_{\tau}\left[\lambda\mathbb{D}(\tau(s,a))/\tau(s,a)\right]}{1-\gamma} \geq 0. \tag{C.4}$$

Combine the inequalities (C.3) and (C.4), it is sufficient to obtain $\sigma_n^*$ satisfying the following condition:

$$\sigma_n^* \geq \frac{\frac{1}{n}\sum_{i=1}^{n}\tau(s_i,a_i)\left(r_i + \gamma q^{\pi}(s'_i,\pi) - q^{\pi}(s_i,a_i)\right)}{1-\gamma} + \lambda\xi_n(\mathbb{D},\tau),$$

for any $\tau \in \Omega$. This could be satisfied by the uncertainty deviation condition in Theorem 3.1 that

$$\sup_{\tau\in\Omega}\left\{\frac{1}{n}\sum_{i=1}^{n}\frac{\tau(s_i,a_i)\left(-r_i - \gamma q^{\pi}(s'_i,\pi) + q^{\pi}(s_i,a_i)\right)}{1-\gamma} + \lambda\xi_n(\mathbb{D},\tau)\right\} \geq -\sigma_n.$$

by taking $\sigma_n^* = \sigma_n$. It then obtains that for any $\tau \in \Omega$,

$$\frac{1}{n}\sum_{i=1}^{n}\frac{\tau(s_i,a_i)\left(-r_i - \gamma q^{\pi}(s'_i,\pi) + q^{\pi}(s_i,a_i)\right)}{1-\gamma} + \lambda\xi_n(\mathbb{D},\tau) \geq -\sigma_n$$

$$\frac{1}{n}\sum_{i=1}^{n}\frac{\tau(s_i,a_i)\left(-r_i - \gamma q^{\pi}(s'_i,\pi) + q^{\pi}(s_i,a_i)\right)}{1-\gamma} + \lambda\xi_n(\mathbb{D},\tau) + (J(\pi) - q^{\pi}(s^0,\pi)) \geq -\sigma_n$$

By some algebra, this implies

$$J(\pi) \leq \frac{1}{n}\sum_{i=1}^{n}\frac{r_i\tau(s_i,a_i)}{1-\gamma} + \sup_{q\in\mathcal{Q}}\underbrace{\frac{\frac{1}{n}\sum_{i=1}^{n}\tau(s_i,a_i)(\gamma q(s'_i,\pi) - q(s_i,a_i)) + (1-\gamma)q(s^0,\pi)}{1-\gamma}}_{\widehat{M}_n(q,\tau)}$$

$$+ \lambda\xi_n(\mathbb{D},\tau) + \sigma_n$$

This completes the proof for the confidence upper bound. $\qquad\square$

## D  Proof of Theorem 3.2

*Proof.* It follows the definition of $\widehat{J}_n^-(\pi;\tau)$, we have

$$\widehat{J}_n^-(\pi;\tau) = \frac{1}{n}\sum_{i=1}^{n}\frac{r_i\tau(s_i,a_i)}{1-\gamma} - \sup_{q\in\mathcal{Q}}\widehat{M}_n(-q,\tau) - \lambda\xi_n(\mathbb{D},\tau) - \sigma_n$$

and we obtain the maximizer $\max_{\pi\in\Pi}\left\{\sup_{\tau\in\Omega}\widehat{J}_n^-(\pi;\tau)\right\}$. Therefore, to provide the equivalence, it suffices to show, for any $\pi \in \Pi$, the optimization

$$\sup_{\tau\in\Omega}\left\{\frac{1}{n}\sum_{i=1}^{n}\frac{r_i\tau(s_i,a_i)}{1-\gamma} - \sup_{q\in\mathcal{Q}}\widehat{M}_n(-q,\tau) - \lambda\xi_n(\mathbb{D},\tau) - \sigma_n\right\} \tag{D.1}$$

is equivalent to the optimization

$$\min_{q\in\mathcal{Q}_{\varepsilon_n}} q(s^0,\pi),$$

$$\mathcal{Q}_{\varepsilon_n} = \left\{q \in \mathcal{Q}: \sup_{\tau\in\widetilde{\Omega}_{\widetilde{\sigma}_n}}\left|n^{-1}\sum_{i=1}^{n}\tau(s_i,a_i)(r_i + \gamma q^{\pi}(s'_i,\pi) - q^{\pi}(s_i,a_i))\right| \leq \varepsilon_n\right\},$$

$$\widetilde{\Omega}_{\widetilde{\sigma}_n} = \left\{\tau_\circ/\sup_{\tau_\circ\in\Omega}\|\tau_\circ\|_\Omega \text{ for } \tau_\circ \in \Omega: \xi_n(\mathbb{D},\tau_\circ)) \leq \widetilde{\sigma}_n\right\}.$$

which can be re-expressed as a prime form:

$\min_{q \in \mathcal{Q}} q(s^0, \pi)$, **s.t.** $q$ satisfies

$$\sup_{\tau \in \widetilde{\Omega}_{\widetilde{\sigma}_n}} \left\{ \left| \frac{1}{n} \sum_{i=1}^{n} \frac{\tau(s_i, a_i)(r_i + \gamma q(s'_i, \pi) - q(s_i, a_i))}{1 - \gamma} \right| \right\} \leq \frac{\varepsilon_n}{1 - \gamma} := \widetilde{\varepsilon}_n, \qquad \text{(D.2)}$$

where

$$\widetilde{\Omega}_{\widetilde{\sigma}_n} := \left\{ \frac{\tau_\circ}{\sup_{\tau_\circ \in \Omega} \|\tau_\circ\|_{\Omega_\circ}}, \tau_\circ \in \Omega : \left| \frac{1}{n} \sum_{i=1}^{n} \frac{\mathbb{D}(\tau_\circ(s_i, a_i))}{1 - \gamma} \right| \leq \widetilde{\sigma}_n \right\}.$$

Note that, it follows the definition of $\mathbb{D}(\cdot)$, the above form of $\widetilde{\Omega}_{\widetilde{\sigma}_n}$ can be further relaxed to

$$\widetilde{\Omega}_{\widetilde{\sigma}_n} := \left\{ \frac{\tau_\circ}{\sup_{\tau_\circ \in \Omega_\circ} \|\tau_\circ\|_\Omega}, \tau_\circ \in \Omega : \frac{1}{n} \sum_{i=1}^{n} \frac{\mathbb{D}(\tau_\circ(s_i, a_i))}{1 - \gamma} \leq \widetilde{\sigma}_n \right\}.$$

Therefore, it is sufficient to show the optimization E.21 is equivalent to the optimization D.1.

First, by the rule of $\sup$ & $\inf$: $\sup\{A_n\} = -\inf\{-A_n\}$ for any sequence $A_n$, we observe that

$$\sup_{\tau \in \widetilde{\Omega}_{\widetilde{\sigma}_n}} \left| \frac{1}{n} \sum_{i=1}^{n} \frac{\tau(s_i, a_i)(r_i + \gamma q(s'_i, \pi) - q(s_i, a_i))}{1 - \gamma} \right| \leq \widetilde{\varepsilon}_n$$

$$\implies \inf_{\tau \in \widetilde{\Omega}_{\widetilde{\sigma}_n}} \left\{ \frac{1}{n} \sum_{i=1}^{n} \frac{\tau(s_i, a_i)(q(s_i, a_i) - r_i - \gamma q(s'_i, \pi))}{1 - \gamma} \right\} \geq -\widetilde{\varepsilon}_n$$

For a fixed $\widetilde{\Omega}_{\widetilde{\sigma}_n}$ and $\Upsilon \geq 0$, the optimization (E.21) is equivalent to,

$$\min_{q \in \mathcal{Q}} q(s^0, \pi), \text{ **s.t.** } q \in \left\{ \inf_{\tau \in \widetilde{\Omega}_{\widetilde{\sigma}_n}} \left\{ -\left( \frac{1}{n} \sum_{i=1}^{n} \frac{\tau(s_i, a_i)(r_i + \gamma q(s'_i, \pi) - q(s_i, a_i))}{1 - \gamma} \right) \right\} \geq -\widetilde{\varepsilon}_n \right\}$$

$$\iff \min_{q \in \mathcal{Q}} q(s^0, \pi), \text{ **s.t.** } q \in \left\{ -\inf_{\tau \in \widetilde{\Omega}_{\widetilde{\sigma}_n}} \left\{ -\left( \frac{1}{n} \sum_{i=1}^{n} \frac{\tau(s_i, a_i)(r_i + \gamma q(s'_i, \pi) - q(s_i, a_i))}{1 - \gamma} \right) \right\} \leq \widetilde{\varepsilon}_n \right\}$$

$$\iff \min_{q \in \mathcal{Q}} q(s^0, \pi), \text{ **s.t.** } q \in \left\{ \sup_{\tau \in \widetilde{\Omega}_{\widetilde{\sigma}_n}} \left\{ \frac{1}{n} \sum_{i=1}^{n} \frac{\tau(s_i, a_i)(r_i + \gamma q(s'_i, \pi) - q(s_i, a_i))}{1 - \gamma} \right\} \leq \widetilde{\varepsilon}_n \right\}$$

$$\iff \min_{q \in \mathcal{Q}} q(s^0, \pi) + \sup_{\Upsilon \geq 0} \Upsilon \left( \sup_{\tau \in \widetilde{\Omega}_{\widetilde{\sigma}_n}} \left\{ \frac{1}{n} \sum_{i=1}^{n} \frac{\tau(s_i, a_i)(r_i + \gamma q(s'_i, \pi) - q(s_i, a_i))}{1 - \gamma} \right\} - \widetilde{\varepsilon}_n \right)$$

Furthermore, we can express the above optimization as a prime form w.r.t. to $\tau$. Also, we have an observation that the space

$$\widetilde{\Omega} := \left\{ \frac{\tau_\circ}{\sup_{\tau_\circ \in \Omega} \|\tau_\circ\|_\Omega} : \tau_\circ \in \Omega \right\}$$

where, for any $\tau \in \widetilde{\Omega}$, $\|\tau\|_{\widetilde{\Omega}} \leq 1$. Then we can further write

$$\min_{q \in \mathcal{Q}} q(s^0, \pi) + \sup_{\Upsilon \geq 0} \Upsilon \left( \sup_{\tau \in \widetilde{\Omega}} \left\{ \frac{1}{n} \sum_{i=1}^{n} \frac{\tau(s_i, a_i)(r_i + \gamma q(s'_i, \pi) - q(s_i, a_i))}{1 - \gamma} \right\} - \widetilde{\varepsilon}_n \right)$$

$$\text{**s.t.** } \left\{ \frac{1}{n} \sum_{i=1}^{n} \frac{\mathbb{D}(\tau(s_i, a_i) \sup_{\tau_\circ \in \Omega} \|\tau_\circ\|_\Omega)}{1 - \gamma} \leq \widetilde{\sigma}_n \right\}$$

It follows that the exchange of variables, $\tau^\circ(s, a) = \tau(s, a) \sup_{\tau_\circ \in \Omega} \|\tau_\circ\|_\Omega \in \Omega$ and $\tau(s, a) = \tau^\circ(s, a) / \sup_{\tau_\circ \in \Omega} \|\tau_\circ\|_\Omega \in \widetilde{\Omega}$, so that $\tau$ and $\tau^\circ$ is bijective.

$$\min_{q \in \mathcal{Q}} q(s^0, \pi) + \sup_{\Upsilon \geq 0} \Upsilon \left( \sup_{\tau \in \widetilde{\Omega}} \left\{ \frac{1}{n} \sum_{i=1}^{n} \frac{\tau(s_i, a_i)(r_i + \gamma q(s'_i, \pi) - q(s_i, a_i))}{1 - \gamma} \right\} - \widetilde{\varepsilon}_n \right)$$

$$\text{**s.t.** } \left\{ \frac{1}{n} \sum_{i=1}^{n} \frac{\mathbb{D}(\tau_\circ(s_i, a_i))}{1 - \gamma} \leq \widetilde{\sigma}_n \right\}.$$

This is further equivalent to the form

$$\min_{q \in \mathcal{Q}} \sup_{\Upsilon \leq 0} \sup_{\tau \in \widetilde{\Omega}} \left\{ q(s^0, \pi) + \Upsilon \left( \left\{ \frac{1}{n} \sum_{i=1}^{n} \frac{\tau(s_i, a_i)\left(r_i + \gamma q(s'_i, \pi) - q(s_i, a_i)\right)}{1 - \gamma} \right\} - \widetilde{\varepsilon}_n \right) \right\}$$

$$\textbf{s.t. } \tau \in \left\{ \left| \frac{1}{n} \sum_{i=1}^{n} \frac{\mathbb{D}(\tau_\circ(s_i, a_i))}{1 - \gamma} \right| \leq \widetilde{\sigma}_n \right\}$$

Next, we transform the above prime form to its duality, for any dual variable $\Psi \geq 0$,

$$\min_{q \in \mathcal{Q}} \sup_{\Upsilon \geq 0} \sup_{\tau \in \Omega} \sup_{\Psi \geq 0} \left\{ q(s^0, \pi) + \Upsilon \left( \left\{ \frac{1}{n} \sum_{i=1}^{n} \frac{\tau(s_i, a_i)\left(r_i + \gamma q(s'_i, \pi) - q(s_i, a_i)\right)}{1 - \gamma} \right\} - \widetilde{\varepsilon}_n \right) \right.$$
$$\left. - \Psi \left( \frac{1}{n} \sum_{i=1}^{n} \frac{\mathbb{D}(\tau_\circ(s_i, a_i))}{1 - \gamma} - \widetilde{\sigma}_n \right) \right\}$$

$$\iff \min_{q \in \mathcal{Q}} q(s^0, \pi) + \sup_{\Upsilon \geq 0} \sup_{\tau \in \Omega} \sup_{\Psi \geq 0} \left\{ \Upsilon \left( \left\{ \frac{1}{n} \sum_{i=1}^{n} \frac{\tau(s_i, a_i)\left(r_i + \gamma q(s'_i, \pi) - q(s_i, a_i)\right)}{1 - \gamma} \right\} - \widetilde{\varepsilon}_n \right) \right.$$
$$\left. - \Psi \left( \frac{1}{n} \sum_{i=1}^{n} \frac{\mathbb{D}(\tau_\circ(s_i, a_i))}{1 - \gamma} - \widetilde{\sigma}_n \right) \right\}$$

$$\iff \min_{q \in \mathcal{Q}} q(s^0, \pi) + \sup_{\Upsilon \geq 0} \sup_{\tau \in \Omega} \sup_{\Psi \geq 0} \left\{ \Upsilon \left( \left\{ \frac{1}{n} \sum_{i=1}^{n} \frac{\tau(s_i, a_i)\left(r_i + \gamma q(s'_i, \pi) - q(s_i, a_i)\right)}{1 - \gamma} \right\} \right) \right.$$
$$\left. - \Psi \left( \frac{1}{n} \sum_{i=1}^{n} \frac{\mathbb{D}(\tau_\circ(s_i, a_i))}{1 - \gamma} \right) \right\} - \Upsilon \widetilde{\varepsilon}_n + \Psi \widetilde{\sigma}_n$$

$$\iff \min_{q \in \mathcal{Q}} q(s^0, \pi) + \sup_{\Upsilon \geq 0} \sup_{\tau \in \Omega} \sup_{\Psi \geq 0} \left\{ \Upsilon \left( \left\{ \frac{1}{n} \sum_{i=1}^{n} \frac{\tau(s_i, a_i)\left(q(s_i, a_i) - r_i - \gamma q(s'_i, \pi)\right)}{1 - \gamma} \right\} \right) \right.$$
$$\left. - \Psi \left( \frac{1}{n} \sum_{i=1}^{n} \frac{\mathbb{D}(\tau_\circ(s_i, a_i))}{1 - \gamma} \right) \right\} - \Upsilon \widetilde{\varepsilon}_n + \Psi \widetilde{\sigma}_n.$$

Let $\tau_\circ(s, a) = \Upsilon \tau(s, a)$ over the space $\Omega_\circ$, such that $\Upsilon$ is replaced by $\sup_{\tau_\circ \in \Omega} \|\tau_\circ\|_\Omega$. Moreover, it is feasible to select $\lambda$ equals to the maximizer of $\Psi$, i.e., $\Psi^* = \lambda$, this directly implies that

$$\min_{q \in \mathcal{Q}} \sup_{\tau \in \Omega} \left\{ q(s^0, \pi) + \frac{1}{n} \sum_{i=1}^{n} \frac{\tau(s_i, a_i)\left(r_i + \gamma q(s'_i, \pi) - q(s_i, a_i)\right)}{1 - \gamma} \right.$$
$$\left. - \frac{\lambda}{n} \sum_{i=1}^{n} \frac{\mathbb{D}(\tau(s_i, a_i))}{1 - \gamma} - \|\tau\|_\Omega \widetilde{\varepsilon}_n + \lambda \widetilde{\sigma}_n \right\}$$

$$\iff \min_{q \in \mathcal{Q}} \sup_{\tau \in \Omega} \left\{ q(s^0, \pi) + \frac{1}{n} \sum_{i=1}^{n} \frac{\tau(s_i, a_i)\left(r_i + \gamma q(s'_i, \pi) - q(s_i, a_i)\right)}{1 - \gamma} \right.$$
$$\left. - \frac{\lambda}{n} \sum_{i=1}^{n} \frac{\mathbb{D}(\tau(s_i, a_i))}{1 - \gamma} - \left( \|\tau\|_\Omega \widetilde{\varepsilon}_n - \lambda \widetilde{\sigma}_n \right) \right\}. \tag{D.3}$$

Denote the inner maximizer of (D.3) as $\tau^*$, then we set up

$$\sigma_n = \|\tau^*\|_\Omega \widetilde{\varepsilon}_n - \lambda \widetilde{\sigma}_n$$
$$= \|\tau^*\|_\Omega (1 - \gamma) \varepsilon_n - \lambda \widetilde{\sigma}_n.$$

Then the above expression is equivalent to

$$\min_{q \in \mathcal{Q}} \sup_{\tau \in \Omega} \left\{ q(s^0, \pi) + \frac{1}{n} \sum_{i=1}^{n} \frac{\tau(s_i, a_i)\left(r_i + \gamma q(s'_i, \pi) - q(s_i, a_i)\right)}{1 - \gamma} - \frac{\lambda}{n} \sum_{i=1}^{n} \frac{\mathbb{D}(\tau(s_i, a_i))}{1 - \gamma} \right\} - \sigma_n$$

We check the Slater's condition [40], as

$$q(s^0, \pi) + \frac{1}{n} \sum_{i=1}^{n} \frac{\tau(s_i, a_i) \left(r_i + \gamma q(s_i', \pi) - q(s_i, a_i)\right)}{1 - \gamma} - \frac{\lambda}{n} \sum_{i=1}^{n} \frac{\mathbb{D}(\tau(s_i, a_i))}{1 - \gamma} \qquad \text{(D.4)}$$

is linear on $q$, and also

$$\inf_{\tau \in \Omega} \left\{ -\left( \frac{1}{n} \sum_{i=1}^{n} \frac{\tau(s_i, a_i) \left(r_i + \gamma q(s_i', \pi) - q(s_i, a_i)\right)}{1 - \gamma} - \frac{\lambda}{n} \sum_{i=1}^{n} \frac{\mathbb{D}(\tau(s_i, a_i))}{1 - \gamma} \right) \right\}$$

is convex on $q$, as it is the supremum of a linear function of $q$, then Slater's condition is satisfied and strong duality holds,

$$\min_{q \in \mathcal{Q}} \sup_{\tau \in \Omega} \left\{ q(s^0, \pi) + \frac{1}{n} \sum_{i=1}^{n} \frac{\tau(s_i, a_i) \left(r_i + \gamma q(s_i', \pi) - q(s_i, a_i)\right)}{1 - \gamma} - \frac{\lambda}{n} \sum_{i=1}^{n} \frac{\mathbb{D}(\tau(s_i, a_i))}{1 - \gamma} \right\} - \sigma_n$$

$$\iff \sup_{\tau \in \Omega} \min_{q \in \mathcal{Q}} \left\{ q(s^0, \pi) + \frac{1}{n} \sum_{i=1}^{n} \frac{\tau(s_i, a_i) \left(r_i + \gamma q(s_i', \pi) - q(s_i, a_i)\right)}{1 - \gamma} - \frac{\lambda}{n} \sum_{i=1}^{n} \frac{\mathbb{D}(\tau(s_i, a_i))}{1 - \gamma} \right\} - \sigma_n$$
$$\text{(D.5)}$$

where the order of $\min_{q \in \mathcal{Q}} \sup_{\tau \in \Omega}$ is exchanged to $\sup_{\tau \in \Omega} \min_{q \in \mathcal{Q}}$. According to the max-min form in (D.5), we have

$$\sup_{\tau \in \Omega} \min_{q \in \mathcal{Q}} \left\{ q(s^0, \pi) + \frac{1}{n} \sum_{i=1}^{n} \frac{\tau(s_i, a_i) \left(r_i + \gamma q(s_i', \pi) - q(s_i, a_i)\right)}{1 - \gamma} - \frac{\lambda}{n} \sum_{i=1}^{n} \frac{\mathbb{D}(\tau(s_i, a_i))}{1 - \gamma} \right\} - \sigma_n$$

$$\iff \sup_{\tau \in \Omega} \min_{q \in \mathcal{Q}} \left\{ \frac{1}{n} \sum_{i=1}^{n} \frac{r_i \tau(s_i, a_i)}{1 - \gamma} + q(s^0, \pi) \right.$$
$$\left. + \frac{1}{n} \sum_{i=1}^{n} \frac{\tau(s_i, a_i) \left(\gamma q(s_i', \pi) - q(s_i, a_i)\right)}{1 - \gamma} - \frac{\lambda}{n} \sum_{i=1}^{n} \frac{\mathbb{D}(\tau(s_i, a_i))}{1 - \gamma} \right\} - \sigma_n$$

$$\iff \sup_{\tau \in \Omega} \left\{ \frac{1}{n} \sum_{i=1}^{n} \frac{r_i \tau(s_i, a_i)}{1 - \gamma} + \min_{q \in \mathcal{Q}} \left\{ q(s^0, \pi) + \frac{1}{n} \sum_{i=1}^{n} \frac{\tau(s_i, a_i) \left(\gamma q(s_i', \pi) - q(s_i, a_i)\right)}{1 - \gamma} \right\} \right.$$
$$\left. - \frac{\lambda}{n} \sum_{i=1}^{n} \frac{\mathbb{D}(\tau(s_i, a_i))}{1 - \gamma} \right\} - \sigma_n$$

$$\iff \sup_{\tau \in \Omega} \left\{ \frac{1}{n} \sum_{i=1}^{n} \frac{r_i \tau(s_i, a_i)}{1 - \gamma} - \max_{q \in \mathcal{Q}} \left\{ \frac{1}{n} \sum_{i=1}^{n} \frac{\tau(s_i, a_i) \left(q(s_i, a_i) - \gamma q(s_i', \pi)\right)}{1 - \gamma} - q(s^0, \pi) \right\} \right.$$
$$\left. - \frac{\lambda}{n} \sum_{i=1}^{n} \frac{\mathbb{D}(\tau(s_i, a_i))}{1 - \gamma} \right\} - \sigma_n.$$

It follows the definition of $\widehat{M}_n(-q, \tau)$ and $\xi_n(\mathbb{D}, \tau)$, we conclude the above form is equivalent to

$$\frac{1}{n} \sum_{i=1}^{n} \frac{r_i \tau(s_i, a_i)}{1 - \gamma} - \sup_{q \in \mathcal{Q}} \widehat{M}_n(-q, \tau) - \lambda \xi_n(\mathbb{D}, \tau) - \sigma_n = \widehat{J}_n^-(\pi; \tau).$$

This completes the proof. $\qquad\square$

# E  Proof of Theorem 4.1

## E.1  Proof of Lemma E.1

**Lemma E.1** (Evaluation error lemma). *For any target policy $\pi$ and $q \in \mathcal{Q}$,*

$$J(\pi) - q\left(s^0, \pi\right) = \frac{\mathbb{E}_{d^\pi}\left[r + \gamma q\left(s', \pi\right) - q(s, a)\right]}{1 - \gamma}.$$

*Proof of Lemma E.1.* We follow the proof of Lemma 1 in [60]. First, we observe that $J(\pi) = \frac{\mathbb{E}_{d^\pi}[r]}{1-\gamma}$, then it suffices to show $\mathbb{E}_{d^\pi}[q(s,a) - \gamma q(s',\pi)] = (1-\gamma)q(s^0,\pi)$. It follows the definition of $d_\pi$ and $q(s',\pi)$, we have

$$\frac{\mathbb{E}_{d_\pi}[q(s,a) - \gamma q(s',\pi)]}{1-\gamma}$$

$$= \int_{s,a} \sum_{t=0}^{\infty} \gamma^t P\left(s^t = s, a^t = a|s^0, \pi\right) q(s,a) - \int_{s,a} \sum_{t=1}^{\infty} \gamma^t P\left(s^t = s|s^0, \pi\right) q(s,\pi)$$

$$= \int_{s,a} \sum_{t=0}^{\infty} \gamma^t P\left(s^t = s, a^t = a|s^0, \pi\right) q(s,a) - \int_{s,a} \sum_{t=1}^{\infty} \gamma^t P\left(s^t = s, a^t = a|s^0, \pi\right) q(s,a)$$

$$= \int_{s,a} P\left(s^0 = s, a^0 = a|s^0, \pi\right) q(s,a) = q(s^0, \pi),$$

where the conditional probability $P(\cdot|s^0, \pi)$ is taken by assuming that the system follows the policy $\pi$ with initial state $s^0$. This completes the proof. $\qquad\square$

## E.2    Proof of Lemma E.2

**Lemma E.2.** *Suppose for $\tau \in \Omega$, $\sup_\tau \|\tau(s,a)\|_{L_2(\mu)} \leq \mathcal{U}_2^\tau$ and $\sup_\tau \|\tau(s,a)\|_{L_\infty} \leq \mathcal{U}_\infty^\tau$ and $\sup_q \|q(s,a)\|_{L_\infty} \leq \bar{V}$, given an offline data $\mathcal{D}_{1:n} = \{s_i, a_i, r_i, s_i'\}_{i=1}^n$, w.p. $\geq 1 - \delta$,*

$$\left|\mathbb{E}_\mu\left[\tau(s,a)(r(s,a) + \gamma q(s',\pi) - q(s,a))\right] - \mathbb{P}_n\left[\tau(s_i,a_i)(r_i + \gamma q(s_i',\pi) - q(s_i,a_i))\right]\right|$$

$$\leq \mathcal{U}_2^\tau \sqrt{\frac{2\bar{V}^2 \ln \frac{(e^D \max\{D_\Omega, D_\mathcal{Q}, D_\Pi\} + 1)^3 (\mathcal{U}_2^\tau)^{2D}}{\delta}}{n}} + \frac{2\mathcal{U}_\infty^\tau \bar{V} \ln \frac{(e^D \max\{D_\Omega, D_\mathcal{Q}, D_\Pi\} + 1)^3 (\mathcal{U}_2^\tau)^{2D}}{\delta}}{3n}.$$

*holds for any $\pi \in \Pi$, $\tau \in \Omega$ and $q \in \mathcal{Q}$, and empirical measure $\mathbb{P}_n$. The terms $D_\Omega$, $D_\mathcal{Q}$ and $D_\Pi$ are the pseudo-dimension of $\Omega$, $\mathcal{Q}$ and $\Pi$, respectively, and $D = D_\Omega + D_\mathcal{Q} + D_\Pi$ is so-called effective pseudo-dimension.*

*Proof.* First, we observe that it suffices to bound provide a uniform deviation bound that applies to all $\pi \in \Pi, \tau \in \Omega$, and $q \in \mathcal{Q}$. According to the definition of $\varepsilon$-covering number in Definition L.1, and we define the $\varepsilon$-covering number with resecp to a weighted $L^2$ norm $\|\cdot\|_{L_2(\mu)}$ in the space of $\Omega$, $\mathcal{Q}$ and $\Pi$ as follows:

$$\|\tau_1 - \tau_2\|_{L_2(\mu)} := \sqrt{\int_{\mathcal{S}\times\mathcal{A}} |\tau_1(s,a) - \tau_2(s,a)|^2 d\mu(\mathcal{S}, \mathcal{A})}$$

$$\|q_1 - q_2\|_{L_2(\mu)} := \sqrt{\int_{\mathcal{S}\times\mathcal{A}} |q_1(s,a) - q_2(s,a)|^2 d\mu(\mathcal{S} \times \mathcal{A})}$$

$$\|\pi_1 - \pi_2\|_{L_2(\mu)} := \sqrt{\int_{\mathcal{S}} |\pi_1(\cdot|s) - \pi_2(\cdot|s)|^2 d\mu(\mathcal{S})}. \tag{E.1}$$

where $\mu(\mathcal{S})$ is the marginal measure for $\mu(\mathcal{S} \times \mathcal{A})$. For the product space $\mathcal{G} := \Omega \times \mathcal{Q} \times \Pi$, where the function $g \in \mathcal{G}$ that

$$g(s,a,r,s') = \tau(s,a)(r + \gamma q(s',\pi) - q(s,a)).$$

We have the $L_2(\mu)$ metric for $g_1, g_2 \in \mathcal{G}$, that $\|g_1 - g_2\|_{L_2(\mu)}$ is upper bounded by

$$\sqrt{\int_{\mathcal{S}\times\mathcal{A}} |\mathbb{E}_{r=r(s,a),s'\sim\mathbb{P}(\cdot|s,a)}[g_1(s,a,r,s')] - \mathbb{E}_{r=r(s,a),s'\sim\mathbb{P}(\cdot|s,a)}[g_2(s,a,r,s')]|^2 d\mu(\mathcal{S} \times \mathcal{A})}. \tag{E.2}$$

Based on this $L_2(\mu)$ metric, and to complete the proof, it is sufficient to establish the supremum bound, w.p. $\geq 1 - \delta$,

$$\sup_{g \in \mathcal{G}} |\mathbb{E}_\mu[g(s,a,r,s')] - \mathbb{P}_n[g(s_i,a_i,r_i,s_i')]|.$$

To apply Bernstein -type concentration inequality, we need to first examine the boundedness. According to the boundedness conditions on the function classes, we have

$$\sup_{\tau \in \Omega} \|\tau(s,a)\|_{L_2(\mu)} \leq \mathcal{U}_2^\tau; \quad \sup_{\tau \in \Omega} \|\tau(s,a)\|_{L_\infty} \leq \mathcal{U}_\infty^\tau$$

$$(r(s,a) + \gamma q(s',\pi) - q(s,a)) \in [-\bar{V}, \bar{V}], \forall s, a, s'.$$

It is easy to conclude that

$$\sup_{g \in \mathcal{G}} \|g(s,a,r,s')\|_{L_2(\mu)} \leq \mathcal{U}_2^\tau \bar{V}$$

$$\sup_{g \in \mathcal{G}} \|g(s,a,r,s')\|_{L_\infty} \leq \mathcal{U}_\infty^\tau \bar{V},$$

To quantify the complexity of the product space $\mathcal{G}$, we first need to calculate the $L_2(\mu)$-distance in $\mathcal{G}$. With some calculations, for $g_1, g_2 \in \mathcal{G}$ corresponding to $\tau_1 \times q_1 \times \pi_1$ and $\tau_2 \times q_2 \times \pi_2$, respectively,

$$\|g_1(s,a,r,s') - g_2(s,a,r,s')\|_{L_2(\mu)}$$
$$=\|\tau_1(s,a)(r + \gamma q_1(s',\pi_1) - q_1(s,a)) - \tau_2(s,a)(r + \gamma q_2(s',\pi_2) - q_2(s,a))\|_{L_2(\mu)}$$
$$\leq \bar{V}\|\tau_1(s,a) - \tau_2(s,a)\|_{L_2(\mu)} + \mathcal{U}_2^\tau \|(r + \gamma q_1(s',\pi_1) - q_1(s,a)) - (r + \gamma q_2(s',\pi_2) - q_2(s,a))\|_{L_2(\mu)}$$
$$\quad + \mathcal{U}_2^\tau \|(r + \gamma q_1(s',\pi_1) - q_1(s,a)) - (r + \gamma q_2(s',\pi_2) - q_2(s,a))\|_{L_2(\mu)}$$
$$\leq \bar{V}\|\tau_1(s,a) - \tau_2(s,a)\|_{L_2(\mu)} + \mathcal{U}_2^\tau (1+\gamma)\|q_1(s,a) - q_2(s,a)\|_{L_2(\mu)}$$
$$\quad + \gamma \bar{V}\mathcal{U}_2^\tau \|\pi_1(\cdot|s') - \pi_2(\cdot|s')\|_{L_2(\mu)} \tag{E.3}$$

which leads to

$$\mathcal{N}(3\widetilde{C}\varepsilon, \mathcal{G}, \|\cdot\|_{L_2(\mu)}) \leq \mathcal{N}(\varepsilon, \Omega, \|\cdot\|_{L_2(\mu)})\mathcal{N}(\varepsilon, \mathcal{Q}, \|\cdot\|_{L_2(\mu)})\mathcal{N}(\varepsilon, \Pi, \|\cdot\|_{L_2(\mu)}), \tag{E.4}$$

where $\widetilde{C} := \mathcal{U}_2^\tau(1 + \gamma + \gamma\bar{V}) + \bar{V}$.

To upper bound these factors, we apply the generalized version of Corollary 2 in [16]. For the pseudo-dimension of $\Omega$, $\mathcal{Q}$ and $\Pi$, i.e., $D_\Omega$, $D_\mathcal{Q}$ and $D_\Pi$, and for some $\varepsilon' > 0$,

$$\mathcal{N}\left(3\widetilde{C}\epsilon', \mathcal{G}, \|\cdot\|_{L_2(\mu)}\right) \leq e^3 \left(D_\Omega + 1\right)\left(D_\mathcal{Q} + 1\right)\left(D_\Pi + 1\right)\left(\frac{4e\widetilde{C}}{\epsilon'}\right)^{D_\Omega + D_\mathcal{Q} + D_\Pi}.$$

This also implies

$$\mathcal{N}\left(\epsilon, \mathcal{G}, \|\cdot\|_{L_2(\mu)}\right) \leq e^3 \left(D_\Omega + 1\right)\left(D_\mathcal{Q} + 1\right)\left(D_\Pi + 1\right)\left(\frac{12e\widetilde{C}^2}{\epsilon'}\right)^{D_\Omega + D_\mathcal{Q} + D_\Pi} = C_1 \left(\frac{1}{\varepsilon}\right)^D. \tag{E.5}$$

where $C_1 = e^3 \left(D_\Omega + 1\right)\left(D_\mathcal{Q} + 1\right)\left(D_\Pi + 1\right)(12e\widetilde{C}^2)^D$ and $D = D_\Omega + D_\mathcal{Q} + D_\Pi$, i.e., the "effective" pseudo-dimension. With the calculated function class complexity, we apply empirical Bernstein inequality and union bound, w.p. $\geq 1 - \delta$ with $Z = g(s,a,r,s')$,

$$|\mathbb{E}_\mu[g(s,a,r,s')] - \mathbb{P}_n[g(s_i,a_i,r_i,s_i')]|$$

$$\leq \frac{1}{n}\sqrt{2\sum_{i=1}^n Var_\mu[Z]\ln\frac{8\mathcal{N}\left(\epsilon,\mathcal{G},\|\cdot\|_{L_2(\mu)}\right)}{\delta}} + \frac{2\|Z\|_{L_\infty}\ln\frac{8\mathcal{N}\left(\epsilon,\mathcal{G},\|\cdot\|_{L_2(\mu)}\right)}{\delta}}{3n}$$

$$\leq \frac{1}{n}\sqrt{2\sum_{i=1}^n \mathbb{E}_\mu[Z^2]\ln\frac{8\mathcal{N}\left(\epsilon,\mathcal{G},\|\cdot\|_{L_2(\mu)}\right)}{\delta}} + \frac{2\mathcal{U}_\infty^\tau\bar{V}\ln\frac{8\mathcal{N}\left(\epsilon,\mathcal{G},\|\cdot\|_{L_2(\mu)}\right)}{\delta}}{3n}$$

$$\leq \sqrt{\frac{2n\mathcal{U}_{\tau,2}^2\bar{V}^2\ln\frac{8\mathcal{N}\left(\epsilon,\mathcal{G},\|\cdot\|_{L_2(\mu)}\right)}{\delta}}{n^2}} + \frac{2\mathcal{U}_\infty^\tau\bar{V}\ln\frac{8\mathcal{N}\left(\epsilon,\mathcal{G},\|\cdot\|_{L_2(\mu)}\right)}{\delta}}{3n}$$

$$= \mathcal{U}_2^\tau\sqrt{\frac{2\bar{V}^2\ln\frac{8\mathcal{N}\left(\epsilon,\mathcal{G},\|\cdot\|_{L_2(\mu)}\right)}{\delta}}{n}} + \frac{2\mathcal{U}_\infty^\tau\bar{V}\ln\frac{8\mathcal{N}\left(\epsilon,\mathcal{G},\|\cdot\|_{L_2(\mu)}\right)}{\delta}}{3n}.$$

We set $\varepsilon = \mathcal{O}(\frac{1}{\sqrt{n}})$, and combine with the upper bound for covering number (E.5) by some algebra, thus we have

$$\left| \mathbb{E}_\mu \left[ \tau(s,a)(r(s,a) + \gamma q\left(s',\pi\right) - q(s,a)) \right] - \mathbb{P}_n \left[ \tau(s_i,a_i)\left(r_i + \gamma q\left(s_i',\pi\right) - q(s_i,a_i)\right) \right] \right|$$

$$\leq \mathcal{U}_2^\tau \sqrt{ \frac{ 2\bar{V}^2 \ln \frac{(e^D \max\{D_\Omega, D_\mathcal{Q}, D_\Pi\} + 1)^3 (\mathcal{U}_2^\tau)^{2D}}{\delta} }{n} } + \frac{ 2\mathcal{U}_\infty^\tau \bar{V} \ln \frac{(e^D \max\{D_\Omega, D_\mathcal{Q}, D_\Pi\} + 1)^3 (\mathcal{U}_2^\tau)^{2D}}{\delta} }{3n}.$$

This completes the proof. $\qquad\square$

### E.3 Proof of Lemma E.3

**Lemma E.3.** *For some admissible probability measure or empirical probability measure, denoted as $\nu$, suppose $\sqrt{\mathbb{E}_\nu[(\tau(s,a))^2]} \leq C$ for some positive constant $C$ and $\tau \in \Omega$, and*

$$\sup_{\tau \in \Omega} |\mathbb{E}_\nu[\tau(s,a)(r(s,a) + \gamma q(s',\pi) - q(s,a))]| \leq \varepsilon,$$

*then it holds that $\sqrt{\mathbb{E}_\nu[(r + \gamma q(s',\pi) - q(s,a))^2]} \leq \varepsilon/C$.*

*Proof.* To facilitate the proof, we first define $\widetilde{\tau} := \sup_{\tau \in \Omega} |\mathbb{E}_\nu\left[\tau(s,a)\Delta(q,\pi)\right]|$, for $q \in \mathcal{Q}$, and denote $\Delta(q,\pi) = r(s,a) + \gamma q(s',\pi) - q(s,a)$. Then for any $\pi \in \Pi$ and $q \in \mathcal{Q}$, it satisfies that

$$
\begin{aligned}
C\sqrt{\mathbb{E}_\nu[(r(s,a) + \gamma q(s',\pi) - q(s,a))^2]} &= C\|\Delta(q,\pi)\|_{L_2(\nu)} \\
&= C \langle \Delta(q,\pi), \Delta(q,\pi) \rangle_\nu^{\frac{1}{2}} \\
&= \langle \Delta(q,\pi), \Delta(q,\pi) \rangle_\nu^{\frac{1}{2}} \cdot \left\langle \frac{C\widetilde{\tau}(s,a)}{\|\widetilde{\tau}(s,a)\|_{L_2(\nu)}}, \frac{C\widetilde{\tau}(s,a)}{\|\widetilde{\tau}(s,a)\|_{L_2(\nu)}} \right\rangle_\nu^{\frac{1}{2}} \\
&= \left\langle \Delta(q,\pi), \frac{C\widetilde{\tau}}{\|\widetilde{\tau}\|_{L_2(\nu)}} \right\rangle_\nu \\
&= \sup_{\tau \in \Omega} |\mathbb{E}_\nu\left[\tau(s,a)\Delta(q,\pi)\right]|,
\end{aligned}
$$

where the third equality comes from that the direction $\widetilde{\tau}/\|\widetilde{\tau}(s,a)\|_{L_2(\nu)}$ is aligned with the direction of the maximizer of inner product, and the fourth equality comes from the exact equality condition for Cauchy-Schwarz inequality. This completes the proof. $\qquad\square$

### E.4 Proof of Lemma E.4

**Lemma E.4.** *For any $\pi \in \Pi$ and any $\tau \in \widetilde{\Omega}_{\widetilde{\sigma}_n}$ where $\sup_{\tau \in \widetilde{\Omega}_{\widetilde{\sigma}_n}} \|\tau(s,a)\|_{L_2(\mu)} \leq \mathcal{U}_2^\star$ and $\sup_{\tau \in \widetilde{\Omega}_{\widetilde{\sigma}_n}} \|\tau(s,a)\|_{L_\infty} \leq \mathcal{U}_\infty^\star$, and any $q_1, q_2 \in \mathcal{Q}_{\varepsilon_n}$, given an offline data $\mathcal{D}_{1:n} = \{s_i, a_i, r_i, s_i'\}_{i=1}^n$, w.p. $\geq 1 - \delta$, it holds that*

$$
\begin{aligned}
&\bigg| \mathbb{E}_\mu \left[ \tau(s,a) \left[ (q_1(s,a) - r(s,a) - \gamma q_1\left(s',\pi\right)) - (q_2(s,a) - r(s,a) - \gamma q_2\left(s',\pi\right)) \right] \right] \\
&\quad - \mathbb{P}_n \tau(s_i,a_i) \left[ (q_1(s_i,a_i) - r_i - \gamma q_1\left(s_i',\pi\right)) - (q_2(s_i,a_i) - r_i - \gamma q_2\left(s_i',\pi\right)) \right] \bigg| \\
&\lesssim \mathcal{U}_2^\star \sqrt{ \frac{ 32\bar{V}^2 \ln \frac{8\mathcal{N}\left(\epsilon, \mathcal{G}_{\varepsilon_n, \widetilde{\sigma}_n}, \|\cdot\|_{L_2(\mu)}\right)}{\delta} }{n} } + \frac{ 8\mathcal{U}_\infty^\star \bar{V} \ln \frac{8\mathcal{N}\left(\epsilon, \mathcal{G}_{\varepsilon_n, \widetilde{\sigma}_n}, \|\cdot\|_{L_2(\mu)}\right)}{\delta} }{3n}.
\end{aligned}
$$

*Proof.* At first, we define a product space

$$\mathcal{G}_{\varepsilon_n, \widetilde{\sigma}_n} := \widetilde{\Omega}_{\widetilde{\sigma}_n} \times \mathcal{Q}_{\varepsilon_n} \times \Pi. \tag{E.6}$$

equipped with the $L_2(\mu)$ weighted metric, so that any $g \in \mathcal{G}_{\varepsilon_n, \widetilde{\sigma}_n}$ can be expressed as

$$g(s,a,r,s') = \tau(s,a)\bigg[ (q_1(s,a) - r - \gamma q_1\left(s',\pi\right)) - (q_2(s,a) - r - \gamma q_2\left(s',\pi\right)) \bigg].$$

With some calculation, the $L_2(\mu)$-distance in $\mathcal{G}_{\varepsilon_n,\widetilde{\sigma}_n}$ can be bounded. That is, for $g_1, g_2 \in \mathcal{G}_{\varepsilon_n,\widetilde{\sigma}_n}$ corresponding to $\tau_1, q_1, q_2, \pi_1$ and $\tau_2, q'_1, q'_2, \pi_2$, the $L_2(\mu)$ distance between $g_1$ and $g_2$ is upper bounded as,

$$\|g_1(s,a,r,s') - g_2(s,a,r,s')\|_{L_2(\mu)}$$
$$\leq \|\tau_1(s,a)(r + \gamma q'_1(s',\pi) - q'_1(s,a)) - \tau_2(s,a)(r + \gamma q'_2(s',\pi) - q'_2(s,a))\|_{L_2(\mu)}$$
$$+ \|\tau_1(s,a)(r + \gamma q_1(s',\pi) - q_1(s,a)) - \tau_2(s,a)(r + \gamma q_2(s',\pi) - q_2(s,a))\|_{L_2(\mu)}$$
$$\leq \bar{V}\|\tau_1(s,a) - \tau_2(s,a)\|_{L_2(\mu)} + \mathcal{U}_2^\tau(1+\gamma)\|q_1(s,a) - q_2(s,a)\|_{L_2(\mu)}$$
$$+ \gamma\bar{V}\mathcal{U}_2^\tau\|\pi_1(\cdot|s') - \pi_2(\cdot|s')\|_{L_2(\mu)}. \tag{E.7}$$

Based on (E.7), and following the definition of the covering number in Definition L.1, then we have

$$\mathcal{N}(6\widetilde{C}\varepsilon, \mathcal{G}_{\varepsilon_n,\widetilde{\sigma}_n}, \|\cdot\|_{L_2(\mu)}) \leq \mathcal{N}(\varepsilon, \widetilde{\Omega}_{\widetilde{\sigma}_n}, \|\cdot\|_{L_2(\mu)})\mathcal{N}(\varepsilon, \mathcal{Q}_{\varepsilon_n}, \|\cdot\|_{L_2(\mu)})\mathcal{N}(\varepsilon, \Pi, \|\cdot\|_{L_2(\mu)})$$

where $\widetilde{C} := \mathcal{U}_2^\star(1 + \gamma + \gamma\bar{V}) + \bar{V}$. Accordingly, follows Corollary 2 in [16], by some algebra,

$$\mathcal{N}\left(\epsilon, \mathcal{G}_{\varepsilon_n,\widetilde{\sigma}_n}, \|\cdot\|_{L_2(\mu)}\right) \leq e^3\left(D_{\widetilde{\Omega}_{\widetilde{\sigma}_n}} + 1\right)\left(D_{\mathcal{Q}_{\varepsilon_n}} + 1\right)(D_\Pi + 1)\left(\frac{24e\widetilde{C}^2}{\epsilon'}\right)^{D_{\widetilde{\Omega}_{\widetilde{\sigma}_n}} + D_{\mathcal{Q}_{\varepsilon_n}} + D_\Pi}$$

$$= C'_1\left(\frac{1}{\varepsilon}\right)^D$$

where $C'_1 = e^3\left(D_{\widetilde{\Omega}_{\widetilde{\sigma}_n}} + 1\right)\left(D_{\mathcal{Q}_{\varepsilon_n}} + 1\right)(D_\Pi + 1)(24e\widetilde{C}^2)^D$ and $D = D_{\widetilde{\Omega}_{\widetilde{\sigma}_n}} + D_{\mathcal{Q}_{\varepsilon_n}} + D_\Pi$. Next, we apply empirical Bernstein concentration inequality and union bound as in the proof of Lemma E.2, we conclude that

$$\left| \mathbb{E}_\mu\left[\tau(s,a)\left[(q_1(s,a) - r(s,a) - \gamma q_1(s',\pi)) - (q_2(s,a) - r(s,a) - \gamma q_2(s',\pi))\right]\right] \right.$$

$$\left. - \mathbb{P}_n\tau(s_i,a_i)\left[(q_1(s_i,a_i) - r_i - \gamma q_1(s'_i,\pi)) - (q_2(s_i,a_i) - r_i - \gamma q_2(s'_i,\pi))\right] \right|$$

$$\lesssim \mathcal{U}_2^\star\sqrt{\frac{32\bar{V}^2 \ln\frac{8\mathcal{N}\left(\epsilon,\mathcal{G}_{\varepsilon_n,\widetilde{\sigma}_n},\|\cdot\|_{L_2(\mu)}\right)}{\delta}}{n}} + \frac{8\mathcal{U}_\infty^\star\bar{V} \ln\frac{8\mathcal{N}\left(\epsilon,\mathcal{G}_{\varepsilon_n,\widetilde{\sigma}_n},\|\cdot\|_{L_2(\mu)}\right)}{\delta}}{3n}. \tag{E.8}$$

This completes the proof. $\qquad\square$

## E.5 Proof of Lemma E.5

**Lemma E.5.** *Define the maximizer* $\overline{q^\pi} := \arg\max_{q\in\mathcal{Q}_{\varepsilon_n}} q(s^0,\pi)$ *and the minimizer* $\underline{q^\pi} := \arg\min_{q\in\mathcal{Q}_{\varepsilon_n}} q(s^0,\pi)$ *over the confidence set* $\mathcal{Q}_{\varepsilon_n}$ *with both* $q = \overline{q^\pi}$ *and* $q = \underline{q^\pi}$ *satisfy that*

$$\left|\frac{1}{n}\sum_{i=1}^n \tau(s_i,a_i)\left(r_i + \gamma q(s'_i,\pi) - q(s_i,a_i)\right)\right| \leq \varepsilon_n, \forall\tau\in\widetilde{\Omega}_{\widetilde{\sigma}_n}.$$

*There must exist an MDP* $\{\mathcal{S}, \mathcal{A}, \mathbb{P}_{\max}(\mathbb{P}_{\min}), \gamma, r_{\max}(r_{\min}), s^0\}$ *which is identical to the true environment MDP with* $\mathbb{P}_{\max}(\mathbb{P}_{\min}) = \mathbb{P}$*: MDP$^\star$ only except that the reward function* $r_{\max}(r_{\min})$ *is redefined as*

$$r_{\max}(s,a) = \overline{q^\pi}(s,a) - \gamma\mathbb{E}_{s'\sim\mathbb{P}_{\max}(\cdot|s,a)}\left[\sum_{a'\in\mathcal{A}} \pi(a'|s')\overline{q^\pi}(s',a')\right],$$

$$r_{\min}(s,a) = \underline{q^\pi}(s,a) - \gamma\mathbb{E}_{s'\sim\mathbb{P}_{\min}(\cdot|s,a)}\left[\sum_{a'\in\mathcal{A}} \pi(a'|s')\underline{q^\pi}(s',a')\right].$$

*In addition, the reward function* $r_{\max}(r_{\min})$ *is approximating the true reward, i.e.,* $\|r_{\max}(s,a) - r(s,a)\|_{L_2(\mu)}$ *or* $\|r_{\min}(s,a) - r(s,a)\|_{L_2(\mu)}$ *is upper bounded by*

$$\sqrt{\frac{2\bar{V}^2 \ln\frac{8\mathcal{N}\left(\epsilon,\mathcal{G}_{\varepsilon_n,\widetilde{\sigma}_n},\|\cdot\|_{L_2(\mu)}\right)}{\delta}}{n}} + \frac{2\mathcal{U}_\infty\bar{V} \ln\frac{8\mathcal{N}\left(\epsilon,\mathcal{G}_{\varepsilon_n,\widetilde{\sigma}_n},\|\cdot\|_{L_2(\mu)}\right)}{\delta}}{3n} + \frac{\varepsilon_n}{\mathcal{U}_2^\star}.$$

*for some constant* $\mathcal{U}_\infty := \frac{\mathcal{U}_\infty^\star}{\mathcal{U}_2^\star}$ *for* $\sup_{\tau \in \widetilde{\Omega}_{\bar{\sigma}_n}} \|\tau(s,a)\|_{L_2(\mu)} \leq \mathcal{U}_2^\star$ *and* $\sup_{\tau \in \widetilde{\Omega}_{\bar{\sigma}_n}} \|\tau(s,a)\|_{L_\infty} \leq \mathcal{U}_\infty^\star$.

*Proof.* Without loss of generality, we prove the lemma for $r_{\max}$, and the results for $r_{\min}$ can be obtained in a similar way. It follows from the definition of $r_{\max}$, i.e.,

$$r_{\max}(s,a) = \overline{q^\pi}(s,a) - \gamma \mathbb{E}_{s' \sim \mathbb{P}_{\max}(\cdot|s,a)} \left[ \sum_{a' \in \mathcal{A}} \pi^k(a'|s')\overline{q^\pi}(s',a') \right], \tag{E.9}$$

we re-arrange the terms as follows:

$$\overline{q^\pi}(s,a) = r_{\max}(s,a) + \gamma \mathbb{E}_{s' \sim \mathbb{P}_{\max}(\cdot|s,a)} \left[ \sum_{a' \in \mathcal{A}} \pi(a'|s')\overline{q^\pi}(s',a') \right]. \tag{E.10}$$

This is exactly the Bellman equation over the MDP $\{\mathcal{S}, \mathcal{A}, \mathbb{P}_{\max}, \gamma, r_{\max}, s^0\}$. It follows the equivalence between $\mathbb{P}_{\max}$ and $\mathbb{P}$, we further observe that $\overline{q^\pi}(\cdot, \cdot)$ is the true $q$-function for policy $\pi$ in the MDP $\{\mathcal{S}, \mathcal{A}, \mathbb{P}, \gamma, r_{\max}, s^0\}$. To show the reward function $r_{\max}$ approximates the true reward function $r$. It follows from Lemma E.2 with $\|\tau(s,a)\|_{L_2(\mu)}^2 \leq \mathcal{U}_{\text{prime},2}^2$ as $\tau \in \widetilde{\Omega}_{\bar{\sigma}_n}$, we have $\sup_{\tau \in \widetilde{\Omega}_{\bar{\sigma}_n}} |\mathbb{E}_\mu [\tau(s,a)\Delta(\overline{q^\pi}, \pi)]|$ for $\Delta(q, \pi) = r(s,a) + \gamma q(s', \pi) - q(s,a)$, is upper bounded by

$$\mathcal{U}_2^\star \sqrt{\frac{2\bar{V}^2 \ln \frac{8\mathcal{N}\left(\epsilon, \mathcal{G}_{\varepsilon_n, \bar{\sigma}_n}, \|\cdot\|_{L_2(\mu)}\right)}{\delta}}{n}} + \frac{2\mathcal{U}_\infty \mathcal{U}_2^\star \bar{V} \ln \frac{8\mathcal{N}\left(\epsilon, \mathcal{G}_{\varepsilon_n, \bar{\sigma}_n}, \|\cdot\|_{L_2(\mu)}\right)}{\delta}}{3n} + \varepsilon_n.$$

With $\sup_{\tau \in \widetilde{\Omega}_{\bar{\sigma}_n}} |\mathbb{E}_\mu [\tau(s,a)\Delta(\overline{q^\pi}, \pi)]|$ and Lemma E.3 on $\|\tau(s,a)\|_{L_2(\mu)}^2 \leq \mathcal{U}_{\text{prime},2}^2$, we obtain

$$\|\Delta(\overline{q^\pi}, \pi)\|_{L_2(\mu)} \leq \sqrt{\frac{2\bar{V}^2 \ln \frac{8\mathcal{N}\left(\epsilon, \mathcal{G}_{\varepsilon_n, \bar{\sigma}_n}, \|\cdot\|_{L_2(\mu)}\right)}{\delta}}{n}} + \frac{2\mathcal{U}_\infty \bar{V} \ln \frac{8\mathcal{N}\left(\epsilon, \mathcal{G}_{\varepsilon_n, \bar{\sigma}_n}, \|\cdot\|_{L_2(\mu)}\right)}{\delta}}{3n} + \varepsilon_n/\mathcal{U}_{\text{prime},2}^2.$$

Since $\mathbb{P}_{\max} = \mathbb{P}$, and follow the definition of $r_{\max}$ in (E.9), we have

$$\|r(s,a) - r_{\max}(s,a)\|_{L_2(\mu)} = \left\| r(s,a) - \overline{q^\pi}(s,a) + \gamma \mathbb{E}_{s' \sim \mathbb{P}_{\max}(\cdot|s,a)} \left[ \sum_{a' \in \mathcal{A}} \pi(a'|s')\overline{q^\pi}(s', \pi) \right] \right\|_{L_2(\mu)}$$

$$= \|\Delta(\overline{q^\pi}, \pi)\|_{L_2(\mu)}.$$

Combining with the upper bound on $\|\Delta(\overline{q^\pi}, \pi)\|_{L_2(\mu)}$, this completes the proof. $\square$

## E.6 Proof of Lemma E.6

**Lemma E.6** (Upper Bound for Version Space Function over $\mu$). *On the notations and definitions in Lemma E.5, where $\overline{q^\pi}(\cdot, \cdot)$ and $\underline{q^\pi}(\cdot, \cdot)$ are the true action-value function under policy $\pi$ for the MDPs $\{\mathcal{S}, \mathcal{A}, \mathbb{P}_{\max}, \gamma, r_{\max}, s^0\}$ and $\{\mathcal{S}, \mathcal{A}, \mathbb{P}_{\min}, \gamma, r_{\min}, s^0\}$, respectively. Then*

$$\|\underline{q^\pi}(s,a) - \overline{q^\pi}(s,a)\|_{L_2(\mu)} \leq \frac{2\varepsilon_r}{1 - \gamma},$$

*where*

$$\varepsilon_r = \sqrt{\frac{2\bar{V}^2 \ln \frac{8\mathcal{N}\left(\epsilon, \mathcal{G}_{\varepsilon_n, \bar{\sigma}_n}, \|\cdot\|_{L_2(\mu)}\right)}{\delta}}{n}} + \frac{2\mathcal{U}_\infty \bar{V} \ln \frac{8\mathcal{N}\left(\epsilon, \mathcal{G}_{\varepsilon_n, \bar{\sigma}_n}, \|\cdot\|_{L_2(\mu)}\right)}{\delta}}{3n} + \varepsilon_n/\mathcal{U}_{prime,2}^2,$$

*for* $\mathcal{U}_\infty \geq 0$.

*Proof.* According to Lemma E.5 and the definitions of $r_{\max}$ and $r_{\min}$, we have the reward functions $r_{\max}$ and $r_{\min}$ are bounded above over $L_2(\mu)$, i.e.,

$$\|r_{\max}(s,a) - r_{\min}(s,a)\|_{L_2(\mu)} = \|r_{\max}(s,a) - r(s,a) + r(s,a) - r_{\min}(s,a)\|_{L_2(\mu)}$$

$$\leq \|r_{\max}(s,a) - r(s,a)\|_{L_2(\mu)} + \|r(s,a) - r_{\min}(s,a)\|_{L_2(\mu)}$$

$$\leq 2\varepsilon_r. \tag{E.11}$$

By the fact that $\overline{q^\pi}(\cdot,\cdot)$ and $\underline{q^\pi}(\cdot,\cdot)$ are the true action-value functions for policy $\pi$ in the MDPs $\{\mathcal{S},\mathcal{A},\mathbb{P}_{\max},\gamma,r_{\max},s^0\}$ and $\{\mathcal{S},\mathcal{A},\mathbb{P}_{\min},\gamma,r_{\min},s^0\}$, respectively. Then by the definition of the reward functions $r_{\max}$ and $r_{\min}$, we have

$$\|r_{\max}(s,a) - r_{\min}(s,a)\|_{L_2(\mu)}$$

$$= \left\| \left( \overline{q^\pi}(s,a) - \gamma\mathbb{E}_{s'\sim\mathbb{P}_{\max}(\cdot|s,a)}\left[ \sum_{a'\in\mathcal{A}} \pi(a'|s')\overline{q^\pi}(s',a') \right] \right) \right.$$

$$\left. - \left( \underline{q^\pi}(s,a) - \gamma\mathbb{E}_{s'\sim\mathbb{P}_{\min}(\cdot|s,a)}\left[ \sum_{a'\in\mathcal{A}} \pi(a'|s')\underline{q^\pi}(s',a') \right] \right) \right\|_{L_2(\mu)}$$

$$=: \|(\overline{q^\pi}(s,a) - \mathbb{P}^\pi\overline{q^\pi}(s,a)) - (\underline{q^\pi}(s,a) - \mathbb{P}^\pi\underline{q^\pi}(s,a))\|_{L_2(\mu)} \qquad (\text{E.12})$$

where $\mathbb{P}^\pi$ is the transition kernel under the policy $\pi$. We re-organize (E.12), and obtain

$$\|(\overline{q^\pi}(s,a) - \gamma\mathbb{P}^\pi\overline{q^\pi}(s,a)) - (\underline{q^\pi}(s,a) - \gamma\mathbb{P}^\pi\underline{q^\pi}(s,a))\|_{L_2(\mu)}$$

$$= \|\overline{q^\pi}(s,a) - \underline{q^\pi}(s,a) + \gamma\mathbb{P}^\pi(\underline{q^\pi}(s,a) - \overline{q^\pi}(s,a))\|_{L_2(\mu)}$$

$$= \|(\mathbb{I} - \gamma\mathbb{P}^\pi)(\underline{q^\pi}(s,a) - \overline{q^\pi}(s,a))\|_{L_2(\mu)}$$

$$\geq \|\underline{q^\pi}(s,a) - \overline{q^\pi}(s,a)\|_{L_2(\mu)} - \gamma\|\mathbb{P}^\pi(\underline{q^\pi}(s,a) - \overline{q^\pi}(s,a))\|_{L_2(\mu)}$$

$$\geq \|\underline{q^\pi}(s,a) - \overline{q^\pi}(s,a)\|_{L_2(\mu)} - \gamma\|(\underline{q^\pi}(s,a) - \overline{q^\pi}(s,a))\|_{L_2(\mu)}$$

$$\geq (1-\gamma)\|\underline{q^\pi}(s,a) - \overline{q^\pi}(s,a)\|_{L_2(\mu)},$$

where the second inequality comes from each element of $\mathbb{P}^\pi$ is a convex average of $\underline{q^\pi}(s,a) - \overline{q^\pi}(s,a)$. Combine with the inequality (E.11), we conclude that

$$(1-\gamma)\|\underline{q^\pi}(s,a) - \overline{q^\pi}(s,a)\|_{L_2(\mu)} \leq 2\varepsilon_r \implies \|\underline{q^\pi}(s,a) - \overline{q^\pi}(s,a)\|_{L_2(\mu)} \leq \frac{2\varepsilon_r}{1-\gamma}.$$

We explicitly express the weighted $L_2(\mu)$ norm on $\underline{q^\pi}(s,a) - \overline{q^\pi}(s,a)$, i.e.,

$$\|\underline{q^\pi}(s,a) - \overline{q^\pi}(s,a)\|^2_{L_2(\mu)} = \sum_{a\in\mathcal{A}}\sum_{s\in\mathcal{S}} \left[\underline{q^\pi}(s,a) - \overline{q^\pi}(s,a)\right]^2 \cdot \mu(s,a) \leq \left(\frac{2\varepsilon_r}{1-\gamma}\right)^2.$$

By the non-negative of the term $\mu(s,a)$, we conclude that

$$\sup_{a\in\mathcal{A},s\in\mathcal{S}} \left|\underline{q^\pi}(s,a) - \overline{q^\pi}(s,a)\right| \leq \frac{2\varepsilon_r}{1-\gamma}, \quad \text{almost surely for } (s,a) \text{ with } \mu(s,a) > 0.$$

This completes the proof. $\qquad\qquad\square$

### E.7 Proof of Lemma E.7

**Lemma E.7.** *Suppose for* $\tau \in \Omega$, $\sup_\tau \|\tau(s,a)\|_{L_2(\mu)} \leq \mathcal{U}_2^\tau$ *and* $\sup_\tau \|\tau(s,a)\|_{L_\infty} \leq \mathcal{U}_\infty^\tau$ *and* $\sup_q \|q(s,a)\|_{L_\infty} \leq \bar{V}$, *given an offline data* $\mathcal{D}_{1:n} = \{s_i,a_i,r_i,s_i'\}_{i=1}^n$, *w.p.* $\geq 1-\delta$,

$$\left| \mathbb{E}_\mu\left[ \tau(s,a)\left(r + \gamma q\left(s',\pi\right) - q(s,a)\right) - \lambda\mathbb{D}(\tau(s,a)) \right] \right.$$

$$\left. - \mathbb{P}_n\left( \tau(s_i,a_i)\left(r_i + \gamma q\left(s_i',\pi\right) - q(s_i,a_i)\right) - \lambda\mathbb{D}(\tau(s_i,a_i)) \right) \right|$$

$$\lesssim \left(\mathcal{U}_2^\tau\bar{V} + \lambda\|\mathbb{D}(\tau(s,a))\|^{UB}_{L_2(\mu)}\right) \sqrt{\frac{2\ln\frac{(e^D \max\{D_\Omega,D_\mathcal{Q},D_\Pi\}+1)^3(L\mathcal{U}_2^\tau)^{2D}}{\delta}}{n}}$$

$$+ \frac{2\left(\mathcal{U}_\infty^\tau\bar{V} + \lambda\|\mathbb{D}(\tau(s,a))\|^{UB}_{L_\infty}\right)\ln\frac{(e^D \max\{D_\Omega,D_\mathcal{Q},D_\Pi\}+1)^3(L\mathcal{U}_2^\tau)^{2D}}{\delta}}{3n},$$

*where* $L < \infty$ *issome bounded Lipschitz constant of* $\mathbb{D}(\cdot)$, *and* $\|\mathbb{D}(\tau(s,a))\|^{UB}_{L_2(\mu)} = \sup_{\tau\in\Omega}\|\mathbb{D}(\tau(s,a))\|_{L_2(\mu)}$ *and* $\|\mathbb{D}(\tau(s,a))\|^{UB}_{L_\infty} = \sup_{\tau\in\Omega}\|\mathbb{D}(\tau(s,a))\|_{L_\infty}$.

*Proof.* Define the product space $\widetilde{\mathcal{G}} := \Omega \times \mathcal{Q} \times \Pi$, and the $L_2$ weighted metric

$$\|g_1 - g_2\|_{L_2(\mu)}$$

$$:= \sqrt{\int_{\mathcal{S} \times \mathcal{A}} |\mathbb{E}_{r \sim r(s,a), s' \sim \mathbb{P}(\cdot|s,a)}[g_1(s,a,r,s')] - \mathbb{E}_{r \sim r(s,a), s' \sim \mathbb{P}(\cdot|s,a)}[g_2(s,a,r,s')]|^2 d\mu(\mathcal{S} \times \mathcal{A})}$$

where $g_1, g_2 \in \widetilde{\mathcal{G}}$ for given $\lambda > 0$ such that $g(s,a,r,s') = \tau(s,a)(r + \gamma q(s',\pi) - q(s,a)) - \lambda\mathbb{D}(\tau(s,a))$ for any $g \in \widetilde{\mathcal{G}}$. To apply empirical Bernstein inequality, we study the boundedness conditions: $(r + \gamma q(s',\pi) - q(s,a)) \in [-\bar{V}, \bar{V}], \forall q \in \mathcal{Q}, \sup_{\tau \in \Omega} \|\tau(s,a)\|_{L_2(\mu)} \leq \mathcal{U}_2^\tau, \sup_{\tau \in \Omega} \|\tau(s,a)\|_{L_\infty} \leq \mathcal{U}_\infty^\tau, \lambda\sup_{\tau \in \Omega} \|\mathbb{D}(\tau(s,a))\|_{L_2(\mu)} \in [0, \lambda\|\mathbb{D}(\tau(s,a))\|_{L_2(\mu)}^{\text{UB}}], \lambda\sup_{\tau \in \Omega}\|\mathbb{D}(\tau(s,a))\|_{L_\infty} \in [0, \lambda\|\mathbb{D}(\tau(s,a))\|_{L_\infty}^{\text{UB}}]$. By some calculation, the $L_2(\mu)$-distance in $\widetilde{\mathcal{G}}$ can be bounded. For $g_1, g_2 \in \widetilde{\mathcal{G}}$ corresponding to $\tau_1 \times q_1 \times \pi_1$ and $\tau_2 \times q_2 \times \pi_2$, respectively, we have

$$\|g_1(s,a,r,s') - g_2(s,a,r,s')\|_{L_2(\mu)}$$

$$= \bigg| \tau_1(s,a)(r + \gamma q_1(s',\pi_1) - q_1(s,a)) - \lambda\mathbb{D}(\tau_1(s,a))$$

$$- (\tau_2(s,a)(r + \gamma q_2(s',\pi_2) - q_2(s,a)) - \lambda\mathbb{D}(\tau_2(s,a))) \bigg|$$

$$\leq 2\bar{V}\|\tau_1(s,a) - \tau_2(s,a)\|_{L_2(\mu)} + \mathcal{U}_2^\tau\|(r + \gamma q_1(s',\pi_1) - q_1(s,a)) - (r + \gamma q_2(s',\pi_2) - q_2(s,a))\|_{L_2(\mu)}$$

$$+ \mathcal{U}_2^\tau\|(r + \gamma q_1(s',\pi) - q_1(s,a)) - (r + \gamma q_2(s',\pi_2) - q_2(s,a))\|_{L_2(\mu)} + \lambda\|\mathbb{D}(\tau_1(s,a)) - \mathbb{D}(\tau_2(s,a))\|_{L_2(\mu)}$$

$$\leq 2\bar{V}\|\tau_1(s,a) - \tau_2(s,a)\|_{L_2(\mu)} + \mathcal{U}_2^\tau\|(r + \gamma q_1(s',\pi_1) - q_1(s,a)) - (r + \gamma q_2(s',\pi_2) - q_2(s,a))\|_{L_2(\mu)}$$

$$+ \mathcal{U}_2^\tau\|(r + \gamma q_1(s',\pi) - q_1(s,a)) - (r + \gamma q_2(s',\pi_2) - q_2(s,a))\|_{L_2(\mu)} + \lambda L\|\tau_1(s,a) - \tau_2(s,a)\|_{L_2(\mu)}$$

$$\leq 2\bar{V}\|\tau_1(s,a) - \tau_2(s,a)\|_{L_2(\mu)} + \gamma\bar{V}\mathcal{U}_2^\tau\|\pi_1 - \pi_2\|_{L_2(\mu)}$$

$$+ \mathcal{U}_2^\tau(1 + \gamma)\|q_1(s,a) - q_2(s,a)\|_{L_2(\mu)} + \lambda L\|\tau_1(s,a) - \tau_2(s,a)\|_{L_2(\mu)}, \tag{E.13}$$

where the last inequality comes from the $\mathbb{D}$ is $M$-strongly convex function and thus locally Lipschitz with a Lipschitz constant $L \leq \infty$. Also, we note that

$$\|\sup_{\tau \in \Omega} \mathbb{D}(\tau(s,a)) - 0\|_{L_2(\mu)} \leq L\|\tau^\star(s,a) - 1\|_{L_2(\mu)} \leq L\max\{1, \mathcal{U}_2^\tau\} \leq L\mathcal{U}_2^\tau,$$

where the last inequality holds for $\mathcal{U}_2^\tau \geq 1$. The metric distance (E.13) directly leads to the upper bound for the covering number over $\widetilde{\mathcal{G}}$:

$$\mathcal{N}(4C^\star\varepsilon, \widetilde{\mathcal{G}}, \|\cdot\|_{L_2(\mu)}) \leq \mathcal{N}(\varepsilon, \Omega, \|\cdot\|_{L_2(\mu)})\mathcal{N}(\varepsilon, \mathcal{Q}, \|\cdot\|_{L_2(\mu)})\mathcal{N}(\varepsilon, \Pi, \|\cdot\|_{L_2(\mu)}),$$

where $C^\star := \bar{V}(2 + \gamma\mathcal{U}_2^\tau) + \mathcal{U}_2^\tau(1 + \gamma) + \lambda L$. Apply the generalize version of Corollary 2 in [16], which implies

$$\mathcal{N}\left(\epsilon, \widetilde{\mathcal{G}}, \|\cdot\|_{L_2(\mu)}\right) \leq e^3 (D_\Omega + 1)(D_\mathcal{Q} + 1)(D_\Pi + 1)\left(\frac{16e(C^\star)^2}{\epsilon}\right)^D, \tag{E.14}$$

where $D = D_\Omega + D_\mathcal{Q} + D_\Pi$. By empirical Bernstein inequality and a union bound, we have that with probability at least $1 - \delta$ and $Z = g(s,a,r,s')$,

$$|\mathbb{E}_\mu[g(s,a,r,s')] - \mathbb{P}_n[g(s_i,a_i,r_i,s_i')]|$$

$$\leq \frac{1}{n}\sqrt{2\sum_{i=1}^n Var_\mu[Z]\ln\frac{8\mathcal{N}\left(\epsilon, \widetilde{\mathcal{G}}, \|\cdot\|_{L_2(\mu)}\right)}{\delta}} + \frac{2\left(\mathcal{U}_\infty^\tau\bar{V} + \lambda\|\mathbb{D}(\tau(s,a))\|_{L_\infty}^{\text{UB}}\right)\ln\frac{8\mathcal{N}\left(\epsilon, \widetilde{\mathcal{G}}, \|\cdot\|_{L_2(\mu)}\right)}{\delta}}{3n}$$

$$\leq \frac{1}{n}\sqrt{2\sum_{i=1}^n \mathbb{E}_\mu[Z^2]\ln\frac{8\mathcal{N}\left(\epsilon, \widetilde{\mathcal{G}}, \|\cdot\|_{L_2(\mu)}\right)}{\delta}} + \frac{2\left(\mathcal{U}_\infty^\tau\bar{V} + \lambda\|\mathbb{D}(\tau(s,a))\|_{L_\infty}^{\text{UB}}\right)\ln\frac{8\mathcal{N}\left(\epsilon, \widetilde{\mathcal{G}}, \|\cdot\|_{L_2(\mu)}\right)}{\delta}}{3n}$$

$$\leq \left(\mathcal{U}_2^\tau\bar{V} + \lambda\|\mathbb{D}(\tau(s,a))\|_{L_2(\mu)}^{\text{UB}}\right)\sqrt{\frac{2\ln\frac{8\mathcal{N}(\epsilon, \widetilde{\mathcal{G}}, \|\cdot\|_{L_2(\mu)})}{\delta}}{n}} + \frac{2\left(\mathcal{U}_\infty^\tau\bar{V} + \lambda\|\mathbb{D}(\tau(s,a))\|_{L_\infty}^{\text{UB}}\right)\ln\frac{8\mathcal{N}(\epsilon, \widetilde{\mathcal{G}}, \|\cdot\|_{L_2(\mu)})}{\delta}}{3n}.$$

$$\tag{E.15}$$

We set $\varepsilon = \mathcal{O}(\frac{1}{\sqrt{n}})$, and combine with the upper bound for covering number in (E.14) and $\|\mathbb{D}(\tau(s,a))\|_{L_2(\mu)}^{\text{UB}} < L\mathcal{U}_2^\tau$, it follows similar arguments in the proof of Lemma E.2, by some algera, we conclude that

$$
\begin{aligned}
&\left| \mathbb{E}_\mu \Big[ \tau(s,a)\,(r + \gamma q(s',\pi) - q(s,a)) - \lambda \mathbb{D}(\tau(s,a)) \Big] \right. \\
&\quad \left. - \mathbb{P}_n \Big( \tau(s_i,a_i)\,(r_i + \gamma q(s'_i,\pi) - q(s_i,a_i)) - \lambda \mathbb{D}(\tau(s_i,a_i)) \Big) \right| \\
&\lesssim \big(\mathcal{U}_2^\tau \bar{V} + \lambda \|\mathbb{D}(\tau(s,a))\|_{L_2(\mu)}^{\text{UB}}\big) \sqrt{\frac{2\ln \frac{(e^D \max\{D_\Omega, D_\mathcal{Q}, D_\Pi\}+1)^3 (L\mathcal{U}_2^\tau)^{2D}}{\delta}}{n}} \\
&\quad + \frac{2\big(\mathcal{U}_\infty^\tau \bar{V} + \lambda \|\mathbb{D}(\tau(s,a))\|_{L_\infty}^{\text{UB}}\big) \ln \frac{(e^D \max\{D_\Omega, D_\mathcal{Q}, D_\Pi\}+1)^3 (L\mathcal{U}_2^\tau)^{2D}}{\delta}}{3n}.
\end{aligned}
$$

This completes the proof. $\qquad\square$

## E.8 Proof of Lemma E.8

**Lemma E.8.** *Given an offline data $\mathcal{D}_{1:n} = \{s_i, a_i, r_i, s'_i\}_{i=1}^n$, for any $\tau \in \Omega$,*

$$
\begin{aligned}
&|\mathbb{P}_n \mathbb{D}(\tau(s_i,a_i)) - \mathbb{E}_\mu[\mathbb{D}(\tau(s,a))]| \\
&\leq \|\mathbb{D}(\tau(s,a))\|_{L_2(\mu)}^{\text{UB}} \sqrt{\frac{2\ln \frac{(e^{D_\Omega}(D_\Omega+1))(\{1\vee L\}\mathcal{U}_2^\tau)^{2D_\Omega}}{\delta}}{n}} + \frac{2\|\mathbb{D}(\tau(s,a))\|_{L_\infty}^{UB} \ln \frac{(e^{D_\Omega}(D_\Omega+1))(\{1\vee L\}\mathcal{U}_2^\tau)^{2D_\Omega}}{\delta}}{3n}.
\end{aligned}
$$

*holds w.p. $\geq 1 - \delta$.*

*Proof.* We equip the space $\mathcal{G}^\mathbb{D}$ such that $g(s,a) = \mathbb{D}(\tau(s,a))$ for any $g \in \mathcal{G}^\mathbb{D}$ with the $L_2$ weighted metric. To apply empirical Bernstein inequality, we study the boundedness conditions: $\sup_{\tau \in \Omega} \|\tau(s,a)\|_{L_2(\mu)} \leq \mathcal{U}_2^\tau, \sup_{\tau \in \Omega} \|\tau(s,a)\|_{L_\infty} \leq \mathcal{U}_\infty^\tau, \sup_{\tau \in \Omega} \|\mathbb{D}(\tau(s,a))\|_{L_2(\mu)} \in [0, \|\mathbb{D}(\tau(s,a))\|_{L_2(\mu)}^{\text{UB}}], \sup_{\tau \in \Omega} \|\mathbb{D}(\tau(s,a))\|_{L_\infty} \in [0, \|\mathbb{D}(\tau(s,a))\|_{L_\infty}^{\text{UB}}]$. By some calculation, the $L_2(\mu)$-distance in $\tilde{\mathcal{G}}$ can be bounded. For $g_1, g_2 \in \mathcal{G}^\mathbb{D}$ corresponding to $\tau_1$ and $\tau_2$, respectively, we have

$$
\|g_1(s,a) - g_2(s,a)\|_{L_2(\mu)} = \left| -\mathbb{D}(\tau_1(s,a)) + \mathbb{D}(\tau_2(s,a)) \right| \leq \lambda L \|\tau_1(s,a) - \tau_2(s,a)\|_{L_2(\mu)} \tag{E.16}
$$

where the last inequality comes from the $\mathbb{D}$ is $M$-strongly convex function and thus locally Lipschitz with a Lipschitz constant $L \leq \infty$.

$$
\mathcal{N}(4C^\star \varepsilon, \mathcal{G}^\mathbb{D}, \|\cdot\|_{L_2(\mu)}) \leq \mathcal{N}(\varepsilon, \Omega, \|\cdot\|_{L_2(\mu)})
$$

where $C^\star := \mathcal{U}_2^\tau L$. Apply the generalized version of Corollary 2 in [16], which implies

$$
\mathcal{N}\left(\epsilon, \mathcal{G}^\mathbb{D}, \|\cdot\|_{L_2(\mu)}\right) \leq e\,(D_\Omega + 1) \left(\frac{4e(C^\star)^2}{\epsilon}\right)^{D_\Omega}, \tag{E.17}
$$

By empirical Bernstein inequality and a union bound, we have that with probability at least $1 - \delta$, following proof of Lemma E.2, by some algebra, we have

$$
\begin{aligned}
&|\mathbb{P}_n \mathbb{D}(\tau(s_i,a_i)) - \mathbb{E}_\mu[\mathbb{D}(\tau(s,a))]| \\
&\leq \|\mathbb{D}(\tau(s,a))\|_{L_2(\mu)}^{\text{UB}} \sqrt{\frac{2\ln \frac{(e^{D_\Omega}(D_\Omega+1))(\{1\vee L\}\mathcal{U}_2^\tau)^{2D_\Omega}}{\delta}}{n}} + \frac{2\|\mathbb{D}(\tau(s,a))\|_{L_\infty}^{UB} \ln \frac{(e^{D_\Omega}(D_\Omega+1))(\{1\vee L\}\mathcal{U}_2^\tau)^{2D_\Omega}}{\delta}}{3n}.
\end{aligned}
$$

$\qquad\square$

## E.9 Proof of Lemma E.9

**Lemma E.9.** *Given an offline data $\mathcal{D}_{1:n} = \{s_i, a_i, r_i, s_i'\}_{i=1}^n$, for any $\tau$ in a subset of $\Omega$, i.e., $\widetilde{\Omega}$ such that $\sup_{\tau \in \widetilde{\Omega}} \|\tau(s,a)\|_{L_2(\mu)} \leq C$ for some constant $C \geq 1$, then it suffices to ensure*

$$\sum_{i=1}^n \mathbb{D}(\tau(s_i,a_i)) \leq \frac{M(C^2-1)}{2} + \|\mathbb{D}(\tau(s,a))\|_{L_2(\mu)}^{UB} \sqrt{\frac{2\ln\frac{\text{Vol}(\mathcal{G}^{\mathbb{D}})}{\delta}}{n}} + \frac{\|\mathbb{D}(\tau(s,a))\|_{\infty}^{UB}\ln\frac{\text{Vol}(\mathcal{G}^{\mathbb{D}})}{\delta}}{3n},$$

*where $\text{Vol}(\mathcal{G}^{\mathbb{D}}) = (e^{D_\Omega}(D_\Omega + 1))(\{1 \vee L\}\mathcal{U}_2^\tau)^{2D_\Omega}$.*

*Proof.* In this proof, we first convert the upper bound from $\sum_{i=1}^n \mathbb{D}(\tau(s_i,a_i))$ to $\mathbb{E}_\mu[\mathbb{D}(\tau(s,a))]$. In the second part, we leverage the strongly-convexity of $\mathbb{D}$ for upper bound $\|\tau(s,a)\|_{L_2(\mu)}$. It follows from Lemma E.8 and apply the norm triangle inequality, we have

$$\mathbb{E}_\mu[\mathbb{D}(\tau(s,a))] \leq \sum_{i=1}^n \mathbb{D}(\tau(s_i,a_i)) - \|\mathbb{D}(\tau(s,a))\|_{L_2(\mu)}^{UB}\sqrt{\frac{2\ln\frac{\text{Vol}(\mathcal{G}^{\mathbb{D}})}{\delta}}{n}} - \frac{\|\mathbb{D}(\tau(s,a))\|_{\infty}^{UB}\ln\frac{\text{Vol}(\mathcal{G}^{\mathbb{D}})}{\delta}}{3n}$$

$$:= \sum_{i=1}^n \mathbb{D}(\tau(s_i,a_i)) - \varepsilon_n^\circ. \tag{E.18}$$

According to the zero value of detection function $\mathbb{D}(\cdot)$ and its non-negative property functions, i.e., $\mathbb{D}(1) = 0$ Then we immediately have

$$\mathbb{E}_\mu\left[\mathbb{D}(\tau(s_i,a_i))\right] \leq \sum_{i=1}^n \mathbb{D}(\tau(s_i,a_i)) - \varepsilon_n \iff \left|\mathbb{E}_\mu\left[\mathbb{D}(\tau(s_i,a_i))\right]\right| \leq \sum_{i=1}^n \mathbb{D}(\tau(s_i,a_i)) - \varepsilon_n^\circ.$$

Furthermore, we have $\left|\mathbb{E}_\mu\left[\mathbb{D}(\tau(s_i,a_i))\right] - \mathbb{E}_\mu\left[\mathbb{D}(\tau_0(s_i,a_i))\right]\right| \leq \sum_{i=1}^n \mathbb{D}(\tau(s_i,a_i)) - \varepsilon_n$, where $\tau_0(\cdot,\cdot) \equiv 1$, such that $\mathbb{E}_\mu\left[\mathbb{D}(\tau_0(s,a))\right] = 0$ Motivated by the Lipschitz continuity of $\mathbb{D}(\cdot)$, we construct two target functions in order to facilitate the proof,

$$\mathbb{D}^\diamond(\tau) := -\mathbb{E}_\mu\left[\mathbb{D}(\tau(s,a))\right]; \widetilde{\mathbb{D}}(\tau) := \mathbb{E}_\mu\left[\frac{M}{2}(\tau(s,a))^2 - \mathbb{D}(\tau(s,a))\right].$$

Since $\mathbb{D}(\tau)$ is $M$-strongly-convex over $\tau$, which implies that $\widetilde{\mathbb{D}}(\tau)$ is concave, which implies that $\widetilde{\mathbb{D}}(\tau)$ is $M$-strongly-concave with respect to $\tau$ and $\sqrt{\mathbb{E}_\mu[(\cdot)^2]}$. Then

$$\mathbb{E}_\mu\left[(\tau_0(s,a) - \tau(s,a))^2\right] \leq \frac{2(\mathbb{D}^\diamond(\tau) - \mathbb{D}^\diamond(\tau_0))}{M}$$

$$\implies \mathbb{E}_\mu\left[(\tau_0(s,a) - \tau(s,a))^2\right] \leq \frac{2(\mathbb{E}_\mu\left[\mathbb{D}(\tau(s,a))\right] - \mathbb{E}_\mu\left[\mathbb{D}(\tau_0(s,a))\right])}{M}. \tag{E.19}$$

According to the definition of $\tau_0$, by some algebra, we have (E.19) is equivalent to

$$\mathbb{E}_\mu\left[(1 - \tau(s,a))^2\right] \leq \frac{2\mathbb{E}_\mu\left[\mathbb{D}(\tau(s,a))\right]}{M} \implies \|\tau(s,a)\|_{L_2(\mu)}^2 \leq \frac{2\mathbb{E}_\mu\left[\mathbb{D}(\tau(s,a))\right] + M}{M}. \tag{E.20}$$

According to (E.18), then we have $\|\tau(s,a)\|_{L_2(\mu)}^2 \leq (2(\sum_{i=1}^n \mathbb{D}(\tau(s_i,a_i)) - \varepsilon_n^\circ) + M)/M$. By some algebra, where we solve for $C = \|\tau(s,a)\|_{L_2(\mu)}^2$, then we conclude that

$$\sum_{i=1}^n \mathbb{D}(\tau(s_i,a_i)) \leq \frac{M(C^2-1)}{2} + \|\mathbb{D}(\tau(s,a))\|_{L_2(\mu)}^{UB}\sqrt{\frac{2\ln\frac{\text{Vol}(\mathcal{G}^{\mathbb{D}})}{\delta}}{n}} + \frac{\|\mathbb{D}(\tau(s,a))\|_{\infty}^{UB}\ln\frac{\text{Vol}(\mathcal{G}^{\mathbb{D}})}{\delta}}{3n}.$$

This completes the proof. $\qquad\square$

## E.10 Proof of Theorem 4.1

*Proof.* In this proof, we aim to bound the regret $J(\pi) - J(\widehat{\pi})$ for $\widehat{\pi}$ is return from (6) in maintext. First, recall that we have a consistent confident set of value estimates as

$$\mathcal{Q}_{\varepsilon_n} = \left\{q \in \mathcal{Q} : \sup_{\tau \in \widetilde{\Omega}_{\tilde{\sigma}_n}} \left|n^{-1}\sum_{i=1}^n \tau(s_i,a_i)(r_i + \gamma q^\pi(s_i',\pi) - q^\pi(s_i,a_i))\right| \leq \varepsilon_n\right\}, \tag{E.21}$$

with the uncertainty control on important-weight class

$$\widetilde{\Omega}_{\widetilde{\sigma}_n} = \left\{ \tau_\circ / \sup_{\tau_\circ \in \Omega} \|\tau_\circ\|_\Omega \text{ for } \tau_\circ \in \Omega : \xi_n(\mathbb{D}, \tau_\circ)) \le \widetilde{\sigma}_n \right\}. \tag{E.22}$$

We can rewrite this confidence set $\mathcal{Q}_{\varepsilon_n}$ as

$$\mathcal{Q}_{\varepsilon_n} = \left\{ \sup_{\tau \in \widetilde{\Omega}_{\widetilde{\sigma}_n}} \left| n^{-1} \sum_{i=1}^n \tau(s_i, a_i)(r_i + \gamma q^\pi(s_i', \pi) - q^\pi(s_i, a_i)) \right| \le \varepsilon_n, \ \forall \ \tau \in \widetilde{\Omega}_{\widetilde{\sigma}_n} \right\}.$$

Now, for any fixed policy $\pi \in \Pi$ and $\varepsilon_n, \widetilde{\sigma}_n$, we define the maximizer and minimizer in $\mathcal{Q}_{\varepsilon_n}$ as $\overline{q^\pi}$ and $\underline{q^\pi}$, i.e., the maximizer $\overline{q^\pi} := \arg\max_{q \in \mathcal{Q}_{\varepsilon_n}} q(s^0, \pi)$ and the minimizer $\underline{q^\pi} := \arg\min_{q \in \mathcal{Q}_{\varepsilon_n}} q(s^0, \pi)$ over the confidence set $\mathcal{Q}_{\varepsilon_n}$, so that the follow inequalites hold, for any $\tau \in \widetilde{\Omega}_{\widetilde{\sigma}_n}$,

$$\left| \frac{1}{n} \sum_{i=1}^n \frac{\tau(s_i, a_i) \left( r_i + \gamma \overline{q^\pi}(s_i', \pi) - \overline{q^\pi}(s_i, a_i) \right)}{1 - \gamma} \right| \le \varepsilon_n \tag{E.23}$$

$$\left| \frac{1}{n} \sum_{i=1}^n \frac{\tau(s_i, a_i) \left( r_i + \gamma \underline{q^\pi}(s_i', \pi) - \underline{q^\pi}(s_i, a_i) \right)}{1 - \gamma} \right| \le \varepsilon_n. \tag{E.24}$$

In addition, it is obvious that, for any $\lambda > 0$ and $\tau \in \widetilde{\Omega}_{\widetilde{\sigma}_n}$ and $q \in \mathcal{Q}_{\varepsilon_n}$,

$$\left| \frac{1}{n} \sum_{i=1}^n \frac{\tau(s_i, a_i) \left( r_i + \gamma q(s_i', \pi) - q(s_i, a_i) \right)}{1 - \gamma} - \lambda \frac{1}{n} \sum_{i=1}^n \frac{\mathbb{D}(\tau(s_i, a_i))}{1 - \gamma} \right|$$

$$\le \left| \frac{1}{n} \sum_{i=1}^n \frac{\tau(s_i, a_i) \left( r_i + \gamma q(s_i', \pi) - q(s_i, a_i) \right)}{1 - \gamma} \right| + \lambda \left| \frac{1}{n} \sum_{i=1}^n \frac{\mathbb{D}(\tau(s_i, a_i))}{1 - \gamma} \right| \le \varepsilon_n + \lambda \widetilde{\sigma}_n. \tag{E.25}$$

where the last inequality comes from the conditions (E.21) and (E.22). According to the definition of the discounted return, $J(\pi) = q^\pi(s^0, \pi)$ for any $\pi \in \Pi$, then we have

$$J(\pi) - J(\widehat{\pi}) = J(\pi) - q^\pi(s^0, \widehat{\pi}) \le J(\pi) - \underline{q^\pi}(s^0, \widehat{\pi}) \le J(\pi) - \underline{q^\pi}(s^0, \pi).$$

where the second equality from $\underline{q^\pi}(s^0, \widehat{\pi})$ is the lower bound of $q^\pi(s^0, \widehat{\pi})$ for $\widehat{\pi} \in \Pi$, and the last inequality comes from $\widehat{\pi}$ is the maximizer with respect to pessimistic value estimate. According the evaluation error Lemma E.1, and note that $\overline{q^\pi}(s^0, \pi)$ is the upper bound of $q^\pi(s^0, \pi)$, thus we have

$$J(\pi) - J(\widehat{\pi})$$
$$= J(\pi) - \underline{q^\pi}(s^0, \pi)$$
$$\le \overline{q^\pi}(s^0, \pi) - \underline{q^\pi}(s^0, \pi)$$
$$= \overline{q^\pi}(s^0, \pi) - J(\pi) + J(\pi) - \underline{q^\pi}(s^0, \pi)$$
$$= \overline{q^\pi}(s^0, \pi) - \left( \underline{q^\pi}(s^0, \pi) + \frac{\mathbb{E}_{d^\pi} \left[ r + \gamma \overline{q^\pi}(s', \pi) - \overline{q^\pi}(s, a) \right]}{1 - \gamma} \right)$$
$$+ \left( \underline{q^\pi}(s^0, \pi) + \frac{\mathbb{E}_{d^\pi} \left[ r + \gamma \underline{q^\pi}(s', \pi) - \underline{q^\pi}(s, a) \right]}{1 - \gamma} \right) - \underline{q^\pi}(s^0, \pi)$$
$$= \frac{\mathbb{E}_{d^\pi} [r(s, a) + \gamma \underline{q^\pi}(s', \pi) - \underline{q^\pi}(s, a)]}{1 - \gamma} - \frac{\mathbb{E}_{d^\pi} [r(s, a) + \gamma \overline{q^\pi}(s', \pi) - \overline{q^\pi}(s, a)]}{1 - \gamma}$$
$$= \frac{\mathbb{E}_{d^\pi} \left[ \left( r(s, a) + \gamma \underline{q^\pi}(s', \pi) - \underline{q^\pi}(s, a) \right) - \left( r(s, a) + \gamma \overline{q^\pi}(s', \pi) - \overline{q^\pi}(s, a) \right) \right]}{1 - \gamma}$$
$$:= \frac{\mathbb{E}_{d^\pi} \left[ \Delta(\overline{q^\pi}, \pi) - \Delta(\underline{q^\pi}, \pi) \right]}{1 - \gamma},$$

where we use the notation $\Delta(q, \pi) = q(s, a) - r(s, a) - \gamma q(s', \pi)$. Based on this, it is sufficient to bound the $\frac{\mathbb{E}_{d^\pi} \left[ \Delta(\overline{q^\pi}, \pi) - \Delta(\underline{q^\pi}, \pi) \right]}{1 - \gamma}$ in order to bound the regret $J(\pi) - J(\widehat{\pi})$. To proceed the proof,

we define admissible implicit exploratory distribution as $\rho$ that satisfies the condition on uncertainty control (7) in maintext, i.e., $\left\|\frac{\rho(s,a)}{\mu(s,a)}\right\|_{L_2(\mu)} := \mathcal{U}_2^\star \leq \sup_{\tau \in \widetilde{\Omega}_{\widetilde{\sigma}_n}} \|\tau(s,a)\|_{L_2(\mu)} := \mathcal{U}_2^\star$. With this implicit exploratory distribution, we can decompose the regret error over $\rho$ as,

$$
\frac{\mathbb{E}_{d^\pi}\left[\Delta(\overline{q^\pi},\pi) - \Delta(\underline{q}^\pi,\pi)\right]}{1-\gamma} = \underbrace{\frac{\mathbb{E}_\mu\left[\left(\frac{\rho(s,a)}{\mu(s,a)} - \tau_{\rho/\mu}^{\widetilde{\Omega}_{\widetilde{\sigma}_n}}(s,a)\right)\left(\Delta(\overline{q^\pi},\pi) - \Delta(\underline{q}^\pi,\pi)\right)\right]}{1-\gamma}}_{\text{err}_1}
$$

$$
+ \underbrace{\frac{\mathbb{E}_\mu\left[\tau_{\rho/\mu}^{\widetilde{\Omega}_{\widetilde{\sigma}_n}}(s,a)\left(\Delta(\overline{q^\pi},\pi) - \Delta(\underline{q}^\pi,\pi)\right)\right]}{1-\gamma}}_{\text{err}_2} + \underbrace{\frac{\mathbb{E}_{d^\pi}\left[\Delta(\overline{q^\pi},\pi) - \Delta(\underline{q}^\pi,\pi)\right] - \mathbb{E}_\rho\left[\Delta(\overline{q^\pi},\pi) - \Delta(\underline{q}^\pi,\pi)\right]}{1-\gamma}}_{\text{err}_3}.
$$

$$\tag{E.26}$$

where $\tau_{\rho/\mu}^{\widetilde{\Omega}_{\widetilde{\sigma}_n}}$ is the importance-weight estimator that $\tau_{\rho/\mu}^{\widetilde{\Omega}_{\widetilde{\sigma}_n}} \in \widetilde{\Omega}_{\widetilde{\sigma}_n}$, with the definition as

$$
\tau_{\rho/\mu}^{\widetilde{\Omega}_{\widetilde{\sigma}_n}}(s,a) := \underset{\tau \in \text{lr-hull}(\widetilde{\Omega}_{\widetilde{\sigma}_n})}{\arg\min} \ \mathbb{E}_\mu\left[\left(\frac{\rho(s,a)}{\mu(s,a)} - \tau(s,a)\right)\left(\Delta(\overline{q^\pi},\pi) - \Delta(\underline{q}^\pi,\pi)\right)\right],
$$

where lr-hull$(\widetilde{\Omega}_{\widetilde{\sigma}_n})$ is the linear hull of the function class $\widetilde{\Omega}_{\widetilde{\sigma}_n}$. Note that, we use the linear hull to enhance the expressivity of the function class over $\tau$ and more robustness to the function approximation error. With the above error decomposition, we bound the major three terms err$_1$, err$_2$ and err$_3$ subsequently.

**Bounding err$_1$.** Intuitively, the term err$_1$ is introduced by the function approximation error. Due to the construction of uncertainty control class $\widetilde{\Omega}_{\widetilde{\sigma}_n}$, the function approximation is well-controlled, almost cannot be detected under small importance-weight class, i.e., $\widetilde{\sigma}_n$ is small. We explain this in the following. According to Cauchy–Schwarz inequality,

$$
\begin{aligned}
\text{err}_1 &= \mathbb{E}_\mu\left[\left(\frac{\rho(s,a)}{\mu(s,a)} - \tau_{\rho/\mu}^{\widetilde{\Omega}_{\widetilde{\sigma}_n}}(s,a)\right)\left[\left(\Delta(\overline{q^\pi},\pi) - \Delta(\underline{q}^\pi,\pi)\right)\right]\right] \\
&\leq \mathbb{E}_\mu\left[\left|\frac{\rho(s,a)}{\mu(s,a)} - \tau_{\rho/\mu}^{\widetilde{\Omega}_{\widetilde{\sigma}_n}}(s,a)\right|\left|\left(\Delta(\overline{q^\pi},\pi) - \Delta(\underline{q}^\pi,\pi)\right)\right|\right] \\
&\leq \sqrt{\mathbb{E}_\mu\left[\left(\frac{\rho(s,a)}{\mu(s,a)} - \tau_{\rho/\mu}^{\widetilde{\Omega}_{\widetilde{\sigma}_n}}(s,a)\right)^2\right]} \cdot \sqrt{\mathbb{E}_\mu\left[\left\{\left(\Delta(\overline{q^\pi},\pi) - \Delta(\underline{q}^\pi,\pi)\right)\right\}^2\right]} \\
&= \underbrace{\left\|\frac{\rho(s,a)}{\mu(s,a)} - \tau_{\rho/\mu}^{\widetilde{\Omega}_{\widetilde{\sigma}_n}}(s,a)\right\|_{L_2(\mu)}}_{\text{err}_{11}} \cdot \underbrace{\left\|\Delta(\overline{q^\pi},\pi) - \Delta(\underline{q}^\pi,\pi)\right\|_{L_2(\mu)}}_{\text{err}_{12}}.
\end{aligned}
$$

On this point, it suffices to bound the terms err$_{11}$ and err$_{12}$.

**Bounding err$_{11}$.** It follows the definition of $\rho$, it observes that $\left\|\frac{\rho(s,a)}{\mu(s,a)}\right\|_{L_2(\mu)} = \mathcal{U}_2^\star \leq \mathcal{U}_2^\star$. Also, since the importance-weight estimator $\tau_{\rho/\mu}^{\widetilde{\Omega}_{\widetilde{\sigma}_n}}(s,a) \in \widetilde{\Omega}_{\widetilde{\sigma}_n}$, so that $\left\|\tau_{\rho/\mu}^{\widetilde{\Omega}_{\widetilde{\sigma}_n}}(s,a)\right\|_{L_2(\mu)} \leq \mathcal{U}_2^\star$. Due to the non-negativity of $\rho(s,a), \mu(s,a)$ and $\tau_{\rho/\mu}^{\widetilde{\Omega}_{\widetilde{\sigma}_n}}(s,a)$ for any $s,a$ over the support of $\mu$, we can obtain the upper bound

$$
\mathbb{E}_\mu\left[\left(\frac{\rho(s,a)}{\mu(s,a)} - \tau_{\rho/\mu}^{\widetilde{\Omega}_{\widetilde{\sigma}_n}}(s,a)\right)^2\right] \leq \min\left\{\mathbb{E}_\mu\left[\left(\frac{\rho(s,a)}{\mu(s,a)}\right)^2\right], \mathbb{E}_\mu\left[\left(\tau_{\rho/\mu}^{\widetilde{\Omega}_{\widetilde{\sigma}_n}}(s,a)\right)^2\right]\right\}
$$
$$
\leq \min\{(\mathcal{U}_2^\circ)^2, \mathcal{U}_{\text{prime},2}^2\}. \tag{E.27}
$$

**Bounding err$_{12}$.** It follows from the norm triangle inequality, we have

$$
\text{err}_{12} = \|\Delta(\overline{q^\pi},\pi) - \Delta(\underline{q}^\pi,\pi)\|_{L_2(\mu)} \leq \|\Delta(\overline{q^\pi},\pi)\|_{L_2(\mu)} + \|\Delta(\underline{q}^\pi,\pi)\|_{L_2(\mu)}.
$$

According to (E.23) and (E.24), and it follows from Lemma E.2 over the product space $\mathcal{G}_{\varepsilon_n,\widetilde{\sigma}_n} := \widetilde{\Omega}_{\widetilde{\sigma}_n} \times \mathcal{Q}_{\varepsilon_n} \times \Pi$. Accordingly, for $q \in \mathcal{Q}_{\varepsilon_n}$ and $\pi \in \Pi$, we have $\sup_{\tau \in \widetilde{\Omega}_{\widetilde{\sigma}_n}} \mathbb{E}_\mu[\tau(s,a)\Delta(q,\pi)] \lesssim \varepsilon_n^1$ for

$$\varepsilon_n^1 = \mathcal{U}_2^\star \sqrt{\frac{\bar{V}^2 \ln \frac{8\mathcal{N}(\epsilon, \mathcal{G}_{\varepsilon_n,\widetilde{\sigma}_n}, \|\cdot\|_{L_2(\mu)})}{\delta}}{n}} + \frac{\mathcal{U}_\infty^\star \bar{V} \ln \frac{8\mathcal{N}(\epsilon, \mathcal{G}_{\varepsilon_n,\widetilde{\sigma}_n}, \|\cdot\|_{L_2(\mu)})}{\delta}}{3n} + \varepsilon_n,$$

The above inequality also holds for $\overline{q^\pi}$ and $\underline{q^\pi}$. Then, as $\|\tau(s,a)\|_{L_2(\mu)} \leq \mathcal{U}_{\text{prime},2}$ for $\tau \in \widetilde{\Omega}_{\widetilde{\sigma}_n}$, it follows from Lemma E.3, we have $\|\Delta(\overline{q^\pi},\pi)\|_{L_2(\mu)} + \|\Delta(\underline{q^\pi},\pi)\|_{L_2(\mu)} \leq 2\varepsilon_n^1/\mathcal{U}_2^\star$, which implies

$$\mathcal{U}_2^\star \text{err}_{12} \lesssim \varepsilon_n^1. \tag{E.28}$$

Combine with the bounds in (E.28) and (E.27), we conclude the upper bound for $\text{err}_1$:

$$\text{err}_1 \leq \frac{1}{(1-\gamma)} \left\{ \mathcal{U}_{\star,2} \sqrt{\frac{\bar{V}^2 \ln \frac{8\mathcal{N}(\epsilon, \mathcal{G}_{\varepsilon_n,\widetilde{\sigma}_n}, \|\cdot\|_{L_2(\mu)})}{\delta}}{n}} + \frac{\mathcal{U}_\infty^\star \bar{V} \ln \frac{8\mathcal{N}(\epsilon, \mathcal{G}_{\varepsilon_n,\widetilde{\sigma}_n}, \|\cdot\|_{L_2(\mu)})}{\delta}}{3n} + \varepsilon_n \right\}. \tag{E.29}$$

**Bounding $\text{err}_2$.** It first observes that $\tau_{\rho/\mu}^{\widetilde{\Omega}_{\widetilde{\sigma}_n}} \in \text{lr-hull}(\widetilde{\Omega}_{\widetilde{\sigma}_n})$, and $(1-\gamma)\text{err}_2 \leq \sup_{\tau \in \text{lr-hull}(\widetilde{\Omega}_{\widetilde{\sigma}_n})} \mathbb{E}_\mu \left[ \tau(s,a) \left( \Delta(\overline{q^\pi},\pi) - \Delta(\underline{q^\pi},\pi) \right) \right]$. Before we proceed to bound, we first shows the equivalence between the $\tau \in \Omega$ and $\tau \in \text{lr-hull}(\Omega)$ when measuring the statistical complexity for any linear functional $h_{\text{linear}}(\cdot)$ with respect to $\tau$. That is $\sup_{\tau \in \text{lr-hull}(\Omega)} |h_{\text{linear}}(\cdot)| = \sup_{\tau \in \Omega} |h_{\text{linear}}(\cdot)|$. Let's consider any $\tau^\dagger \in \text{lr-hull}(\Omega)$, i.e., $\tau^\dagger = \sum_i \beta_i \tau_i$, where $\tau_i \in \Omega$ for any $i$ and $\sum_i |\beta_i| = 1$. For any $h_{\text{linear}}(\cdot)$ and any $\tau^\dagger \in \text{lr-hull}(\Omega)$, we have that

$$|h_{\text{linear}}(\tau^\dagger)| = \left| h\left( \sum_i \beta_i \tau_i \right) \right| = \left| \sum_i \beta_i h(\tau_i) \right| \leq \sum_i |\beta_i| |h(\tau_i)| \leq \sup_{\tau \in \Omega} |h(\tau)|. \tag{E.30}$$

As (E.30) holds for any $\tau^\dagger \in \text{lr-hull}(\Omega)$. Take maximum over $\tau^\dagger \in \text{lr-hull}(\Omega)$ on the LHS, we have

$$\sup_{\tau^\dagger \in \text{lr-hull}(\Omega)} \left| h_{\text{linear}}(\tau^\dagger) \right| \leq \sup_{\tau \in \Omega} |h(\tau)|. \tag{E.31}$$

On the other side, as $\Omega \subset \text{lr-hull}(\Omega)$, it is easy to observe that

$$\sup_{\tau^\dagger \in \text{lr-hull}(\Omega)} \left| h_{\text{linear}}(\tau^\dagger) \right| \geq \sup_{\tau \in \Omega} |h_{\text{linear}}(\tau)|. \tag{E.32}$$

Combine (E.31) and (E.32), we conclude for $h_{\text{linear}}(\cdot)$, $\sup_{\tau^\dagger \in \text{lr-hull}(\Omega)} \left| h_{\text{linear}}(\tau^\dagger) \right| = \sup_{\tau \in \Omega} |h_{\text{linear}}(\tau)|$. Note that $\frac{1}{1-\gamma} \mathbb{E}_\mu \left[ \tau(s,a) \left[ \Delta(\underline{q^\pi},\pi) - \Delta(\overline{q^\pi},\pi) \right] \right]$ is linear in $\tau$ due to $\tau$ is the weights over average Bellman error $\Delta(\cdot)$ which enjoys the linearity, and belongs to $h_{\text{linear}}(\cdot)$, therefore the above derivation can be applied. According to the equivalence between $\Omega$ and $\text{lr-hull}(\Omega)$, to quantify the statistical error is sufficient to bound

$$\sup_{\tau \in \text{lr-hull}(\widetilde{\Omega}_{\widetilde{\sigma}_n})} \mathbb{E}_\mu \left[ \tau(s,a)\Delta(\overline{q^\pi},\pi) \right] - \inf_{\tau \in \text{lr-hull}(\widetilde{\Omega}_{\widetilde{\sigma}_n})} \mathbb{E}_\mu \left[ \tau(s,a)\Delta(\underline{q^\pi},\pi) \right]$$

$$= \underbrace{\sup_{\tau \in \widetilde{\Omega}_{\widetilde{\sigma}_n}} \mathbb{E}_\mu \left[ \tau(s,a)\Delta(\overline{q^\pi},\pi) \right]}_{\text{err}_{21}} - \underbrace{\inf_{\tau \in \widetilde{\Omega}_{\widetilde{\sigma}_n}} \mathbb{E}_\mu \left[ \tau(s,a)\Delta(\underline{q^\pi},\pi) \right]}_{\text{err}_{22}}.$$

Let $\tau_{\max}$ and $\tau_{\min}$ be the optimizer of $\text{err}_{21}$ and $\text{err}_{22}$, respectively. And we define an auxiliary objective function, which leverages the convexity of $\mathbb{D}(\cdot)$ over $\tau$. That is,

$$\mathbb{P}_n \mathcal{L}_\mathbb{D}(\tau, q) := \frac{1}{n} \sum_{i=1}^n \left[ \tau(s_i, a_i) \left( r_i + \gamma q(s_i', \pi) - q(s_i, a_i) \right) - \lambda \mathbb{D}(\tau(s_i, a_i)) \right],$$

for $\tau \in \widetilde{\Omega}_{\widetilde{\sigma}_n}$, $q \in \mathcal{Q}_{\varepsilon_n}$. We make the decomposition:

$$\mathrm{err}_{21} - \mathrm{err}_{22} = \underbrace{\mathbb{E}_\mu\left[\tau_{\max}(s,a)\Delta(\overline{q^\pi},\pi)\right] - \mathbb{E}_\mu\left[\tau_{\min}(s,a)\Delta(\overline{q^\pi},\pi)\right]}_{\mathrm{err}_{21}}$$
$$+ \underbrace{\mathbb{E}_\mu\left[\tau_{\min}(s,a)\Delta(\overline{q^\pi},\pi)\right] - \mathbb{E}_\mu\left[\tau_{\min}(s,a)\Delta(\underline{q^\pi},\pi)\right]}_{\mathrm{err}_{22}}.$$

And therefore, to bound $\mathrm{err}_{21} - \mathrm{err}_{22}$, we are sufficient to bound $\mathrm{err}_{21}$ and $\mathrm{err}_{22}$.

**Bounding $\mathrm{err}_{21}$.** It follows from Cauchy-Schwarz inequality, then we have

$$\mathrm{err}_{21} \leq \sqrt{\underbrace{\mathbb{E}_\mu\left[(\tau_{\max}(s,a) - \tau_{\min}(s,a))^2\right]}_{\mathrm{err}_{211}}\underbrace{\mathbb{E}_\mu\left[(\Delta(\overline{q^\pi},\pi))^2\right]}_{\mathrm{err}_{212}}}. \tag{E.33}$$

**Bounding $\mathrm{err}_{211}$.** Recall $\mathcal{L}_{\mathbb{D}}(\tau,q) := \mathbb{E}_\mu\left[\tau(s,a)\left(r + \gamma q\left(s',\pi\right) - q(s,a)\right) - \lambda\mathbb{D}(\tau(s,a))\right]$. For fixed $q \in \mathcal{Q}_{\varepsilon_n}$ and $\tau \in \widetilde{\Omega}_{\widetilde{\sigma}_n}$, we show $\mathcal{L}_{\mathbb{D}}(\tau,q)$, is $\lambda M$-strongly con-cave with respect to $\tau$ and $\mathbb{E}_\mu[(\cdot)^2]$. Let us consider an counterpart for $\mathcal{L}_{\mathbb{D}}(\tau,q)$, i.e., $\mathcal{L}_{\mathbb{D}}^\circ(\tau,q) := \mathcal{L}_{\mathbb{D}}(\tau,q) + \frac{\lambda M}{2}\mathbb{E}_\mu\left[(\tau(s,a))^2\right]$, so that we have $\mathcal{L}_{\mathbb{D}}^\circ(\tau,q) = \mathbb{E}_\mu\left[\tau(s,a)\left(r + \gamma q\left(s',\pi\right) - q(s,a)\right) - \lambda[\mathbb{D}(\tau(s,a)) - \frac{M}{2}(\tau(s,a))^2]\right]$. Since $\mathbb{D}(\cdot)$ is $M$-strongly convex with respect to $\tau$, so $\mathcal{L}_{\mathbb{D}}^\circ(\tau,q)$ is concave, which implies that $\mathcal{L}_{\mathbb{D}}(\tau,q)$ is $\lambda M$-strongly-concave with respect to $\tau$ and $\mathbb{E}_\mu[(\cdot)^2]$. It follows from the strongly-concavity, and plug-in $\tau_{\max}$, $\tau_{\min}$ and $\overline{q^\pi}$,

$$\mathbb{E}_\mu\left[(\tau_{\max}(s,a) - \tau_{\min}(s,a))^2\right] \leq \frac{2(\mathcal{L}_{\mathbb{D}}(\tau_{\max},\overline{q^\pi}) - \mathcal{L}_{\mathbb{D}}(\tau_{\min}^\circ,\overline{q^\pi}))}{\lambda M}. \tag{E.34}$$

This implies it is sufficient to bound $\mathcal{L}_{\mathbb{D}}(\tau_{\max},\overline{q^\pi}) - \mathcal{L}_{\mathbb{D}}(\tau_{\min},\overline{q^\pi})$ for bound $\mathrm{err}_{211}$.

$$\mathcal{L}_{\mathbb{D}}(\tau_{\max},\overline{q^\pi}) - \mathcal{L}_{\mathbb{D}}(\tau_{\min},\overline{q^\pi}) = \underbrace{\mathcal{L}_{\mathbb{D}}(\tau_{\max},\overline{q^\pi}) - \mathbb{P}_n\mathcal{L}_{\mathbb{D}}(\tau_{\max},\overline{q^\pi})}_{\mathrm{err}_{2111}} + \underbrace{\mathbb{P}_n\mathcal{L}_{\mathbb{D}}(\tau_{\max},\overline{q^\pi}) - \mathbb{P}_n\mathcal{L}_{\mathbb{D}}(\tau_{\min},\overline{q^\pi})}_{\mathrm{err}_{2112}}$$
$$+ \underbrace{\mathbb{P}_n\mathcal{L}_{\mathbb{D}}(\tau_{\min},\overline{q^\pi}) - \mathcal{L}_{\mathbb{D}}(\tau_{\min},\overline{q^\pi})}_{\mathrm{err}_{2113}}.$$

It follows from Lemma E.7, with the defintion on the boundedness of $\mathbb{D}$ class terms, i.e., $\|\mathbb{D}(\tau(s,a))\|_{L_2(\mu)}^{\mathrm{prime}} = \sup_{\tau \in \widetilde{\Omega}_{\widetilde{\sigma}_n}}\|\mathbb{D}(\tau(s,a))\|_{L_2(\mu)}$ and $\|\mathbb{D}(\tau(s,a))\|_{L_\infty}^{\mathrm{prime}} = \sup_{\tau \in \widetilde{\Omega}_{\widetilde{\sigma}_n}}\|\mathbb{D}(\tau(s,a))\|_{L_\infty}$, the terms $\mathrm{err}_{2111}$ and $\mathrm{err}_{2112}$ is upper bounded by

$$\varepsilon_2 := \left(\mathcal{U}_2^\star\bar{V} + \lambda\|\mathbb{D}(\tau(s,a))\|_{L_2(\mu)}^{\mathrm{prime}}\right)\sqrt{\frac{2\ln\frac{(e^D\max\{D_\Omega,D_\mathcal{Q},D_\Pi\}+1)^3(L\mathcal{U}_2^\tau)^{2D}}{\delta}}{n}}$$
$$+ \frac{2\left(\mathcal{U}_{\mathrm{prime},\infty}\bar{V} + \lambda\|\mathbb{D}(\tau(s,a))\|_\infty^{\mathrm{prime}}\right)\ln\frac{(e^D\max\{D_\Omega,D_\mathcal{Q},D_\Pi\}+1)^3(L\mathcal{U}_2^\tau)^{2D}}{\delta}}{3n},$$

According to (E.25), as $\overline{q^\pi} \in \mathcal{Q}_{\varepsilon_n}$ and $\tau \in \widetilde{\Omega}_{\widetilde{\sigma}_n}$, thus $\mathrm{err}_{2112} \leq 2(\varepsilon_n + \lambda\widetilde{\sigma}_n)$. Therefore, combine with $\mathrm{err}_{2112}$, we conclude that

$$\mathbb{E}_\mu\left[(\tau_{\max}(s,a) - \tau_{\min}(s,a))^2\right] \leq \frac{2}{\lambda M}\left(\varepsilon_2 + \varepsilon_n + \lambda\widetilde{\sigma}_n\right) := \frac{2\varepsilon_3}{\lambda M}. \tag{E.35}$$

**Bounding $\mathrm{err}_{212}$.** First, it observes that the density ratio class $\widetilde{\Omega}_{\widetilde{\sigma}_n}$ is upper bounded with respect to weighted $L_2(\mu)$ norm, i.e., $\sup_{\tau \in \widetilde{\Omega}_{\widetilde{\sigma}_n}}\|\tau(s,a)\|_{L_2(\mu)} \leq \mathcal{U}_2^\star$. It follows from Lemma E.3, we have $\mathcal{U}_2^\star\|\Delta(\overline{q^\pi},\pi)\|_{L_2(\mu)} = \sup_{\tau \in \widetilde{\Omega}_{\widetilde{\sigma}_n}}|\mathbb{E}_\mu\left[\tau(s,a)\Delta(\overline{q^\pi},\pi)\right]| \leq \varepsilon_n$, where the last inequality comes from $\overline{q^\pi} \in \mathcal{Q}_{\varepsilon_n}$ and (E.21). Therefore, we conclude that

$$\sqrt{\mathrm{err}_{212}} \leq \frac{1}{\mathcal{U}_2^\star}\varepsilon_n. \tag{E.36}$$

It combines with (E.35), (E.33) and (E.36), we have

$$\mathrm{err}_{21} \leq \frac{\varepsilon_1}{\mathcal{U}_2^\star}\sqrt{\frac{2\varepsilon_3}{\lambda M}}. \tag{E.37}$$

**Bounding err$_{22}$.** We first observe that $\mathbb{E}_\mu\left[\tau_{\min}(s,a)\Delta(\overline{q^\pi},\pi)\right] - \mathbb{E}_\mu\left[\tau_{\min}(s,a)\Delta(\underline{q^\pi},\pi)\right] = \mathbb{E}_\mu\left[\tau_{\min}(s,a)\left(\Delta(\overline{q^\pi},\pi) - \Delta(\underline{q^\pi},\pi)\right)\right]$. Then it follows from Lemma E.4 and I(E.21), with the norm triangle inequality, we can conclude that

$$\text{err}_{22} \le \varepsilon_n^1. \tag{E.38}$$

Now we summarize the bound for err$_2$ throught combining the upper bounds on (E.37) and (E.38), we have

$$\text{err}_2 \le \frac{1}{1-\gamma}\left(\mathcal{U}_2^\star\sqrt{\frac{32\bar{V}^2\ln\frac{8\mathcal{N}\left(\epsilon,\mathcal{G}_{\varepsilon_n},\tilde{\sigma}_n,\|\cdot\|_{L_2(\mu)}\right)}{\delta}}{n}} + \frac{8\mathcal{U}_\infty^\star\bar{V}\ln\frac{8\mathcal{N}\left(\epsilon,\mathcal{G}_{\varepsilon_n},\tilde{\sigma}_n,\|\cdot\|_{L_2(\mu)}\right)}{\delta}}{3n}\right.$$
$$\left. + \frac{\varepsilon_n}{\mathcal{U}_2^\star}\sqrt{\frac{2\varepsilon_3}{\lambda M}} + 2\varepsilon_n\right). \tag{E.39}$$

**Bounding err$_3$.** In the following, we proceed to bound the term err$_3$. First, we make the decomposition as follows:

$$(1-\gamma)\text{err}_3 = \sum_{a\in\mathcal{A},s\in\mathcal{S}}[d_\pi(s,a) - \rho(s,a)][\Delta(\overline{q^\pi},\pi) - \Delta(\underline{q^\pi},\pi)]$$

$$= \sum_{a\in\mathcal{A},s\in\mathcal{S}}\mathbb{1}_{\{d_\pi(s,a)-\rho(s,a)\ge 0\}}[d_\pi(s,a) - \rho(s,a)][\Delta(\overline{q^\pi},\pi) - \Delta(\underline{q^\pi},\pi)]$$

$$+ \sum_{a\in\mathcal{A},s\in\mathcal{S}}\mathbb{1}_{\{d_\pi(s,a)-\rho(s,a)< 0\}}[d_\pi(s,a) - \rho(s,a)][\Delta(\overline{q^\pi},\pi) - \Delta(\underline{q^\pi},\pi)]$$

$$\le \underbrace{\sum_{a\in\mathcal{A},s\in\mathcal{S}}\mathbb{1}_{\{d_\pi(s,a)-\rho(s,a)< 0\}}[\rho(s,a) - d_\pi(s,a)]|\Delta(\overline{q^\pi},\pi) - \Delta(\underline{q^\pi},\pi)|}_{\text{err}_{31}}$$

$$+ \underbrace{\sum_{a\in\mathcal{A},s\in\mathcal{S}}\mathbb{1}_{\{d_\pi(s,a)-\rho(s,a)\ge 0\}}[d_\pi(s,a) - \rho(s,a)][\Delta(\overline{q^\pi},\pi) - \Delta(\underline{q^\pi},\pi)]}_{\text{err}_{32}}.$$

Next, we bound the terms err$_{31}$ and err$_{32}$ separately.

**Bounding err$_{31}$.** We first observe that $\mathbb{1}_{\{d_\pi(s,a)-\rho(s,a)<0\}}[\rho(s,a) - d_\pi(s,a)] = (\rho(s,a) - d_\pi(s,a))^+$, and we have

$$\text{err}_{31} = \sum_{a\in\mathcal{A},s\in\mathcal{S}}\mathbb{1}_{\{d_\pi(s,a)-\rho(s,a)<0\}}[\rho(s,a) - d_\pi(s,a)][\Delta(\overline{q^\pi},\pi) - \Delta(\underline{q^\pi},\pi)]$$

$$= \sum_{a\in\mathcal{A},s\in\mathcal{S}}(\rho(s,a) - d_\pi(s,a))^+[\Delta(\overline{q^\pi},\pi) - \Delta(\underline{q^\pi},\pi)]$$

$$= \mathbb{E}_{(\rho(s,a)-d_\pi(s,a))^+}[\Delta(\overline{q^\pi},\pi) - \Delta(\underline{q^\pi},\pi)].$$

By the condition that $\|\tau(s,a)\|_{L_2(\mu)} \le \mathcal{U}_2^\star$, and it follows from Lemma E.3,

$$\sqrt{\mathbb{E}_\mu[(\Delta(\overline{q^\pi},\pi))^2]} \text{ or } \sqrt{\mathbb{E}_\mu[(\Delta(\underline{q^\pi},\pi))^2]}$$

$$\lesssim \frac{\mathcal{U}_2^\star\sqrt{\frac{\bar{V}^2\ln\frac{8\mathcal{N}\left(\epsilon,\mathcal{G}_{\varepsilon_n},\tilde{\sigma}_n,\|\cdot\|_{L_2(\mu)}\right)}{\delta}}{n}} + \frac{\mathcal{U}_\infty^\star\bar{V}\ln\frac{8\mathcal{N}\left(\epsilon,\mathcal{G}_{\varepsilon_n},\tilde{\sigma}_n,\|\cdot\|_{L_2(\mu)}\right)}{\delta}}{3n} + \mathcal{U}_2^\star\sqrt{\varepsilon_\mathcal{Q}}}{\mathcal{U}_2^\star}. \tag{E.40}$$

Since $\|\frac{\rho(s,a)}{\mu(s,a)}\|_{L_2(\mu)} \le \mathcal{U}_2^\star$ and $(\rho(s,a) - d_\pi(s,a))^+ \in [0, \rho(s,a)]$ for any $(s,a)$, we have

$$
\begin{aligned}
&\mathbb{E}_{(\rho(s,a) - d_\pi(s,a))^+} |\Delta(\overline{q^\pi}, \pi) - \Delta(\underline{q^\pi}, \pi)| \\
&\le \mathbb{E}_\rho[|\Delta(\overline{q^\pi}, \pi)|] + \mathbb{E}_\rho[|\Delta(\underline{q^\pi}, \pi)|] \\
&= \mathbb{E}_\mu\left[\frac{\rho(s,a)}{\mu(s,a)} |\Delta(\overline{q^\pi}, \pi)|\right] + \mathbb{E}_\mu\left[\frac{\rho(s,a)}{\mu(s,a)} |\Delta(\underline{q^\pi}, \pi)|\right] \\
&\le \|\frac{\rho(s,a)}{\mu(s,a)}\|_{L_2(\mu)} \|\Delta(\overline{q^\pi}, \pi)\|_{L_2(\mu)} + \|\frac{\rho(s,a)}{\mu(s,a)}\|_{L_2(\mu)} \|\Delta(\underline{q^\pi}, \pi)\|_{L_2(\mu)} \\
&\lesssim \mathcal{U}_2^\star \sqrt{\frac{\bar{V}^2 \ln \frac{8\mathcal{N}(\epsilon, \mathcal{G}_{\varepsilon_n, \tilde{\sigma}_n}, \|\cdot\|_{L_2(\mu)})}{\delta}}{n}} + \frac{2\mathcal{U}_\infty^\star \bar{V} \ln \frac{8\mathcal{N}(\epsilon, \mathcal{G}_{\varepsilon_n, \tilde{\sigma}_n}, \|\cdot\|_{L_2(\mu)})}{\delta}}{3n} + \mathcal{U}_2^\star \sqrt{\varepsilon_\mathcal{Q}}. \quad \text{(E.41)}
\end{aligned}
$$

**Bounding err$_{32}$.** We now bound the term err$_{32}$. It observes that

$$
\begin{aligned}
\text{err}_{32} &= \sum_{a \in \mathcal{A}, s \in \mathcal{S}} (d_\pi(s,a) - \rho(s,a))^+ [\Delta(\underline{q^\pi}, \pi) - \Delta(\overline{q^\pi}, \pi)] \\
&= \sum_{a \in \mathcal{A}, s \in \mathcal{S}} (d_\pi(s,a) - \rho(s,a))^+ (\mathbb{I} - \gamma \mathbb{P}^\pi) \Delta_{\overline{q^\pi} - \underline{q^\pi}},
\end{aligned}
$$

where $\Delta_{\overline{q^\pi} - \underline{q^\pi}} = \overline{q^\pi}(s,a) - \underline{q^\pi}(s,a)$. We make the decomposition with respect to $\mu(s,a) > 0$ and $\mu(s,a) = 0$, that is

$$
\begin{aligned}
\text{err}_{32} &= \sum_{a \in \mathcal{A}, s \in \mathcal{S}} \mathbb{1}_{\mu(s,a) > 0} (d_\pi(s,a) - \rho(s,a))^+ (\mathbb{I} - \gamma \mathbb{P}^\pi) \Delta_{\overline{q^\pi} - \underline{q^\pi}} \\
&\quad + \sum_{a \in \mathcal{A}, s \in \mathcal{S}} \mathbb{1}_{\mu(s,a) = 0} (d_\pi(s,a) - \rho(s,a))^+ (\mathbb{I} - \gamma \mathbb{P}^\pi) \Delta_{\overline{q^\pi} - \underline{q^\pi}} \\
&\le \sum_{a \in \mathcal{A}, s \in \mathcal{S}} \mathbb{1}_{\mu(s,a) > 0} (d_\pi(s,a) - \rho(s,a))^+ \left|(\mathbb{I} - \gamma \mathbb{P}^\pi) \Delta_{\overline{q^\pi} - \underline{q^\pi}}\right| \\
&\quad + \sum_{a \in \mathcal{A}, s \in \mathcal{S}} \mathbb{1}_{\mu(s,a) = 0} (d_\pi(s,a) - \rho(s,a))^+ (\mathbb{I} - \gamma \mathbb{P}^\pi) \Delta_{\overline{q^\pi} - \underline{q^\pi}}.
\end{aligned}
$$

It observes that, for state-action pairs $(s,a) \in \mathcal{S} \times \mathcal{A}$ with $\mu(s,a) > 0$,

$$
\begin{aligned}
|(\mathbb{I} - \gamma \mathbb{P}^\pi) \Delta_{\overline{q^\pi} - \underline{q^\pi}}| &= |\gamma \overline{q^\pi}(s', \pi) - \overline{q^\pi}(s,a) - (\gamma \underline{q^\pi}(s', \pi) - \underline{q^\pi}(s,a)| \\
&\le |\gamma(\overline{q^\pi}(s', \pi) - \underline{q^\pi}(s', \pi))| + |\overline{q^\pi}(s,a) - \underline{q^\pi}(s,a)| \\
&\le \sup_{a \in \mathcal{A}, s \in \mathcal{S}, \mu(s,a) > 0} (1 + \gamma) \left|\overline{q^\pi}(s,a) - \underline{q^\pi}(s,a)\right| \lesssim \frac{2\varepsilon_r(1 + \gamma)}{1 - \gamma},
\end{aligned}
$$

where the last inequality comes from Lemma E.6, and $\varepsilon_r$ is defined in Lemma E.6 with some modifications adapting to $\varepsilon_1$ and $\varepsilon_2$ in (E.21) and (E.22), and the function class $\mathcal{G}_{\varepsilon_n, \tilde{\sigma}_n}$. Therefore, we have

$$
\varepsilon_r = \sqrt{\frac{2\bar{V}^2 \ln \frac{8\mathcal{N}(\epsilon, \mathcal{G}_{\varepsilon_n, \tilde{\sigma}_n}, \|\cdot\|_{L_2(\mu)})}{\delta}}{n}} + \frac{2\mathcal{U}_\infty \bar{V} \ln \frac{8\mathcal{N}(\epsilon, \mathcal{G}_{\varepsilon_n, \tilde{\sigma}_n}, \|\cdot\|_{L_2(\mu)})}{\delta}}{3n} + \frac{\varepsilon_n}{\mathcal{U}_2^\star}.
$$

where $\mathcal{U}_\infty \geq 0$. Next, we have that

$$
\begin{aligned}
\text{err}_{32} &\leq \sum_{a \in \mathcal{A}, s \in \mathcal{S}} \mathbb{1}_{\mu(s,a)>0} \left(d_\pi(s,a) - \rho(s,a)\right)^+ \frac{2\varepsilon_r(1+\gamma)}{1-\gamma} \\
&\quad + \sum_{a \in \mathcal{A}, s \in \mathcal{S}} \mathbb{1}_{\mu(s,a)=0} \left(d_\pi(s,a) - \rho(s,a)\right)^+ (\mathbb{I} - \gamma \mathbb{P}^\pi) \Delta_{\overline{q^\pi} - \underline{q^\pi}} \\
&\leq \sum_{a \in \mathcal{A}, s \in \mathcal{S}} \left(d_\pi(s,a) - \rho(s,a)\right)^+ \frac{2\varepsilon_r(1+\gamma)}{1-\gamma} \\
&\quad + \sum_{a \in \mathcal{A}, s \in \mathcal{S}} \mathbb{1}_{\mu(s,a)=0} \left(d_\pi(s,a) - \rho(s,a)\right)^+ (\mathbb{I} - \gamma \mathbb{P}^\pi) \Delta_{\overline{q^\pi} - \underline{q^\pi}} \\
&= \frac{2\varepsilon_r(1+\gamma)}{1-\gamma} \sum_{a \in \mathcal{A}, s \in \mathcal{S}} \left(d_\pi(s,a) - \rho(s,a)\right)^+ \\
&\quad + \sum_{a \in \mathcal{A}, s \in \mathcal{S}} \mathbb{1}_{\mu(s,a)=0} \left[ \left(d_\pi(s,a) - \rho(s,a)\right)^+ (\mathbb{I} - \gamma \mathbb{P}^\pi) \Delta_{\overline{q^\pi} - \underline{q^\pi}} \right] \\
&\lesssim \frac{\varepsilon_r}{1-\gamma} \sum_{a \in \mathcal{A}, s \in \mathcal{S}} \left(d_\pi(s,a) - \rho(s,a)\right)^+ \\
&\quad + \sum_{a \in \mathcal{A}, s \in \mathcal{S}} \mathbb{1}_{\mu(s,a)=0} \left[ \left(d_\pi(s,a) - \rho(s,a)\right)^+ (\mathbb{I} - \gamma \mathbb{P}^\pi) \Delta_{\overline{q^\pi} - \underline{q^\pi}} \right].
\end{aligned}
$$

Optimizing $\text{err}_{32}$ for the set contains $\{\rho : \|\frac{\rho(s,a)}{\mu(s,a)}\|_{L_2(\mu)} \leq \mathcal{U}_2^\star\}$, we obtain the tight bound that

$$
\begin{aligned}
\text{err}_{32} &\lesssim \min_{\left\{\rho : \left\|\frac{\rho(s,a)}{\mu(s,a)}\right\|_{L_2(\mu)} \leq \mathcal{U}_2^\star\right\}} \left\{ \sum_{a \in \mathcal{A}, s \in \mathcal{S}} \mathbb{1}_{\mu(s,a)=0} \left( \left(d_\pi(s,a) - \rho(s,a)\right)^+ (\mathbb{I} - \gamma \mathbb{P}^\pi) \Delta_{\overline{q^\pi} - \underline{q^\pi}} \right) \right. \\
&\quad \left. + \frac{\varepsilon_r}{1-\gamma} \sum_{a \in \mathcal{A}, s \in \mathcal{S}} \left(d_\pi(s,a) - \rho(s,a)\right)^+ \right\}. \quad\quad (\text{E.42})
\end{aligned}
$$

Combine the upper bounds (E.41) and (E.42), and we summarize the upper bound for $\text{err}_3$ as follows:

$$
\begin{aligned}
\text{err}_3 &\lesssim \frac{1}{1-\gamma} \left( \mathcal{U}_2^\star \sqrt{\frac{\bar{V}^2 \ln \frac{8\mathcal{N}\left(\epsilon, \mathcal{G}_{\varepsilon_n, \tilde{\sigma}_n}, \|\cdot\|_{L_2(\mu)}\right)}{\delta}}{n}} + \frac{2\mathcal{U}_\infty^\star \bar{V} \ln \frac{8\mathcal{N}\left(\epsilon, \mathcal{G}_{\varepsilon_n, \tilde{\sigma}_n}, \|\cdot\|_{L_2(\mu)}\right)}{\delta}}{3n} \right. \\
&\quad + \min_{\left\{\rho : \left\|\frac{\rho(s,a)}{\mu(s,a)}\right\|_{L_2(\mu)} \leq \mathcal{U}_2^\star\right\}} \left\{ \sum_{a \in \mathcal{A}, s \in \mathcal{S}} \mathbb{1}_{\mu(s,a)=0} \left( \left(d_\pi(s,a) - \rho(s,a)\right)^+ (\mathbb{I} - \gamma \mathbb{P}^\pi) \Delta_{\overline{q^\pi} - \underline{q^\pi}} \right) \right. \\
&\quad \left. \left. + \frac{\varepsilon_r}{1-\gamma} \sum_{a \in \mathcal{A}, s \in \mathcal{S}} \left(d_\pi(s,a) - \rho(s,a)\right)^+ \right\} + \mathcal{U}_2^\star \sqrt{\varepsilon_\mathcal{Q}} \right). \\
&\quad\quad\quad\quad (\text{E.43})
\end{aligned}
$$

According to the calculation of $\mathcal{N}\left(\epsilon, \mathcal{G}_{\varepsilon_n, \tilde{\sigma}_n}, \|\cdot\|_{L_2(\mu)}\right)$, and use the notation $\text{Vol}(\Theta)$ for the function class complexity, i.e., $\text{Vol}(\Theta) = (e^D \max\{D_\Omega, D_\mathcal{Q}, D_\Pi\} + 1)^3 (\{1 \vee L\}\mathcal{U}_2^\tau)^{2D}$ where $D = D_\Omega + D_\mathcal{Q} + D_\Pi$, we set $\varepsilon_n$ as: $\varepsilon_n = \widetilde{\mathcal{O}}(n^{-1/2}\mathcal{U}_2^\tau(\sqrt{\ln\{\text{Vol}(\Theta)/\delta\}} + \mathcal{U}_\infty^\tau \sqrt{\varepsilon_\mathcal{Q}})$ for ensuring the best approximator for $q^\pi$ is in $\mathcal{Q}_{\varepsilon_n}$. According to Lemma E.9, we set $\tilde{\sigma}_n = \widetilde{\mathcal{O}}(n^{-1/2}\mathcal{U}_2^\star L \sqrt{\ln\{\text{Vol}(\Theta)/\delta\}} + M(\mathcal{U}_2^\tau - 1)^2)$ The set up for $\tilde{\sigma}_n$ is to ensure $\sup_{\tau \in \tilde{\Omega}_{\tilde{\sigma}_n}} \|\tau(s,a)\|_{L_2(\mu)} \leq \mathcal{U}_2^\star$, where $\mathcal{U}_2^\star \in [1, \mathcal{U}_2^\tau)$. And Then it follows from the regret decomposition (E.26), and the upper error bounds for $\text{err}_1, \text{err}_2$ and $\text{err}_3$ in (E.29), (E.39) and (E.43), by some algebra and if we ignore the high-order fast terms, we conclude that, w.p. $\geq 1 - \delta$, Then it follows from the regret decomposition (E.26), and the upper error bounds for $\text{err}_1, \text{err}_2$ and $\text{err}_3$ in (E.29), (E.39) and (E.43), by some algebra, w.p. $\geq 1 - \delta$, we

have

$$J(\pi) - J(\widehat{\pi}) \leq \frac{1}{1-\gamma}\mathcal{O}\Bigg(\mathcal{E}_1^n + \sqrt{\left(1 + \mathcal{U}_\infty^\tau + \frac{\mathcal{U}_\infty^\tau}{M}\right)}\max\{(\varepsilon_\mathcal{Q})^{1/2}, (\varepsilon_\mathcal{Q})^{3/4}\}$$

$$+ \min_{\left\{\rho:\left\|\frac{\rho}{\mu}\right\|_{L_2(\mu)} \leq \mathcal{U}_2^\star\right\}}\left\{\mathbb{E}_{(d_\pi - \rho)^+}\left[\mathbb{1}_{\mu=0}(\mathbb{I} - \gamma\mathbb{P}^\pi)\Delta_{\overline{q^\pi} - \underline{q^\pi}}(s,a) + \mathbb{1}_{\mu>0}\mathcal{E}_2^n\right]\right\}\Bigg),$$

$$\text{(E.44)}$$

where $\mathcal{E}_1^n = \mathcal{U}_2^\star(\bar{V} + L)\sqrt{\ln\{\mathrm{Vol}(\Theta)/\delta\}/(nM)} + \sqrt{\mathcal{U}_2^\tau(\bar{V}^3 + \bar{V}^2 L)/M}(\ln\{\mathrm{Vol}(\Theta)/\delta\}/n)^{\frac{3}{4}} + \mathcal{U}_\infty^\tau(\bar{V} + L)\ln\{\mathrm{Vol}(\Theta)/\delta\}/n$ and $\mathcal{E}_2^n = (1-\gamma)^{-1}((\bar{V} + L)\sqrt{\ln\{\mathrm{Vol}(\Theta)/\delta\}/n} + (\mathcal{U}_\infty^\tau\bar{V}/\mathcal{U}_2^\star)\ln\{\mathrm{Vol}(\Theta)/\delta\}/n)$. Furthermore, if we ignore the high-order fast terms using a big-Oh notation $\widetilde{\mathcal{O}}$, by some algebra, we conclude that

$$J(\pi) - J(\widehat{\pi}) \leq \frac{1}{1-\gamma}\widetilde{\mathcal{O}}\Bigg(\mathcal{U}_2^\star\mathfrak{C}_{\bar{V},L}\sqrt{\frac{\ln\{\mathrm{Vol}(\Theta)/\delta\}}{nM}} + \sqrt{\frac{\mathfrak{C}_{\mathcal{U}_\infty^\tau}}{M}}\max\{(\varepsilon_\mathcal{Q})^{1/2}, (\varepsilon_\mathcal{Q})^{3/4}\}$$ (E.45)

$$+ \min_{\left\{\rho:\left\|\frac{\rho}{\mu}\right\|_{L_2(\mu)} \leq \mathcal{U}_2^\star\right\}}\left\{\mathbb{E}_{(d_\pi - \rho)^+}\left[\mathbb{1}_{\mu=0}(\mathbb{I} - \gamma\mathbb{P}^\pi)\Delta_{\overline{q^\pi} - \underline{q^\pi}}(s,a) + \mathbb{1}_{\mu>0}\mathfrak{C}_{\bar{V},\gamma}\sqrt{\frac{\ln\{\mathrm{Vol}(\Theta)/\delta\}}{n}}\right]\right\}\Bigg),$$

$$\text{(E.46)}$$

where we use $\mathfrak{C}_x$ denote constant terms depending on $x$. This completes the proof. $\qquad\square$

## F   Proof of Corollary 4.1

*Proof.* To complete the proof, it is sufficient to choose a particular $\rho^\diamond$ such that $\left\|\frac{\rho}{\mu}\right\|_{L_2(\mu)} \leq \mathcal{U}_2^\star$ to obtain a regret as the upper bound for the regret in (E.44). For any comparator policy $\pi^\diamond$, since

$$\min_{\left\{\rho:\left\|\frac{\rho}{\mu}\right\|_{L_2(\mu)} \leq \mathcal{U}_2^\star\right\}}\left\{\mathbb{E}_{(d_{\pi^\diamond} - \rho)^+}\left[\mathbb{1}_{\mu=0}(\mathbb{I} - \gamma\mathbb{P}^\pi)\Delta_{\overline{q^\pi} - \underline{q^\pi}}(s,a) + \mathbb{1}_{\mu>0}\mathcal{E}_2^n\right]\right\}\Bigg)$$

$$\leq \mathbb{E}_{(d_{\pi^\diamond} - \rho^\diamond)^+}\left[\mathbb{1}_{\mu=0}(\mathbb{I} - \gamma\mathbb{P}^\pi)\Delta_{\overline{q^\pi} - \underline{q^\pi}}(s,a) + \mathbb{1}_{\mu>0}\mathcal{E}_2^n\right].$$

therefore when we set $\rho^\diamond = d_{\pi^\diamond}$ which satisfies the condition that $\left\|\frac{\rho}{\mu}\right\|_{L_2(\mu)} \leq \mathcal{U}_2^\star$, because $\pi^\diamond \in \Pi(\mathcal{U}_2^\tau)$ based on the definition of $\Pi(\mathcal{U}_2^\tau)$. In this case, it observes that

$$\mathbb{E}_{(d_{\pi^\diamond} - \rho^\diamond)^+}\left[\mathbb{1}_{\mu=0}(\mathbb{I} - \gamma\mathbb{P}^\pi)\Delta_{\overline{q^\pi} - \underline{q^\pi}}(s,a) + \mathbb{1}_{\mu>0}\mathcal{E}_2^n\right] = 0$$

Then we have

$$J(\pi^\diamond) - J(\widehat{\pi}) \leq \frac{1}{1-\gamma}\mathcal{O}\Bigg(\mathcal{E}_1^n + \sqrt{\left(1 + \mathcal{U}_\infty^\tau + \frac{\mathcal{U}_\infty^\tau}{M}\right)}\max\{(\varepsilon_\mathcal{Q})^{1/2}, (\varepsilon_\mathcal{Q})^{3/4}\}\Bigg) \qquad \text{(F.1)}$$

with the assumption that $\varepsilon_\mathcal{Q} \in (0, 1]$, we have $\max\{(\varepsilon_\mathcal{Q})^{1/2}, (\varepsilon_\mathcal{Q})^{3/4}\} = (\varepsilon_\mathcal{Q})^{1/2}$, and therefore if we ignore the high-order fast terms, we conclude that

$$J(\pi^\diamond) - J(\widehat{\pi}) \leq \frac{1}{1-\gamma}\widetilde{\mathcal{O}}\Bigg(\mathcal{U}_2^\star(\bar{V} + L)\sqrt{\frac{\ln\{\mathrm{Vol}(\Theta)/\delta\}}{nM}} + \sqrt{(1 + \mathcal{U}_\infty^\tau + \mathcal{U}_\infty^\tau/M)\,\varepsilon_\mathcal{Q}}\Bigg).$$

$$\square$$

## G   Proof of Corollary 4.2

*Proof.* On the condition of Corollary 4.1 and set $\varepsilon_\mathcal{Q} = 0$, we can follow the proof of Corollary 4.1, and obtain the regret bound in (F.1) but with the modification as, w.p. $1 - \delta$,

$$J(\pi^\diamond) - J(\widehat{\pi}) \leq \frac{1}{1-\gamma}\mathcal{O}(\mathcal{E}_1^n),$$

for $\mathcal{E}_1^n = \mathcal{U}_2^\star(\bar{V} + L)\sqrt{\ln\{\text{Vol}(\Theta)/\delta\}/(nM)} + \sqrt{\mathcal{U}_2^\tau(\bar{V}^3 + \bar{V}^2 L)/M}(\ln\{\text{Vol}(\Theta)/\delta\}/n)^{\frac{3}{4}} + \mathcal{U}_\infty^\tau(\bar{V} + L)\ln\{\text{Vol}(\Theta)/\delta\}/n$. We let $\varepsilon = \mathcal{E}_1^n/(1 - \gamma)$, and we solve this equation for $n$, by some algebra, we obtain the sample complexity:

$$n = \mathcal{O}\left(\left(\frac{(\mathcal{U}_2^\star(\bar{V} + L)/\sqrt{M})^2}{\varepsilon^2(1-\gamma)^2} + \frac{(\mathcal{U}_2^\tau \bar{V}^2(\bar{V} + L)/M)^{0.67}}{\varepsilon^{1.33}(1-\gamma)^{1.33}} + \frac{\mathcal{U}_\infty^\tau(\bar{V} + L)}{\varepsilon(1 - \gamma)}\right)\ln\frac{\text{Vol}(\Theta)}{\delta}\right).$$

This completes the proof. $\qquad\square$

## H    Proof of Theorem 4.2

*Proof.* To complete the proof, it is sufficient to set $\rho = d^{\pi_b}$ and we can obtain the regret following the proof of Corollary 4.1 with $\varepsilon_{\mathcal{Q}} = 0$, i.e.,

$$J(\pi^\diamond) - J(\widehat{\pi}) \leq \frac{1}{1 - \gamma}\mathcal{O}(\mathcal{E}_1^n).$$

where $\mathcal{E}_1^n = \mathcal{U}_2^\star(\bar{V} + L)\sqrt{\ln\{\text{Vol}(\Theta)/\delta\}/(nM)} + \sqrt{\mathcal{U}_2^\tau(\bar{V}^3 + \bar{V}^2 L)/M}(\ln\{\text{Vol}(\Theta)/\delta\}/n)^{\frac{3}{4}} + \mathcal{U}_\infty^\tau(\bar{V} + L)\ln\{\text{Vol}(\Theta)/\delta\}/n$. As $d^{\pi_b} = \mu$, so that $\|d^{\pi_b}/\mu(s,a)\|_{L_2(\mu)} = \tau_{d^{\pi_b}/\mu} = 1$. Therefore, it is feasible to set $\mathcal{U}_2^\star = \mathcal{U}_\infty^\tau = 1$, and this completes the proof. $\qquad\square$

## I    Proof of Theorem 5.1

### I.1    Proof of Lemma I.1

**Lemma I.1.** *For $k \in [\bar{K}]$, suppose $q^k \in \mathcal{Q}$ and $\tau^k \in \Omega$ such that $\|\tau^k(s,a)\|_{L_2(\mu)} \leq C_1$ and $\|\tau^k(s,a)\|_{L_\infty} \leq C_2$ where the constants $C_2 \geq C_1 > 0$, it satisfies that*

$$\left|\frac{\frac{1}{n}\sum_{i=1}^n \tau^k(s_i, a_i)\left(r_i + \gamma q^k\left(s_i', \pi^k\right) - q^k(s_i, a_i)\right)}{1 - \gamma}\right| \leq \varepsilon.$$

*for some $\varepsilon \geq 0$. There must exist an MDP $\{\mathcal{S}, \mathcal{A}, \mathbb{P}_k, \gamma, r_k, s^0\}$ which is identical to the true environment MDP $\{\mathcal{S}, \mathcal{A}, \mathbb{P}, \gamma, r, s^0\}$: MDP$^\star$ only except the reward function $r_k(s, a)$ is iterative based when $\mathbb{P}_k = \mathbb{P}$, which is defined as*

$$r_k(s, a) = q^k(s, a) - \gamma\mathbb{E}_{s'\sim\mathbb{P}(\cdot|s,a)}\left[\sum_{a'\in\mathcal{A}}\pi^k(a'|s')q^k(s', a')\right].$$

*In addition, such reward functions, for any $k \in [\bar{K}]$, are approximating to the true reward,*

$$\|r_k(s, a) - r(s, a)\|_{L^2(\mu)} \leq C_1\sqrt{\frac{2\bar{V}^2\ln\frac{8\mathcal{N}\left(\epsilon, \mathcal{G}^k, \|\cdot\|_{L_2(\mu)}\right)}{\delta}}{n}} + \frac{2C_2\bar{V}\ln\frac{8\mathcal{N}\left(\epsilon, \mathcal{G}^k, \|\cdot\|_{L_2(\mu)}\right)}{\delta}}{3n} + \varepsilon,$$

*where $\mathcal{G}^k := \widetilde{\Omega}^k \times \mathcal{Q} \times \Pi$ for $\tau^k \in \widetilde{\Omega}^k$. And $q^k$ is the true action-value function under the policy $\pi^k$ in the MDP $\{\mathcal{S}, \mathcal{A}, \mathbb{P}, \gamma, r_k, s^0\}$.*

*Proof.* Following the definition of $r_k$, we observe that

$$q^k(s, a) = r_k(s, a) + \gamma\mathbb{E}_{s'\sim\mathbb{P}(\cdot|s,a)}\left[\sum_{a'\in\mathcal{A}}\pi^k(a'|s')q^k(s', a')\right]$$

$$= r_k(s, a) + \gamma\mathbb{E}_{s'\sim\mathbb{P}_k(\cdot|s,a)}\left[\sum_{a'\in\mathcal{A}}\pi^k(a'|s')q^k(s', a')\right]. \tag{I.1}$$

The second equality comes from $\mathbb{P}_k = \mathbb{P}$ for any $k$. Thus the equation realizes a Bellman equation over the MDP $\{\mathcal{S}, \mathcal{A}, \mathbb{P}_k, \gamma, r_k, s^0\}$ for the policy $\pi^k$. Further, this implies that $q^k(s, a)$ is the corresponding true action-value function. Following the proof of Lemma E.2, for the MDP

$\{\mathcal{S}, \mathcal{A}, \mathbb{P}_k, \gamma, r_k, s^0\}$ and we define the subset of $\tau^k$ with $\|\tau^k(s,a)\|_{L_2(\mu)} \leq C_1$, $\|\tau(s,a)\|_{L_\infty} \leq C_2$ as $\widetilde{\Omega}^k$. Then we have

$$\sup_{\tau^k \in \widetilde{\Omega}^k} \left| \mathbb{E}_\mu \left[ \tau^k(s,a) \left( r(s,a) + \gamma q^k\left(s', \pi^k\right) - q^k(s,a) \right) \right] \right|$$

$$\leq C_1 \sqrt{\frac{2\bar{V}^2 \ln \frac{8\mathcal{N}\left(\epsilon, \mathcal{G}^k, \|\cdot\|_{L_2(\mu)}\right)}{\delta}}{n}} + \frac{2C_2 \bar{V} \ln \frac{8\mathcal{N}\left(\epsilon, \mathcal{G}^k, \|\cdot\|_{L_2(\mu)}\right)}{\delta}}{3n} + \varepsilon.$$

Since $\|\tau^k(s,a)\|_{L_2(\mu)} \leq C_1$, it follows from Lemma E.3, we have

$$C_1 \sqrt{\mathbb{E}_\mu[(r(s,a) + \gamma q^k\left(s', \pi^k\right) - q^k(s,a))^2]} \tag{I.2}$$

$$\leq C_1 \sqrt{\frac{\bar{V}^2 \ln \frac{8\mathcal{N}\left(\epsilon, \mathcal{G}^k, \|\cdot\|_{L_2(\mu)}\right)}{\delta}}{n}} + \frac{2C_2 \bar{V} \ln \frac{8\mathcal{N}\left(\epsilon, \mathcal{G}^k, \|\cdot\|_{L_2(\mu)}\right)}{\delta}}{3n} + \varepsilon. \tag{I.3}$$

Due to the equivalence $\mathbb{P}_k = \mathbb{P}$, we have

$$\|r(s,a) - r_k(s,a)\|_{L^2(\mu)} = \left\| r(s,a) - q^k(s,a) + \gamma \mathbb{E}_{s' \sim \mathbb{P}_k(\cdot|s,a)} \left[ \sum_{a' \in \mathcal{A}} \pi^k(a'|s') q^k(s', \pi^k) \right] \right\|_{L^2(\mu)}$$

$$= \sqrt{\mathbb{E}_\mu[(r(s,a) + \gamma q^k\left(s', \pi^k\right) - q^k(s,a))^2]}$$

$$\leq C_1 \sqrt{\frac{\bar{V}^2 \ln \frac{8\mathcal{N}\left(\epsilon, \mathcal{G}^k, \|\cdot\|_{L_2(\mu)}\right)}{\delta}}{n}} + \frac{2C_2 \bar{V} \ln \frac{8\mathcal{N}\left(\epsilon, \mathcal{G}^k, \|\cdot\|_{L_2(\mu)}\right)}{\delta}}{3n} + \varepsilon.$$

This completes the proof. $\qquad \square$

## I.2 Proof of Lemma I.2

**Lemma I.2.** *Define*

$$\widetilde{q}^\pi := \inf_{q \in \mathcal{Q}} \sup_\rho \mathbb{E}_\rho[(q(s,a) - \mathcal{B}^\pi q(s,a))^2].$$

*Under Assumption 1-3 in maintext, for any $\pi \in \Pi, \tau \in \Omega$ and $q \in \mathcal{Q}$, given an offline data $\mathcal{D}_{1:n} = \{s_i, a_i, r_i, s_i'\}_{i=1}^n$, then w.p. $\geq 1 - \delta$,*

$$\left| \mathbb{P}_n \tau(s_i, a_i) \left( r_i + \gamma \widetilde{q}^\pi\left(s_i', \pi\right) - \widetilde{q}^\pi(s_i, a_i) \right) \right| \leq \varepsilon_n^\diamond$$

*for*

$$\varepsilon_n^\diamond = \left( 3\sqrt{2} \mathcal{U}_2^\tau \bar{V} + 2\sqrt{2} \lambda \|\mathbb{D}(\tau(s,a))\|_{L_2(\mu)}^{UB} \right) \sqrt{\frac{\ln \frac{\mathrm{Vol}(\Theta^\dagger)}{\delta}}{n}}$$

$$+ \frac{\left( 6\mathcal{U}_\infty^\tau \bar{V} + 4\lambda \|\mathbb{D}(\tau(s,a))\|_{L_\infty}^{UB} \right) \ln \frac{\mathrm{Vol}(\Theta^\dagger)}{\delta}}{3n} + \mathcal{U}_2^\tau \sqrt{\varepsilon_\mathcal{Q}},$$

*where $\mathrm{Vol}(\Theta^\dagger) = (e^D \max\{D_\Omega, D_\mathcal{Q}, D_\Pi\} + 1)^3 (\{1 \vee L\} \mathcal{U}_2^\tau)^{2D}$ for $D = D_\Omega + D_\mathcal{Q} + D_\Pi$.*

*Proof.* First, we plug-in $\widetilde{q}^\pi$ into $\mathbb{E}_\mu[\tau(s,a) \Delta(\widetilde{q}^\pi, \pi)]$ for any $\tau \in \Omega$, where $\Delta(q, \pi) := r(s,a) + \gamma q^\pi\left(s', \pi\right) - q^\pi(s,a)$. According to Cauchy–Schwarz inequality,

$$\left| \mathbb{E}_\mu[\tau(s,a) \Delta(\widetilde{q}^\pi, \pi)] \right| \leq \mathbb{E}_\mu[|\tau(s,a)| |\Delta(\widetilde{q}^\pi, \pi)|] \leq \sqrt{\mathbb{E}_\mu[\tau^2(s,a)] \mathbb{E}_\mu\left[\Delta^2(\widetilde{q}^\pi, \pi)\right]} \leq \mathcal{U}_2^\tau \sqrt{\varepsilon_\mathcal{Q}}. \tag{I.4}$$

where the last inequality comes from the weightd $L_2(\mu)$ boundedness over $\tau$ and Assumption 1 in maintext on realizibility error over $\mathcal{Q}$. It then follows from Lemma E.7,

$$\left| \mathbb{E}_\mu \left[ \tau(s,a) \Delta(q, \pi) - \lambda \mathbb{D}(\tau(s,a)) \right] - \mathbb{P}_n \left( \tau(s_i, a_i) \Delta_i(q, \pi) - \lambda \mathbb{D}(\tau(s_i, a_i)) \right) \right| \leq \varepsilon_{1,n}$$

where $\Delta_i(q, \pi) := r_i + \gamma q\left(s'_i, \pi\right) - q(s_i, a_i)$ and $\varepsilon_{1,n}$ denotes the upper bound of the inequality in Lemma E.7. With the norm triangle inequality,

$$\left|\mathbb{E}_\mu\Big[\tau(s,a)\Delta_i(\widetilde{q}^\pi, \pi) - \lambda\mathbb{D}(\tau(s,a))\Big] - \mathbb{P}_n\Big(\tau(s_i,a_i)\Delta_i(\widetilde{q}^\pi, \pi) - \lambda\mathbb{D}(\tau(s_i,a_i))\Big)\right|$$

$$\geq \left|\mathbb{P}_n\Big(\tau(s_i,a_i)\Delta_i(\widetilde{q}^\pi, \pi) - \lambda\mathbb{D}(\tau(s_i,a_i))\Big) + \mathbb{E}_\mu\big[\lambda\mathbb{D}(\tau(s,a))\big]\right| - \left|\mathbb{E}_\mu\big[\tau(s,a)\Delta_i(\widetilde{q}^\pi, \pi)\big]\right|$$

This indicates

$$\left|\mathbb{P}_n\Big(\tau(s_i,a_i)\Delta_i(\widetilde{q}^\pi, \pi) - \lambda\mathbb{D}(\tau(s_i,a_i))\Big) + \mathbb{E}_\mu\big[\lambda\mathbb{D}(\tau(s,a))\big]\right| \leq \varepsilon_{1,n} + \mathcal{U}_2^\tau\sqrt{\varepsilon_\mathcal{Q}}$$

where the inequality comes from (I.4). Apply Triangle inequality again, the above inequality implies

$$\left|\mathbb{P}_n\Big(\tau(s_i,a_i)\Delta_i(\widetilde{q}^\pi, \pi)\Big)\right| \leq \left|\mathbb{P}_n\lambda\mathbb{D}(\tau(s_i,a_i)) - \mathbb{E}_\mu\big[\lambda\mathbb{D}(\tau(s,a))\big]\right| + \mathcal{U}_2^\tau\sqrt{\varepsilon_\mathcal{Q}} + \varepsilon_{1,n}$$

If follow Lemma E.8 and plug-in $\varepsilon_{1,n}$ from Lemma E.7, by some algebra, we conclude that

$$\left|\mathbb{P}_n\Big(\tau(s_i,a_i)\Delta_i(\widetilde{q}^\pi, \pi)\Big)\right| \leq \big(3\sqrt{2}\mathcal{U}_2^\tau\bar{V} + 2\sqrt{2}\lambda\|\mathbb{D}(\tau(s,a))\|_{L_2(\mu)}^{\mathrm{UB}}\big)\sqrt{\frac{\ln\frac{\mathrm{Vol}(\Theta^\dagger)}{\delta}}{n}}$$

$$+ \frac{\big(6\mathcal{U}_\infty^\tau\bar{V} + 4\lambda\|\mathbb{D}(\tau(s,a))\|_{L_\infty}^{\mathrm{UB}}\big)\ln\frac{\mathrm{Vol}(\Theta^\dagger)}{\delta}}{3n} + \mathcal{U}_2^\tau\sqrt{\varepsilon_\mathcal{Q}}.$$

This completes the proof. $\qquad\square$

## I.3  Proof of Lemma I.3

**Lemma I.3.** *Define*

$$\widetilde{q}^\pi := \inf_{q\in\mathcal{Q}}\sup_\rho \mathbb{E}_\rho[(q(s,a) - \mathcal{B}^\pi q(s,a))^2],$$

*for some admissible distribution $\rho$. Under Assumption 1-3 in maintext, for any $\pi \in \Pi, \tau \in \Omega$ and $q \in \mathcal{Q}$, given an offline data $\mathcal{D}_{1:n} = \{s_i, a_i, r_i, s'_i\}_{i=1}^n$, then w.p. $\geq 1 - \delta$,*

$$\left|\mathbb{E}_\mu\left[\tau(s,a)\left(r(s,a) + \gamma\widetilde{q}^\pi\left(s',\pi\right) - \widetilde{q}^\pi(s,a)\right)\right]\right| \leq 2\big(2\mathcal{U}_2^\tau\bar{V} + 2\sqrt{2}\lambda\|\mathbb{D}(\tau(s,a))\|_{L_2(\mu)}^{\mathrm{UB}}\big)\sqrt{\frac{2\ln\frac{\mathrm{Vol}(\Theta^\dagger)}{\delta}}{n}}$$

$$+ \frac{4\big(2\mathcal{U}_\infty^\tau\bar{V} + 4\lambda\|\mathbb{D}(\tau(s,a))\|_{L_\infty}^{\mathrm{UB}}\big)\ln\frac{\mathrm{Vol}(\Theta^\dagger)}{\delta}}{3n} + \mathcal{U}_\infty^\tau\sqrt{\varepsilon_\mathcal{Q}}.$$

*where* $\mathrm{Vol}(\Theta^\dagger) = (e^D\max\{D_\Omega, D_\mathcal{Q}, D_\Pi\} + 1)^3(\{1\vee L\}\mathcal{U}_2^\tau)^{2D}$ *for* $D = D_\Omega + D_\mathcal{Q} + D_\Pi$.

*Proof.* According to Lemma E.2, for any $q \in \mathcal{Q}, \pi \in \Pi, \tau \in \Omega$, we have

$$\left|\mathbb{E}_\mu\left[\tau(s,a)\left(r(s,a) + \gamma q\left(s',\pi\right) - q(s,a)\right)\right] - \mathbb{P}_n\left[\tau(s_i,a_i)\left(r_i + \gamma q\left(s'_i,\pi\right) - q(s_i,a_i)\right)\right]\right|$$

$$\lesssim \mathcal{U}_2^\tau\sqrt{\frac{2\bar{V}^2\ln\frac{\mathrm{Vol}(\Theta^\dagger)}{\delta}}{n}} + \frac{2\mathcal{U}_\infty^\tau\bar{V}\ln\frac{\mathrm{Vol}(\Theta^\dagger)}{\delta}}{3n}.$$

As this holds for any $q \in \mathcal{Q}$, thus it must hold for $\widetilde{q}$ which is in $\mathcal{Q}$ with approximation erorr $\varepsilon_\mathcal{Q}$. Then by trainagle inequality, we have

$$\left|\mathbb{E}_\mu\left[\tau(s_i,a_i)\left(r(s,a) + \gamma\widetilde{q}^\pi\left(s',\pi\right) - \widetilde{q}^\pi(s,a)\right)\right]\right| \leq \left|\mathbb{P}_n\left[\tau(s_i,a_i)\left(r_i + \gamma\widetilde{q}^\pi\left(s'_i,\pi\right) - \widetilde{q}^\pi(s_i,a_i)\right)\right]\right|$$

$$+ \mathcal{U}_2^\tau\sqrt{\frac{2\bar{V}^2\ln\frac{\mathrm{Vol}(\Theta^\dagger)}{\delta}}{n}} + \frac{2\mathcal{U}_\infty^\tau\bar{V}\ln\frac{\mathrm{Vol}(\Theta^\dagger)}{\delta}}{3n}.$$

According to Lemma I.2, we conclude that

$$\left|\mathbb{E}_\mu\left[\tau(s,a)\left(r(s,a) + \gamma\widetilde{q}^\pi\left(s',\pi\right) - \widetilde{q}^\pi(s,a)\right)\right]\right| \leq 2\big(2\mathcal{U}_2^\tau\bar{V} + 2\sqrt{2}\lambda\|\mathbb{D}(\tau(s,a))\|_{L_2(\mu)}^{\mathrm{UB}}\big)\sqrt{\frac{2\ln\frac{\mathrm{Vol}(\Theta^\dagger)}{\delta}}{n}}$$

$$+ \frac{4\big(2\mathcal{U}_\infty^\tau\bar{V} + 4\lambda\|\mathbb{D}(\tau(s,a))\|_{L_\infty}^{\mathrm{UB}}\big)\ln\frac{\mathrm{Vol}(\Theta^\dagger)}{\delta}}{3n} + \mathcal{U}_\infty^\tau\sqrt{\varepsilon_\mathcal{Q}}.$$

$\qquad\square$

## I.4 Proof of Lemma I.4

**Lemma I.4.** *Suppose $\mathbb{P}_n\mathbb{D}(\tau(s_i, a_i)) \leq \varepsilon_n^{\mathbb{D}}$ for some $\tau \in \Omega$ and $\varepsilon_n^{\mathbb{D}}$ depends on $n$ but it is not necessary to be $0$. Then, w.p., $\geq 1 - \delta$,*

$$\|\tau(s,a)\|_{L_2(\mu)} \leq \frac{1}{\sqrt{M}}\left\{ L\mathcal{U}_2^\tau \sqrt{\frac{2\ln\frac{\text{Vol}(\mathcal{G}^{\mathbb{D}})}{\delta}}{n}} + 2\sqrt{\frac{L\mathcal{U}_\infty\mathcal{U}_2^\tau \ln\frac{\text{Vol}(\mathcal{G}^{\mathbb{D}})}{\delta}}{3n}} + \sqrt{2\varepsilon_n^{\mathbb{D}}} + \sqrt{M}\right\}.$$

*where $L$ is the local Lipschitz constant.*

*Proof.* To proceed the proof, we first convert the upper bound for $\mathbb{P}_n\mathbb{D}(\tau(s_i, a_i))$ to the upper bound for $\mathbb{E}_\mu[\mathbb{D}(\tau(s_i, a_i))]$. According to Lemma E.8, we have

$$\mathbb{E}_\mu[\mathbb{D}(\tau(s,a))] \lesssim \mathbb{P}_n\mathbb{D}(\tau(s_i, a_i)) + \varepsilon_n^\diamond, \tag{I.5}$$

where

$$\varepsilon_n^\diamond = \|\mathbb{D}(\tau(s,a))\|_{L_2(\mu)}^{\text{UB}}\sqrt{\frac{2\ln\frac{\text{Vol}(\mathcal{G}^{\mathbb{D}})}{\delta}}{n}} + \frac{2\|\mathbb{D}(\tau(s,a))\|_{L_\infty}^{\text{UB}}\ln\frac{\text{Vol}(\mathcal{G}^{\mathbb{D}})}{\delta}}{3n}.$$

for $\text{Vol}(\mathcal{G}^{\mathbb{D}}) = (e^{D_\Omega}(D_\Omega + 1))(\{1 \vee L\}\mathcal{U}_2^\tau)^{2D_\Omega}$. It follows the inequality in (E.20) and combine with (I.5), we have $\|\tau(s,a)\|_{L_2(\mu)}^2 \leq \frac{2(\varepsilon_n^{\mathbb{D}} + \varepsilon_n^\diamond) + M}{M}$ To simplify the notation, we define

$$\varepsilon_n^{\diamond,1} = \sqrt{\frac{2\ln\frac{\text{Vol}(\mathcal{G}^{\mathbb{D}})}{\delta}}{n}}; \varepsilon_n^{\diamond,2} = \frac{2\|\mathbb{D}(\tau(s,a))\|_{L_\infty}^{\text{UB}}\ln\frac{\text{Vol}(\mathcal{G}^{\mathbb{D}})}{\delta}}{3n}.$$

According to the Lipschitz continuity of $\mathbb{D}(\cdot)$ with local Lipschitz constant $L$, we have

$$\|\tau(s,a)\|_{L_2(\mu)}^2 \leq \frac{\|\mathbb{D}(\tau(s,a))\|_{L_2(\mu)}^{\text{UB}}\varepsilon_n^{\diamond,1} + 2(\varepsilon_n^{\diamond,2} + \varepsilon_n^{\mathbb{D}}) + M}{M} \leq \frac{L\mathcal{U}_2^\tau\varepsilon_n^{\diamond,1} + 2(\varepsilon_n^{\diamond,2} + \varepsilon_n^{\mathbb{D}}) + M}{M}$$

$$\implies \|\tau(s,a)\|_{L_2(\mu)}^2 + \left(\frac{L\mathcal{U}_2^\tau\varepsilon_n^{\diamond,1}}{M}\right)\|\tau(s,a)\|_{L_2(\mu)} - \frac{2}{M}(\varepsilon_n^{\diamond,2} + \varepsilon_n^{\mathbb{D}}) - \frac{M}{M} \leq 0.$$

Therefore, it suffices to solve the root of

$$\|\tau(s,a)\|_{L_2(\mu)}^2 + \left(\frac{L\mathcal{U}_2^\tau\varepsilon_n^{\diamond,1}}{M}\right)\|\tau(s,a)\|_{L_2(\mu)} + \frac{2}{M}(\varepsilon_n^{\diamond,2} + \varepsilon_n^{\mathbb{D}}) + \frac{M}{M} = 0,$$

and we conclude that

$$\sqrt{M}\|\tau(s,a)\|_{L_2(\mu)} \leq L\mathcal{U}_2^\tau\varepsilon_n^{\diamond,1} + \sqrt{2\varepsilon_n^{\diamond,2}} + \sqrt{2\varepsilon_n^{\mathbb{D}}} + \sqrt{M}$$

$$\leq L\mathcal{U}_2^\tau\sqrt{\frac{2\ln\frac{\text{Vol}(\mathcal{G}^{\mathbb{D}})}{\delta}}{n}} + 2\sqrt{\frac{L\mathcal{U}_\infty\mathcal{U}_2^\tau\ln\frac{\text{Vol}(\mathcal{G}^{\mathbb{D}})}{\delta}}{3n}} + \sqrt{2\varepsilon_n^{\mathbb{D}}} + \sqrt{M}.$$

where the last inequality comes from the Lipschitz continuity of $\mathbb{D}(\cdot)$. This completes the proof. $\square$

## I.5 Proof of Lemma I.5

**Lemma I.5.** *Define*

$$\tau_*^k := \arg\max_\tau\left\{ q^k(s^0, \pi^k) + \frac{c^*}{(1-\gamma)n}\Big|\sum_{i=1}^n \tau(s_i, a_i)\left(q^k(s_i, a_i) - r_i - \gamma q^k(s_i', \pi^k)\right)\Big| - \lambda\xi_n\left(\tau(s_i, a_i)\right)\right\},$$

*for $\widetilde{q}^{\pi^k} := \inf_{q \in \mathcal{Q}}\mathbb{E}_\mu\left[\left(q(s,a) - \mathcal{B}^{\pi^k}q(s,a)\right)^2\right]$. Then for any each iteration $k \in [\bar{K}]$, the maximizer $\tau_*^k(s_i, a_i)$ at $k$-th iteration satisfies that*

$$\mathbb{P}_n\mathbb{D}(\tau_*^k(s_i, a_i)) \leq \frac{(1-\gamma)}{\lambda}\Bigg(2\bar{V} + \frac{c^*}{1-\gamma}\bigg\{\left(3\mathcal{U}_2^\tau\bar{V} + 2\sqrt{2}\lambda\|\mathbb{D}(\tau(s,a))\|_{L_2(\mu)}^{UB}\right)\sqrt{\frac{2\ln\frac{\text{Vol}(\Theta^\dagger)}{\delta}}{n}}$$

$$+ \frac{2\left(3\mathcal{U}_\infty^\tau\bar{V} + 2\lambda\|\mathbb{D}(\tau(s,a))\|_{L_\infty}^{UB}\right)\ln\frac{\text{Vol}(\Theta^\dagger)}{\delta}}{3n} + \mathcal{U}_\infty^\tau\sqrt{\varepsilon_\mathcal{Q}}\bigg\}\Bigg),$$

*where $\text{Vol}(\Theta^\dagger) = (e^D\max\{D_\Omega, D_\mathcal{Q}, D_\Pi\} + 1)^3(\{1 \vee L\}\mathcal{U}_2^\tau)^{2D}$ for $D = D_\Omega + D_\mathcal{Q} + D_\Pi$.*

*Proof.* To proceed the proof, we first define a constant $\tau_0(s,a) := 1$ for any $s, a$, and we observe that

$$q^k(s^0, \pi^k) + \sup_\tau \left\{ \frac{c^*}{(1-\gamma)n} \Big| \sum_{i=1}^n \tau(s_i, a_i) \big( q^k(s_i, a_i) - r_i - \gamma q^k(s'_i, \pi^k) \big) \Big| \right\} - \lambda \xi_n \big( \tau_*^k(s_i, a_i) \big)$$

$$\geq q^k(s^0, \pi^k) + \sup_\tau \left\{ \frac{c^*}{(1-\gamma)n} \Big| \sum_{i=1}^n \tau(s_i, a_i) \big( q^k(s_i, a_i) - r_i - \gamma q^k(s'_i, \pi^k) \big) \Big| - \lambda \xi_n \big( \tau(s_i, a_i) \big) \right\}$$

$$\geq \widetilde{q}^{\pi^k}(s^0, \pi^k) + \sup_\tau \left\{ \frac{c^*}{(1-\gamma)n} \Big| \sum_{i=1}^n \tau(s_i, a_i) \big( \widetilde{q}^{\pi^k}(s_i, a_i) - r_i - \gamma \widetilde{q}^{\pi^k}(s'_i, \pi^k) \big) \Big| - \lambda \xi_n \big( \tau(s_i, a_i) \big) \right\}$$

$$= \widetilde{q}^{\pi^k}(s^0, \pi^k) + \frac{c^*}{(1-\gamma)n} \Big| \sum_{i=1}^n \tau_*^k(s_i, a_i) \big( \widetilde{q}^{\pi^k}(s_i, a_i) - r_i - \gamma \widetilde{q}^{\pi^k}(s'_i, \pi^k) \big) \Big| - \lambda \xi_n \big( \tau_*^k(s_i, a_i) \big)$$

$$\geq \widetilde{q}^{\pi^k}(s^0, \pi^k) + \frac{c^*}{(1-\gamma)n} \Big| \sum_{i=1}^n \tau_0(s_i, a_i) \big( \widetilde{q}^{\pi^k}(s_i, a_i) - r_i - \gamma \widetilde{q}^{\pi^k}(s'_i, \pi^k) \big) \Big| - \lambda \xi_n \big( \tau_0(s_i, a_i) \big)$$

$$= \widetilde{q}^{\pi^k}(s^0, \pi^k) + \frac{c^*}{(1-\gamma)n} \Big| \sum_{i=1}^n \big( \widetilde{q}^{\pi^k}(s_i, a_i) - r_i - \gamma \widetilde{q}^{\pi^k}(s'_i, \pi^k) \big) \Big| \geq -\bar{V},$$

where the last inequality comes from the boundedness condition on $\widetilde{q}^{\pi^k}$, and the non-negativity of the second term. Based on this, we further have

$$\lambda \xi_n \big( \tau_*^k(s_i, a_i) \big) - \widetilde{q}^{\pi^k}(s^0, \pi^k) - \sup_\tau \left\{ \frac{c^*}{(1-\gamma)n} \Big| \sum_{i=1}^n \tau(s_i, a_i) \big( \widetilde{q}^{\pi^k}(s_i, a_i) - r_i - \gamma \widetilde{q}^{\pi^k}(s'_i, \pi^k) \big) \Big| \right\} \leq \bar{V},$$

which directly implies with Lemma I.2,

$$\xi_n \big( \tau_*^k(s_i, a_i) \big) \leq \frac{1}{\lambda} \left( 2\bar{V} + \frac{c^*}{1-\gamma} \left\{ \big( 3\mathcal{U}_2^\tau \bar{V} + 2\sqrt{2}\lambda \|\mathbb{D}(\tau(s,a))\|_{L_2(\mu)}^{\mathrm{UB}} \big) \sqrt{\frac{2 \ln \frac{\mathrm{Vol}(\Theta^\dagger)}{\delta}}{n}} \right.\right.$$

$$\left.\left. + \frac{2 \big( 3\mathcal{U}_\infty^\tau \bar{V} + 2\lambda \|\mathbb{D}(\tau(s,a))\|_{L_\infty}^{\mathrm{UB}} \big) \ln \frac{\mathrm{Vol}(\Theta^\dagger)}{\delta}}{3n} + \mathcal{U}_\infty^\tau \sqrt{\varepsilon_\mathcal{Q}} \right) \right).$$

$\square$

## I.6 Proof of Lemma I.6

**Lemma I.6.** *For $k \in [\bar{K}]$, the following inequality holds, w.p. $\geq 1 - \delta$,*

$$\frac{1}{\bar{K}} \sum_{k=1}^{\bar{K}} \left\{ J(\pi^k; \{\mathcal{S}, \mathcal{A}, \mathbb{P}_k, \gamma, r_k, s^0\}) - J(\pi^k) \right\}$$

$$\leq \frac{\sqrt{\varepsilon_\mathcal{Q}}}{1-\gamma} + \frac{c^*}{1-\gamma} \left\{ \big( 3\mathcal{U}_2^\star \bar{V} + 2\lambda \|\mathbb{D}(\tau(s,a))\|_{L_2(\mu)}^{prime} \big) \sqrt{\frac{2 \ln \frac{\mathrm{Vol}(\Theta^\dagger)}{\delta}}{n}} \right.$$

$$\left. + \frac{2 \big( 3\mathcal{U}_\infty \mathcal{U}_2^\star \bar{V} + 2\lambda \|\mathbb{D}(\tau(s,a))\|_{L_\infty}^{prime} \big) \ln \frac{\mathrm{Vol}(\Theta^\dagger)}{\delta}}{3n} + \mathcal{U}_\infty \mathcal{U}_2^\star \sqrt{\varepsilon_\mathcal{Q}} \right\} + \mathcal{O}\left( \lambda \mathcal{U}_\infty \mathcal{U}_2^\star \frac{\sqrt{\varepsilon_\mathcal{Q}}}{1-\gamma} \right) + \mathcal{O}(\frac{1}{\sqrt{n}}).$$

*where $\mathcal{U}_2^\star \in [1, \mathcal{U}_2^\tau)$, can be choose via controlling $\lambda$ and $c^*$, and $\|\mathbb{D}(\tau(s,a))\|_{L_2(\mu)}^{prime} = \sup_{\tau: \|\tau(s,a)\|_{L_2(\mu)} \leq \mathcal{U}_2^\star} \|\mathbb{D}(\tau(s,a))\|_{L_2(\mu)}$, and $\|\mathbb{D}(\tau(s,a))\|_{L_\infty}^{prime} = \sup_{\tau: \|\tau(s,a)\|_{L_2(\mu)} \leq \mathcal{U}_2^\star} \|\mathbb{D}(\tau(s,a))\|_{L_\infty}$, and $\mathrm{Vol}(\Theta^\dagger) = (e^D \max\{D_\Omega, D_\mathcal{Q}, D_\Pi\} + 1)^3 (\{1 \vee L\}\mathcal{U}_2^\tau)^{2D}$ for $D = D_\Omega + D_\mathcal{Q} + D_\Pi$.*

*Proof.* To facilitate the proof, we first define some useful optimizers as follows:

$$\widetilde{q}^{\pi^k} := \inf_{q \in \mathcal{Q}} \mathbb{E}_\mu\left[\left(q(s,a) - \mathcal{B}^{\pi^k}q(s,a)\right)^2\right]$$

$$\tau^k_* := \arg\max_\tau \left\{ q^k(s^0, \pi^k) + \frac{c^*}{(1-\gamma)n}\Big|\sum_{i=1}^n \tau(s_i, a_i)\left(q^k(s_i, a_i) - r_i - \gamma q^k(s'_i, \pi^k)\right)\Big| - \lambda\xi_n\left(\tau(s_i, a_i)\right)\right\}$$

$$\widetilde{\tau}^k := \arg\max_\tau \left\{ \widetilde{q}^{\pi^k}(s^0, \pi^k) + \frac{c^*}{(1-\gamma)}\mathbb{E}_\mu\left[\tau(s,a)\left(\widetilde{q}^{\pi^k}(s,a) - r_i - \gamma\widetilde{q}^{\pi^k}(s', \pi^k)\right)\right] - \lambda\mathbb{E}_\mu[\mathbb{D}(\tau(s,a))]\right\}$$

$$\tau^k_\star := \arg\max_\tau \left\{ \min_q \left\{ q(s^0, \pi^k) + \left\{\frac{c^*}{(1-\gamma)n}\Big|\sum_{i=1}^n \tau_\psi(s_i, a_i)\left(q(s_i, a_i) - r_i - \gamma q(s'_i, \pi^k)\right)\Big| - \lambda\xi_n\left(\tau(s_i, a_i)\right)\right\}\right\}\right\}$$

According to Lemma I.1, $q^k$ is the true action-value function with respect to the MDP $\{\mathcal{S}, \mathcal{A}, \mathbb{P}_k, \gamma, r_k, s^0\}$, therefore we have,

$$J(\pi^k; \{\mathcal{S}, \mathcal{A}, \mathbb{P}_k, \gamma, r_k, s^0\}) - J(\pi^k) = q^k(s^0, \pi^k) - J(\pi^k).$$

Based on this observation, it suffices to upper-bound the following term,

$$q^k(s^0, \pi^k) - J(\pi^k)$$

$$\leq q^k(s^0, \pi^k) + \sup_\tau \frac{c^*}{(1-\gamma)n}\Big|\sum_{i=1}^n \tau(s_i, a_i)\left(q^k(s_i, a_i) - r_i - \gamma q^k(s'_i, \pi^k)\right)\Big| - \lambda\xi_n\left(\widetilde{\tau}^k(s_i, a_i)\right)$$

$$+ \lambda\xi_n\left(\widetilde{\tau}^k(s_i, a_i)\right) - J(\pi^k)$$

$$\leq q^k(s^0, \pi^k) + \sup_\tau \left\{\frac{c^*}{(1-\gamma)n}\Big|\sum_{i=1}^n \tau(s_i, a_i)\left(q^k(s_i, a_i) - r_i - \gamma q^k(s'_i, \pi^k)\right)\Big| - \lambda\xi_n\left(\tau(s_i, a_i)\right)\right\}$$

$$+ \lambda\xi_n\left(\widetilde{\tau}^k(s_i, a_i)\right) - J(\pi^k), \tag{I.6}$$

Follow the definition of $q^k$ which is the minimizer of

$$q(s^0, \pi^k) + \sup_\psi \left\{\frac{c^*}{(1-\gamma)n}\Big|\sum_{i=1}^n \tau(s_i, a_i)\left(q(s_i, a_i) - r_i - \gamma q(s'_i, \pi^k)\right)\Big| - \lambda\xi_n\left(\tau_\psi(s_i, a_i)\right)\right\}.$$

Therefore, for $\widetilde{q}^{\pi^k}$ in function class, we have

$$q^k(s^0, \pi^k) + \sup_\tau \left\{\frac{c^*}{(1-\gamma)n}\Big|\sum_{i=1}^n \tau(s_i, a_i)\left(q^k(s_i, a_i) - r_i - \gamma q^k(s'_i, \pi^k)\right)\Big| - \lambda\xi_n\left(\tau(s_i, a_i)\right)\right\}$$

$$= \min_q \left\{ q(s^0, \pi^k) + \sup_\tau \left\{\frac{c^*}{(1-\gamma)n}\Big|\sum_{i=1}^n \tau_\psi(s_i, a_i)\left(q(s_i, a_i) - r_i - \gamma q(s'_i, \pi^k)\right)\Big| - \lambda\xi_n\left(\tau(s_i, a_i)\right)\right\}\right\}$$

$$\leq \widetilde{q}^{\pi^k}(s^0, \pi^k) + \frac{c^*}{(1-\gamma)n}\Big|\sum_{i=1}^n \tau^k_\star(s_i, a_i)\left(\widetilde{q}^{\pi^k}(s_i, a_i) - r_i - \gamma\widetilde{q}^{\pi^k}(s'_i, \pi^k)\right)\Big| - \lambda\xi_n\left(\tau^k_\star(s_i, a_i)\right)$$

$$\leq \widetilde{q}^{\pi^k}(s^0, \pi^k) + \sup_\tau \left\{\frac{c^*}{(1-\gamma)n}\Big|\sum_{i=1}^n \tau(s_i, a_i)\left(\widetilde{q}^{\pi^k}(s_i, a_i) - r_i - \gamma\widetilde{q}^{\pi^k}(s'_i, \pi^k)\right)\Big| - \lambda\xi_n\left(\tau(s_i, a_i)\right)\right\}$$

By this, we have ([I.6](#)) is upper bounded by

$$
\widetilde{q}^{\pi^k}(s^0, \pi^k) + \sup_\tau \left\{ \frac{c^*}{(1-\gamma)n} \Big| \sum_{i=1}^n \tau(s_i, a_i) \left( \widetilde{q}^{\pi^k}(s_i, a_i) - r_i - \gamma \widetilde{q}^{\pi^k}(s_i', \pi^k) \right) \Big| - \lambda \xi_n \big( \tau(s_i, a_i) \big) \right\}
$$

$$
+ \lambda \xi_n \big( \widetilde{\tau}^k(s_i, a_i) \big) - J(\pi^k)
$$

$$
\leq \widetilde{q}^{\pi^k}(s^0, \pi^k) + \sup_\tau \left\{ \frac{c^*}{(1-\gamma)n} \Big| \sum_{i=1}^n \tau_\psi(s_i, a_i) \left( \widetilde{q}^{\pi^k}(s_i, a_i) - r_i - \gamma \widetilde{q}^{\pi^k}(s_i', \pi^k) \right) \Big| \right\}
$$

$$
+ \lambda \xi_n \big( \widetilde{\tau}^k(s_i, a_i) \big) - J(\pi^k)
$$

$$
= \underbrace{\widetilde{q}^{\pi^k}(s^0, \pi^k) - J(\pi^k)}_{\Delta_1} + \underbrace{\sup_\tau \left\{ \frac{c^*}{(1-\gamma)n} \Big| \sum_{i=1}^n \tau(s_i, a_i) \left( \widetilde{q}^{\pi^k}(s_i, a_i) - r_i - \gamma \widetilde{q}^{\pi^k}(s_i', \pi^k) \right) \Big| \right\}}_{\Delta_2}
$$

$$
+ \underbrace{\lambda \xi_n \big( \widetilde{\tau}^k(s_i, a_i) \big)}_{\Delta_3}.
$$

**Bounding $\Delta_1$.** According to Lemma [E.1](#), we have

$$
\Delta_1 = \frac{\mathbb{E}_{d^{\pi^k}}[\widetilde{q}^{\pi^k}(s,a) - r(s,a) - \gamma \widetilde{q}^{\pi^k}(s', \pi^k)]}{1-\gamma} \leq \frac{\| \widetilde{q}^{\pi^k}(s,a) - r(s,a) - \gamma \widetilde{q}^{\pi^k}(s', \pi^k) \|_{L(d^{\pi^k})}}{1-\gamma}.
$$

As $\pi^k \in \Pi$ and thus $d^{\pi^k}$ is an admissible data distribution, it follows Assumption 1 in maintext, it follows

$$
\frac{\| \widetilde{q}^{\pi^k}(s,a) - r(s,a) - \gamma \widetilde{q}^{\pi^k}(s', \pi^k) \|_{L(d^{\pi^k})}}{1-\gamma} \leq \frac{\sqrt{\varepsilon_\mathcal{Q}}}{1-\gamma}.
$$

Thus, we have $\Delta_1 \leq \frac{\sqrt{\varepsilon_\mathcal{Q}}}{1-\gamma}$.

**Bounding $\Delta_2$.** According to Lemma [I.2](#) and [I.4](#), with a proper choice of $\lambda$ and $c^*$ in the proof of Theorem 5.1, we can have a well-controlled uncertainty concentrability coefficient $\mathcal{U}_2^\star$. Replacing $\pi$ with $\pi^k$, we then have

$$
\sup_\tau \left\{ \frac{c^*}{(1-\gamma)n} \Big| \sum_{i=1}^n \tau(s_i, a_i) \left( \widetilde{q}^{\pi^k}(s_i, a_i) - r_i - \gamma \widetilde{q}^{\pi^k}(s_i', \pi^k) \right) \Big| \right\}
$$

$$
\leq \frac{c^*}{1-\gamma} \left\{ \left( 3\sqrt{2} \mathcal{U}_2^\star \bar{V} + 2\sqrt{2} \lambda \| \mathbb{D}(\tau(s,a)) \|_{L_2(\mu)}^{\text{prime}} \right) \sqrt{\frac{\ln \frac{\text{Vol}(\Theta^\dagger)}{\delta}}{n}} \right.
$$

$$
\left. + \frac{\left( 6\mathcal{U}_\infty^\star \bar{V} + 4\lambda \| \mathbb{D}(\tau(s,a)) \|_{L_\infty}^{\text{prime}} \right) \ln \frac{\text{Vol}(\Theta^\dagger)}{\delta}}{3n} + \mathcal{U}_\infty^\star \sqrt{\varepsilon_\mathcal{Q}} \right\}.
$$

where $\mathcal{U}_2^\star \in [1, \mathcal{U}_2^\tau)$ and $\| \mathbb{D}(\tau(s,a)) \|_{L_2(\mu)}^{\text{prime}} = \sup_{\tau : \| \tau(s,a) \|_{L_2(\mu)} \leq \mathcal{U}_2^\star} \| \mathbb{D}(\tau(s,a)) \|_{L_2(\mu)}$, and $\| \mathbb{D}(\tau(s,a)) \|_{L_\infty}^{\text{prime}} = \sup_{\tau : \| \tau(s,a) \|_{L_2(\mu)} \leq \mathcal{U}_2^\star} \| \mathbb{D}(\tau(s,a)) \|_{L_\infty}$.

**Bounding $\Delta_3$.** Follow a similar argument in Proposition 1 of [28] and the definition of $\widetilde{\tau}^k$, by some algebra, we have, for any $(s,a)$,

$$
\widetilde{\tau}^k(s,a) = \left[ (\mathbb{D}')^{-1} \left( \frac{c^* \left( r(s,a) + \gamma \widetilde{q}^{\pi^k}(s', \pi^k) - \widetilde{q}^{\pi^k}(s,a) \right)}{\lambda(1-\gamma)} \right) \right]^+.
$$

It follows from Lemma [I.3](#), for any $(s,a)$ such that $\mu(s,a) > 0$, we have

$$
\left| r(s,a) + \gamma \widetilde{q}^{\pi^k}(s', \pi^k) - \widetilde{q}^{\pi^k}(s,a) \right|
$$

$$
\lesssim (\bar{V} + \lambda L) \sqrt{\frac{2 \ln \frac{\text{Vol}(\Theta^\dagger)}{\delta}}{n}} + \frac{\mathcal{U}_\infty (\bar{V} + L\lambda) \ln \frac{\text{Vol}(\Theta^\dagger)}{\delta}}{3n} + \mathcal{U}_\infty \sqrt{\varepsilon_\mathcal{Q}} := \varepsilon_n^{\widetilde{q}^{\pi^k}}.
$$

And thus for any $(s_i, a_i)$ where $i = 1, ..., n$, we have

$$\frac{1}{1-\gamma} c^* \left( r(s_i, a_i) + \gamma \widetilde{q}^{\pi^k}(s_i', \pi^k) - \widetilde{q}^{\pi^k}(s_i, a_i) \right) \lesssim \frac{1}{1-\gamma} c^* \varepsilon_n^{\widetilde{q}^{\pi^k}}.$$

This directly leads to

$$(1-\gamma)\lambda \xi_n \left( \widetilde{\tau}^k(s_i, a_i) \right) \leq \frac{\lambda}{n} \sum_{i=1}^n \mathbb{D}\left( \left[ (\mathbb{D}')^{-1} \left( \frac{c^* \varepsilon_n^{\widetilde{q}^{\pi^k}}}{\lambda(1-\gamma)} \right) \right]^+ \right) \leq \lambda \mathbb{D}\left( \left[ (\mathbb{D}')^{-1} \left( \frac{c^* \varepsilon_n^{\widetilde{q}^{\pi^k}}}{\lambda(1-\gamma)} \right) \right]^+ \right).$$

We note that for sufficient large $n$, we have $c^* \varepsilon_n^{\widetilde{q}^{\pi^k}} \to c^* \mathcal{U}_\infty \mathcal{U}_2^\star \sqrt{\varepsilon_\mathcal{Q}}$ and thus

$$\left[ (\mathbb{D}')^{-1} \left( \frac{c^* \varepsilon_n^{\widetilde{q}^{\pi^k}}}{\lambda(1-\gamma)} \right) \right]^+ \xrightarrow{n \uparrow \infty} \left[ (\mathbb{D}')^{-1} \left( \frac{c^* \mathcal{U}_\infty \mathcal{U}_2^\star \sqrt{\varepsilon_\mathcal{Q}}}{\lambda(1-\gamma)} \right) \right]^+.$$

As $\mathbb{D}'$ is monotonic increasing, and since $\frac{c^* \mathcal{U}_\infty \mathcal{U}_2^\star \sqrt{\varepsilon_\mathcal{Q}}}{\lambda(1-\gamma)} \geq 0$, so $(\mathbb{D}')^{-1} \left( \frac{c^* \mathcal{U}_\infty \mathcal{U}_2^\star \sqrt{\varepsilon_\mathcal{Q}}}{\lambda(1-\gamma)} \right) \geq 1$, and according the property of divergence function $\mathbb{D}'(1) = 0$ thus

$$\left[ (\mathbb{D}')^{-1} \left( \frac{c^* \mathcal{U}_\infty \mathcal{U}_2^\star \sqrt{\varepsilon_\mathcal{Q}}}{\lambda(1-\gamma)} \right) \right]^+ = (\mathbb{D}')^{-1} \left( \frac{c^* \mathcal{U}_\infty \mathcal{U}_2^\star \sqrt{\varepsilon_\mathcal{Q}}}{\lambda(1-\gamma)} \right),$$

for sure. This immediately implies that

$$\mathbb{D}\left( (\mathbb{D}')^{-1} \left( \frac{c^* \varepsilon_n^{\widetilde{q}^{\pi^k}}}{\lambda(1-\gamma)} \right) \right) \xrightarrow{n \uparrow \infty} \mathbb{D}\left( (\mathbb{D}')^{-1} \left( \frac{c^* \mathcal{U}_\infty \mathcal{U}_2^\star \sqrt{\varepsilon_\mathcal{Q}}}{\lambda(1-\gamma)} \right) \right).$$

Therefore we conclude that

$$\Delta_3 \lesssim \lambda \mathbb{D}\left( (\mathbb{D}')^{-1} \left( \frac{c^* \mathcal{U}_\infty \mathcal{U}_2^\star \sqrt{\varepsilon_\mathcal{Q}}}{\lambda(1-\gamma)} \right) \right) + \mathcal{O}(\frac{1}{\sqrt{n}}).$$

For sufficient small realizibility error $\sqrt{\varepsilon_\mathcal{Q}}$, and $c^* \asymp \lambda$, for bounded $\mathcal{U}_\infty$ and $\mathcal{U}_2^\star$, and also by the monotonicity of $(\mathbb{D}')^{-1}(\cdot)$, we have

$$(\mathbb{D}')^{-1} \left( \frac{c^* \mathcal{U}_\infty \mathcal{U}_2^\star \sqrt{\varepsilon_\mathcal{Q}}}{\lambda(1-\gamma)} \right) \asymp (\mathbb{D}')^{-1} \left( \frac{\mathcal{U}_\infty \mathcal{U}_2^\star \sqrt{\varepsilon_\mathcal{Q}}}{(1-\gamma)} \right).$$

According to the local Lipschitz continuity of $\mathbb{D}$ and $\mathbb{D}'$ due to strongly convexity, we have

$$\begin{aligned}
\lambda \mathbb{D}\left( (\mathbb{D}')^{-1} \left( \frac{\mathcal{U}_\infty \mathcal{U}_2^\star \sqrt{\varepsilon_\mathcal{Q}}}{(1-\gamma)} \right) \right) &\leq \lambda \left| \mathbb{D}\left( (\mathbb{D}')^{-1} \left( \frac{\mathcal{U}_\infty \mathcal{U}_2^\star \sqrt{\varepsilon_\mathcal{Q}}}{(1-\gamma)} \right) \right) \right| \\
&= \lambda \left| \mathbb{D}\left( (\mathbb{D}')^{-1} \left( \frac{\mathcal{U}_\infty \mathcal{U}_2^\star \sqrt{\varepsilon_\mathcal{Q}}}{(1-\gamma)} \right) \right) - 0 \right| \\
&\leq \lambda L \left| (\mathbb{D}')^{-1} \left( \frac{\mathcal{U}_\infty \mathcal{U}_2^\star \sqrt{\varepsilon_\mathcal{Q}}}{(1-\gamma)} \right) - 1 \right| \\
&\lesssim \lambda L \left| \frac{\mathcal{U}_\infty \mathcal{U}_2^\star \sqrt{\varepsilon_\mathcal{Q}}}{(1-\gamma)} \right|.
\end{aligned}$$

where $L$ is some Lipschtiz constants, and then we conclude that

$$\Delta_3 \lesssim \mathcal{O}\left( \lambda \mathcal{U}_\infty \mathcal{U}_2^\star \frac{\sqrt{\varepsilon_\mathcal{Q}}}{1-\gamma} \right) + \mathcal{O}(\frac{1}{\sqrt{n}}). \tag{I.7}$$

Combine the results on bounding $\Delta_1$, $\Delta_2$ and $\Delta_3$, we have

$$J(\pi^k; \{\mathcal{S}, \mathcal{A}, \mathbb{P}_k, \gamma, r_k, s^0\}) - J(\pi^k) \leq \frac{\sqrt{\varepsilon_\mathcal{Q}}}{1-\gamma} + \frac{c^*}{1-\gamma} \left\{ \left( 3\mathcal{U}_2^\star \bar{V} + 2\lambda \|\mathbb{D}(\tau(s,a))\|_{L_2(\mu)}^{\mathrm{prime}} \right) \sqrt{\frac{2 \ln \frac{\mathrm{Vol}(\Theta^\dagger)}{\delta}}{n}} \right.$$

$$+ \frac{2\left( 3\mathcal{U}_\infty \mathcal{U}_2^\star \bar{V} + 2\lambda \|\mathbb{D}(\tau(s,a))\|_{L_\infty}^{\mathrm{prime}} \right) \ln \frac{\mathrm{Vol}(\Theta^\dagger)}{\delta}}{3n} + \mathcal{U}_\infty \mathcal{U}_2^\star \sqrt{\varepsilon_\mathcal{Q}} \left. \right\} + \mathcal{O}\left( \lambda \mathcal{U}_\infty \mathcal{U}_2^\star \frac{\sqrt{\varepsilon_\mathcal{Q}}}{1-\gamma} \right) + \mathcal{O}(\frac{1}{\sqrt{n}}).$$

As the above upper bound holds for any $k \in [\bar{K}]$, this completes the proof for $\frac{1}{\bar{K}} \sum_{k=1}^{\bar{K}} \left\{ J(\pi^k; \{\mathcal{S}, \mathcal{A}, \mathbb{P}_k, \gamma, r_k, s^0\}) - J(\pi^k) \right\}$. $\qquad \square$

## I.7 Proof of Lemma I.7

**Lemma I.7.** *For any $\pi \in \Pi$, and define the normalized negative entropy as*

$$\mathcal{H}_{NegEnt}(\pi(\cdot|s)) = \sum_{a \in \mathcal{A}} \pi(a|s) \log(\pi(a|s)).$$

*Then we have*

$$\sum_{k=1}^{\bar{K}} \left\langle \pi(\cdot|s) - \pi^{k+1}(\cdot|s), q^k(s, \cdot) \right\rangle \leq \mathcal{H}_{NegEnt}(\pi(\cdot|s)) - \mathcal{H}_{NegEnt}(\pi^0(\cdot|s))$$

*where $\pi^0(\cdot|s)$ is the initial random policy for algorithm run.*

*Proof.* To prove this Lemma, we use mathematical induction. Suppose the inequality holds for the round $(\bar{K} - 1)$, i.e.,

$$\sum_{k=1}^{\bar{K}-1} \left\langle \pi(\cdot|s) - \pi^{k+1}(\cdot|s), q^k(s, \cdot) \right\rangle \leq \mathcal{H}_{\text{NegEnt}}(\pi(\cdot|s)) - \mathcal{H}_{\text{NegEnt}}(\pi^0(\cdot|s))$$

Then we verify the inequality for $\bar{K}$ in the following,

$$\sum_{k=1}^{\bar{K}} \left\langle \pi(\cdot|s) - \pi^{k+1}(\cdot|s), q^k(s, \cdot) \right\rangle$$

$$= \sum_{k=1}^{\bar{K}} \left\langle \pi(\cdot|s), q^k(s, \cdot) \right\rangle - \sum_{k=1}^{\bar{K}} \left\langle \pi^{k+1}(\cdot|s), q^k(s, \cdot) \right\rangle - \mathcal{H}_{\text{NegEnt}}(\pi(\cdot|s)) + \mathcal{H}_{\text{NegEnt}}(\pi(\cdot|s))$$

$$\leq \sum_{k=1}^{\bar{K}} \left\langle \pi_{\bar{K}+1}(\cdot|s), q^k(s, \cdot) \right\rangle - \sum_{k=1}^{\bar{K}} \left\langle \pi^{k+1}(\cdot|s), q^k(s, \cdot) \right\rangle$$
$$- \mathcal{H}_{\text{NegEnt}}(\pi_{\bar{K}+1}(\cdot|s)) + \mathcal{H}_{\text{NegEnt}}(\pi(\cdot|s))$$

$$= \sum_{k=1}^{\bar{K}-1} \left\langle \pi_{\bar{K}+1}(\cdot|s), q^k(s, \cdot) \right\rangle - \sum_{k=1}^{\bar{K}} \left\langle \pi^{k+1}(\cdot|s), q^k(s, \cdot) \right\rangle + \left\langle \pi_{\bar{K}+1}(\cdot|s), q^k(s, \cdot) \right\rangle$$
$$- \mathcal{H}_{\text{NegEnt}}(\pi_{\bar{K}+1}(\cdot|s)) + \mathcal{H}_{\text{NegEnt}}(\pi(\cdot|s))$$

$$= \sum_{k=1}^{\bar{K}-1} \left\langle \pi_{\bar{K}+1}(\cdot|s), q^k(s, \cdot) \right\rangle - \sum_{k=1}^{\bar{K}-1} \left\langle \pi^{k+1}(\cdot|s), q^k(s, \cdot) \right\rangle + \left\langle \pi_{\bar{K}+1}(\cdot|s), q^k(s, \cdot) \right\rangle$$
$$- \left\langle \pi_{\bar{K}+1}(\cdot|s), q^k(s, \cdot) \right\rangle - \mathcal{H}_{\text{NegEnt}}(\pi_{\bar{K}+1}(\cdot|s)) + \mathcal{H}_{\text{NegEnt}}(\pi(\cdot|s))$$

$$\leq \mathcal{H}_{\text{NegEnt}}(\pi_{\bar{K}+1}(\cdot|s)) - \mathcal{H}_{\text{NegEnt}}(\pi^0(\cdot|s)) - \mathcal{H}_{\text{NegEnt}}(\pi_{\bar{K}+1}(\cdot|s)) + \mathcal{H}_{\text{NegEnt}}(\pi(\cdot|s))$$

$$= \mathcal{H}_{\text{NegEnt}}(\pi(\cdot|s)) - \mathcal{H}_{\text{NegEnt}}(\pi^0(\cdot|s)).$$

This completes the proof. $\qquad \square$

## I.8 Proof of Lemma I.8

**Lemma I.8.** *For any policy $\pi$, it satisfies that*

$$\sum_{k=1}^{\bar{K}} \left\langle \pi - \pi^k(\cdot|s), q^k(s, \cdot) \right\rangle \leq 2\sqrt{2\bar{V}\bar{K}\log|\mathcal{A}|}.$$

*Proof.* Following the definition of the Bergman divergence in terms of negative entropy, we have that

$$D_{\text{NegEntropy}}(\pi^k, \pi^{k+1}) = \mathcal{H}_{\text{NegEnt}}(\pi^k(\cdot|s)) - \mathcal{H}_{\text{NegEnt}}(\pi^{k+1}(\cdot|s)) \tag{I.8}$$

$$- \left\langle \nabla \mathcal{H}_{\text{NegEnt}}(\pi^{k+1}(\cdot|s)), \pi^k(\cdot|s) - \pi^{k+1}(\cdot|s) \right\rangle. \tag{I.9}$$

By the second-order Taylor expansion on $\mathcal{H}_{\text{NegEnt}}(\pi^k(\cdot|s))$ and evaluated at $\pi^{k+1}(\cdot|s)$, we have

$$\mathcal{H}_{\text{NegEnt}}(\pi^k(\cdot|s)) = \mathcal{H}_{\text{NegEnt}}(\pi^{k+1}(\cdot|s)) - \left\langle \nabla\mathcal{H}_{\text{NegEnt}}(\pi^{k+1}(\cdot|s)), \pi^k(\cdot|s) - \pi^{k+1}(\cdot|s)\right\rangle$$
$$+ \frac{1}{2}(\pi^k(\cdot|s) - \pi^{k+1}(\cdot|s))^T \nabla^2\mathcal{H}_{\text{NegEnt}}(\pi_{k||k-1}(\cdot|s))(\pi^k(\cdot|s) - \pi^{k+1}(\cdot|s)),$$
(I.10)

where $\pi_{k||k-1}(\cdot|s)$ lies on the line connecting $\pi^k(\cdot|s)$ and $\pi^{k+1}(\cdot|s)$. With (I.9) and (I.10), we do the subtraction, then it obtains

$$D_{\text{NegEntropy}}(\pi^k, \pi^{k+1}) = \frac{1}{2}(\pi^k(\cdot|s) - \pi^{k+1}(\cdot|s))^T \nabla^2\mathcal{H}_{\text{NegEnt}}(\pi_{k||k-1}(\cdot|s))(\pi^k(\cdot|s) - \pi^{k+1}(\cdot|s)).$$

Then we proceed to bound

$$\left\langle \pi^k(\cdot|s) - \pi^{k+1}(\cdot|s), q^k(s, \cdot)\right\rangle \tag{I.11}$$

$$\leq \sqrt{q^k(s, \cdot)^T \nabla^{-2}\mathcal{H}_{\text{NegEnt}}(\pi_{k||k-1}(\cdot|s))q^k(s, \cdot)}$$

$$\cdot \sqrt{(\pi^k(\cdot|s) - \pi^{k+1}(\cdot|s))^T \nabla^2\mathcal{H}_{\text{NegEnt}}(\pi_{k||k-1}(\cdot|s))(\pi^k(\cdot|s) - \pi^{k+1}(\cdot|s))}$$

$$= \sqrt{2q^k(s, \cdot)^T \nabla^{-2}\mathcal{H}_{\text{NegEnt}}(\pi_{k||k-1}(\cdot|s))q^k(s, \cdot)D_{\text{NegEntropy}}(\pi^k, \pi^{k+1})}$$

$$\leq \sqrt{2\zeta}\|q^k\|_{L_\infty}\sqrt{D_{\text{NegEntropy}}(\pi^k, \pi^{k+1})}, \tag{I.12}$$

where $\zeta$ is defined in Algorithm 1 maintext, which indicates the projection rate. Next, we aim to upper bound $\sqrt{D_{\text{NegEntropy}}(\pi^k, \pi^{k+1})}$. Follow the soft policy improvement Lemma 2 in [15] that $\pi^{k+1}$ is the global maximizer of $\sum_{k'=1}^k \langle\pi(\cdot|s), q^{k'}(s, \cdot)\rangle - \mathcal{H}_{\text{NegEnt}}(\pi(\cdot|s))$. By

$$0 = \sum_{k'=1}^k \langle\pi^k(\cdot|s), q^{k'}(s, \cdot)\rangle - \mathcal{H}_{\text{NegEnt}}(\pi^k(\cdot|s)) - \left(\sum_{k'=1}^k \langle\pi^{k+1}(\cdot|s), q^{k'}(s, \cdot)\rangle - \mathcal{H}_{\text{NegEnt}}(\pi^{k+1}(\cdot|s))\right)$$

$$- \left\langle \pi^k(\cdot|s) - \pi^{k+1}(\cdot|s), \nabla_\pi \sum_{k'=1}^k \langle\pi^{k+1}(\cdot|s), q^{k'}(s, \cdot)\rangle - \mathcal{H}_{\text{NegEnt}}(\pi^{k+1}(\cdot|s))\right\rangle$$

$$- \left(\sum_{k'=1}^k \langle\pi^k(\cdot|s), q^{k'}(s, \cdot)\rangle - \mathcal{H}_{\text{NegEnt}}(\pi^k(\cdot|s)) - \left(\sum_{k'=1}^k \langle\pi^{k+1}(\cdot|s), q^{k'}(s, \cdot)\rangle - \mathcal{H}_{\text{NegEnt}}(\pi^{k+1}(\cdot|s))\right)\right.$$

$$\left. - \left\langle \pi^k(\cdot|s) - \pi^{k+1}(\cdot|s), \nabla_\pi \sum_{k'=1}^k \langle\pi^{k+1}(\cdot|s), q^{k'}(s, \cdot)\rangle - \mathcal{H}_{\text{NegEnt}}(\pi^{k+1}(\cdot|s))\right\rangle\right).$$

By $\pi^{k+1}$ is the maximizer of $\sum_{k'=1}^k \langle\pi(\cdot|s), q^{k'}(s, \cdot)\rangle - \mathcal{H}_{\text{NegEnt}}(\pi(\cdot|s))$, then

$$\nabla_\pi \sum_{k'=1}^k \langle\pi^{k+1}(\cdot|s), q^{k'}(s, \cdot)\rangle - \mathcal{H}_{\text{NegEnt}}(\pi^{k+1}(\cdot|s)) = 0.$$

This implies that

$$\sum_{k'=1}^k \langle\pi^k(\cdot|s), q^{k'}(s, \cdot)\rangle - \mathcal{H}_{\text{NegEnt}}(\pi^k(\cdot|s)) - \left(\sum_{k'=1}^k \langle\pi^{k+1}(\cdot|s), q^{k'}(s, \cdot)\rangle - \mathcal{H}_{\text{NegEnt}}(\pi^{k+1}(\cdot|s))\right)$$

$$= \left(\sum_{k'=1}^k \langle\pi^k(\cdot|s), q^{k'}(s, \cdot)\rangle - \mathcal{H}_{\text{NegEnt}}(\pi^k(\cdot|s)) - \left(\sum_{k'=1}^k \langle\pi^{k+1}(\cdot|s), q^{k'}(s, \cdot)\rangle - \mathcal{H}_{\text{NegEnt}}(\pi^{k+1}(\cdot|s))\right)\right.$$

$$\left. - \left\langle \pi^k(\cdot|s) - \pi^{k+1}(\cdot|s), \nabla_\pi \sum_{k'=1}^k \langle\pi^{k+1}(\cdot|s), q^{k'}(s, \cdot)\rangle - \mathcal{H}_{\text{NegEnt}}(\pi^{k+1}(\cdot|s))\right\rangle\right)$$

$$= \sum_{k'=1}^k \langle\pi^k(\cdot|s), q^{k'}(s, \cdot)\rangle - \sum_{k'=1}^k \langle\pi^{k+1}(\cdot|s), q^{k'}(s, \cdot)\rangle + \mathcal{H}_{\text{NegEnt}}(\pi^{k+1}(\cdot|s)) - \mathcal{H}_{\text{NegEnt}}(\pi^k(\cdot|s))$$

$$- \left\langle \pi^k(\cdot|s) - \pi^{k+1}(\cdot|s), \nabla_\pi \sum_{k'=1}^k \langle\pi^{k+1}(\cdot|s), q^{k'}(s, \cdot)\rangle\right\rangle - \left\langle \pi^k(\cdot|s) - \pi^{k+1}(\cdot|s), \nabla_\pi\mathcal{H}_{\text{NegEnt}}(\pi^{k+1}(\cdot|s))\right\rangle.$$

Since

$$\sum_{k'=1}^{k} \langle \pi^k(\cdot|s), q^{k'}(s, \cdot) \rangle - \sum_{k'=1}^{k} \langle \pi^{k+1}(\cdot|s), q^{k'}(s, \cdot) \rangle \leq \left\langle \pi^k(\cdot|s) - \pi^{k+1}(\cdot|s), \nabla_\pi \sum_{k'=1}^{k} \langle \pi^{k+1}(\cdot|s), q^{k'}(s, \cdot) \rangle \right\rangle.$$

Thus we have

$$\sum_{k'=1}^{k} \langle \pi^k(\cdot|s), q^{k'}(s, \cdot) \rangle - \mathcal{H}_{\text{NegEnt}}(\pi^k(\cdot|s)) - \left( \sum_{k'=1}^{k} \langle \pi^{k+1}(\cdot|s), q^{k'}(s, \cdot) \rangle - \mathcal{H}_{\text{NegEnt}}(\pi^{k+1}(\cdot|s)) \right)$$

$$\leq \mathcal{H}_{\text{NegEnt}}(\pi^{k+1}(\cdot|s)) - \mathcal{H}_{\text{NegEnt}}(\pi^k(\cdot|s)) - \langle \pi^k(\cdot|s) - \pi^{k+1}(\cdot|s), \nabla_\pi \mathcal{H}_{\text{NegEnt}}(\pi^{k+1}(\cdot|s)) \rangle$$

$$= - D_{\text{NegEntropy}}(\pi^k, \pi^{k+1}).$$

This implies that

$$D_{\text{NegEntropy}}(\pi^k(\cdot|s), \pi^{k+1}(\cdot|s)) \leq \sum_{k'=1}^{k} \langle \pi^{k+1}(\cdot|s), q^{k'}(s, \cdot) \rangle - \mathcal{H}_{\text{NegEnt}}(\pi^{k+1}(\cdot|s))$$

$$- \left( \sum_{k'=1}^{k} \langle \pi^k(\cdot|s), q^{k'}(s, \cdot) \rangle - \mathcal{H}_{\text{NegEnt}}(\pi^k(\cdot|s)) \right)$$

$$= \sum_{k'=1}^{k-1} \langle \pi^{k+1}(\cdot|s), q^{k'}(s, \cdot) \rangle - \mathcal{H}_{\text{NegEnt}}(\pi^{k+1}(\cdot|s))$$

$$- \left( \sum_{k'=1}^{k-1} \langle \pi^k(\cdot|s), q^{k'}(s, \cdot) \rangle - \mathcal{H}_{\text{NegEnt}}(\pi^k(\cdot|s)) \right)$$

$$- \langle \pi^k(\cdot|s) - \pi^{k+1}(\cdot|s), q^k(s, \cdot) \rangle.$$

Since

$$\sum_{k'=1}^{k-1} \langle \pi^{k+1}(\cdot|s), q^{k'}(s, \cdot) \rangle - \mathcal{H}_{\text{NegEnt}}(\pi^{k+1}(\cdot|s))$$

$$- \left( \sum_{k'=1}^{k-1} \langle \pi^k(\cdot|s), q^{k'}(s, \cdot) \rangle - \mathcal{H}_{\text{NegEnt}}(\pi^k(\cdot|s)) \right) \leq 0$$

Then we have

$$D_{\text{NegEntropy}}(\pi^k(\cdot|s), \pi^{k+1}(\cdot|s)) \leq - \langle \pi^k(\cdot|s) - \pi^{k+1}(\cdot|s), q^k(s, \cdot) \rangle$$

Combine with (I.12) and boundedness condition on $q$-function, we have

$$- \langle \pi^k(\cdot|s) - \pi^{k+1}(\cdot|s), q^k(s, \cdot) \rangle \leq \sqrt{2\zeta} \bar{V} \sqrt{\langle \pi^{k+1}(\cdot|s) - \pi^k(\cdot|s), q^k(s, \cdot) \rangle}$$

$$\leq \sqrt{2\zeta \bar{V} \langle \pi^{k+1}(\cdot|s) - \pi^k(\cdot|s), q^k(s, \cdot) \rangle}.$$

Solving the equation

$$\langle \pi^k(\cdot|s) - \pi^{k+1}(\cdot|s), q^k(s, \cdot) \rangle^2 - 2\zeta \bar{V}^2 \langle \pi^k(\cdot|s) - \pi^{k+1}(\cdot|s), q^k(s, \cdot) \rangle = 0.$$

We obtain that

$$- \langle \pi^k(\cdot|s) - \pi^{k+1}(\cdot|s), q^k(s, \cdot) \rangle \leq \zeta 2\bar{V}. \tag{I.13}$$

Then we proceed to bound

$$\sum_{k=1}^{\bar{K}} \left\langle \pi - \pi^k(\cdot|s), q^k(s,\cdot) \right\rangle \leq \sum_{k=1}^{\bar{K}} \left\langle \pi(\cdot|s), q^k(s,\cdot) \right\rangle - \mathcal{H}_{\text{NegEnt}}(\pi(\cdot|s))$$

$$- \left( \sum_{k=1}^{\bar{K}} \left\langle \pi^k(\cdot|s), q^k(s,\cdot) \right\rangle - \mathcal{H}_{\text{NegEnt}}(\pi(\cdot|s)) \right)$$

$$= \sum_{k=1}^{\bar{K}} \left\langle \pi(\cdot|s) - \pi^{k+1}(\cdot|s), q^k(s,\cdot) \right\rangle$$

$$+ \sum_{k=1}^{\bar{K}} \left\langle \pi^{k+1}(\cdot|s), q^k(s,\cdot) \right\rangle - \mathcal{H}_{\text{NegEnt}}(\pi(\cdot|s))$$

$$- \left( \sum_{k=1}^{\bar{K}} \left\langle \pi^k(\cdot|s), q^k(s,\cdot) \right\rangle - \mathcal{H}_{\text{NegEnt}}(\pi(\cdot|s)) \right).$$

Leverage Lemma I.7,

$$\sum_{k=1}^{\bar{K}} \left\langle \pi - \pi^k(\cdot|s), q^k(s,\cdot) \right\rangle \leq \mathcal{H}_{\text{NegEnt}}(\pi(\cdot|s)) - \mathcal{H}_{\text{NegEnt}}(\pi^0(\cdot|s))$$

$$+ \sum_{k=1}^{\bar{K}} \left\langle \pi^{k+1}(\cdot|s), q^k(s,\cdot) \right\rangle - \mathcal{H}_{\text{NegEnt}}(\pi(\cdot|s))$$

$$- \left( \sum_{k=1}^{\bar{K}} \left\langle \pi^k(\cdot|s), q^k(s,\cdot) \right\rangle - \mathcal{H}_{\text{NegEnt}}(\pi(\cdot|s)) \right)$$

$$= \sum_{k=1}^{\bar{K}} \left\langle \pi^{k+1}(\cdot|s), q^k(s,\cdot) \right\rangle - \sum_{k=1}^{\bar{K}} \left\langle \pi^k(\cdot|s), q^k(s,\cdot) \right\rangle$$

$$- \mathcal{H}_{\text{NegEnt}}(\pi^0(\cdot|s)) + \mathcal{H}_{\text{NegEnt}}(\pi(\cdot|s))$$

$$\leq \sum_{k=1}^{\bar{K}} \left\langle \pi^{k+1}(\cdot|s), q^k(s,\cdot) \right\rangle - \sum_{k=1}^{\bar{K}} \left\langle \pi^k(\cdot|s), q^k(s,\cdot) \right\rangle$$

$$- \mathcal{H}_{\text{NegEnt}}(\pi^0(\cdot|s)).$$

Combine with the inequality (I.13), we have

$$\sum_{k=1}^{\bar{K}} \left\langle \pi - \pi^k(\cdot|s), q^k(s,\cdot) \right\rangle \leq \sum_{k=1}^{\bar{K}} \zeta 2\bar{V} - \frac{1}{\bar{K}} \mathcal{H}_{\text{NegEnt}}(\pi^0(\cdot|s))$$

$$\leq \zeta^{-1} (\zeta^2 2\bar{V}\bar{K} - \log \frac{1}{|\mathcal{A}|}).$$

Minimizing the $\zeta^{-1}(2\zeta^2 \bar{V}\bar{K} + \log|\mathcal{A}|)$ over $\zeta$, we set $\zeta = \sqrt{\frac{\log|\mathcal{A}|}{\bar{K}2\bar{V}}}$ and thus

$$\sum_{k=1}^{\bar{K}} \left\langle \pi - \pi^k(\cdot|s), q^k(s,\cdot) \right\rangle \leq \sqrt{2\bar{V}\bar{K}\log|\mathcal{A}|} + \frac{\sqrt{2\bar{V}\bar{K}}\log|\mathcal{A}|}{\sqrt{\log|\mathcal{A}|}}$$

$$= 2\sqrt{2\bar{V}\bar{K}\log|\mathcal{A}|}. \tag{I.14}$$

$\square$

## I.9   Proof of Lemma I.9

**Lemma I.9.** *For any policy $\pi$, the average regret*

$$\frac{1}{\bar{K}}\sum_{k=1}^{\bar{K}}\left\{J(\pi;\{\mathcal{S},\mathcal{A},\mathbb{P}_k,\gamma,r_k,s^0\})-J(\pi^k;\{\mathcal{S},\mathcal{A},\mathbb{P}_k,\gamma,r_k,s^0\})\right\}\leq\frac{2\sqrt{2\bar{V}\log|\mathcal{A}|}}{\sqrt{\bar{K}}(1-\gamma)}.$$

*Proof.* To faciliate the proof, we denote $\mathbb{E}^k[\cdot]$ is the expectation taken to the system of iterated MDP $\{\mathcal{S},\mathcal{A},\mathbb{P}_k,\gamma,r_k,s^0\}$. It follows the definitions of the discounted return, we have

$$\frac{1}{\bar{K}}\sum_{k=1}^{\bar{K}}J(\pi;\{\mathcal{S},\mathcal{A},\mathbb{P}_k,\gamma,r_k,s^0\})-J(\pi^k;\{\mathcal{S},\mathcal{A},\mathbb{P}_k,\gamma,r_k,s^0\})$$

$$=\frac{\frac{1}{\bar{K}}\sum_{k=1}^{\bar{K}}\mathbb{E}^k_{d^\pi}[q^k(s,\pi)-q^k(s,\pi^k)]}{1-\gamma}=\frac{\frac{1}{\bar{K}}\sum_{k=1}^{\bar{K}}\mathbb{E}^k_{d^\pi}\left[\langle q^k(s,\cdot),\pi(\cdot|s)-\pi^k(\cdot|s)\rangle\right]}{1-\gamma}$$

As the dynamics of $\{\mathcal{S},\mathcal{A},\mathbb{P}_k,\gamma,r_k,s^0\}$ is identical to $\{\mathcal{S},\mathcal{A},\mathbb{P},\gamma,r,s^0\}$ except for the reward functions. Therefore,

$$\frac{\frac{1}{\bar{K}}\sum_{k=1}^{\bar{K}}\mathbb{E}^k_{d^\pi}\left[\langle q^k(s,\cdot),\pi(\cdot|s)-\pi^k(\cdot|s)\rangle\right]}{1-\gamma}$$

$$=\mathbb{E}_{d^\pi}\left[\frac{\frac{1}{\bar{K}}\sum_{k=1}^{\bar{K}}\langle q^k(s,\cdot),\pi(\cdot|s)-\pi^k(\cdot|s)\rangle}{1-\gamma}\right]\leq\frac{2\sqrt{2\bar{V}\log|\mathcal{A}|}}{\sqrt{\bar{K}}(1-\gamma)}.$$

where the last inequality comes from Lemma I.8. $\qquad\square$

## I.10   Proof of Lemma I.10

**Lemma I.10.** *For any $k\in[\bar{K}]$, we have*

$$\sup_\tau\left\{\frac{1}{(1-\gamma)}\mathbb{E}_\mu\left[\tau(s,a)\left(r(s,a)+\gamma q^k(s',\pi^k)-q^k(s,a)\right)\right]\right\}$$

$$\leq\frac{1}{1-\gamma}\left\{2\left(2\mathcal{U}_2^\star\bar{V}+2\lambda\|\mathbb{D}(\tau(s,a))\|_{L_2(\mu)}^{prime}\right)\sqrt{\frac{2\ln\frac{\mathrm{Vol}(\Theta^\dagger)}{\delta}}{n}}\right.$$

$$\left.+\frac{4\left(2\mathcal{U}_\infty\mathcal{U}_2^\star\bar{V}+2\lambda\|\mathbb{D}(\tau(s,a))\|_{L_\infty}^{prime}\right)\ln\frac{\mathrm{Vol}(\Theta^\dagger)}{\delta}}{3n}+\mathcal{U}_\infty\mathcal{U}_2^\star\sqrt{\varepsilon_\mathcal{Q}}\right\}$$

$$+\mathcal{O}\left(\frac{\lambda\mathcal{U}_\infty\mathcal{U}_2^\star\frac{\sqrt{\varepsilon_\mathcal{Q}}}{1-\gamma}}{c^*}\right)+\frac{\bar{V}}{c^*}+\mathcal{O}(\frac{\bar{V}\mathcal{U}_\infty\mathcal{U}_2^\star}{n})+\mathcal{O}(\frac{\bar{V}\mathcal{U}_\infty\mathcal{U}_2^\star(1+\gamma)}{n})+\mathcal{O}(\frac{\gamma\bar{V}}{n})+\mathcal{O}\left(\frac{1}{\sqrt{n}}\right),$$

*where $\mathrm{Vol}(\Theta^\dagger)=(e^D\max\{D_\Omega,D_\mathcal{Q},D_\Pi\}+1)^3(\{1\vee L\}\mathcal{U}_2^\tau)^{2D}$ for $D=D_\Omega+D_\mathcal{Q}+D_\Pi$.*

*Proof.* To complete the proof, it suffices to show

$$\sup_\tau\left\{\frac{1}{(1-\gamma)n}\Big|\sum_{i=1}^n\tau(s_i,a_i)\left(q^k(s_i,a_i)-r_i-\gamma q^k(s'_i,\pi^k)\right)\Big|\right\},$$

is upper bounded for any $k\in[\bar{K}]$. To facilitate the proof, we define

$$\widetilde{\tau}^k:=\arg\max_\tau\left\{\widetilde{q}^{\pi^k}(s^0,\pi^k)+\frac{c^*}{(1-\gamma)}\mathbb{E}_\mu\left[\tau(s,a)\left(\widetilde{q}^{\pi^k}(s,a)-r_i-\gamma\widetilde{q}^{\pi^k}(s',\pi^k)\right)\right]-\frac{\lambda}{1-\gamma}\mathbb{E}_\mu[\mathbb{D}(\tau(s,a))]\right\},$$

and define

$$\Delta_1:=q^k(s^0,\pi^k)+\sup_\tau\left\{\frac{c^*}{(1-\gamma)n}\Big|\sum_{i=1}^n\tau(s_i,a_i)\left(q^k(s_i,a_i)-r_i-\gamma q^k(s'_i,\pi^k)\right)\Big|\right\}.$$

Then obviously we have

$$\Delta_1 = \Delta_1 - \lambda\xi_n\big(\widetilde{\tau}^k(s_i,a_i)\big) + \lambda\xi_n\big(\widetilde{\tau}^k(s_i,a_i)\big)$$

$$\leq q^k(s^0,\pi^k) + \sup_\tau\left\{\frac{c^*}{(1-\gamma)n}\Big|\sum_{i=1}^n \tau(s_i,a_i)\big(q^k(s_i,a_i) - r_i - \gamma q^k(s_i',\pi^k)\big)\Big| - \lambda\xi_n\big(\tau(s_i,a_i)\big)\right\}$$

$$+ \lambda\xi_n\big(\widetilde{\tau}^k(s_i,a_i)\big).$$

where the first inequality comes from $\widetilde{\tau}^k$ is not the maximizer of

$$\frac{c^*}{(1-\gamma)n}\Big|\sum_{i=1}^n \tau(s_i,a_i)\big(q^k(s_i,a_i) - r_i - \gamma q^k(s_i',\pi^k)\big)\Big| - \lambda\xi_n\big(\tau(s_i,a_i)\big).$$

By the definition of $q^k(\cdot,\cdot)$, then we have

$$q^k(s^0,\pi^k) + \sup_\tau\left\{\frac{c^*}{(1-\gamma)n}\Big|\sum_{i=1}^n \tau(s_i,a_i)\big(q^k(s_i,a_i) - r_i - \gamma q^k(s_i',\pi^k)\big)\Big| - \lambda\xi_n\big(\tau(s_i,a_i)\big)\right\}$$

$$+ \lambda\xi_n\big(\widetilde{\tau}^k(s_i,a_i)\big)$$

$$= \min_q\left\{q(s^0,\pi^k) + \sup_\tau\left\{\frac{c^*}{(1-\gamma)n}\Big|\sum_{i=1}^n \tau(s_i,a_i)\big(q_\theta(s_i,a_i) - r_i - \gamma q(s_i',\pi^k)\big)\Big|\right.\right.$$

$$\left.\left. - \lambda\xi_n\big(\tau(s_i,a_i)\big)\right\}\right\} + \lambda\xi_n\big(\widetilde{\tau}^k(s_i,a_i)\big)$$

As $\widetilde{q}^{\pi^k}(\cdot,\cdot)$ belongs to the function space associated with $q$, so

$$\min_q\left\{q(s^0,\pi^k) + \sup_\tau\left\{\frac{c^*}{(1-\gamma)n}\Big|\sum_{i=1}^n \tau(s_i,a_i)\big(q(s_i,a_i) - r_i - \gamma q(s_i',\pi^k)\big)\Big| - \lambda\xi_n\big(\tau(s_i,a_i)\big)\right\}\right\}$$

$$\leq \widetilde{q}^{\pi^k}(s^0,\pi^k) + \sup_\tau\left\{\frac{c^*}{(1-\gamma)n}\Big|\sum_{i=1}^n \tau(s_i,a_i)\big(\widetilde{q}^{\pi^k}(s_i,a_i) - r_i - \gamma\widetilde{q}^{\pi^k}(s_i',\pi^k)\big)\Big| - \lambda\xi_n\big(\tau(s_i,a_i)\big)\right\}$$

$$\leq \widetilde{q}^{\pi^k}(s^0,\pi^k) + \sup_\tau\left\{\frac{c^*}{(1-\gamma)n}\Big|\sum_{i=1}^n \tau(s_i,a_i)\big(\widetilde{q}^{\pi^k}(s_i,a_i) - r_i - \gamma\widetilde{q}^{\pi^k}(s_i',\pi^k)\big)\Big|\right\}$$

where the last inequality comes from the second inequality comes from the non-negativity of $\xi_n(\cdot)$. This immediately implies that

$$\Delta_1 \leq \widetilde{q}^{\pi^k}(s^0,\pi^k) + \sup_\tau\left\{\frac{c^*}{(1-\gamma)n}\Big|\sum_{i=1}^n \tau_\tau(s_i,a_i)\big(\widetilde{q}^{\pi^k}(s_i,a_i) - r_i - \gamma\widetilde{q}^{\pi^k}(s_i',\pi^k)\big)\Big|\right\} + \lambda\xi_n\big(\widetilde{\tau}^k(s_i,a_i)\big)$$

$$\leq \bar{V} + \sup_\tau\left\{\frac{c^*}{(1-\gamma)n}\Big|\sum_{i=1}^n \tau_\tau(s_i,a_i)\big(\widetilde{q}^{\pi^k}(s_i,a_i) - r_i - \gamma\widetilde{q}^{\pi^k}(s_i',\pi^k)\big)\Big|\right\} + \lambda\xi_n\big(\widetilde{\tau}^k(s_i,a_i)\big).$$

Follow the proof of Lemma I.6, we have

$$\sup_\tau\left\{\frac{c^*}{(1-\gamma)n}\Big|\sum_{i=1}^n \tau(s_i,a_i)\big(\widetilde{q}^{\pi^k}(s_i,a_i) - r_i - \gamma\widetilde{q}^{\pi^k}(s_i',\pi^k)\big)\Big|\right\}$$

$$\leq \frac{c^*}{1-\gamma}\left\{\big(3\mathcal{U}_2^\star\bar{V} + 2\lambda\|\mathbb{D}(\tau(s,a))\|_{L_2(\mu)}^{\text{prime}}\big)\sqrt{\frac{2\ln\frac{\text{Vol}(\Theta^\dagger)}{\delta}}{n}}\right.$$

$$\left. + \frac{2\big(3\mathcal{U}_\infty\mathcal{U}_2^\star\bar{V} + 2\lambda\|\mathbb{D}(\tau(s,a))\|_{L_\infty}^{\text{prime}}\big)\ln\frac{\text{Vol}(\Theta^\dagger)}{\delta}}{3n} + \mathcal{U}_\infty\mathcal{U}_2^\star\sqrt{\varepsilon_\mathcal{Q}}\right\} := \varepsilon_{3,n},$$

for $\mathrm{Vol}(\Theta^\dagger) = (e^D \max\{D_\Omega, D_\mathcal{Q}, D_\Pi\} + 1)^3 (\{1 \vee L\}\mathcal{U}_2^\tau)^{2D}$ for $D = D_\Omega + D_\mathcal{Q} + D_\Pi$. And by inequality (I.7), we have $\lambda \xi_n\big(\tilde{\tau}^k(s_i, a_i)\big) \lesssim \mathcal{O}\big(\lambda \mathcal{U}_\infty \mathcal{U}_2^\star \sqrt{\varepsilon_\mathcal{Q}}\big) + \mathcal{O}(\frac{1}{\sqrt{n}})$. Then, we conclude that

$$q^k(s^0, \pi^k) + \sup_\tau \left\{ \frac{c^*}{(1-\gamma)n} \Big| \sum_{i=1}^n \tau(s_i, a_i)\big(q^k(s_i, a_i) - r_i - \gamma q^k(s_i', \pi^k)\big) \Big| \right\}$$

$$\lesssim \varepsilon_{3,n} + \mathcal{O}\left(\lambda \mathcal{U}_\infty \mathcal{U}_2^\star \frac{\sqrt{\varepsilon_\mathcal{Q}}}{1-\gamma}\right) + \mathcal{O}(\frac{1}{\sqrt{n}}).$$

With the boundedness condition on $q^k(s^0, \pi^k) \in [-\bar{V}, \bar{V}]$, by some algebra, then

$$\sup_\tau \left\{ \frac{c^*}{(1-\gamma)n} \Big| \sum_{i=1}^n \tau(s_i, a_i)\big(q^k(s_i, a_i) - r_i - \gamma q^k(s_i', \pi^k)\big) \Big| \right\}$$

$$\leq \varepsilon_{3,n} + \mathcal{O}\left(\lambda \mathcal{U}_\infty \mathcal{U}_2^\star \frac{\sqrt{\varepsilon_\mathcal{Q}}}{1-\gamma}\right) + \mathcal{O}(\frac{1}{\sqrt{n}}) + \bar{V}. \tag{I.15}$$

Therefore, we can conclude that

$$\sup_\tau \left\{ \frac{1}{(1-\gamma)n} \Big| \sum_{i=1}^n \tau(s_i, a_i)\big(q^k(s_i, a_i) - r_i - \gamma q^k(s_i', \pi^k)\big) \Big| \right\}$$

$$\leq \frac{1}{1-\gamma} \Bigg\{ \big(3\mathcal{U}_2^\star \bar{V} + 2\lambda \|\mathbb{D}(\tau(s,a))\|_{L_2(\mu)}^{\mathrm{prime}}\big) \sqrt{\frac{2 \ln \frac{\mathrm{Vol}(\Theta^\dagger)}{\delta}}{n}}$$

$$+ \frac{2\big(3\mathcal{U}_\infty \mathcal{U}_2^\star \bar{V} + 2\lambda \|\mathbb{D}(\tau(s,a))\|_{L_\infty}^{\mathrm{prime}}\big) \ln \frac{\mathrm{Vol}(\Theta^\dagger)}{\delta}}{3n} + \mathcal{U}_\infty \mathcal{U}_2^\star \sqrt{\varepsilon_\mathcal{Q}} \Bigg\} \tag{I.16}$$

$$+ \mathcal{O}\left(\frac{\lambda \mathcal{U}_\infty \mathcal{U}_2^\star \frac{\sqrt{\varepsilon_\mathcal{Q}}}{1-\gamma}}{c^*}\right) + \mathcal{O}(\frac{1}{\sqrt{n}}) + \frac{\bar{V}}{c^*}. \tag{I.17}$$

According to Lemma E.2, we have for any $\tau$,

$$\frac{1}{(1-\gamma)} \Big| \mathbb{E}_\mu \big[\tau(s,a)\big(q^k(s,a) - r(s,a) - \gamma q^k(s', \pi^k)\big)\big] \Big|$$

$$\leq \frac{1}{(1-\gamma)n} \Big| \sum_{i=1}^n \tau(s_i, a_i)\big(q^k(s_i, a_i) - r_i - \gamma q^k(s_i', \pi^k)\big) \Big|$$

$$+ \frac{1}{1-\gamma} \left\{ \mathcal{U}_2^\star \sqrt{\frac{2\bar{V}^2 \ln \frac{\mathrm{Vol}(\Theta^\dagger)}{\delta}}{n}} + \frac{2\mathcal{U}_\infty \mathcal{U}_2^\star \bar{V} \ln \frac{\mathrm{Vol}(\Theta^\dagger)}{\delta}}{3n} \right\}$$

$$+ \mathcal{O}(\frac{\bar{V} \mathcal{U}_\infty \mathcal{U}_2^\star}{n}) + \mathcal{O}(\frac{\bar{V} \mathcal{U}_\infty \mathcal{U}_2^\star (1+\gamma)}{n}) + \mathcal{O}(\frac{\gamma \bar{V}}{n}).$$

Combine with the bound (I.17), we conclude that

$$\sup_\psi \left\{ \frac{1}{(1-\gamma)} \mathbb{E}_\mu \big[\tau_\psi(s,a)\big(r(s,a) + \gamma q^k(s', \pi^k) - q^k(s,a)\big)\big] \right\}$$

$$\leq \sup_\psi \left\{ \frac{1}{(1-\gamma)} \Big| \mathbb{E}_\mu \big[\tau_\psi(s,a)\big(q^k(s,a) - r(s,a) - \gamma q^k(s', \pi^k)\big)\big] \Big| \right\}$$

$$\leq \frac{1}{1-\gamma} \Bigg\{ 2\big(2\mathcal{U}_2^\star \bar{V} + 2\lambda \|\mathbb{D}(\tau(s,a))\|_{L_2(\mu)}^{\mathrm{prime}}\big) \sqrt{\frac{2 \ln \frac{\mathrm{Vol}(\Theta^\dagger)}{\delta}}{n}}$$

$$+ \frac{4\big(2\mathcal{U}_\infty \mathcal{U}_2^\star \bar{V} + 2\lambda \|\mathbb{D}(\tau(s,a))\|_{L_\infty}^{\mathrm{prime}}\big) \ln \frac{\mathrm{Vol}(\Theta^\dagger)}{\delta}}{3n} + \mathcal{U}_\infty \mathcal{U}_2^\star \sqrt{\varepsilon_\mathcal{Q}} \Bigg\}$$

$$+ \mathcal{O}\left(\frac{\lambda \mathcal{U}_\infty \mathcal{U}_2^\star \frac{\sqrt{\varepsilon_\mathcal{Q}}}{1-\gamma}}{c^*}\right) + \frac{\bar{V}}{c^*} + \mathcal{O}(\frac{\bar{V} \mathcal{U}_\infty \mathcal{U}_2^\star}{n}) + \mathcal{O}(\frac{\bar{V} \mathcal{U}_\infty \mathcal{U}_2^\star (1+\gamma)}{n}) + \mathcal{O}(\frac{\gamma \bar{V}}{n}) + \mathcal{O}\left(\frac{1}{\sqrt{n}}\right).$$

This completes the proof $\qquad\qquad\qquad\qquad\qquad\qquad\qquad\qquad\qquad\qquad\qquad\qquad\qquad\qquad\square$

## I.11 Proof of Theorem 5.1

*Proof.* Let the policy $\widehat{\pi}$ be the output of the penalized adversarial in Algorithm 1 of maintext. In this proof, we aim to bound the regret

$$J(\pi) - J(\widehat{\pi}).$$

First, we note that $\widehat{\pi}$ is a mixed policy over $\{\pi^k\}_{k=1}^{\bar{K}}$, then we follow Theorem 1 in [57] to decompose the discounted return of $\widehat{\pi}$, i.e., $J(\widehat{\pi})$, that is, $J(\widehat{\pi}) = \frac{1}{\bar{K}} \sum_{k=1}^{\bar{K}} J(\pi^k)$. Based on this, it suffices to bound

$$J(\pi) - \frac{1}{\bar{K}} \sum_{k=1}^{\bar{K}} J(\pi^k) = \frac{1}{\bar{K}} \sum_{k=1}^{\bar{K}} \left( J(\pi) - J(\pi^k) \right). \tag{I.18}$$

By Lemma I.1 for that the learned $q$-function at the $k$-th iteration $q^k$ is the true action-value function under the policy $\pi^k$ in the iterative MDP $\{ \mathcal{S}, \mathcal{A}, \mathbb{P}_k, \gamma, r_k, s^0 \}$. The regret (I.18) can be further decomposed as follows:

$$\frac{1}{\bar{K}} \sum_{k=1}^{\bar{K}} \left( J(\pi) - J(\pi^k) \right) = \underbrace{\frac{1}{\bar{K}} \sum_{k=1}^{\bar{K}} \left( J(\pi^k; \{ \mathcal{S}, \mathcal{A}, \mathbb{P}_k, \gamma, r_k, s^0 \}) - J(\pi^k) \right)}_{\Delta_1}$$

$$+ \underbrace{\frac{1}{\bar{K}} \sum_{k=1}^{\bar{K}} \left( J(\pi; \{ \mathcal{S}, \mathcal{A}, \mathbb{P}_k, \gamma, r_k, s^0 \}) - J(\pi^k; \{ \mathcal{S}, \mathcal{A}, \mathbb{P}_k, \gamma, r_k, s^0 \}) \right)}_{\Delta_2}$$

$$+ \underbrace{\frac{1}{\bar{K}} \sum_{k=1}^{\bar{K}} \left( J(\pi) - J(\pi; \{ \mathcal{S}, \mathcal{A}, \mathbb{P}_k, \gamma, r_k, s^0 \}) \right)}_{\Delta_3} \tag{I.19}$$

Based on this error decomposition, it suffices to upper-bound the above three terms. In analysis, first, $\Delta_1$ is the regret over true MDP and iterative MDP for policy $\pi^k$. Second, $\Delta_2$ is the regret over policy $\pi^k$ and $\pi$ under iterative MDP. Third, $\Delta_3$ is the regret over true MDP and iterative MDP for policy $\pi$. In the following, we bound each term subsequently.

**Bounding $\Delta_1$.** According to Lemma I.6, we have $\Delta_1$ is upper bounded by

$$\Delta_1 \leq \frac{c^*}{1-\gamma} \Bigg\{ \left( 3\mathcal{U}_2^\star \bar{V} + 2\lambda \| \mathbb{D}(\tau(s,a)) \|_{L_2(\mu)}^{\text{prime}} \right) \sqrt{\frac{2\ln \frac{\text{Vol}(\Theta^\dagger)}{\delta}}{n}}$$

$$+ \frac{2\left( 3\mathcal{U}_\infty \mathcal{U}_2^\star \bar{V} + 2\lambda \| \mathbb{D}(\tau(s,a)) \|_{L_\infty}^{\text{prime}} \right) \ln \frac{\text{Vol}(\Theta^\dagger)}{\delta}}{3n} + \mathcal{U}_\infty \mathcal{U}_2^\star \sqrt{\varepsilon_{\mathcal{Q}}} \Bigg\} \tag{I.20}$$

$$+ \mathcal{O}\left( (1 + \lambda \mathcal{U}_\infty \mathcal{U}_2^\star) \frac{\sqrt{\varepsilon_{\mathcal{Q}}}}{1-\gamma} \right) + \mathcal{O}(\frac{1}{\sqrt{n}}). \tag{I.21}$$

**Bounding $\Delta_2$.** For this term, it is concerned with the optimization error. According to Lemma I.9, our algorithm achieves a no-regret oracle, and the optimization error can be minimized by increasing the rounds of optimization, i.e., increasing $bar K$.

$$\Delta_2 \leq \frac{2\sqrt{2\bar{V} \log |\mathcal{A}|}}{\sqrt{\bar{K}}(1-\gamma)}. \tag{I.22}$$

**Bounding $\Delta_3$.** To bound $\Delta_3$, it suffices to bound $J(\pi) - J(\pi; \{ \mathcal{S}, \mathcal{A}, \mathbb{P}_k, \gamma, r_k, s^0 \})$, for any $k \in [\bar{K}]$. we define admissible implicit exploratory distribution as $\rho_k$ that satisfies, which essentially can be

induced and controlled via penalization on the detection function through $\lambda$ in Algorithm 1, i.e.,

$$J(\pi) - J(\pi; \{\mathcal{S}, \mathcal{A}, \mathbb{P}_k, \gamma, r_k, s^0\}) \tag{I.23}$$

$$= q^\pi(s^0, \pi) - J(\pi; \{\mathcal{S}, \mathcal{A}, \mathbb{P}_k, \gamma, r_k, s^0\}) = \frac{\mathbb{E}_{d^\pi}\left[q^\pi(s,a) - r_k(s,a) - \gamma q^\pi(s,\pi)\right]}{1-\gamma}$$

$$= \underbrace{\frac{1}{1-\gamma}\mathbb{E}_\mu\left[\frac{\rho_k(s,a)}{\mu(s,a)}\left[q^\pi(s,a) - r_k(s,a) - \gamma q^\pi(s,\pi)\right]\right]}_{\Delta_{31}} - \underbrace{\frac{1}{1-\gamma}\mathbb{E}_{\rho_k}\left[q^\pi(s,a) - r_k(s,a) - \gamma q^\pi(s,\pi)\right]}_{\Delta_{32}}$$

$$+ \underbrace{\frac{1}{1-\gamma}\mathbb{E}_{d^\pi}\left[q^\pi(s,a) - r_k(s,a) - \gamma q^\pi(s,\pi)\right]}_{\Delta_{33}}. \tag{I.24}$$

Accordingly, we can make a mirror decomposition as in the proof of Theorem 4.1. The difference is, instead of controlling the uncertainty level through constrained set $\widetilde{\Omega}_{\widetilde{\sigma}_n}$, in this penalization adversarial algorithm, the uncertainty level is controlled via penalization. To proceed with the proof, we first study and connect the penalized uncertainty control to constrained uncertainty control. According to Lemma I.4, we have

$$\|\tau(s,a)\|_{L_2(\mu)} \leq \frac{1}{\sqrt{M}}\left\{L\mathcal{U}_2^\tau\sqrt{\frac{2\ln\frac{\mathrm{Vol}(\mathcal{G}^\mathbb{D})}{\delta}}{n}} + 2\sqrt{\frac{L\mathcal{U}_\infty\mathcal{U}_2^\tau\ln\frac{\mathrm{Vol}(\mathcal{G}^\mathbb{D})}{\delta}}{3n}} + \sqrt{2\varepsilon_n^\mathbb{D}} + \sqrt{M}\right\}. \tag{I.25}$$

where we can determine $\varepsilon_n^\mathbb{D}$ using Lemma I.5. This implies that we can well control $\lambda$ even in the penalization adversarial estimation to control the uncertainty level in the form of $\|\tau(s,a)\|_{L_2(\mu)}$ for $\tau \in \Omega$, i.e., $\|\tau(s,a)\|_{L_2(\mu)} \leq \mathcal{U}_2^\star$ for $\mathcal{U}_2^\star$ depending on $\lambda$. Throughout the rest of the proof, it is sufficient to proceed with the condition on $\|\tau(s,a)\|_{L_2(\mu)} \leq \mathcal{U}_2^\star$.

**Bounding $\Delta_{31}$.** We define the $\Omega$ sub-class that $\widetilde{\Omega} = \{\tau : \|\tau(s,a)\|_{L_2(\mu)} \leq \mathcal{U}_2^\star, \tau \in \Omega\}$, and define a importance-weight estimator over $\rho_k$:

$$\tau_{\rho_k/\mu}(s,a) := \argmin_{\tau \in \text{lr-hull}(\widetilde{\Omega})} \frac{1}{1-\gamma}\mathbb{E}_\mu\left[\left(\frac{\rho_k(s,a)}{\mu(s,a)} - \tau(s,a)\right)\left[q^\pi(s,a) - r_k(s,a) - \gamma q^\pi(s,\pi)\right]\right],$$

Then we make the following error decomposition for $\Delta_{31}$ as

$$\Delta_{31} = \frac{1}{1-\gamma}\mathbb{E}_\mu\left[\left(\frac{\rho_k(s,a)}{\mu(s,a)} - \tau_{\rho_k/\mu}(s,a)\right)\left[q^\pi(s,a) - r_k(s,a) - \gamma q^\pi(s,\pi)\right]\right]$$

$$+ \frac{1}{1-\gamma}\mathbb{E}_\mu\left[\tau_{\rho_k/\mu}(s,a)\left[q^\pi(s,a) - r_k(s,a) - \gamma q^\pi(s,\pi)\right]\right]$$

$$\leq \underbrace{\frac{\mathbb{E}_\mu\left[\left(\frac{\rho_k(s,a)}{\mu(s,a)} - \tau_{\rho_k/\mu}(s,a)\right)\left[q^\pi(s,a) - r_k(s,a) - \gamma q^\pi(s,\pi)\right]\right]}{1-\gamma}}_{\Delta_{311}}$$

$$+ \underbrace{\sup_{\tau \in \text{lr-hull}(\Omega, C_{\tau,\rho_k}, \mathcal{U}_2^\star)} \frac{1}{1-\gamma}\mathbb{E}_\mu\left[\tau(s,a)\left[q^\pi(s,a) - r_k(s,a) - \gamma q^\pi(s,\pi)\right]\right]}_{\Delta_{312}}.$$

**Bounding $\Delta_{311}$.** Follow the definition of $r_k$ in Lemma I.1, it observes that

$$q^\pi(s,a) - r_k(s,a) - \gamma q^\pi(s,\pi) = r(s,a) - r_k(s,a)$$

$$= r(s,a) - q^k(s,a) + \gamma\mathbb{E}_{s'\sim\mathbb{P}(\cdot|s,a)}\left[\sum_{a'\in\mathcal{A}}\pi^k(a'|s')q^k(s',a')\right]$$

$$= r(s,a) + \gamma q^k(s',\pi^k) - q^k(s,a). \tag{I.26}$$

Then to bound $\Delta_{311}$ it suffices to bound

$$\frac{\mathbb{E}_\mu\left[\left(\frac{\rho_k(s,a)}{\mu(s,a)} - \tau_{\rho_k/\mu}(s,a)\right)\left[r(s,a) + \gamma q^k(s',\pi^k) - q^k(s,a)\right]\right]}{1-\gamma}.$$

It observes that

$$
\sup_{\tau} \left\{ \frac{\left| \mathbb{E}_\mu \left[ \tau(s,a) \left( q^k(s,a) - r(s,a) - \gamma q^k(s',\pi^k) \right) \right] \right|}{1-\gamma} \right\}
$$

$$
= \sup_{\tau} \left\{ \frac{\left| -\mathbb{E}_\mu \left[ \tau(s,a) \left( q^k(s,a) - r(s,a) - \gamma q^k(s',\pi^k) \right) \right] \right|}{1-\gamma} \right\}
$$

$$
= \sup_{\tau} \left\{ \frac{\left| \mathbb{E}_\mu \left[ \tau(s,a) \left( r(s,a) + \gamma q^k(s',\pi^k) - q^k(s,a) \right) \right] \right|}{1-\gamma} \right\}.
$$

Then we apply Lemma I.10, with control on $\tau$ as $\mathcal{U}_2^\star$ for $L_2$ boundedness and $\mathcal{U}_\infty \mathcal{U}_2^\star$ for $L_\infty$. Then, we have

$$
\sup_{\tau} \left\{ \frac{\mathbb{E}_\mu \left[ \tau(s,a) \left( r(s,a) + \gamma q^k(s',\pi^k) - q^k(s,a) \right) \right]}{1-\gamma} \right\} \leq \varepsilon_n^4.
$$

where $\varepsilon_n^4$ is the upper bound as in Lemma I.10. Now, as $\|\tau(s,a)\|_{L_2(\mu)} \leq \mathcal{U}_2^\star$. According to Lemma E.3, we have

$$
\frac{\left\| r(s,a) + \gamma q^k(s',\pi^k) - q^k(s,a) \right\|_{L_2(\mu)}}{(1-\gamma)} \leq \frac{\varepsilon_n^4}{\mathcal{U}_2^\star}. \tag{I.27}
$$

Also, due to the non-negativity of $\frac{\rho_k(s,a)}{\mu(s,a)}$ and $\tau_{\rho_k/\mu}(s,a)$ for any $(s,a)$ over the support on $\mu$, we have

$$
\mathbb{E}_\mu \left[ \left( \frac{\rho_k(s,a)}{\mu(s,a)} - \tau_{\rho_k/\mu}(s,a) \right)^2 \right] \leq \min \left\{ \mathbb{E}_\mu \left[ \left( \frac{\rho_k(s,a)}{\mu(s,a)} \right)^2 \right], \mathcal{U}_2^\star \right\}. \tag{I.28}
$$

By Cauchy-schwarz inequality, and combine with inequalities (I.27) and (I.28), we conclude that

$$
\Delta_{311} \leq \frac{\sqrt{\min \left\{ \mathbb{E}_\mu \left[ \left( \frac{\rho_k(s,a)}{\mu(s,a)} \right)^2 \right], \mathcal{U}_2^\star \right\}}}{1-\gamma} \frac{\varepsilon_n^4}{\mathcal{U}_2^\star} := \varepsilon_n^{4,\prime}. \tag{I.29}
$$

We can plug-in $\varepsilon_n^4$ to complete the upper bound for $\Delta_{311}$.

**Bounding $\Delta_{312}$.** According to (I.25). The supermum boundedness condition for $\tau$ can be identified over the class lr-hull($\widetilde{\Omega}$):

$$
\|\tau(s,a)\|_{L_\infty} \leq \mathcal{U}_\infty \mathcal{U}_2^\star. \tag{I.30}
$$

for some constant $\mathcal{U}_\infty$ as described in [56]. It follows from Lemma I.10, under the boundedness conditions on $\tau$ in (I.30), we can conclude that

$$
\Delta_{312} \leq \varepsilon_n^5. \tag{I.31}
$$

where $\varepsilon_n^5$ is defined as the upper bound term in Lemma I.10.

Combine with the upper bounds on $\Delta_{311}$ and $\Delta_{312}$ in (I.29) and (I.31), we have

$$
\Delta_{31} \leq \varepsilon_n^{4,\prime} + \varepsilon_n^5.
$$

This completes upper bounding $\Delta_{31}$.

**Bounding $\Delta 33 - \Delta 32$.** According to the error decomposition (I.24), it remains to bound $-\Delta 32 + \Delta 33$. We have the upper bound on

$$
(1-\gamma)(\Delta_{33} - \Delta_{32})
$$
$$
= \underbrace{\mathbb{E}_{d^\pi}\left[q^\pi(s,a) - r_k(s,a) - \gamma q^\pi(s,\pi)\right]}_{\Delta_{33}} - \underbrace{\mathbb{E}_{\rho_k}\left[q^\pi(s,a) - r_k(s,a) - \gamma q^\pi(s,\pi)\right]}_{\Delta_{32}}
$$
$$
= \sum_{a\in\mathcal{A}, s\in\mathcal{S}} \mathbb{1}_{\{d_\pi(s,a)-\rho_k(s,a)\geq 0\}}[d_\pi(s,a) - \rho_k(s,a)]\left[q^\pi(s,a) - r_k(s,a) - \gamma q^\pi(s,\pi)\right]
$$
$$
+ \sum_{a\in\mathcal{A}, s\in\mathcal{S}} \mathbb{1}_{\{d_\pi(s,a)-\rho_k(s,a)< 0\}}[d_\pi(s,a) - \rho_k(s,a)]\left[q^\pi(s,a) - r_k(s,a) - \gamma q^\pi(s,\pi)\right]
$$
$$
\leq \underbrace{\sum_{a\in\mathcal{A}, s\in\mathcal{S}} \mathbb{1}_{\{d_\pi(s,a)-\rho_k(s,a)< 0\}}[\rho_k(s,a) - d_\pi(s,a)]\left[q^\pi(s,a) - r_k(s,a) - \gamma q^\pi(s,\pi)\right]}_{\Delta_{331}}
$$
$$
+ \underbrace{\sum_{a\in\mathcal{A}, s\in\mathcal{S}} \mathbb{1}_{\{d_\pi(s,a)-\rho_k(s,a)\geq 0\}}[d_\pi(s,a) - \rho_k(s,a)]\left[q^\pi(s,a) - r_k(s,a) - \gamma q^\pi(s,\pi)\right]}_{\Delta_{332}}.
$$

**Bounding $\Delta_{331}$.** First, we observe that

$$
\Delta_{331} = \sum_{a\in\mathcal{A}, s\in\mathcal{S}} \mathbb{1}_{\{\rho_k(s,a)-d_\pi(s,a)>0\}}[\rho_k(s,a) - d_\pi(s,a)]\left[q^\pi(s,a) - r_k(s,a) - \gamma q^\pi(s,\pi)\right],
$$

which is equivalent to $\sum_{a\in\mathcal{A}, s\in\mathcal{S}} (\rho_k(s,a) - d_\pi(s,a))^+ \left[q^\pi(s,a) - r_k(s,a) - \gamma q^\pi(s,\pi)\right]$. We apply change of measure for shifting to the distribution over $\mu$, i.e,

$$
\sum_{a\in\mathcal{A}, s\in\mathcal{S}} (\rho_k(s,a) - d_\pi(s,a))^+ \left[q^\pi(s,a) - r_k(s,a) - \gamma q^\pi(s,\pi)\right]
$$
$$
= \sum_{a\in\mathcal{A}, s\in\mathcal{S}} \left\{ \frac{(\rho_k(s,a) - d_\pi(s,a))^+}{\mu(s,a)} \left[q^\pi(s,a) - r_k(s,a) - \gamma q^\pi(s,\pi)\right]\cdot\mu(s,a) \right\}
$$
$$
\leq \left\| \frac{(\rho_k(s,a) - d_\pi(s,a))^+}{\mu(s,a)} \right\|_{L_2(\mu)} \|q^\pi(s,a) - r_k(s,a) - \gamma q^\pi(s,\pi)\|_{L_2(\mu)} \tag{I.32}
$$

As for any implicit exploratory distribution $\rho_k$, we have $(\rho_k(s,a) - d_\pi(s,a))^+ \leq \rho_k(s,a)$ for any $s, a$, thus we have

$$
\left\| \frac{(\rho_k(s,a) - d_\pi(s,a))^+}{\mu(s,a)} \right\|_{L_2(\mu)} \leq \left\| \frac{\rho_k(s,a)}{\mu(s,a)} \right\|_{L_2(\mu)} \leq \min\{\mathcal{U}_{2,\rho_k}, \mathcal{U}_2^\star\}, \tag{I.33}
$$

where $\mathcal{U}_{2,\rho_k} := \left\| \frac{\rho_k(s,a)}{\mu(s,a)} \right\|_{L_2(\mu)}$. Upon the inequalities (I.27), (I.26), (I.32), and the observation (I.33), we conclude that

$$
\Delta_{331} \leq \frac{\min\{\mathcal{U}_{2,\rho_k}, \mathcal{U}_2^\star\}\varepsilon_n^4}{\mathcal{U}_2^\star}.
$$

**Bounding $\Delta_{332}$.** With respect to the on-support and off-supprot region: $\mu(s,a) = 0$ and $\mu(s,a) > 0$, we have the following decomposition,

$$
\Delta_{332} = \underbrace{\sum_{a\in\mathcal{A}, s\in\mathcal{S}} \mathbb{1}_{\mu(s,a)>0} (d_\pi(s,a) - \rho(s,a))^+ \left[q^\pi(s,a) - r_k(s,a) - \gamma q^\pi(s,\pi)\right]}_{\Delta_{3321}}
$$
$$
+ \underbrace{\sum_{a\in\mathcal{A}, s\in\mathcal{S}} \mathbb{1}_{\mu(s,a)=0} (d_\pi(s,a) - \rho(s,a))^+ \left[q^\pi(s,a) - r_k(s,a) - \gamma q^\pi(s,\pi)\right]}_{\Delta_{3322}}.
$$

**Bounding** $\Delta_{3321}$**.** According to (I.26) and (I.27), we have

$$\sup_{\{(s,a)\in\mathcal{S}\times\mathcal{A}:\mu(s,a)>0\}}\frac{|r(s,a)+\gamma q^k(s',\pi^k)-q^k(s,a)|}{(1-\gamma)}\leq\frac{\varepsilon_n^4}{\mathcal{U}_2^\star}.$$

Then we conclude that

$$\Delta_{3321}\leq\sum_{a\in\mathcal{A},s\in\mathcal{S}}(d_\pi(s,a)-\rho(s,a))^+\frac{\varepsilon_n^4}{\mathcal{U}_2^\star}.$$

The term $\Delta_{3322}$ is the off-support extrapolation error. Therefore, we conclude that

$$\Delta_{332}\leq\sum_{a\in\mathcal{A},s\in\mathcal{S}}(d_\pi(s,a)-\rho(s,a))^+\frac{\varepsilon_n^4}{\mathcal{U}_2^\star}+\Delta_{3322}.$$

In the following, we conclude that

$$\begin{aligned}
&\Delta_{33}-\Delta_{32}\\
\leq&\frac{\min\{\mathcal{U}_{2,\rho_k},\mathcal{U}_2^\star\}\varepsilon_n^4}{\mathcal{U}_2^\star}+\sum_{a\in\mathcal{A},s\in\mathcal{S}}(d_\pi(s,a)-\rho(s,a))^+\cdot\frac{\varepsilon_n^4}{\mathcal{U}_2^\star}\\
&+\sum_{a\in\mathcal{A},s\in\mathcal{S}}\mathbb{1}_{\mu(s,a)=0}\,(d_\pi(s,a)-\rho(s,a))^+\left[q^\pi(s,a)-r_k(s,a)-\gamma q^\pi(s,\pi)\right]\\
=&\frac{\min\{\mathcal{U}_{2,\rho_k},\mathcal{U}_2^\star\}\varepsilon_n^4}{\mathcal{U}_2^\star}+\sum_{a\in\mathcal{A},s\in\mathcal{S}}(d_\pi(s,a)-\rho(s,a))^+\cdot\frac{\varepsilon_n^4}{\mathcal{U}_2^\star}\\
&+\sum_{a\in\mathcal{A},s\in\mathcal{S}}\mathbb{1}_{\mu(s,a)=0}d_\pi(s,a)\left[q^\pi(s,a)-r_k(s,a)-\gamma q^\pi(s,\pi)\right]
\end{aligned}$$

Combine with the bound on $\Delta_{31}$, we have

$$\begin{aligned}
\Delta_3\leq&\varepsilon_n^{4,\prime}+\varepsilon_n^5+\frac{\min\{\mathcal{U}_{2,\rho_k},\mathcal{U}_2^\star\}\varepsilon_n^4}{\mathcal{U}_2^\star}+\sum_{a\in\mathcal{A},s\in\mathcal{S}}(d_\pi(s,a)-\rho(s,a))^+\cdot\frac{\varepsilon_n^4}{\mathcal{U}_2^\star}\\
&+\sum_{a\in\mathcal{A},s\in\mathcal{S}}\mathbb{1}_{\mu(s,a)=0}\,(d_\pi(s,a)-\rho(s,a))^+\left[q^\pi(s,a)-r_k(s,a)-\gamma q^\pi(s,\pi)\right]\quad\text{(I.34)}
\end{aligned}$$

According to the regret decomposition in (I.19), and the upper bound on $\Delta_1$ in (I.21), the upper bound on $\Delta_2$ in (I.22), and the upper bound on $\Delta_3$ in (I.34), by some algebra, we set $c^*=\widetilde{\mathcal{O}}\big(\sqrt{n\bar{V}/(\lambda L\mathcal{U}_2^\tau\ln\{\mathrm{Vol}(\Theta^\dagger)/\delta\})}\big)$. we set $\lambda$ as the solution of $\lambda=2\mathcal{E}_2/\mathcal{E}_1$ which depends on $\mathcal{U}_2^\star$ in order to ensure the $L_2(\mu)$ norm for uncertainty control. It follows from Lemma I.4 and Lemma I.5, we have $\mathcal{E}_1=\mathcal{O}\big((\sqrt{M}\mathcal{U}_2^\star+(\max\{\mathcal{U}_2^\star,\mathcal{U}_\infty^\tau\}\bar{V}+L\mathcal{U}_2^\star)\sqrt{\ln\{\mathrm{Vol}(\Theta^\dagger)/\delta\}/n})^2\big)$ and $\mathcal{E}_2=\mathcal{O}\big(\bar{V}+(1-\gamma)^{-1/2}\sqrt{\mathcal{U}_2^\star(\bar{V}^2+\lambda L\bar{V})}\sqrt[4]{\ln\{\mathrm{Vol}(\Theta^\dagger)/\delta\}/n}+\sqrt{\mathcal{U}_\infty^\tau(\bar{V}^3+\lambda L\bar{V})}\sqrt{\ln\{\mathrm{Vol}(\Theta^\dagger)/\delta\}/n}+(((\lambda\mathcal{U}_\infty^\tau\varepsilon_\mathcal{Q}+\bar{V}\mathcal{U}_\infty^\tau(1-\gamma)\varepsilon_\mathcal{Q}^{0.5}))/(1-\gamma)^2)^{0.5}\big)$. Plug-in the choice of $\lambda$ and $c^*$, by some algebra, if we further ignore the high-order fast terms using a big-Oh notation $\widetilde{\mathcal{O}}$ and set $\varepsilon_\mathcal{Q}=0$, we conclude that

$$\begin{aligned}
J(\pi)-J(\widehat{\pi})\leq&\frac{1}{1-\gamma}\widetilde{\mathcal{O}}\bigg(\sqrt[4]{\frac{(\mathcal{U}_2^\star)^2\mathfrak{C}_{\bar{V},\lambda,L}^1\ln\{\mathrm{Vol}(\Theta^\dagger)/\delta\}}{n}}+\sqrt{\frac{\bar{V}\log|\mathcal{A}|}{\bar{K}}}\\
&+\frac{1}{\bar{K}}\sum_{k=1}^{\bar{K}}\min_{\rho_k\in\Delta_{\mathcal{U}_2^\star}}\mathbb{E}_{(d_\pi-\rho_k)^+}\bigg[\mathbb{1}_{\mu=0}\Big(\mathcal{B}^{\pi^k}q^k(s,a)-q^k(s,a)\Big)+\mathbb{1}_{\mu>0}\sqrt{\frac{\mathfrak{C}_{\bar{V},\lambda,L}^2\ln\{\mathrm{Vol}(\Theta^\dagger)/\delta\}}{n}}\bigg]\bigg),
\end{aligned}$$

where $\Delta_{\mathcal{U}_2^\star}:=\{\rho_k:\|\frac{\rho_k}{\mu}\|_{L_2(\mu)}<\mathcal{U}_2^\star\}$, $\mathfrak{C}_{\bar{V},\lambda,L}^1,\mathfrak{C}_{\bar{V},\lambda,L}^2$ are some constant terms, and the function class complexity $\mathrm{Vol}(\Theta^\dagger)=(e^D\max\{D_\Omega,D_\mathcal{Q},D_\Pi\}+1)^3(\{1\vee L\}\mathcal{U}_2^\tau)^{2D}$ for $D=D_\Omega+D_\mathcal{Q}+D_\Pi$. This completes the proof. □

# J Proof of Theorem 5.2

## J.1 Proof of Lemma J.1

**Lemma J.1** (Covering number for $\mathcal{Q}_\theta$, $\Pi_\omega$ and $\Omega_\psi$). *For any $\varepsilon \in (0,1]$, the covering number for $\mathcal{Q}_\theta$, $\Pi_\omega$ and $\Omega_\psi$ satisfy the following conditions:*

$$\mathcal{N}\left(\varepsilon/\bar{V}; \mathcal{Q}_\theta\left(diam_\theta\right), \|\cdot\|_{L_2}\right) \leq \left(\frac{2\bar{V}}{\varepsilon} + 1\right)^d,$$

$$\mathcal{N}\left(\varepsilon; \Omega_\psi\left(diam_\psi\right), \|\cdot\|_{L_2}\right) \leq \left(\frac{2diam_\psi}{\varepsilon} + 1\right)^d,$$

$$\mathcal{N}\left(\varepsilon; \Pi_\omega\left(diam_\omega\right), \|\cdot\|_{L_2}\right) \leq \left(\frac{4ediam_\omega}{\varepsilon} + 1\right)^d.$$

*Proof.* In this proof, we calculate the covering number over the class $\mathcal{Q}_\theta$, $\Pi_\omega$ and $\Omega_\psi$.

**For $\mathcal{Q}_\theta$.** It follows the definition of $\mathcal{Q}_\theta$ $\mathcal{Q}_\theta\left(\mathrm{diam}_\theta\right) \overset{\text{def}}{=} \{(s,a) \mapsto \langle\phi(s,a),\theta\rangle\}$. with $\|\theta\|_{L_2} \leq \bar{V}$. Thus $\mathcal{Q}_\theta\left(\mathrm{diam}_\theta\right)$ is a Euclidean ball with radis $\bar{V}$. As $\phi(s,a)$ is a $d$-dimensional feature space and $\theta \in \mathbb{R}^d$, it follows Lemma 5.7 in [56], we have, for any $\varepsilon > 0$,

$$\mathcal{N}\left(\varepsilon/\bar{V}; \mathcal{Q}_\theta\left(\mathrm{diam}_\theta\right), \|\cdot\|'_{L_2}\right) \leq \frac{\left(\frac{2}{\varepsilon} + 1\right)^d \mathrm{Vol}(\mathcal{Q}_\theta\left(\mathrm{diam}_\theta\right))}{\mathrm{Vol}\left(\mathcal{Q}'_\theta\left(\mathrm{diam}_\theta\right)\right)},$$

where $\|\cdot\|'_{L_2}$ is the pair norm of $\|\cdot\|_{L_2}$ and $\mathcal{Q}'_\theta\left(\mathrm{diam}_\theta\right)$ is the corresponding ball in $\|\cdot\|'_{L_2}$ norm. We take the balls $\mathcal{Q}'_\theta\left(\mathrm{diam}_\theta\right) = \mathcal{Q}_\theta\left(\mathrm{diam}_\theta\right)$, then we obtain

$$\mathcal{N}\left(\varepsilon; \mathcal{Q}_\theta\left(\mathrm{diam}_\theta\right), \|\cdot\|_{L_2}\right) \leq \left(\frac{2\bar{V}}{\varepsilon} + 1\right)^d.$$

**For $\Omega_\psi$.** It follows the definition $\Omega_\psi\left(\mathrm{diam}_\psi\right) \overset{\text{def}}{=} \{(s,a) \mapsto \langle\phi(s,a),\psi\rangle \mid \|\psi\|_{L_2} \leq \mathrm{diam}_\psi\}$ and a similar argument as in the calculation on $\Omega_\psi$, we have

$$\mathcal{N}\left(\varepsilon; \Omega_\psi\left(\mathrm{diam}_\psi\right), \|\cdot\|_{L_2}\right) \leq \left(\frac{2\mathrm{diam}_\psi}{\varepsilon} + 1\right)^d.$$

**For $\Pi_\theta$.** To apply the standard results in a Euclidean ball, we need bound, for any $\omega_1, \omega_2$ where $\|\omega_1 - \omega_2\|_{L_2} \leq 0.5$, $\|\pi_{\omega_1} - \pi_{\omega_2}\|_{L_2} := \sqrt{\int_{\mathcal{S}} |\pi_1(\cdot|s) - \pi_2(\cdot|s)|^2 dP(\mathcal{S})}$. with respect to some probability measure $P$. First, we observe, for any $(s,a) \in \mathcal{S} \times \mathcal{A}$:

$$\pi_{\omega_1}(a|s) - \pi_{\omega_2}(a|s) = \exp(\log(\pi_{\omega_1}(a|s) - \pi_{\omega_2}(a|s))) = \exp(\log(\pi_{\omega_1}(a|s)/\pi_{\omega_2}(a|s))).$$

Follow the definition of the policy class $\Pi_\omega$, we denote $\iota(a,s,\omega) = \exp(\langle\phi(s,a),\omega\rangle)$ and $\iota(\cdot,s,\omega) = \int_{a \in \mathcal{A}} \exp(\langle\phi(s,a),\omega\rangle)$, for any $s,a \in \mathcal{S} \times \mathcal{A}$. For any $s,a$, as $\|\phi(s,a)\|_{L_2} \leq 1$, by Cauchy-Schewarz inequality, then we have

$$\exp(\log(\pi_{\omega_1}(a|s)) - \log(\pi_{\omega_2}(a|s)))$$
$$= \exp(\log(\pi_{\omega_1}(a|s)/\pi_{\omega_2}(a|s)))$$
$$= \exp(\log(\frac{\iota(a,s,\omega_1)}{\iota(\cdot,s,\omega_1)}(\frac{\iota(\cdot,s,\omega_2)}{\iota(a,s,\omega_2)})))$$
$$= \exp\left(\log\left(\iota(a,s,\omega_1-\omega_2)\int_{\widetilde{a}\in\mathcal{A}}\left(\iota(\widetilde{a},s,\omega_1-\omega_2)\frac{\iota(\cdot,s,\omega_1)}{\iota(\cdot,s,\omega_2)}\right)\right)\right)$$
$$= \exp\left(\log\left(\iota(a,s,\omega_1-\omega_2)\int_{\widetilde{a}\in\mathcal{A}}\{\pi_{\omega_2}(\widetilde{a}|s)\iota(\widetilde{a},s,\omega_1-\omega_2)\}\right)\right)$$
$$\leq \exp\left(\log\left(\exp(\|\omega_1-\omega_2\|_{L_2})\int_{\widetilde{a}\in\mathcal{A}}\{\pi_{\omega_2}(\widetilde{a}|s)\exp(\|\omega_1-\omega_2\|_{L_2})\}\right)\right)$$
$$= \exp\left(\log\left(\exp(\|\omega_1-\omega_2\|_{L_2})\exp(\|\omega_1-\omega_2\|_{L_2})\}\right)\right).$$

Now, it suffices to bound $\exp\left(\log\left(\exp(\|\omega_1 - \omega_2\|_{L_2})\exp(\|\omega_1 - \omega_2\|_{L_2})\})\right)\right)$, and by the exponential inequality, we have for $\|\omega_1 - \omega_2\|_{L_2} \leq 0.5$,

$$\exp\left(\log\left(\exp(\|\omega_1 - \omega_2\|_{L_2})\exp(\|\omega_1 - \omega_2\|_{L_2})\})\right)\right)$$
$$\leq \exp\left(\log\left(\exp(\|\omega_1 - \omega_2\|_{L_2})\left(1 + (e/2)\|\omega_1 - \omega_2\|_{L_2})\}\right)\right)\right)$$
$$= \|\omega_1 - \omega_2\|_{L_2}\left(1 + (e/2)\|\omega_1 - \omega_2\|_{L_2}\right).$$

This directly implies that $\pi_{\omega_1}(a|s) - \pi_{\omega_2}(a|s) \leq e\|\omega_1 - \omega_2\|_{L_2}\pi_{\omega_2}(\widetilde{a}|s)$ Shuffle $\omega_1$ and $\omega_2$, we have $\pi_{\omega_2}(a|s) - \pi_{\omega_1}(a|s) \leq e\|\omega_2 - \omega_1\|_{L_2}\pi_{\omega_1}(\widetilde{a}|s)$ and therefore we obtain

$$\|\pi_{\omega_1} - \pi_{\omega_2}\|_{L_2(\mu)} \leq \sup_{s,a\in\mathcal{S}\times A} |\pi_{\omega_1}(a|s) - \pi_{\omega_2}(a|s)| \leq 2e\|\omega_1 - \omega_2\|_{L_2}.$$

as $\pi_{\omega_1}, \pi_{\omega_2}$ are probability density function with integration 1. This completes the proof. Now, we apply the standard covering number arguments in Lemma 7 of [56] over Euclidean ball of $\omega$, we have

$$\mathcal{N}\left(\varepsilon; \Pi_\omega\left(\text{diam}_\omega\right), \|\cdot\|_{L_2}\right) \leq \left(\frac{4e\text{diam}_\omega}{\varepsilon} + 1\right)^d.$$

This completes the proof. $\qquad\square$

## J.2 Proof of Theorem 5.2

*Proof.* We follow the error decomposition as in the proof of Theorem 5.1, according to identical MDP Lemma I.1, we have the error decomposition

$$\frac{1}{\bar{K}}\sum_{k=1}^{\bar{K}}\left(J(\pi) - J(\pi^k)\right)$$

$$= \underbrace{\frac{1}{\bar{K}}\sum_{k=1}^{\bar{K}}\left(J(\pi^k; \{\mathcal{S}, \mathcal{A}, \mathbb{P}_k, \gamma, r_k(s,a), s^0\}) - J(\pi^k)\right)}_{\text{err}_1}$$

$$\underbrace{\frac{1}{\bar{K}}\sum_{k=1}^{\bar{K}}\left(J(\pi; \{\mathcal{S}, \mathcal{A}, \mathbb{P}_k, \gamma, r_k(s,a), s^0\}) - J(\pi^k; \{\mathcal{S}, \mathcal{A}, \mathbb{P}_k, \gamma, r_k(s,a), s^0\})\right)}_{\text{err}_2}$$

$$\underbrace{\frac{1}{\bar{K}}\sum_{k=1}^{\bar{K}}\left(J(\pi) - J(\pi; \{\mathcal{S}, \mathcal{A}, \mathbb{P}_k, \gamma, r_k(s,a), s^0\})\right)}_{\text{err}_3}.$$

As in the analysis in the proof of Theorem 5.1, we can well control $\lambda$ even in the penalization adversarial estimation to control the uncertainty level in the form of $\|\phi(s,a)^\top\psi\|_{L_2(\mu)} \leq \mathcal{U}_2^{\text{lr}}$ for $\tau_\psi \in \Omega_\psi$ for some constant $\mathcal{U}_2^{\text{lr}}$. Also, we define the produce space $\ddot{\mathcal{G}} = \Pi_\omega \times \mathcal{Q}_\theta \times \Omega_\psi$, so that

$$g(s,a,r,s') = \tau_\psi(s,a)(r(s,a) + \gamma q_\theta(s', \pi_\omega) - q_\theta(s,a)) - \lambda\mathbb{D}(\tau_\psi(s,a))$$

for any $g \in \ddot{\mathcal{G}}$. It follows the steps on calculating the complexity of the product space, e.g., (E.14) in the proof of Lemma E.7, we plug-in the covering number in Lemma J.1 and apply Corollary 2 in [16], by some algebra we have

$$\mathcal{N}(\varepsilon, \ddot{\mathcal{G}}, \|\cdot\|_{L_2(\mu)}) \lesssim \left(1 + \frac{48e\widetilde{C}^2\bar{V}\text{diam}_\psi\text{diam}_\omega}{\varepsilon}\right)^d.$$

We set $\varepsilon = \mathcal{O}(\widetilde{C}^2/\sqrt{n})$ for some $\widetilde{C} \geq 0$, and obtain

$$\mathcal{N}(\widetilde{C}^2/\sqrt{n}, \bar{\mathcal{G}}, \|\cdot\|_{L_2(\mu)}) \lesssim \left(1 + e\sqrt{n}\bar{V}\text{diam}_\psi\text{diam}_\omega\right)^d. \tag{J.1}$$

which implies that

$$\mathcal{N}(\varepsilon, \ddot{\mathcal{G}}, \|\cdot\|_{L_2(\mu)}) \lesssim \left(1 + e\sqrt{n}(1 \vee L)\bar{V}\text{diam}_\psi\text{diam}_\omega\right)^d. \tag{J.2}$$

**Bounding err$_1$.** According to Lemma I.6 and follow the (I.21), we obtain

$$\mathrm{err}_1 \le \frac{c^*}{1-\gamma}\Bigg\{ \big(3\mathcal{U}_2^{\mathrm{lr}}\bar{V} + 2\lambda\|\mathbb{D}(\tau(s,a))\|_{L_2(\mu)}^{\mathcal{U}_2^{\mathrm{lr}}}\big)\sqrt{\frac{2\ln\frac{8\mathcal{N}(\epsilon,\ddot{\mathcal{G}},\|\cdot\|_{L_2(\mu)})}{\delta}}{n}}$$

$$+ \frac{2\big(3d\mathrm{diam}_\psi\bar{V} + 2\lambda\|\mathbb{D}(\tau(s,a))\|_\infty^{\mathrm{diam}_\psi}\big)\ln\frac{8\mathcal{N}(\epsilon,\ddot{\mathcal{G}},\|\cdot\|_{L_2(\mu)})}{\delta}}{3n}, \tag{J.3}$$

where $\|\mathbb{D}(\tau(s,a))\|_{L_2(\mu)}^{\mathcal{U}_2^{\mathrm{lr}}} = \sup_{\tau:\|\tau(s,a)\|_{L_2(\mu)}\le\mathcal{U}_2^{\mathrm{lr}}}\|\mathbb{D}(\tau(s,a))\|_{L_2(\mu)}$, and $\|\mathbb{D}(\tau(s,a))\|_{L_\infty}^{\mathrm{diam}_\psi} = \sup_{\tau:\|\tau(s,a)\|_{L_2(\mu)}\le\mathcal{U}_2^{\mathrm{lr}}}\|\mathbb{D}(\tau(s,a))\|_{L_\infty}$. As $\mathbb{D}$ is $M$-strongly convex function and thus locally Lipschitz with a bounded Lipschitz constant $L \le \infty$, then we have

$$\mathrm{err}_1 \lesssim \frac{c^*}{1-\gamma}\Bigg( \big(\mathcal{U}_2^{\mathrm{lr}}\bar{V} + \lambda L\mathcal{U}_2^{\mathrm{lr}}\big)\sqrt{\frac{\ln\frac{8\mathcal{N}(\epsilon,\ddot{\mathcal{G}},\|\cdot\|_{L_2(\mu)})}{\delta}}{n}}$$

$$+ \frac{2\big(d\mathrm{diam}_\psi\bar{V} + \lambda Ld\mathrm{diam}_\psi\big)\ln\frac{8\mathcal{N}(\epsilon,\ddot{\mathcal{G}},\|\cdot\|_{L_2(\mu)})}{\delta}}{3n}\Bigg). \tag{J.4}$$

Plug-in the covering number in (J.2), we conclude

$$\mathrm{err}_1 \lesssim \frac{c^*}{1-\gamma}\Bigg( \big(\mathcal{U}_2^{\mathrm{lr}}\bar{V} + \lambda L\mathcal{U}_2^{\mathrm{lr}}\big)\sqrt{\frac{\ln\{8\big(1+e\sqrt{n}(1\vee L)\bar{V}\mathrm{diam}_\psi\mathrm{diam}_\omega\big)^d/\delta\}}{n}}$$

$$+ \frac{2\big(\mathrm{diam}_\psi d\bar{V} + \lambda Ld\mathrm{diam}_\psi\big)\ln\{8\big(1+e\sqrt{n}(1\vee L)\bar{V}\mathrm{diam}_\psi\mathrm{diam}_\omega\big)^d/\delta\}}{3n}\Bigg). \tag{J.5}$$

**Bounding err$_2$.** According to Lemma I.9, we achieve no-regret policy optimization oracle, and thus

$$\mathrm{err}_2 \le \frac{2\sqrt{2\bar{V}\log|\mathcal{A}|}}{\sqrt{\bar{K}}(1-\gamma)}. \tag{J.6}$$

**Bounding err$_3$.** For any $\pi \in \Pi_\omega$, following the definition of err$_3$ we have

$$\mathrm{err}_3 = \frac{1}{\bar{K}}\sum_{k=1}^{\bar{K}}\big(J(\pi) - J(\pi;\{\mathcal{S},\mathcal{A},\mathbb{P}_k,\gamma,r_k(s,a),s^0\})\big)$$

$$= \frac{1}{\bar{K}}\sum_{k=1}^{\bar{K}}\big(q^\pi(s^0,\pi) - J(\pi;\{\mathcal{S},\mathcal{A},\mathbb{P}_k,\gamma,r_k,s^0\})\big)$$

$$= \frac{1}{\bar{K}}\sum_{k=1}^{\bar{K}}\frac{\mathbb{E}_{d^\pi}[q^\pi(s,\pi) - r_k(s,a) - q^\pi(s,\pi)]}{1-\gamma}$$

$$= \frac{1}{\bar{K}}\sum_{k=1}^{\bar{K}}\frac{\mathbb{E}_{d^\pi}\big[\phi(s,a)^\top\theta_k - \mathbb{P}^{\pi^k}\phi(s,a)^\top\theta_k\big]}{1-\gamma}, \tag{J.7}$$

where the last equality comes from a similar derivation as in (I.26). Therefore, it suffices to bound

$$\frac{\mathbb{E}_{d^\pi}\big|\phi(s,a)^\top\theta_k - \mathbb{P}^{\pi^k}\phi(s,a)^\top\theta_k\big|}{1-\gamma}, \tag{J.8}$$

for any $k \in [\bar{K}]$. According to Lemma I.10, and the covering number Lemma J.1, it immediately obtains

$$\sup_\psi \frac{1}{1-\gamma}\mathbb{E}_\mu\Big[\phi(s,a)^\top\psi\big(r(s,a) + \gamma\phi(s',\pi^k)^\top\theta_k - \phi(s,a)^\top\theta_k\big)\Big]$$

$$\le \frac{1}{1-\gamma}\Bigg( \big(\mathcal{U}_2^{\mathrm{lr}}\bar{V} + \lambda L\mathcal{U}_2^{\mathrm{lr}}\big)\sqrt{\frac{\ln\{8\big(1+e\sqrt{n}(1\vee L)\bar{V}\mathrm{diam}_\psi\mathrm{diam}_\omega\big)^d/\delta\}}{n}}$$

$$+ \frac{2\big(d\mathrm{diam}_\psi\bar{V} + \lambda Ld\mathrm{diam}_\psi\big)\ln\{8\big(1+e\sqrt{n}(1\vee L)\bar{V}\mathrm{diam}_\psi\mathrm{diam}_\omega\big)^d/\delta\}}{3n}\Bigg) + \frac{\bar{V}}{c^*}.$$

As $\|\phi(s,a)^\top \psi\|_{L_2(\mu)} \le \mathcal{U}_2^{\mathrm{lr}} < \infty$, we apply Lemma E.3, then

$$\frac{\mathbb{E}_\mu \left| \phi(s,a)^\top \theta_k - \mathbb{P}^{\pi^k} \phi(s,a)^\top \theta_k \right|}{1-\gamma} \le \frac{1}{1-\gamma} \left( (\bar{V} + \lambda L) \sqrt{\frac{\ln\{8\left(1 + e\sqrt{n}(1 \vee L)\bar{V}\mathrm{diam}_\psi \mathrm{diam}_\omega\right)^d / \delta\}}{n}} \right.$$

$$\left. + \frac{\mathrm{diam}_\psi}{\mathcal{U}_2^{\mathrm{lr}}} \frac{2(\bar{V} + \lambda L)\ln\{8\left(1 + e\sqrt{n}(1 \vee L)\bar{V}\mathrm{diam}_\psi \mathrm{diam}_\omega\right)^d / \delta\}}{3n} \right) + \frac{\bar{V}}{c^* \mathcal{U}_2^{\mathrm{lr}}} := \mathcal{E}(c^*, \lambda, n).$$

According to Lemma E.2 and (I.15) with covering number arguments in Lemma J.1, it observes that

$$\frac{\sum_{i=1}^n \left[ \phi(s_i, a_i)^\top \theta_k - \mathbb{P}^{\pi^k} \phi(s_i, a_i)^\top \theta_k \right]}{n(1-\gamma)} \le \mathcal{E}(c^*, \lambda, n)$$

$$\iff \frac{\sum_{i=1}^n \left[ \phi(s_i, a_i)^\top \left( \theta_k - \theta' - \gamma \sum_{s',a'} \varphi(s') \pi^k(a' \mid s') \phi(s', a')^\top \theta_k \right) \right]}{n(1-\gamma)} \le \mathcal{E}(c^*, \lambda, n)$$

$$:= \frac{\sum_{i=1}^n \left[ \phi(s_i, a_i)^\top \mathbb{G}^{\pi^k} \right]}{n(1-\gamma)} \le \mathcal{E}(c^*, \lambda, n).$$

where $\theta'$ is the coefficients for linear representation of $r(s,a)$, and $\varphi(s')$ is the low rank decomposition for transition kernel [19]. This implies

$$\left\| \sqrt{\frac{1}{n} \sum_{i=1}^n [\phi(s_i, a_i)\phi(s_i, a_i)^\top]} \mathbb{G}^{\pi^k} \right\|_{L_2} \le (1-\gamma)\mathcal{E}(c^*, \lambda, n).$$

Then by Cauchy-Schwarz inequality, for any $s, a$, we have

$$|\phi(s,a)^\top \mathbb{G}^{\pi^k}| \le \|\phi(s,a)^\top \mathbb{G}^{\pi^k}\|_{L_2}$$

$$= \|\phi(s,a)^\top \sqrt{(\frac{1}{n}\sum_{i=1}^n [\phi(s_i,a_i)\phi(s_i,a_i)^\top])^{-1}} \sqrt{\frac{1}{n}\sum_{i=1}^n [\phi(s_i,a_i)\phi(s_i,a_i)^\top] \mathbb{G}^{\pi^k}} \|_{L_2}$$

$$\le \|\phi(s,a)^\top \sqrt{(\frac{1}{n}\sum_{i=1}^n [\phi(s_i,a_i)\phi(s_i,a_i)^\top])^{-1}} \|_{L_2} \| \sqrt{\frac{1}{n}\sum_{i=1}^n [\phi(s_i,a_i)\phi(s_i,a_i)^\top] \mathbb{G}^{\pi^k}} \|_{L_2}$$

$$\le \|\phi(s,a)^\top \sqrt{(\frac{1}{n}\sum_{i=1}^n [\phi(s_i,a_i)\phi(s_i,a_i)^\top])^{-1}} \|_{L_2} (1-\gamma)\mathcal{E}(c^*, \lambda, n).$$

Then we have an upper bound for (J.8),

$$\frac{\mathbb{E}_{d^\pi} \left| \phi(s,a)^\top \theta_k - \mathbb{P}^{\pi^k} \phi(s,a)^\top \theta_k \right|}{1-\gamma}$$

$$\le \mathbb{E}_{d^\pi}[\|\phi(s,a)^\top \sqrt{(\frac{1}{n}\sum_{i=1}^n [\phi(s_i,a_i)\phi(s_i,a_i)^\top])^{-1}} \|_{L_2}]\mathcal{E}(c^*, \lambda, n)$$

$$\le \mathbb{E}_{d^\pi} \left[ \sqrt{\phi(s,a)^\top (\frac{1}{n}\sum_{i=1}^n [\phi(s_i,a_i)\phi(s_i,a_i)^\top])^{-1}\phi(s,a)} \right] \mathcal{E}(c^*, \lambda, n).$$

To facilitate the proof, we use the notation $\|x\|_\Sigma \overset{\text{def}}{=} \sqrt{x^\top (\Sigma)^{-1} x}$. According to Lemma 32 in [63], we obtain the upper bound for $\mathbb{E}_{d^\pi}\left[\|\phi(s,a)\|_{\Sigma_n^{-1}}\right]$ as follows:

$$
\begin{aligned}
\mathbb{E}_{d^\pi}\left[\|\phi(s,a)\|_{\Sigma_n^{-1}}\right] =& \mathbb{E}_{d^\pi}\left[\sqrt{\phi(s,a)^\top (\frac{1}{n}\sum_{i=1}^n [\phi(s_i,a_i)\phi(s_i,a_i)^\top])^{-1}\phi(s,a)}\right] \\
\leq& \sqrt{\mathbb{E}_{d^\pi}\left[\phi(s,a)^\top (\frac{1}{n}\sum_{i=1}^n [\phi(s_i,a_i)\phi(s_i,a_i)^\top])^{-1}\phi(s,a)\right]} \\
\leq& \sqrt{\text{trace}\left(\mathbb{E}_{d^\pi}\left[\phi(s,a)\phi(s,a)^\top\right](\frac{1}{n}\sum_{i=1}^n [\phi(s_i,a_i)\phi(s_i,a_i)^\top])^{-1}\right)} \\
\leq& \sqrt{\iota(d_\pi,\mu)\text{trace}(\frac{1}{n}\sum_{i=1}^n [\phi(s_i,a_i)\phi(s_i,a_i)^\top](\frac{1}{n}\sum_{i=1}^n [\phi(s_i,a_i)\phi(s_i,a_i)^\top]))^{-1}} \\
=& \sqrt{\iota(d_\pi,\mu)d}. \tag{J.9}
\end{aligned}
$$

where $\iota(d_\pi,\mu) = \sup_{x\in\mathbb{R}^d} \frac{x^T \mathbb{E}_{d\pi}[\phi(s,a)\phi(s,a)^\top]x}{x^\top\mathbb{E}_\mu[\phi(s,a)\phi(s,a)^\top]x}$ Based on this, we conclude that

$$\text{err}_3 \leq \sqrt{\iota(d_\pi,\mu)d}\mathcal{E}(c^*,\lambda,n). \tag{J.10}$$

We combine the upper bounds in (J.5), (J.6) and (J.10), and we set $c^* = \widetilde{\mathcal{O}}\big(\sqrt[4]{n/d\ln\{(1+e\sqrt{n}(1\vee L)\breve{V}c_\psi c_\omega)/\delta\}}\big)$ and set $\lambda = \lambda(c_\psi(\mathcal{U}_2^{\text{lr}}))$ for $c_\psi\{\mathcal{U}_2^{\text{lr}}\} = \sup_{\{\psi:\|\phi(s,a)^\top\psi\|_{L_2(\mu)}=\mathcal{U}_2^{\text{lr}}\}}\|\psi\|_{L_\infty}$, by some algebra, we conclude that

$$
J(\pi) - J(\widehat{\pi}^{\text{lr}}) \leq \frac{1}{\sqrt{1-\gamma}}\sqrt[4]{\frac{\iota(d_\pi,\mu)d(\bar{V}^2+\bar{V}\lambda L)^2\ln\{8\left(1+e\sqrt{n}(1\vee L)\bar{V}\text{diam}_\psi\text{diam}_\omega\right)^d/\delta\}}{n}}
$$

$$
+\frac{1}{\sqrt{1-\gamma}}\sqrt{\frac{(\iota(d_\pi,\mu)d)^{0.5}\bar{V}^2\left(d\text{diam}_\psi\bar{V}+\lambda Ld\text{diam}_\psi\right)/\mathcal{U}_2^{\text{lr}}\ln\{8\left(1+e\sqrt{n}(1\vee L)\bar{V}\text{diam}_\psi\text{diam}_\omega\right)^d/\delta\}}{n}}
$$

$$
+\frac{\sqrt{\iota(d_\pi,\mu)d}}{1-\gamma}\frac{d\text{diam}_\psi}{\mathcal{U}_2^{\text{lr}}}\frac{2\left(\bar{V}+\lambda L\right)\ln\{8\left(1+e\sqrt{n}(1\vee L)\bar{V}\text{diam}_\psi\text{diam}_\omega\right)^d/\delta\}}{3n}+\frac{2\sqrt{2\bar{V}\log|\mathcal{A}|}}{\sqrt{\bar{K}}(1-\gamma)}.
$$

If we ignoring the fast term and let $\bar{K}\gg\log|\mathcal{A}|$, we have

$$
J(\pi) - J(\widehat{\pi}^{\text{lr}}) \lesssim \frac{1}{\sqrt{1-\gamma}}\sqrt[4]{\frac{\iota(d_\pi,\mu)d(\bar{V}^2+\bar{V}\lambda L)^2\ln\{8\left(1+e\sqrt{n}(1\vee L)\bar{V}\text{diam}_\psi\text{diam}_\omega\right)^d/\delta\}}{n}}.
\tag{J.11}
$$

For $\psi\in\{\psi:\|\phi(s,a)^\top\psi\|_{L_2(\mu)}\leq\mathcal{U}_2^{\text{lr}}\}$, we can observe that

$$\|\phi(s,a)^\top\psi\|_{L_2(\mu)} \leq \text{trace}(\mathbb{E}_\mu[\phi(s,a)\phi(s,a)^\top])\|\theta\|_{L_2} \tag{J.12}$$

$$\leq \text{trace}(\mathbb{E}_\mu[\phi(s,a)\phi(s,a)^\top])dc_\psi\{\mathcal{U}_2^{\text{lr}}\}. \tag{J.13}$$

It combines with (J.13) and (J.11), we conclude that

$$
J(\pi) - J(\widehat{\pi}^{\text{lr}})
$$

$$
\lesssim \frac{\min\{\text{trace}(\mathbb{E}_\mu[\phi(s,a)\phi(s,a)^\top])dc_\psi\{\mathcal{U}_2^{\text{lr}}\}, \sqrt{\iota(d_\pi,\mu)d}\}}{1-\gamma}
$$

$$
\cdot\sqrt[4]{\frac{(\bar{V}^2+\bar{V}\lambda L)^2\ln\{(1+e\sqrt{n}(1\vee L)\bar{V}\text{diam}_\psi\text{diam}_\omega)^d/\delta\}}{n}}
$$

$$
\leq \frac{\sqrt{\min\{\kappa^2 c_\psi^2\{\mathcal{U}_2^{\text{lr}}\}d^2, \iota(d_\pi,\mu)d\}}}{1-\gamma}\sqrt[4]{\frac{(\bar{V}^2+\bar{V}\lambda L)^2\ln\{(1+e\sqrt{n}(1\vee L)\bar{V}\text{diam}_\psi\text{diam}_\omega)^d/\delta\}}{n}},
$$

where $\kappa = \text{trace}(\mathbb{E}_\mu[\phi(s,a)\phi(s,a)^\top])$. This completes the proof. $\qquad\square$

# K   Additional Related Works

**Offline RL.** The domain approaches of offline RL include fitted Q-iteration (FQI; [10, 46] ), fitted policy iteration [2, 26], Bellman Residual Minimization (BRM; [3, 11, 9], and actor-critic [21, 22, 15]. We refer the reader to [29] for more comprehensive discussions on the topics of the offline RL. In the aforementioned mainstreams of works, ours is closely related to the actor-critic. Actor-critic methods are a hybrid class of methods that mitigate some deficiencies of methods that are either purely policy or purely value-based; in modern RL, they are widely used in practice [58, 59]. A standard framework in actor-critic methods is that actor supervises the policy to improve in order to maximize its values estimated by the critic, value function. From a high-level point of view, this connects our bi-level structured optimization to actor-critic methods. In our framework, the upper-level components make decisions, i.e., searching for a policy maximizing the pessimistic evaluation based on the lower-level outputs, i.e., the uncertainty-controlled confidence set of value estimates. Therefore, our works demonstrate the advantages of actor-critic-type methods in offline RL from a bi-level reformulation perspective.

**Minimax learning.** In a seminal work, [32] proposed the first minimax estimation procedure requiring two function approximators, one for modeling the marginalized importance-weight function, and the other for modeling the value function. This method becomes particularly efficient in estimating the discounted return when the offline data-generating distribution aptly encompasses the distribution invoked by the evaluation policy, thereby avoiding the significant issue of exponential variance in the horizon, a notable drawback of importance sampling [43, 30]. The ripple effect of this method has led to a surge of interest within the RL community [61, 54, 18, 38, 33, 49, 65]. Intriguingly, our bi-level policy optimization aligns with this trend of minimax learning, where we build a confidence set for policy evaluation using the marginalized importance-weight. In particular, algorithmically, our low-level component is most related to the value interval learning in [18]. They provide a minimax interval for quantifying the value bias involved in the discounted return under function approximation settings. However, they only handle the function approximation errors but do not quantify the statistical uncertainty, as well as no uncertainty control is performed. In contrast, our work yields a confidence interval that concurrently incorporates the bias introduced by function approximation and uncertainty stemming from sampling. This also provides a basis for the operations at the upper level in our bi-level structured optimization.

**Conservative value estimation.** Following the principle of pessimism in the face of uncertainty, a significant portion of recently proposed offline RL methods rely on on estimating conservative $q$-values for optimizing the target policy, with the constraint or regularizer serving to limit deviation from the behavior policy [24, 25, 39, 23, 14, 27, 28]. For our work, we also following the pessimistic principle for value estimation. The major differences between ours and the existing works in this mainstream are two-fold. First, with uncertainty control through favoring the policy close to the behavior policy, our algorithm also ensures the consistency of the value estimates. This consistency guarantee plays a key role in our method to ensure no overly pessimistic reasoning. Second, from a high-level point of view, our algorithm has a bi-level structure, and more close to actor-critic-based methods. In contrast, the aforementioned works are more close to approximate dynamic programming [23].

# L   Statistical Learning Tools

In this section, we introduce fundamental concepts from statistical learning theory, as outlined in [1, 55]. We begin with the concept of the covering number. This metric quantifies the number of spherical balls of a specified size required to encompass a designated space, allowing for potential overlaps.

**Definition L.1** (Covering number)**.** *Let $(\mathcal{C}, \|\cdot\|)$ be a $\|\cdot\|$ normed space, and $\mathcal{H} \subseteq \mathcal{C}$. The set $\{b_1, b_2, \ldots, b_m\}$ is a $\epsilon$-covering over $\mathcal{H}$ if $\mathcal{H} \subseteq \cup_{i=1}^m \mathbb{B}(b_i, \varepsilon)$, where $\mathbb{B}(b_i, \varepsilon)$ is the sup-norm-ball centered at $b_i$ with radius $\varepsilon$. Then the covering number of $\mathcal{H}$ is defined as $\mathcal{N}(\epsilon, \mathcal{H}, \|\cdot\|_{L_2}) = \min\{n : \exists \epsilon$-covering over $\mathcal{H}$ of size $m\}$.*

A widely recognized method for examining the generalization capability of statistical learning models involves the use of the *VC-dimension*. This dimension not only characterizes uniform convergence, as detailed in [55], but also asymptotically dictates the sample complexity of PAC learning [5].

**Definition L.2** (growth function, VC-dimension, shattering). *Let $\mathcal{H}$ denote a class of functions from $\mathcal{X}$ to $\{0, 1\}$. For any non-negative integer $m$, we define the growth function of $\mathcal{H}$ as*

$$\Pi_{\mathcal{H}}(m) := \max_{x_1, \dots, x_m \in \mathcal{X}} |\{(h(x_1), \dots, h(x_m)) : h \in \mathcal{H}\}|.$$

*If $|\{(h(x_1), \dots, h(x_m)) : h \in \mathcal{H}\}| = 2^m$, we say $H$ shatters the set $\{x_1, \dots, x_m\}$. The Vapnik-Chervonenkis dimension of $\mathcal{H}$, denoted $\mathrm{VCdim}(\mathcal{H})$, is the size of the largest shattered set, i.e. the largest $m$ such that $\Pi_{\mathcal{H}}(m) = 2^m$. If there is no largest $m$, we define $\mathrm{VCdim}(\mathcal{H}) = \infty$*

For a set of real-valued functions, like those produced by neural networks, the *pseudo dimension* serves as an intuitive measure of complexity. This dimension also suggests similar uniform convergence properties and was introduced by [42].

**Definition L.3** (Pollard's pseudo dimension). *Let $\mathcal{F}$ be a class of functions from $\mathcal{X}$ to $\Re$. The pseudodimension of $\mathcal{F}$, written $D_{\mathcal{F}}$, is the largest integer $m$ for which there exists $(x_1, \dots, x_m, y_1, \dots, y_m) \in \mathcal{X}^m \times \Re^m$ such that for any $(b_1, \dots, b_m) \in \{0, 1\}^m$ there exists $f \in \mathcal{F}$ such that*

$$\forall i : f(x_i) > y_i \iff b_i = 1$$

In the end, it's worth noting that the pseudo dimension extends the concept of the VC-dimension to real-valued functions.