# OpenReview forum: "Bi-Level Offline Policy Optimization with Limited Exploration"
_NeurIPS.cc/2023/Conference — NeurIPS 2023 poster_

### Official Review · Reviewer_q6tp · 2023-07-05

**Soundness:** 4 excellent
**Presentation:** 3 good
**Contribution:** 3 good
**Rating:** 7
**Confidence:** 3

**Summary:**

This manuscript proposes a bi-level structure for solving offline RL problem. The authors design the lower level constructing a set of value estimates, while the upper level maximizing the conservative value estimate. Specifically, the author provides a penalized adversarial estimation algorithm. The superiority of the proposed method is shown from both real world and synthetic data.

**Strengths:**

1. The hierarchical structure for solving offline RL problem is very interesting.
2. The proposed method considers a confidence set of value estimates as well as the uncertainty, and it effectively prevents from being too pessimistic.
3. The manuscript provides solid theoretical results, and it shows the proposed method could work without requiring data-coverage, which sounds quite useful in offline RL scenario.

**Weaknesses:**

1.  It is suggested to add some important steps in Algorithm 1 from Line 270 ~ Line 281, otherwise the Algorithm 1 is too simple while there are many details in 270 ~ Line 281, which is somehow difficult to read.

2. The authors considers the synthetic data (CarPole) and real-world data (Type I Diabetes), as a common benchmark in offline RL research, the D4RL tasks are widely studied in offline RL experiments. It is suggested to consider some D4RL task in the experiment.

**Questions:**

1. There are some confusion in understanding the definition of Consistency  above line 154, should the $\mathcal{Q}_{\epsilon_n}$ be related with policy $\pi$?
2. In line 3 of Algorithm 1, it is required to solve a maxmin problem with iterative method, so how is the time consumption of the proposed method? Could the author provides some experiment study or conceptual analysis on this issue?
3. Any practical suggestions on selecting $\mathbb{G}(x)$ and the initial function of $\tau(s,a)$? The author also mentions the proposed method does not require data-coverage, could this issue be demonstrated from experiments? Say using a biased policy as behavior policy to generate data to learn offline RL policies.

**Limitations:**

see questions and limitations

---

> ### Author Rebuttal · Authors · 2023-08-09
>
>
> We thank the reviewer for the suggestive feedback, which greatly improves our paper! We address the reviewer's comments point by point below.
>
> > **Weakness 1: Algorithm 1**
>
> We feel sorry for the misleading. Due to the page limit, we actually provide a comprehensive discussion of Algorithm 1 in Appendix A of the supplementary materials. It includes a detailed version of Algorithm 1, a discussion on solving maximin problem (line 3 of Algorithm 1), a closed-form update rule for mirror descent (line 4 of Algorithm 1), as well as a convergence analysis of the algorithm and its technical proofs.
>
>
> > **Weakness 2: D4RL tasks**
>
> Thank you for your suggestions! We follow your suggestion to evaluate our algorithm on walker2d, hopper, halfcheetah, and maze2D D4RL benchmarks. The results are reported in **Table 1** of the global response PDF. Our algorithm achieves the best performance in 7 tasks and is comparable to the baselines in the remaining tasks.
>
> > **Question 1:  $\mathcal{Q}_{\varepsilon_n}$ indexed by $\pi$**
>
>  Thanks for pointing this out. Yes, the Bellman consistency set $\mathcal{Q}_{\varepsilon_n}$ defined in line 154 is dependent on $\pi$. We aim to construct a set of functions that is consistent with the q-function under the policy $\pi$, i.e., $q^{\pi}$.
>
> To avoid any confusion, we modify the notation to $\mathcal{Q}^\pi_{\varepsilon_n},$ to reflect its dependency on $\pi$.
>
> > **Question 2: Time consumption of the maximin problem**
>
> Thank you for arising this point. In the following response, we follow your suggestions to study the time complexity/consumption from both theoretical and numerical perspectives.
>
> For the maximin problem, the inner optimization is easy to solve. On the one hand, it could yield a closed-form global solver using the maximum mean discrepancy principle [1,2]. On the other hand, one can utilize the feature mapping class for approximation, which is computationally efficient and can be solved by various algorithms [3]. From this perspective, the computational burden for the inner optimization is light, and the maximin problem is essentially a maximization problem. For more details, please see Appendix A of the supplementary materials.
>
> Theoretically, we study the time complexity of the optimation presented by Theorem A.1 in Appendix A, which shows that our maximin optimization realizes a fast sublinear rate $\mathcal{O}(1/T)$ even under non-linear settings. Numerically, we test the runtime of our algorithm in comparison to the prior works. We report the results in **Table 2** of the global response PDF,
> which shows our algorithm is comparable to other baselines.
>
> > **Question 3 (a): Practical choices on $\mathbb{G}(x)$ and $\tau(s,a)$**
>
> Thanks for your question. There are many choices for $\mathbb{G}(x)$, e.g., Bhattacharyya distance [4], $f$-divergence [5], as well as simple quadratic form functions which we used in our work (see Appendix B.2 in the supplementary materials). In practice, we prefer to select a simple but optimization-friendly choice for the $\mathbb{G}(x)$. The quadratic form function is one of the best choices and has been widely used in the machine learning community [6,7,8].
>
> For the choice of $\tau \in \Omega$, we first note that $\tau$ is for modeling the importance weights. Due to the complex structure of the marginalized importance weight, it demands a sufficiently flexible non-linear function approximation class, e.g., Reproducing Kernel Hilbert Spaces (RKHS), deep neural networks, for optimization [9]. In most of our experiments, to ensure the flexibility of $\tau$, we model $\tau$ by a two-layer feedforward neural network with the layer width $256$ and use ReLU as the activation function. Additionally, the weights are initialized in accordance with the Xavier initialization [10].
>
> > **Question 3 (b): Data-coverage experiments**
>
> Thank you for your suggestions. In our original experiments, we control the $\alpha$ for simulating the scenarios with varying data coverage levels (poor, medium, relatively high). In practice, it's unfeasible to generate an offline dataset with absolute zero data coverage. This is because whenever a policy is rolled out in an interactive environment, the generated datasets inevitably exhibit some coverage. Nevertheless, by setting $\alpha =0.1$, our "poor coverage" scenario is quite close to a "no coverage" situation.
>
> We guess that by "biased policy," you might be referring to a random or non-optimal policy. In our new experiments on the D4RL benchmark, we used offline datasets generated by a randomly initialized policy (which can be considered a biased policy), e.g., halfcheetah-random. These results can be found in **Table 1** of the global response PDF. We can observe that our algorithm is comparable to other baselines in such settings.
>
> **References:**
>
> [1] Gretton, A., et al. (2012), A kernel two-sample test, The Journal of Machine Learning Research.
>
> [2] Shi, C., et al. (2022), A minimax learning approach to off-policy
> evaluation in confounded partially observable Markov decision processes, ICML.
>
> [3] Sriperumbudur, B. K., et al. (2011), Universality, characteristic kernels and rkhs embedding of measures, JMLR.
>
> [4] Choi, E. and Lee, C. (2003), Feature extraction based on Bhattacharyya distance, Pattern Recognition.
>
> [5] Sason, I. and Verdu, S. (2016), f-divergence inequalities, IEEE Transactions on Information
> Theory.
>
> [6] Belousov, B. and Peters, J. (2017), f-divergence constrained policy improvement, arXiv preprint
> arXiv:1801.00056.
>
> [7] Tao, C., et al. (2018), Chi-square generative adversarial
> network, ICML.
>
> [8] Nachum, O., et al. (2019), Algaedice: Policy gradient from arbitrary experience. arXiv preprint arXiv:1912.02074.
>
> [9] Uehara, M., Huang, J. and Jiang, N. (2020), Minimax weight and q-function learning for off-policy
> evaluation, ICML.
>
> [10] Glorot, X. and Bengio, Y. (2010), Understanding the difficulty of training deep feedforward neural
> networks, AISTAT.

---

> > ### Comment · Reviewer_q6tp · 2023-08-16
> > **Thank you for your response**
> >
> > Thank you for your detailed reply. Your response has addressed most of my concerns, and I really appreciate your explanations.

---

> > > ### Author Response · Authors · 2023-08-16
> > > **Response to Reviewer q6tp**
> > >
> > > Thank you for your positive feedback and recognition. We are again grateful for your time and efforts to review our paper and give constructive comments!

---

### Official Review · Reviewer_C4A2 · 2023-07-05

**Soundness:** 2 fair
**Presentation:** 3 good
**Contribution:** 4 excellent
**Rating:** 7
**Confidence:** 3

**Summary:**

Offline Reinforcement Learning suffers from distribution shift, arising as a consequence of limited exploration and function approximation. The paper aims to address these two challenges by proposing a bi-level hierarchical framework for policy optimization. The upper-level optimizes a policy improvement objective using a pessimistic value function. The lower level learns a conservative estimate of the value function by maintaining a confidence set for the upper level. Development of the framework is carried out by first constructing robust intervals for addressing approximation bias and then using these intervals in uncertainty quantification. Motivated by its theoretical results, the work further proposes an empirical penalized adversarial estimation algorithm which incorporates the bi-level objective using an adversarial estimation loss. While the lower level is optimized efficiently using maximum mean discrepancy principle, the upper level utilizes function approximation for learning the proximal mapping.


**Strengths:**

* The paper is well written and organized. Specifically, the paper builds the framework in a structured way.
* While the paper borrows from past results in tackling distribution shift and constrained optimization, the bi-level approach and its abstraction between policy and value objectives is novel.


**Weaknesses:**

* **Empirical Evaluation:** My main concerns are centered at the empirical evaluation and experiments carried out to assess the bi-level framework. Experiments do not validate the theoretical claims of the work and only evaluate the performance of the framework. Ideally the empirical evaluation should validate whether regret bounds of Theorem 4.1, Theorem 5.1 or Theorem 5.2 would hold or not. Authors can compare empirically whether the bound of Theorem 5.2 is tight and that if $\lambda$ and $c^{*}$ are chosen as per the set criteria, regret is close to the bound of Theorem 5.2. Similarly, experiments do not demonstrate the benefits of bi-level approach in addressing distribution shift. It would be worthwhile to compare the performance with and without the upper/lower levels and assess if regret bounds still hold.
* **Significance of Information-Theoretic Results:** Section 4 appears to be disconnected from the rest of the paper. While the Information Theoretic algorithm does not provide an empirically feasible method, it produces a baseline policy improvement scheme using bias and uncertainty errors. However, the significance of these results is unclear. To the best of my knowledge, the information theoretic results are not used to generate the penalized adversarial estimation algorithm. Neither they are used in the empirical evaluation or in producing a tighter regret bound. It would be helpful if the authors could explain the significance and utility of these results and how they should be interpreted. It would also be helpful if the paper could throw light on the benefit of near-optimal regret in Corollary 4.1.
* **Utility of $\tilde{\sigma_{n}}$:** From my understanding, $\tilde{\sigma_{n}}$ trades off pessimism for policy improvement. This indicates that the utility of $\tilde{\sigma_{n}}$ is not well understood. Setting $\tilde{\sigma_{n}}$ to a lower value can hinder policy performance as the lower level will yield an overly conservative value function. On the other hand, setting $\tilde{\sigma_{n}}$ to a higher value leads to distribution shift. It would be helpful to have an optimal value of $\tilde{\sigma_{n}}$ characterized or a criterion for selecting $\tilde{\sigma_{n}}$ based on policy performance or pessimism.

## Minors

* line 29: amplifying -> amplifies
* line 32: to contain -> contains
* line 79: discounted -> discount
* line 125: sepcified -> specified
* line 137: established the CI for $J(\pi)$ is presented in Theorem 3.1.
* line 234: concentrability -> concentratibility
* line 275: conditional -> conditioned
* line 329: set propertly choose -> properly choose


**Questions:**

* Can the empirical evaluation demonstrate the bounds of Theorem 4.1 and Theorem 5.2? Can experiments show the benefit of bi-level approach in tackling distribution shift (i.e- comparing with and without upper/lower levels)?
* Can you please explain how are the Information Theoretic results useful? Are these results used to generate the penalized adversarial estimation algorithm? If not, then how should one interpret Theorem 4.1 and Corollary 4.1?
* What is the optimal $\tilde{\sigma_{n}}$? Is there a criterion for setting $\tilde{\sigma_{n}}$?


**Limitations:**

Authors have discussed limitations and future work.

---

> ### Author Rebuttal · Authors · 2023-08-09
>
> We thank the reviewer for the suggestive feedback, which greatly improves our paper! We address the reviewer's comments point by point below.
>
>
> > **Weakness and Question 1:  Empirical evaluation**
>
> Thanks for your great suggestions!
>
> **For regret validation**:
> We first would like to clarify that it is hard to conduct empirical evaluations for Theorem 4.1 since it is for our information-theoretic algorithm. Even when considering our practical algorithm, empirically validating the tightness of the bound is still not easy. This is because some key components, e.g., concentrability coefficient, universal constants, and function class complexity in the regret bound, are difficult to (accurately) estimate from datasets. Nevertheless, following your suggestion, we empirically verify the convergence rate of regret bound in Theorem 5.2:
>
> Typically, we lack information regarding the optimal policy and its coverage within offline datasets, making it challenging to accurately compute the regret. Therefore, we carefully design a synthetic environment. We only briefly describe the environment due to word limit: the reward $r(s, a)=(a-\beta s)^{\top} \Lambda(a-\beta s)$ with coefficient matrix $\beta$ and the negative definite matrix $\Lambda$. Therefore, the optimal policy has an analytical form $\pi^{\star}(s)=s^{\top} \beta$, which is important to calculate precise regret. The dataset is generated following $\pi_b$ such that $a = \beta s + \mathcal{N}\left(0, \sigma_0^2 I \right)$, indicating the data is less explored when $\sigma_0$ is small. In **Figure 1** of the global response PDF, we study the convergence rate of regret, which validates the  $\mathcal{O}(n^{-1/4})$ rate in Theorem 5.2.
>
> **For framework validation**: Given that the upper level is performed for policy updating, it cannot be removed.
> Thus, we guess that you may suggest studying the effectiveness of the lower-level components (line 153): uncertainty control (UC) and Bellman consistency (BC). To this end, we evaluate the policy performance across the optimization epochs and the regret. The results are provided in **Figure 3** of the global response PDF. **Figure 3 (a)** shows that the algorithm only with UC is quite unstable due to inconsistent estimation on $q^{\pi}$; the algorithm only with BC is relatively stable but suffers performance degradation due to over-estimation issues and insufficient model extrapolation. **Figure 3 (b)** indicates the sub-optimal regret and slow convergence rate  when either UC or BC is absent.
>
> > **Weakness and Question 2: Significance of Information-Theoretic Results**
>
> This is a great point! Yes, the information-theoretic results are not directly used to generate the later practical algorithm. However, their utility is threefold. First, they provide rigorous guarantees for our information-theoretic algorithm, particularly under the scenario where no coverage and completeness-type conditions hold.
> Second, in the development of the theorems, we propose a novel error decomposition analysis (Equations I.19 to I.24 on p.53-p.54 of Appendix), which is one of the theoretic contributions of our work. In comparison to existing works, e.g., [1,2], our decomposition analysis makes the off-support errors explicitly analyzed in the regret, and better characterizes the roles of on-supports and off-support errors.
> Third, with these results, we can well-understand and uncover some hidden nice properties of our algorithm: enhanced off-support model extrapolation. This motivates us to design and implement our practical algorithm. In particular, Corollary 4.1
> provide a stronger regret guarantee than Theorem 4.1, and position our work within the existing literature [1,2]. Under a weak partial coverage assumption, our algorithm achieves ``best efforts'' guarantee, indicating no other policy covered by the dataset is better than our learned policy. To the best of our knowledge, this is the strongest regret result in the existing literature.
>
> We also note that Theorem 4.1 do have a tighter bound than the bound in Theorem 5.1 (for our practical algorithm). Theorem 4.1 has a fast rate $\mathcal{O}(n^{-1/2})$ compared to $\mathcal{O}(n^{-1/4})$ in Theorem 5.1. Essentially, our practical algorithm trade-off some sample efficiency to gain great computational efficiency.
>
> > **Weakness and Question 3: Utility of of $\widetilde{\sigma}_n$**
>
> Thanks for raising this point. Theoretically speaking, the $\widetilde{\sigma}_n$ can be set by the condition in Theorem 4.1. However, in practice, it is quite challenging to find a universal rule to identify the optimal threshold. As you mentioned, it depends on how much pessimism the users would like to consider and how well the dataset is explored. We follow the convention [2,3] to set $\widetilde{\sigma}_n$ with a certain rate according to our theoretical results. For implementing our practical algorithm, we set $\lambda$ be of order $\mathcal{O}(\frac{n^{1/4}}{d\log(\bar{V}\sqrt{n})})$ (the effect of $\widetilde{\sigma}_n$ is transferred to $\lambda$ in the practical algorithm). Fortunately, our algorithm has desired robustness to choices of hyperparameters. It has been shown from our
> sensitivity analyses on the walker2d, hopper, and halfcheetah datasets in **Figure 2** of the global response PDF (we only report halfcheetah due to page limits). The policy performance is robust over a wide value range ($\lambda,c^*$ in $[1,0.01]$), and the performance under our choice $(c^*=0.1,\lambda=0.1)$ is close to the best.
>
> > **Minors**
>
> Thanks for your careful review, we have corrected the typos.
>
> **References:**
>
> [1] Jin, Y., Yang, Z. and Wang, Z. (2021), Is pessimism provably efficient for offline rl?, ICML
>
> [2] Lee, J., et al. (2021), Optidice: Offline policy optimization via stationary distribution correction estimation, ICML
>
> [3] Ma, J. Y., et al. (2022), Offline goal-conditioned reinforcement learning via f-advantage regression, Neurips.

---

> > ### Comment · Reviewer_C4A2 · 2023-08-13
> > **Response to Author's Comments**
> >
> > I thank the authors for providing a detailed response and addressing most of my concerns. After going through authors' responses and other reviewers' comments, my concern regarding the significance of information theoretic results still remains unaddressed.
> >
> > * **Significance of Information-Theoretic Results**- While the information-theoretic bound is used for decomposition analysis, I still struggle to see its direct connection with the penalized adversarial estimation algorithm. If the central contribution of this result is in characterizing the role of off-support and on-support errors, then the result can be considered as an intermediate step in the overall algorithm development. At best, we can conclude that the result leads to a theoretically tighter bound of Theorem 4.1 whose practical implementation still remains an open question.

---

> > > ### Author Response · Authors · 2023-08-14
> > > **Response to Reviewer C4A2**
> > >
> > > Thanks for your timely response and for agreeing with our explanations to most of your questions!
> > >
> > > To better answer your question and avoid any unnecessary confusion, we first briefly summarize the logical flow of the paper.
> > >
> > > 1. We propose a robust OPE procedure.
> > > 2. We leverage the above procedure to construct a bi-level policy optimization algorithm, which is our information-theoretic algorithm.
> > > 3. We establish strong regret guarantees for the information-theoretic algorithm, e.g., Theorem 4.1, and we name the results as information-theoretic results.
> > > 4. We find our information-theoretic algorithm is computationally inefficient due to solving constrained optimization, while the information-theoretic results demonstrate desirable theoretical properties of the algorithm.
> > > 5. We propose a practical algorithm, i.e., the penalized adversarial estimation, for implementation and empirical evaluation.
> > > 6. We study the theoretical properties of the practical algorithm, i.e., Theorem 5.1, and find it inherits desirable characteristics of the information-theoretic algorithm but with sample complexity degradation.
> > > 7. We perform a study of the practical algorithm in linear MDPs.
> > >
> > > > **The importance of the information-theoretic results**:
> > >
> > > By the logical flow, we first develop the information-theoretical algorithm and it is natural to study its theoretical properties. In addition, there are several major reasons that explain why the information-theoretic results matter.
> > >
> > > - As a theoretical-based paper (while we still provide a practical algorithm with comprehensive empirical evaluation), our information-theoretic results help to position our work in the existing literature and highlight our theoretical contributions in offline RL. We can fairly compare our results to the existing information-theoretic results, e.g., [1,2,3,4],
> > > in terms of the function approximation and coverage conditions, sample complexity, horizon dependency, and bound tightness. Following the convention, all these existing algorithms are purely information-theoretic without practical implementation.
> > >
> > > - The information-theoretic findings enhance the understanding of the practical algorithm. Specifically, a comparison between Theorem 4.1 and 5.1 reveals that Theorem 4.1 has a tighter bound than Theorem 5.1. This delivers an important message: to attain computational efficiency, the practical algorithm compromises on sample efficiency, i.e.,  $O(n^{-1/4})$ vs $O(n^{-1/2})$.
> > >
> > > - The information-theoretic results provide a well theoretical justification for the information-theoretic algorithm. While the algorithm currently grapples with computational efficiency challenges, resolving these issues could offer significant advantages. This paves the way for developing a more refined algorithm in offline RL.
> > >
> > > > **The connection between the information-theoretic results and the practical algorithm**:
> > >
> > > By the logical flow, although the information-theoretic results are not directly developed for the practical algorithm, they do have some indirect relationships. The error decomposition technique is shared between the information-theoretic results and the results for the practical algorithm, while the development of the latter is much more involved (optimization error, mirror-descent update, and a trajectory-dependent bound). Second, following a similar proofing strategy, the results in Theorem 4.2, Corollary 4.1, 4.2 can be established for the practical algorithm as well. To keep completeness, we will add these new results in the updated supplementary materials.
> > >
> > > > **Bound tightness**:
> > >
> > > In terms of the dependency on $n$ or horizon, Theorem 4.1 is an ultimate regret bound and has achieved the minimax results [5]. Furthermore, as indicated by Theorem 4.1, it observes a trade-off between bias $\epsilon_b$ and variance $\epsilon_\sigma$, affecting the bound tightness. In fact, Theorem 4.1 intrinsically chooses the best bias-variance decomposition through the best choice of $\mathcal{U}^\star_2$. It's important to note that this decomposition is solely a proof technique, not a knob in the algorithm. Combine with the adaptation $\rho$ via $\min\{\rho: ||\rho/\mu||_{L_2(\mu)} \leq \mathcal{U}^\star_2\}$, we can conclude that our regret bound is hard to be further improved unless with additional conditions e.g., coverage or explicit MDP structure.
> > >
> > >
> > > We hope these explanations adequately address your concern. We remain dedicated to refining our paper and welcome any further comments. Once again, thank you for your invaluable contribution to our work!
> > >
> > > **References:**
> > >
> > > [1] Jin, Y., et al. (2021), Is pessimism provably efficient for offline rl, ICML
> > >
> > > [2] Uehara, M., et al. (2022), Pessimistic model-based offline rl, ICLR
> > >
> > > [3] Zhan, W., et al. (2022), Offline rl with realizability and single-policy concentrability, COLT
> > >
> > > [4] Chen, J., et al. (2022), Offline rl under value and realizability, UAI
> > >
> > > [5] Xie, T., ta al, (2020), Q* approximation for batch rl, UAI

---

> > > > ### Comment · Reviewer_C4A2 · 2023-08-18
> > > > **Response to Comments**
> > > >
> > > > Thank you for the response. The discussion on tightness of bound and importance of results above clarifies my concern. I encourage the authors to add the above discussion as part of the Appendix. It would also be worthwhile to tailor Section 4 in order to highlight the tightness of bound and its role in motivating the empirical algorithm.
> > > >
> > > > Since the authors have addressed all of my concerns pertaining to the validation of theoretical results (regret bound, ablation study and information-theoretic results) and utility of the most important hyperparameter, I would like to raise my score and support acceptance of the work. I thank the authors for their efforts.

---

> > > > > ### Author Response · Authors · 2023-08-18
> > > > > **Thank you for your constructive and positive feedback**
> > > > >
> > > > > We are grateful for your recognition and support for the acceptance of our work! It's glad to know our response sufficiently addressed all your concerns. We really enjoy having such insightful discussions with the reviewer. Your invaluable feedback during the discussion and rebuttal phase has definitely enhanced the quality of our paper.
> > > > >
> > > > > Following your constructive suggestions, we will incorporate a discussion section in the updated Appendix and emphasize the significance of the regret bound and its influence on the practical algorithm in Section 4.
> > > > >
> > > > > Once again, thank you for the time and contributions you've devoted to our work!

---

### Official Review · Reviewer_UsZq · 2023-07-07

**Soundness:** 3 good
**Presentation:** 3 good
**Contribution:** 3 good
**Rating:** 6
**Confidence:** 2

**Summary:**

This paper presents a bi-level structured policy optimization algorithm for offline reinforcement learning (RL). The algorithm aims to address the distributional shift and function approximation challenges in offline RL by modeling a hierarchical interaction between the policy and the value function. The lower level focuses on constructing a confidence set of value estimates to control uncertainty arising from distribution mismatch, while the upper level aims to maximize a conservative value estimate. The paper provides theoretical guarantees for the algorithm and evaluates its performance on both synthetic and real-world datasets.


**Strengths:**

- The paper proposes a novel bi-level structured policy optimization algorithm that addresses the challenges of distributional shift and function approximation in offline RL. This algorithm offers a comprehensive solution for real-world scenarios.
- The algorithm preserves the flexibility of the exploratory data distribution and enables model extrapolation, enhancing its power.
- The paper provides theoretical regret guarantees for the proposed algorithm without relying on data-coverage and completeness-type assumptions, increasing its applicability.


**Weaknesses:**

- There is only a weak connection between several parts of the paper. It is hard to find a central claim that supports the logical flow.
- There remain some issues unsolved in the paper. See the questions for details.


**Questions:**

- At the lower level, how can the confidence set, which constraints model learning, effectively prevent overly pessimistic evaluation?
- How can the proposed algorithm build on weaker conditions than the global data coverage? Does it benefit from the occupancy distribution estimation?


**Limitations:**

Yes

---

> ### Author Rebuttal · Authors · 2023-08-09
>
> We thank the reviewer for the suggestive feedback, which greatly improves our paper! We address the reviewer's comments point by point below.
>
> **Weakness 1: Connection and central claim**
>
> We feel sorry for the confusion and thanks for helping us improve the readability of the paper. The offline policy optimization with limited exploration we study is a very challenging and complicated problem, especially when requiring sample efficiency and strong theoretic regret guarantees.  Reaching the ultimate algorithms necessarily involves numerous intermediary steps and thorough analysis. The central claim of our paper is that we propose a novel framework with strong theoretical guarantees without requiring coverage and Bellman completeness conditions, demonstrating the potential applicability in the limited-explored real-world datasets. The logical flow of the paper can be shortly summarized as: 1) we propose a bi-level structured algorithm in limited data exploration scenarios, which is bridged from a novel robust off-policy confidence interval evaluation. 2) We make great efforts to establish the theoretic regret guarantee for this algorithm, and analyze its regret without requiring coverage and completeness-type assumptions. 3) To implement the developed algorithm and practically solve it, we further construct a penalized adversarial estimation procedure. Within this framework, we still provide the regret analysis for this practical algorithm as well as obtain a strong theoretical guarantee with the relaxed assumption (relative condition number) in a linear MDP case study. 4) We empirically evaluate our practical algorithm via synthetic and real datasets, and compete with the state-of-the-art models. In the updated manuscript, we have made revisions to further improve the readability and strength the connections between several parts, we add and polish the transitional paragraphs that clearly show how one part leads to the next.
> Throughout the paper, we regularly remind the reader of how each part is connected to the central claim. We would also try to include more motivation for each support step and model. Due to the page limit, we might not be fully able to include them, but we tried our best to give more motivation in this revision.
>
>
> > **Question 1: Prevent overly pessimistic evaluation**
>
> Thanks for raising this good question. The reason for preventing overly pessimistic evaluation is two-fold: First, we implement pessimism on a consistent confidence set, regarding a sufficiently small weighted average Bellman error (line 153: Consistency). This is ensured via $\varepsilon_{n}$ as well as the concentration of empirical measures
> This procedure ensures any function in the set is consistent with the Bellman equations, which limits the possibility of picking up a function with an overly pessimistic evaluation. More importantly, according to Theorem 3.1, the confidence set at the lower level is evaluated directly for $J(\pi)$,  i.e., the value function at the initial state $q^{\pi}(s^{0},\pi)$. This distinguished us from the pointwise pessimistic penalties used in some close works [1]. They require the uncertainty lower bound of value function holds for any $(s,a) \in \mathcal{S} \times \mathcal{A}$ (see Definition 4.1 in [1]) instead of the initial state, and thus is more likely to cause overly pessimistic reasoning. Numerically, we conduct an ablation study on the lower-level components in **Figure 3** of the global response PDF. Figure 3 (a) shows that when we remove the Bellman consistency component, our algorithm with only uncertainty control (Algo: UC) is quite unstable without a consistent estimation guarantee on $q^{\pi}$,  and suffers great performance degradation due to potentially overly pessimistic.
>
>
> > **Question 2: Weaker assumption and occupancy distribution**
>
> Thank you for raising that point. Yes, the concept of the occupancy distribution estimation closely connects the marginalized importance weights (the ratio between the occupancy distribution between offline dataset and target/exploration policy) [2]. They play important roles in our proposed algorithms. First, by leveraging the marginalized importance weights, we propose a robust confidence interval estimation (Theorem 3.1) for $J(\pi)$ to construct a Bellman consistency set (line 153: Consistency). Once this consistency set is realized, we can then aim to maximize the rewards the trained agent would obtain in the worst possible MDP that is consistent with the observed dataset (Equation 6), i.e., performing pessimistic extrapolation from offline data without global coverage [3]. Second, the key component in our algorithm: the uncertainty control, essentially constrains the deviation from the occupancy distribution of the offline dataset to the occupancy distribution of the exploration policies via the developed discrimination function which takes the marginalized important weights as input. Thanks to this, our algorithm essentially eliminates the need to evaluate out-of-distribution actions, thus avoiding the problematic overestimation of values in poor (non-global) data coverage scenarios [5]. Note that if performing the above-mentioned uncertainty control in the space of policies [6] rather than in the space of occupancy distribution, it may fail to avoid over-estimation issues.
>
> **References:**
>
> [1] Jin, Y., Yang, Z. and Wang, Z. (2021), Is pessimism provably efficient for offline rl?, ICML.
>
> [2] Liu, Q., et al. (2018), Breaking the curse of horizon: Infinite-horizon off-policy estimation, Neurips.
>
> [3] Uehara, M. and Sun, W. (2022), Pessimistic model-based offline reinforcement learning under
> partial coverage, ICLR.
>
> [4] Lee, J., et al. (2021), Optidice: Offline policy optimization via stationary distribution correction estimation, in ICML.
>
> [5] Wu, Y., Tucker, G. and Nachum, O. (2019), Behavior regularized offline reinforcement learning,
> arXiv preprint arXiv:1911.11361.

---

> > ### Comment · Reviewer_UsZq · 2023-08-19
> > **Response**
> >
> > I acknowledge the authors' rebuttal and I remain my rating towards acceptance.

---

> > > ### Author Response · Authors · 2023-08-19
> > > **Thank you for your recognition!**
> > >
> > > Thanks for your support towards acceptance of our work. We greatly appreciate your time and effort to review our paper and provide us with constructive comments and insightful suggestions!

---

> ### Author Response · Authors · 2023-08-16
> **Follow-up on Our Response to Your Feedback**
>
> We greatly appreciate your valuable and constructive comments, as well as the time and effort devoted!
>
> Following your insightful suggestions, we have conducted ablation studies and additional experiments to clarify the point of preventing overly pessimistic evaluation. Furthermore, we have provided justification for our data-coverage conditions, and the utility of the occupancy distribution. Additionally, we have also strengthened the central claim, ensuring coherence across the paper. All the details can be founded in our response and the attached one-page PDF file.
>
> We sincerely hope our additional clarifications and experiments can fully address your questions and can be helpful in the evaluation of our results. We are eagerly looking forward to your kind feedback.

---

### Official Review · Reviewer_7QtD · 2023-07-09

**Soundness:** 3 good
**Presentation:** 3 good
**Contribution:** 3 good
**Rating:** 7
**Confidence:** 4

**Summary:**

This paper proposed a pessimistic offline RL method. Different from the previous method in the function approximation setting, the method proposed in this paper does not require coverage and completeness assumption. Moreover, in terms of algorithm design, this paper derives a lower confidence bound for RL objective function and explicitly optimizes it to obtain the policy, which is the first attempt in this direction. Both theoretical and empirical results are presented to support the effectiveness of this algorithm design.

**Strengths:**

(1) This paper proposes a novel algorithm design to address the offline policy optimization problem by explicitly optimizing a lower bound of RL objective function. This design is first proposed to my best knowledge, which is valuable.
(2) The algorithm design proposed in this paper has a very global optimality guarantee under weaker assumptions compared with previous works.
(3) The author also derives a practical version of the algorithm which also has a theoretical guarantee. Moreover, to demonstrate the effectiveness of this practical version, several experiment results are presented.

**Weaknesses:**

The experiment is not very strong:
(1) Most of the standard simulation experiments are not considered thus the practical effectiveness of the algorithm is not convincing. The author might need to include more experiments are also included in OptiDICE paper (hopper, walker, halfcheetah, maze etc)
(2) I was also wondering how the algorithm proposed in this paper compared most recent baseline such as TD3 + BC?

**Questions:**

(1) If $\mathcal{U}^\tau_{\infty}=+\infty$, then how would the constant $\mathcal{U}^\tau_2$ and $\mathcal{U}^*_2$ look like, will them become unbounded as well? If that is the case, then the proposed algorithm still needs completeness assumption and coverage assumption to derive the bound in Theorem 5.1.? I don't think requiring coverage assumption for this paper should be a reason for rejection since this paper already made a lot of contributions, but the author needs to provide a clear discussion for this part otherwise it will be somehow misleading since it is not obvious how to achieve global optimal policy without any coverage requirements.

(2) In linear MDP the algorithm complexity is $n^{-1/4}$? If so then why the algorithm has worse dependence on $n$? Can the author also provide some high-level explanation?

(3) OptDICE has been proven to outperform CQL and other baselines (see the original paper of OptiDICE, why OptiDICE performs much worse than CQL?

I will raise my score if the author can answer the above questions appropriately.

**Limitations:**

No limitation.

---

> ### Author Rebuttal · Authors · 2023-08-09
>
> We thank the reviewer for the suggestive feedback, which greatly improves our paper! We address the reviewer's comments point by point below.
>
> > **Weakness 1: Experiments**
>
> Thanks for the suggestion! We evaluate our algorithm on walker2d, hopper, halfcheetah, and maze2D D4RL benchmarks. The results are reported in **Table 1** of the global response PDF. In addition, we add the baseline TD3+BC for comparison in both D4RL datasets and the original datasets (**Table 3** in the global response PDF).
>
> > **Question 1 (a): Boundedness of $\mathcal{U}^\tau_{2}$ and $\mathcal{U}^\star_{2}$ when $\mathcal{U}^{\tau}_{\infty} = +\infty$**
>
> Thanks for raising this point. First, we would like to clarify that $\Omega$-class is user-specified, intending to model the true marginalized importance weight (MIW). Thus, the boundedness of $\Omega$-class, i.e., $\mathcal{U}^\tau_{\infty}$ is determined by users. In practice, the existing works tend to consider $\Omega$ in a bounded class  ($\mathcal{U}^\tau_{\infty} < \infty$) [1]. Second, unlike the $\Omega$-class, the MIW might go to $+\infty$, e.g., in poor coverage scenarios. In this case, using a bounded class to model an unbounded MIW, may cause trouble due to model misspecification. Fortunately, our robust interval learning is motivated to solve this issue, and robust to the misspecification of $\Omega$ (lines 128-133). Third, even if setting $\mathcal{U}^\tau_{\infty} = +\infty$, the $\mathcal{U}^\tau_{2}$ and $\mathcal{U}^\star_{2}$ could be still finite. Essentially,  $||\tau(s,a)||_{L_2(\mu)}$ is indirectly controlled by the $\widetilde{\sigma}_n$
>  via $\xi_n(\mathbb{G}, \tau)$. The user can specify a finite $\widetilde{\sigma}_n$ to make $\mathcal{U}^\tau_2$ finite as well in any case.
>
> > **Question 1 (b): Necessities of the completeness and coverage assumption for Theorem 5.1.**
>
> Thanks for this insightful question.
>
> **For Bellman-completeness**: Intuitively, the key point of why our algorithm can relax the Bellman-completeness condition is due to our developed weighted average Bellman error. We use this type of Bellman error to establish the Bellman consistency (line 153: Consistency), instead of using a squared or minimax Bellman error in the famous API/AVI-type algorithms, e.g., [2,3,4]. Their algorithm require finding $f \in \mathcal{F}$ to minimize  $||f-\mathcal{B}f ||^{2}_{L_2(\mu)} \approx 0$ to obtain $f \approx q^{\pi}$. Unfortunately, the empirical squared Bellman error is biased due to the "double-sampling" issue. They need a separate helper class $\mathcal{G}$ to model $\mathcal{B}f$. It has been shown by [4], only when the class $\mathcal{B}f \in \mathcal{G}$ (Bellman-completeness), their estimation is unbiased. In contrast, thanks to not using the squared loss, our weighted average Bellman error can be estimated from an unbiased estimate without concern about the "double sampling" issue.
>
> **For coverage assumptions**: To derive Theorem 5.1, we indeed do not make any coverage assumptions. This is attributed to our novel error decomposition analysis (Equations I.19 to I.24 on p.53-p.54 of Appendix), which is one of the theoretic contributions of our work. In comparison to prior works, e.g., [5,6], our decomposition analysis makes the off-support errors explicitly analyzed in the regret bound, and further facilitate uncovering nice properties of our algorithm, e.g., the enhanced off-support model extrapolation which improves the applicability of our algorithm in poor coverage scenarios. Moreover, we should notice that the regret in Theorem 5.1 might not be near-optimal since the off-support error is not accessible without any coverage information. To achieve near-optimal regret with a "best-effort" guarantee (for global optimal policy), we still require some minimum partial coverage assumption, e.g., as used in Corollary 4.1. Note that our partial coverage condition (Definition 4.1) is weaker than the existing ones. Numerically, in **Figure 1** of the global response PDF, we study our regret convergence rate when competing with the global optimal policy.
>
>
> > **Question 2: Sample complexity**
>
> This is a good point! Yes, the sample complexity of our practical algorithm is $\mathcal{O}(n^{-1/4})$ which is worse than $\mathcal{O}(n^{-1/2})$ in the information-theoretic algorithm.
>  The worse dependence arises from the use of regularization. To achieve computational efficiency and tractability, our practical algorithm performs uncertainty control and Bellman consistency (line 153) via regularization instead of constrained optimization. The strength of regularization needs to be controlled for balancing the bias-variance trade-off. When setting the regularization coefficients of order ${\mathcal{O}}\big(\sqrt[4]{{n}}\big)$ as stated in Theorem 5.2, the sample complexity degrades from a minimax rate $\mathcal{O}(n^{-1/2})$ to $\mathcal{O}(n^{-1/4})$. The sample complexity degradation due to the regularization can be also found in prior works [3,6]. In **Figure 1** of the global response PDF, our numerical regret analysis confirms the $\mathcal{O}(n^{-1/4})$ rate.
>
> > **Question 3: OptDICE vs CQL**
>
> It seems there may have been a misunderstanding regarding the results. In our paper, the average policy performance of OptiDICE is consistently better than the performance of CQL in both synthetic and real datasets.
>
> **Reference:**
>
> [1] Uehara, M., et al. (2020), Minimax weight and q-function learning for off-policy evaluation, ICML.
>
> [2] Munos, R. and Szepesvari, C. (2008), Finite-time bounds for fitted value iteration, JMLR.
>
> [3] Xie, T., et al. (2021), Bellman-consistent pessimism for offline reinforcement learning, Neurips.
>
> [4] Chen, J. and Jiang, N. (2019), Information-theoretic considerations in batch reinforcement learning, ICML.
>
> [5] Jin, Y., et al. (2021), Is pessimism provably efficient for offline rl, ICML.
>
> [6] Zhan, W., et al. (2022), Offline reinforcement learning with realizability and single-policy concentrability, COLT.

---

> ### Author Response · Authors · 2023-08-16
> **Eagerly Looking Forward to Your Feedback on Our Response**
>
> We are very grateful for your time and effort devoted to reviewing our work, as well as for giving us insightful, valuable, and encouraging comments!
>
> Following your constructive suggestions, we have conducted additional four sets of experiments on D4RL benchmarks and comparing with the most recent baseline TD3+BC. Furthermore, we have clarified the boundedness on $\mathcal{U}^\tau_2 (\mathcal{U}^\star_2)$, the coverage and completeness-type assumptions, and the sample complexity, among other points. All these details are provided in our response and the accompanying one-page PDF file.
>
> We sincerely hope our response can appropriately address your concerns and can be helpful in the evaluation of our work. We are eagerly looking forward to your kind feedback!

---

### Official Review · Reviewer_75WN · 2023-07-31

**Soundness:** 2 fair
**Presentation:** 3 good
**Contribution:** 2 fair
**Rating:** 4
**Confidence:** 4

**Summary:**

This paper addresses the problem of offline reinforcement learning (RL). Offline RL involves computing a policy based on a static/pre-collected dataset, with no further interaction with the environment permitted. A significant hurdle in offline RL is the distributional shift resulting from insufficient exploration within the dataset. Moreover, previous work may necessitate the assumption of Bellman completeness for the utilized function class.  To overcome these challenges, the authors propose a bi-level structured policy optimization algorithm. This algorithm models a hierarchical interaction between the policy and the value function. At the lower level, the focus is on constructing a set of confident value estimates. The upper level, in contrast, aims to maximize a conservative estimate derived from this confidence set. The algorithm can be solved efficiently through a penalized adversarial estimation procedure. The authors provide theoretical regret guarantees for their algorithm and assess its performance using both synthetic and real-world datasets. Performance is compared against several well-known offline RL algorithms, such as CQL, IQL, and COMBO.

**Strengths:**

Strengths:

1. The concept of optimizing the confidence lower and upper bounds in equation (4) over τ within Ω, with the aim of tightening the Confidence Interval (CI), is intriguing and holds potential for generating a more precise confidence interval.

2. The paper, overall, is well-written and well-structured, demonstrating a clear and logical progression of ideas.

3. The authors have given a comprehensive view of their algorithm by providing both theoretical analysis and empirical versions. Additionally, the experimental results demonstrate the effectiveness of the proposed algorithm, as shown on various synthetic datasets as well as a real-world dataset.

**Weaknesses:**


Weaknesses:

1. Complexity: The algorithm, in comparison to alternatives such as IQL and COMBO, is notably more complex. It employs a game-theoretical formulation and requires the resolution of two optimization problems at each iteration. This results in an increase in both the number of hyperparameters that require tuning and the overall computational cost, and potentially instability in the optimization (i.e., convergence).

2. Ambiguities and Clarifications in Section 3 (Lines 119 to 126):
   - Shouldn't $G (x_∗) = \sup_x \{x \cdot x_∗ − G(x)\}$ be revised to $G_* (x_∗) = \sup_x \{x \cdot x_∗ − G(x)\}$?
   - For $J(\pi) = q_\pi(s_0, a) + E_\mu[\lambda G_*((B^\pi q^\pi (s, a) − q^\pi (s, a))/(1 − \gamma))]$, where $\lambda \geq 0$, the initial term appears as if it should be $q_{\pi}(s_0, \pi)$. Additionally, could you explain how $\lambda$ is typically chosen? Does the choice of $\lambda$ significantly impact the algorithm's performance?
   - The derivation from $J(\pi) = q_\pi(s_0, a) + E_\mu[\lambda G_*((B^\pi q^\pi (s, a) − q^\pi (s, a))/(1 − \gamma))]$ to equation (1) is unclear. Why is $\lambda$ merely a coefficient of $G(x)$? Shouldn't it also be a coefficient of  $x · (B^\pi q^\pi (s, a) − q^\pi(s, a))$?
   - The terms L(τ,q,π) and ξ(G,τ) require explicit definitions.

3. Justification for Assumption 2: The justifiability of Assumption 2 is uncertain. It appears to be more of a requirement for feasible derivation rather than a naturally occurring condition. Can it be guaranteed that $\tau(s,a)$ won't reach infinity in real-world problems? More justification on this point would be appreciated.

4. Need for Intuition: More insight into why your framework doesn't require the Bellman completeness assumption would be beneficial.

5. Additional Evaluations: It would be helpful if you could evaluate your proposed algorithm using more standard benchmarks, such as D4RL. This would significantly aid in positioning your work within the existing literature.

**Questions:**

See Weakness.

**Limitations:**

See Weakness.

---

> ### Author Rebuttal · Authors · 2023-08-09
>
>
> We thank the reviewer for the suggestive feedback, which greatly improves our paper! We address the reviewer's comments point by point below.
>
> > **Weakness 1: Complexity of the algorithm**
>
> Thanks for raising this point. We would like to make a discussion on the three aspects.
>
> **Complexity**: we agree with the reviewer that our algorithm is more complex than the minimalist baselines, e.g., IQL, but the complexity of our algorithm is still handleable due to a two-fold reason. First, the mirror descent (step 4 of Algorithm 1) has a closed-form
> exponential update rule. It follows from [1], we have the analytical update rule as  $\pi^{{k}}(\cdot|s) \propto \pi^{{k-1}}\exp(\zeta q^{k}(s,\cdot))$. Thus, the computational cost for the policy update is minimal. Second, the inner optimization in step 3 of Algorithm 1 is easy to solve. On the one hand, it could yield a closed-form global solver using the maximum mean discrepancy principle [2]. On the other hand, one can utilize the feature mapping class for approximation, which is computationally efficient and can be solved by various algorithms [3]. From this perspective, the computational burden for the inner optimization is light, making the minimax problem predominantly a maximization problem. For more details, please refer to Appendix A in the supplementary materials. Numerically, in **Table 2** of the global response PDF, we test the runtime of our algorithm and prior works, which shows our algorithm is comparable to other baselines.
>
> **Convergence**: Theoretically, in Theorem A.1 of Appendix A, we show that our algorithm is provably convergent and realizes a fast sublinear rate $\mathcal{O}(1/T)$ even under non-linear settings. Numerically, we plot the policy performance across optimization epochs of our algorithm in **Figure 3 (a)** in the global response PDF, illustrating the numerical convergence and stability of our algorithm (Algo: BC+UC).
>
> **Hyperparameter-tuning**: Indeed, tuning parameters selection is an open problem in offline policy optimization. Fortunately, our algorithm has desired robustness to choices of hyperparameters, when we set the hyperparameters satisfying the conditions in Theorem 5.2, i.e., $\mathcal{O}(\frac{n^{1/4}}{d\log(\bar{V}\sqrt{n})})$. We follow your suggestion to conduct sensitivity analyses on the walker2d, hopper, and halfcheetah datasets. In **Figure 2** of the global response PDF (we only report for halfcheetah in the PDF due to page limit), it shows that our policy performance is robust over a wide value range of parameters ($\lambda, c^* \in [1,0.01]$), and the performance of under our choice $(c^*=0.1,\lambda=0.1)$ is close the best.
>
> > **Weakness 2: Clarifications in Section 3**
>
> Thanks for your careful reading. Accordingly, we have corrected the typos: $q^\pi(s^0,\pi)$, $\mathbb{G}_{*}(x)$, and
>
> $$E_\mu[\lambda\mathbb{G}_*((\mathcal{B}^{\pi}q^{\pi}(s,a)- q^{\pi}(s,a))/\lambda)/(1-\gamma)].$$
>
> In addition, we explicitly define $L(\tau,q,\pi)$ and $\lambda \xi(\mathbb{G},\tau)$. For the choice and sensitivity analysis of $\lambda$, please see our last response (hyperparameter-tuning).
>
>
> > **Weakness 3: Assumption 2**
>
> Thank you for pointing this out! First, we note that this boundedness condition on $\Omega$-class is standard in literature when modeling marginalized importance weight (MIW), e.g., [2]. Also, since the $\Omega$-class is user-specified, intending to model the true MIW, the boundedness condition is not really an assumption
> because the users can make a choice of $\Omega$-class to satisfy it.
> In our paper, we leave it as a formal assumption to keep the analysis general. Second, unlike the $\Omega$-class, the MIW
> indeed might go to $+\infty$, e.g., in poor coverage scenarios. In this case, using a bounded class to model an unbounded MIW may cause trouble due to model misspecification. Fortunately, our robust interval learning is motivated to solve this issue, and robust to the misspecification of $\Omega$ class (lines 128-133 of the main paper).
>
> > **Weakness 4: Bellman completeness**
>
> Intuitively, the key insight is that we develop a weighted average Bellman error to establish the Bellman consistency (line 153: Consistency). This distinguished our algorithm from the famous API/AVI-type algorithms, e.g., [4,5,6], which rely on a squared or minimax Bellman error. It requires finding $f \in \mathcal{F}$ to minimize  $||f-\mathcal{B}f ||^{2}_{L_2(\mu)} \approx 0$  to obtain $f \approx q^{\pi}$. Unfortunately, the empirical squared Bellman error is biased due to the *double-sampling* issue. They need a separate helper function class $\mathcal{G}$ to model $\mathcal{B}f$, and it has been shown by [6] that only when the class $\mathcal{B}f \in \mathcal{G}$ (Bellman-completeness), the estimation is consistent and unbiased. In contrast, thanks to not using the squared loss, our weighted average Bellman error (use MIW as the weight to mimic the data collected by different exploration policies) is allowed to be estimated by an unbiased sample estimate (i.e., no *double-sampling* issue).
>
> > **Weakness 5: D4RL benchmarks**
>
> Thanks for your great suggestion!  We evaluate our algorithm on D4RL benchmarks (walker2d, hopper, halfcheetah, and maze2d). The results of our algorithm and baselines are summarized in **Table 1** of the global response PDF.
>
> **References:**
>
> [1] Parikh, et al. (2014), Proximal algorithms, Foundations and trend in Optimization.
>
> [2] Uehara, M., Huang, J. and Jiang, N. (2020), Minimax weight and q-function learning for off-policy evaluation, ICML.
>
> [3] Sriperumbudur, B. K., et al. (2011), Universality, characteristic kernels and RKHS embedding of measures, JMLR.
>
> [4] Munos, R. et al. (2008), Finite-time bounds for fitted value iteration, JMLR.
>
> [5] Xie, T., et al. (2021), Bellman-consistent pessimism for offline reinforcement learning, Neurips.
>
> [6] Chen, J. and Jiang, N. (2019), Information-theoretic considerations in batch reinforcement learning, ICML.

---

> > ### Comment · Reviewer_75WN · 2023-08-14
> >
> >
> > Thank you for the response. I appreciate the additional results.
> >
> > However, it seems that the performance was significantly influenced by the hyperparameters $ \lambda $ and $ c^\star $, with the best results achieved around 50, and the lowest at 30. I wonder if the authors could provide more details about the experimental setup for the D4RL experiments, as the devil is often in the details.
> >
> > Also, did you report the results with the best hyperparameters tuned online? Did you tune the hyperparameters for each task as well? If that's the case, it might be unfair to claim the algorithm performs the best, as methods like COMBO, they did not tune the hyperparamters online.

---

> > > ### Author Response · Authors · 2023-08-15
> > > **Response to Reviewer 75WN**
> > >
> > >
> > > Thank you for your kind reply! We think many of your suggestions and comments are of great value to make our paper stronger.
> > >
> > > > **Sensitivity Analysis:**
> > >
> > > Thanks for your careful observation! In the sensitivity analysis, we intentionally challenge our algorithm and test whether it works in extreme cases (over-pessimism and inadequate-pessimism) **by setting** $\lambda, c^*$ **with extremely abnormal values**. We can find that the hyperparameter values that result in the $30$ return are set at $\lambda  = c^*=10$ (over-pessimism). These values are **scaled 100 times larger than the** $\lambda=c^*=0.1$ **yielded by our selection rule**. A similar pattern can be observed when we set $\lambda=c^*=0.0001$ (inadequate pessimism), which is 100 times smaller than our selected choice. In contrast, when the $\lambda, c^*$ are set in a relatively normal range, e.g., $\lambda, c^* \in [1,0.001]$, the policy performance is **quite stable** with the highest around $50$ and the lowest around $47$ return.
> > >
> > > > **Experiments details and hyperparameter tuning:**
> > >
> > > For all D4RL tasks, we **fully offline** tune our hyperparameters following our selection rule (discussed later), instead of online tuning. We provide the experiment details as follows.
> > >
> > > **Our algorithm:**
> > >
> > > We use separate $2$-layer fully connected neural networks for function approximation, where each
> > > hidden layer width is
> > > $256$ with ReLU activation. The policy is Gaussian, with the mean and the standard deviation parameterized by the neural network. We use Adam as the optimizer with a minibatch size $256$. We set the learning rate $\eta^t$ for the $t$-th iteration be $\frac{\eta^0}{1+0.3\cdot t^{1/4}}$, where the $\eta^0$ is fixed to be $1e-3$. This learning rate follows the stepsize rule in Theorem A.1. We set the stepsize $\zeta = 3e-4$, which was selected offline with a heuristic: we consider the set {$1e-4, 3e-4, 1e-5$}, and found $\zeta = 3e-4$ works well for almost all halfcheetah and maze2d tasks. For each D4RL task, our hyperparameters $c^{*}$ and $\lambda$ are set **fully offline** by following the selection rule satisfying the conditions in Theorem 5.2, i.e., $\frac{2\cdot n^{1/4}}{3\cdot d\log(\bar{V}\sqrt{n})}$. We provide our selected hyperparameters in each task as follows.
> > >
> > > | Gym  locomotion    | Hypereparameters |
> > > |----------------------------|------------------|
> > > | walker2d-medium            | 0.25             |
> > > | walker2d-medium-replay     | 0.1              |
> > > | walker2d-medium-expert     | 0.35             |
> > > | walker2d-random            | 0.25             |
> > > | hopper-medium              | 0.4              |
> > > | hopper-medium-replay       | 0.25             |
> > > | hopper-medium-expert       | 0.5              |
> > > | hopper-random              | 0.4              |
> > > | halfcheetah-medium         | 0.25             |
> > > | halfcheetah-medium-replay  | 0.1              |
> > > | halfcheetah-medium-expert  | 0.35             |
> > > | halfcheetah-random         | 0.25             |
> > > | **Maze2d**               |  |
> > > | maze2d-umaze               | 2                |
> > > | maze2d-medium              | 2.25             |
> > > | maze2d-large               | 2.5              |
> > >
> > > **We provide the experiment details for the competing methods as follows:**
> > >
> > > TD3+BC: We mostly follow default setups in the original paper but we set $\lambda=\frac{\alpha}{\frac{1}{n}\sum_{(s, a)}|q(s, a)|}$ as suggested in [1], to improve the policy performance.
> > >
> > > CQL: We follow the author-released github default setup but we modify the actor learning rate and use a fixed $\alpha$ instead of the Lagrange variant. This modification is to match the hyperparameters defined in the original paper, and [1] found the original hyperparameters performed better.
> > >
> > > IQL: We follow the author-released default setups.
> > >
> > > COMBO: We take the results for Gym locomotion from the original paper, and re-run the algorithm in Maze2d. In Maze2d tasks, we apply the author-released selection rule to the set {$1e-4, 3e-4$} for the $q$-function learning rate and the set {$1e-5, 3e-5, 1e-4$} for the policy learning rate: $1e-4$ and $3e-5$ are selected. We consider $\beta \in \{0.01, 0.1, 0.5, 1, 2.5\}$ for conservatism degree, and select $\beta = 1$ for implementation. We leave others as default setups.
> > >
> > > ATAC: We take the results for Gym locomotion from the original paper, and re-run the algorithm in Maze2d. We follow the author-released online selection rule for $\beta$ in the set {$4^{-4}, 4^{-3}, 4^{-2}, 4^{-1}, 4,4^2, 4^3$}. And we finally select $\beta = 4$, $4^{-1}$ and $4^{-1}$ for the maze2d-umaze, medium and large, respectively. We leave others as default setups.
> > >
> > > OptiDCIE, BCQ, BEAR: Results are from the original or D4RL paper.
> > >
> > > $~$
> > >
> > > We hope these explanations adequately address your concern. We really enjoy having such insightful discussions with the reviewer, and thank you once again for your invaluable contributions to our work!
> > >
> > > **Reference**
> > >
> > > [1] Fujimoto, S., et al. 2021, A minimalist approach to offline rl. Neurips

---

> > > ### Author Response · Authors · 2023-08-18
> > > **Additional Response to Reviewer 75WN**
> > >
> > > We performed **additional sensitivity analyses on more D4RL environments** (we just exhausted our HPC resources:) ) and hope the new experiment results can help to resolve the reviewer's concern about hyperparameter-tunning.
> > >
> > > $~$
> > >
> > > We again greatly appreciate the reviewer for your valuable comments and time devoted! All your suggestions have been reflected in our revised paper. We kindly ask the reviewer to consider raising their evaluation if their concerns were appropriately addressed, and we always welcome further feedback!
> > >
> > > $~$
> > >
> > > The following tables provide the experiments of sensitivity analyses on the values of the hyperparameters vs policy performance on the additional D4RL benchmarks (**hopper, walker2d, maze2d**), in addition to the results (halfcheetah) we previously presented in the one-page PDF.  Each number in the following tables is the normalized score of the policy at the last iteration of training, averaged over $3$ random seeds.
> > >
> > > $~$
> > >
> > > **(Gym) Hopper-medium-replay**: Our selection rule chooses $\lambda=c^*=0.25$ with the policy performance $114.0 \pm 2.4$.
> > >
> > > | $c^{*}(\text{col}), \lambda(\text{row})$ | 2.5          | 1            | 0.1          | 0.01          | 0.0025         | 0.001       |
> > > |-----------------------------------------|--------------|--------------|--------------|--------------|--------------|--------------|
> > > | **2.5**                                     | $108.1\pm2.7$| $109.7\pm2.4$| $111.6\pm3.1$| $111.1\pm2.7$| $109.5\pm2.1$| $108.3\pm4.4$|
> > > | **1**                                       | $109.3\pm1.7$| $111.8\pm2.4$| $113.2\pm2.6$| $112.9\pm2.2$| $112.0\pm3.0$| $110.7\pm3.3$|
> > > | **0.1**                                     | $112.6\pm2.1$| $113.3\pm2.0$| $114.4\pm2.9$| $114.6\pm2.1$| $113.2\pm2.9$| $112.5\pm3.4$|
> > > | **0.01**                                    | $111.8\pm2.8$| $112.0\pm3.6$| $114.6\pm2.9$| $114.2\pm3.3$| $113.1\pm2.7$| $110.2\pm3.4$|
> > > | **0.0025**                                   | $109.7\pm2.6$| $111.5\pm3.2$| $113.8\pm2.6$| $113.3\pm4.4$| $112.6\pm3.7$| $110.1\pm3.7$|
> > > | **0.001**                                  | $108.2\pm3.0$| $109.5\pm3.8$| $111.9\pm3.5$| $111.2\pm2.6$| $109.8\pm3.1$| $108.4\pm4.6$|
> > >
> > > $~$
> > >
> > > **(Gym) Walker2d-medium-replay**: Our selection rule chooses $\lambda=c^*=0.1$ with the policy performance $101.2\pm3.2$.
> > >
> > > | $c^{*}(\text{col}), \lambda(\text{row})$ | 2.5          | 1            | 0.1          | 0.01          | 0.0025         | 0.001       |
> > > |-----------------------------------------|--------------|--------------|--------------|--------------|--------------|--------------|
> > > | **2.5**                                     | $95.8\pm2.5$ | $97.4\pm2.8$ | $98.4\pm2.5$ | $99.1\pm3.2$ | $97.8\pm3.0$ | $97.9\pm3.4$ |
> > > | **1**                                       | $97.3\pm2.7$ | $98.0\pm3.1$ | $98.8\pm2.8$ | $99.4\pm3.4$ | $98.7\pm2.7$ | $98.1\pm3.2$ |
> > > |**0.1**                                     | $97.4\pm2.8$ | $98.3\pm2.9$ | $101.2\pm3.2$| $101.3\pm3.4$| $98.9\pm3.6$ | $97.5\pm4.2$ |
> > > | **0.01**                                    | $98.2\pm2.8$ | $99.5\pm2.9$ | $101.7\pm3.9$| $102.6\pm3.4$| $100.2\pm3.1$| $98.4\pm3.3$ |
> > > | **0.0025**                                   | $98.0\pm3.6$ | $97.5\pm4.2$ | $100.1\pm3.6$| $100.8\pm3.5$| $99.2\pm4.2$ | $97.4\pm4.0$ |
> > > |**0.001**                                  | $97.2\pm3.8$ | $98.5\pm3.3$ | $98.2\pm3.8$ | $99.3\pm3.6$ | $97.8\pm5.2$ | $98.2\pm4.1$ |
> > >
> > > $~$
> > >
> > > **Maze2d-medium**: Our selection rule chooses $\lambda=c^*=2.25$ with the policy performance $138.1\pm 7.6$.
> > >
> > > | $c^{*}(\text{col}), \lambda(\text{row})$ | 15            | 10            | 5              | 2.5            | 1              | 0.5            |
> > > |----------------------------------------|---------------|--------------|---------------|---------------|---------------|---------------|
> > > |**15**                                  | $134.5\pm4.6$  | $133.9\pm5.8$ | $134.8\pm4.5$  | $136.7\pm6.2$  | $134.8\pm6.0$  | $134.9\pm5.4$  |
> > > |**10**                                  | $133.7\pm4.2$  | $136.7\pm5.1$ | $135.8\pm6.8$  | $138.4\pm7.4$  | $135.7\pm12.2$ | $137.5\pm8.2$  |
> > > |**5**                                   | $133.9\pm5.8$  | $137.3\pm6.9$ | $136.6\pm5.5$  | $138.3\pm9.2$  | $134.9\pm7.0$  | $135.1\pm5.2$  |
> > > |**2.5**                                 | $137.5\pm6.3$  | $135.8\pm5.9$ | $140.7\pm10.9$ | $138.9\pm9.2$  | $132.2\pm8.1$  | $133.7\pm6.5$  |
> > > |**1**                                   | $134.0\pm4.2$  | $137.2\pm10.7$| $133.8\pm6.9$  | $137.3\pm9.5$  | $138.2\pm5.2$  | $137.6\pm8.0$  |
> > > |**0.5**                                 | $135.2\pm11.8$ | $133.7\pm8.3$ | $136.1\pm7.8$  | $134.5\pm6.7$  | $137.2\pm9.2$  | $135.5\pm7.1$  |
> > >
> > > $~$
> > >
> > > From the tables, we can see that, our algorithm demonstrates robustness over a wide value range of hyperparameters. Also, the policy performance under our hyperparameter choice is close to the best performance in the table, which indicates the effectiveness of our proposed hyperparameter selection rule.

---

### Author Rebuttal · Authors · 2023-08-09

We are grateful to all reviewers for taking the time to review our paper and sharing insightful and positive feedback. We are encouraged that they found the submitted work to be novel (7QtD,UsZq,C4A2) and interesting (75WN, q6tp), addressing a challenging and meaningful research problem (7QtD,UsZq), with a well-written and structured presentation (75WN, C4A2), solid and useful theoretical analyses (7QtD,75WN,q6tp,UsZq), as well as demonstrating algorithm's effectiveness in various datasets empirically (75WN, 7QtD).

We appreciate all reviewers’ constructive comments and questions; We have revised our paper based on their suggestions, which greatly improve our paper. We summarize all the major changes that we have made below. All changes have been reflected in our updated main manuscript and supplementary materials.

* We have conducted additional experiments on D4RL benchmarks (walker2d, hopper, halfcheetah, and maze2D) for better positioning our work within the existing literature. Additionally, we have added a strong baseline TD3+BC [1] for comparison. The results are provided in Table 1 and Table 3 of the attached one-page PDF, respectively.
* We have performed ablation studies on investigating the effect of algorithm components: uncertainty control and Bellman consistency. Please see Figure 3 of the attached one-page PDF for detailed results.
* We have strengthened arguments on the choice of hyperparameters and conducted the corresponding sensitivity analysis. The results are included in Figure 2 of the attached one-page PDF.
* We have conducted numerical experiments to empirically validate our theoretical results. Please see Figure 1 of the attached one-page PDF for detailed results.
* We have extended the explanation and interpretation of assumptions, conditions, and theoretical results.

**Reference:**

[1] Fujimoto, S., and Gu, S. S. (2021). A minimalist approach to offline reinforcement learning. Advances in neural information processing systems, 34, 20132-20145.

---

### Decision · Program_Chairs · 2023-09-21

**Decision:**

Accept (poster)

**Comment:**

This paper introduces a bi-level framework for offline reinforcement learning (RL) to tackle the challenges of distributional shift and function approximation. Unlike previous methods, it does not require coverage and completeness assumptions. The framework consists of two levels: the lower level focuses on creating a confidence set for value estimates to manage uncertainty due to distribution mismatch, while the upper level aims to optimize a policy using a pessimistic value function. Theoretical guarantees are provided, and the effectiveness of the algorithm is empirically validated. The authors have provided a detailed response to the concerns raised by the reviewers. We encourage the authors to add the new discussion regarding the significance of the regret and new empirical results into the revised final version of the paper.